

# Non-invertible symmetry webs

Lakshya Bhardwaj[1], Lea E. Bottini[1], Sakura Schäfer-Nameki[1] and Apoorv Tiwari[2]

**1** Mathematical Institute, University of Oxford, Andrew-Wiles Building,
Woodstock Road, Oxford, OX2 6GG, UK
**2** Department of Physics, KTH Royal Institute of Technology, Stockholm, Sweden

## Abstract

Non-invertible symmetries have by now seen numerous constructions in higher dimensional Quantum Field Theories (QFT). In this paper we provide an in depth study of gauging 0-form symmetries in the presence of non-invertible symmetries. The starting point of our analysis is a theory with $G$ 0-form symmetry, and we propose a description of sequential partial gaugings of sub-symmetries. The gauging implements the theta-symmetry defects of the companion paper [1]. The resulting network of symmetry structures related by this gauging will be called a *non-invertible symmetry web*. Our formulation makes direct contact with fusion 2-categories, and we uncover numerous interesting structures such as symmetry fractionalization in this categorical setting. The complete symmetry web is derived for several groups $G$, and we propose extensions to higher dimensions. The highlight of this analysis is the complete categorical symmetry web, including non-invertible symmetries, for 3d pure gauge theories with orthogonal gauge groups and its extension to arbitrary dimensions.



# 1  Introduction

In light of the rapid advances in the development of non-invertible symmetries in higher-dimensional quantum field theories, the uninitiated may find themselves on the backfoot, trying to grasp and keep up with the many constructions, examples and multitude of papers on this subject [2–32]. In [1] we propose a unified perspective on such matters, which organizes a variety of these constructions into a single overarching framework. The main actors of this

unified construction are referred to as *(twisted) theta defects*. These are produced by gauging topological defects (stacked with TQFTs) of another theory, along with choices of gauge-invariant couplings for these topological defects to bulk gauge fields for the symmetry that is being gauged. This perspective leads to a unified approach for characterizing non-invertible symmetries and their categorical description.

The goal of this paper is to develop a computational framework to exposit this construction of non-invertible defects as twisted theta-symmetries. Concretely, we will develop the tools to gauge arbitrary 0-form symmetries in 2-categories, and determine the 2-category after gauging. The key here is that the initial, 'pre-gauged', category is not necessarily only given in terms of invertible defects. In this sense it extends the gauging of 0-form symmetries of non-normal subgroups in [5] and the gauging of the full 0-form symmetry groups in [14, 15].

The construction is mostly relevant for quantum field theories (QFTs) in 3d, but applicable to a subsector of the symmetry category also in higher dimensions. Very broadly speaking, we consider the topological defects of a theory, which generate its symmetries [33]. The collection of topological surfaces, lines and point operators has the structure of a fusion 2-category [34]. In a 3d QFT, the surface defects generate the 0-form symmetry and lines the 1-form symmetry, however the fusion of these may not necessarily obey a group-like multiplication law. This forces the extension of the symmetry to a fusion 2-category. In the following we will develop how to gauge invertible 0-form symmetries, even if the underlying 2-category is non-invertible, i.e. does not have group-like fusions.

Gauging 0-form symmetries in general 2-categories, in particular in the presence of non-trivial topological lines, is far more subtle for various reasons:

1. More options for implementing the 0-form symmetry: The presence of topological lines results in a much richer way of implementing the 0-form symmetry action. Furthermore, the 1-form symmetry generated by line defects can be gauged on a surface, thus resulting in condensation defects.

2. Symmetry fractionalization: In the presence of lines, symmetries can fractionalize, which in the categorical setting results in the presence of certain non-trivial associators, characterized by 4-cocycles. This symmetry fractionalization results in subtle constraints on the gauging process.

Symmetry fractionalization in higher-dimensional QFTs has recently been discussed in [35–37]. We provide a framework to study the full 2-categorical structure, determining surface defects (objects in the category), including condensation defects, the fusion of surfaces, and fusion of lines.

The construction is developed starting with a category that is fully invertible, with only a 0-form symmetry given by a finite group $G$. This is realized by a 2-category which has topological surfaces (objects) that satisfy the $G$-fusion rules and no non-trivial topological lines

$$2\text{-Vec}(G): \qquad \text{Obj} = \{D_2^{(g)}, \ g \in G\}. \tag{1}$$

Throughout the paper we denote topological defects of dimension $p$ by $D_p$.

We then gauge at first a subgroup $H \subseteq G$ – this can be normal or not – which results in a category with topological lines and non-invertible fusions for some of its topological surfaces. This is an extension of the full gauging of $G$ in [14, 15]. Gauging normal $H$ is performed in section 2 (with a detailed analysis of the so-called bimodules that are relevant in this gauging in appendix A).

The non-trivial further step is to then gauge a subgroup of the remaining invertible 0-form symmetry (which would be $G/H$ if $H$ is normal) in this already once-gauged category. In general, one can consider gauging subsequently all the possible subgroups of the 0-form

Table 1: The categorical symmetry webs that we discuss in depth in this paper: $G$ labels the web and the 0-form symmetry of the starting point theory $\mathfrak{T}_G$.

| $G$ | Web in figure | $\mathfrak{T}_G$ |
|:---:|:---:|:---:|
| $\mathbb{Z}_2 \times \mathbb{Z}_2$ | 2 | PSO(4N) |
| $\mathbb{Z}_4$ | 2 | PSO($4N+2$) |
| $S_3$ | 11 | PSU(3) |
| $D_8$ | 1 and 13 | PSO($2N$) |
| $D_8$(any dim) | 19 and 20 | PSO($2N$) |

symmetry $G$, and proceeding in this way one obtains a categorical symmetry web of theories related by *invertible* gauging operations.[1]

The central new development here is the subsequent gauging, which requires us to study the implications which the presence of lines and potential symmetry fractionalization can entail. We illustrate the main conceptual points through the exploration of several symmetry webs, which are labeled by the 0-form symmetry of the starting point, $G$, and example theories $\mathfrak{T}_G$ with such symmetry. These are summarized in table 1.

In figure 1 we present the $D_8$-web, which exemplifies well the complexity of these structures. This can be realized on the class of 3d gauge theories with gauge algebra $\mathfrak{so}(4N)$. The same web also occurs for $\mathfrak{so}(4N+2)$, where the categorical web gets overlaid over a different assignment of global forms of the gauge group – see figure 13. The arrows in both figures correspond to gaugings of 0-form symmetries. Only the category associated to PSO($2N$) has solely 0-form symmetry – given by $G = D_8$. All subsequent categories are obtained by partial gauging of 0-form symmetry subgroups of $D_8$.

Let us discuss some of the salient features by considering the two examples $G = \mathbb{Z}_2 \times \mathbb{Z}_2$ and $\mathbb{Z}_4$ briefly as an appetizer. Gauging the full group $G$ results in 2-Rep($G$) as the symmetry category [14,15,38]. What we wish to understand now is how to obtain the same result by gauging stepwise, i.e. by subsequently gauging $\mathbb{Z}_2$s.

The first puzzle that arises is how the two categories, one coming from $G = \mathbb{Z}_2 \times \mathbb{Z}_2$ and the other from $G = \mathbb{Z}_4$, are differentiated after gauging one $\mathbb{Z}_2$ subgroup. The two groups are distinguished in terms of the extension class

$$\epsilon \in H^2(\mathbb{Z}_2, \mathbb{Z}_2) = \mathbb{Z}_2, \tag{2}$$

where the trivial elements describes $G = \mathbb{Z}_2 \times \mathbb{Z}_2$ and non-trivial one $G = \mathbb{Z}_4$. Field theoretically, it is clear [39] that a non-trivial (non-split) extension yields a mixed anomaly between the dual 1-form symmetry $\widehat{\mathbb{Z}_2}$ and the remaining $\mathbb{Z}_2$ 0-form symmetry. We will show how this is implemented in the 2-categorical framework, and identify its imprint as a symmetry fractionalization of the 0-form symmetry. This has an extensive history in the math and condensed matter literature (see [40–47] and references therein), which we revisit in the light of fusion 2-categories.

This plays a crucial role in the subsequent gauging of the remaining $\mathbb{Z}_2$, where the presence of the symmetry fractionalization results in the absence of certain topological defects for $G = \mathbb{Z}_4$ and lands us correctly on 2-Rep($\mathbb{Z}_4$) as the final category (which indeed has less simple objects than 2-Rep($\mathbb{Z}_2 \times \mathbb{Z}_2$)).

We should note that additional generalizations of the webs that we discuss exist: for instance adding discrete theta angles, and it may be interesting to consider these as an additional

---

[1]We leave to future work the investigation of the possible gaugings of non group-like 0-form symmetries, which are associated to non-invertible codimension-1 defects. These will typically arise in the categorical symmetry webs as condensation defects or from the gauging of non-normal subgroups.



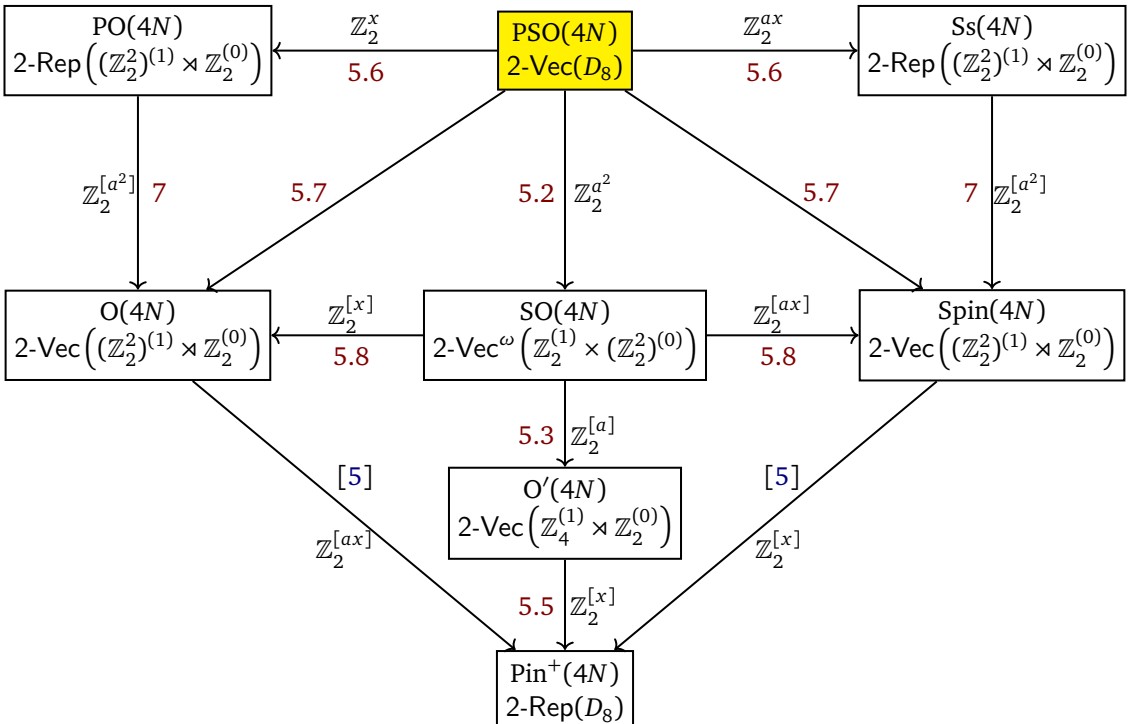

Figure 1: Categorical symmetry web for 3d gauge theories with gauge algebra $\mathfrak{so}(4N)$. We label each theory by its global gauge group, and the 2-category which is its symmetry category. The arrows denote gaugings of 0-form symmetries and each arrow is labelled by the subgroup of $D_8$ that is gauged along that arrow. We discuss each of these steps in the text. The notation for the groups $\mathbb{Z}_2^{[g]}$ is explained in the text around (130). The section labels indicate where the particular gauging is discussed.

refinement of the theories we have studied (for a field theory discussion and the possible theta angles in the setup of $\mathfrak{so}$ gauge theories in 3d see [48]).

In figure 2 we summarize the categorical web for $\mathbb{Z}_4$ (and contrast it with the one for $\mathbb{Z}_2 \times \mathbb{Z}_2$). The three 3d QFTs with $G = \mathbb{Z}_4$ here could e.g. be $\mathfrak{T}_1 = PSU(4)$, $\mathfrak{T}_2 = SU(4)/\mathbb{Z}_2$ and $\mathfrak{T}_3 = SU(4)$ gauge theory. Doing this step-wise gauging agrees with the gauging of the $\mathbb{Z}_4$ group directly, which results in 2-Rep($\mathbb{Z}_4$) [14] (we have detailed this in appendix B.2).

We begin the analysis by discussing some of the basics of 2-categories, the gauging of normal subgroups and the symmetry fractionalization in section 2. This section can be read in tandem with the appendix A, where a bimodule analysis of the gauging is provided. In section 3 we then turn to gauging the remaining group $G/H$, in the presence of already non-invertible symmetries. The two examples $G = \mathbb{Z}_2 \times \mathbb{Z}_2$ and $\mathbb{Z}_4$ accompany the general discussion in these two sections. In section 4 and 5 we derive the full categorical symmetry webs for $G = S_3, D_8$ including non-normal and multiple sequential gaugings. Extensions to higher dimensions are discussed in section 6 and we conclude in section 7. Appendices provide further details: appendix A discusses the 2-representation and bimodule perspective. Appendix B summarizes the full gauging of $G$, expanding on the analysis in [14]. In appendix C we discuss 1-morphisms in various categories that can e.g. implement subtle isomorphisms between various objects.

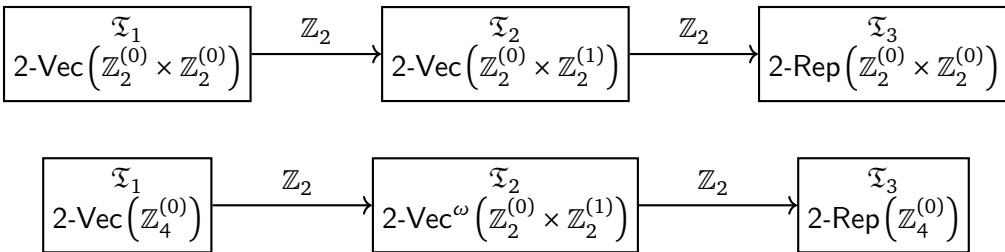

Figure 2: Categorical symmetry web for 3d gauge theories starting with the theory $\mathfrak{T}_1$ with 0-form symmetry $\mathbb{Z}_2 \times \mathbb{Z}_2$ (top) and $\mathbb{Z}_4$ (bottom), and gauging subsequently $\mathbb{Z}_2$. The category in the middle for theory $\mathfrak{T}_2$ has a trivial (top) and non-trivial (bottom) cocycle $\omega$, which is encoded in the group extension class $\epsilon$. This cocycle is key in order to obtain 2-Rep($\mathbb{Z}_4$) after gauging the remaining $\mathbb{Z}_2$ to reach theory $\mathfrak{T}_3$. For $G = \mathbb{Z}_2 \times \mathbb{Z}_2$ the main difference is in the absence of the non-trivial cocycle, i.e. $\epsilon$ is trivial.

**Notation.**

We will use $G$ throughout to refer to a finite group, which acts as the 0-form symmetry. A higher-group symmetry will be generally denoted by $\Gamma$, and $\Gamma^{(p)}$ is the $p$-form symmetry part of the higher-group $\Gamma$.

All topological defects of dimension $p$ are denoted by $D_p$ (possibly with extra embellishments). For a fusion 2-category $\mathcal{C}$ we denote the simple objects by $\mathrm{Obj}(\mathcal{C})$. The 1-endomorphisms of an object $D_2$ in $\mathcal{C}$ will be denoted by $\mathrm{End}_{\mathcal{C}}(D_2)$.

## 2  Gauging normal subgroups

In this paper we apply the methods discussed in the companion paper [1] to obtain the symmetry 2-categories of 3d QFTs comprising of *twisted theta defects* obtainable from the symmetry 2-category $\mathcal{C}_G := 2\text{-Vec}(G)$ of a 3d QFT $\mathfrak{T}_G$ with a non-anomalous $G$ 0-form symmetry by gauging[2] various (possibly non-normal) subgroups of $G$. Here $G$ will always be a finite (possibly non-abelian) group. Let us denote the 3d QFT obtained by gauging a subgroup $H$ of $G$ as $\mathfrak{T}_{G/H}$ (even if $H$ is not a normal subgroup) and the resulting symmetry 2-category $\mathcal{C}_{G/H}$.

It should be noted that we also discuss 'sequential gaugings'. The simplest example of sequential gauging arises if $\mathfrak{T}_{G/H}$ can be obtained from $\mathfrak{T}_{G/H'}$ by gauging a $H''$ 0-form symmetry sitting inside its symmetry 2-category $\mathcal{C}_{G/H'}$, or schematically if we have $\mathfrak{T}_{G/H} = \mathfrak{T}_{G/H'/H''}$. Then we also describe how $H''$ gauging can be implemented to convert the 2-category $\mathcal{C}_{G/H'}$ into the 2-category $\mathcal{C}_{G/H}$. We will see that this sequential gauging procedure involves many interesting subtleties due to the phenomenon of $H''$ *symmetry fractionalization* in the 2-category $\mathcal{C}_{G/H'}$.

Thus, incorporating such and more complicated sequential gaugings involving multiple steps in the gauging sequence, we can describe the most general goal of this paper as follows: this paper studies all possible invertible 0-form symmetry gaugings relating different symmetry 2-categories of the form $\mathcal{C}_{G/H}$ for different values of $H$ (but a fixed $G$). In other words, this paper explains all the arrows involving 0-form gaugings in the symmetry web formed by the symmetry 2-categories $\mathcal{C}_{G/H}$ for fixed $G$.

In this section, we discuss the first gauging $G \to G/H$ in a gauging sequence[3] for $H$ a

---

[2]All gaugings are performed without any possible $H^3(G, U(1))$ valued torsion.

[3]Note that the full gauging $G \to G/G$ leading to the symmetry 2-category $\mathcal{C}_{G/G} = 2\text{-Rep}(G)$ from the symmetry

normal[4] subgroup of $G$. We will sketch the general procedures and concretely exemplify them with the examples of $G = \mathbb{Z}_2 \times \mathbb{Z}_2$, $H = \mathbb{Z}_2$ and $G = \mathbb{Z}_4$, $H = \mathbb{Z}_2$.

In this section, we will focus mostly on the identification of simple objects (upto isomorphism) of $\mathcal{C}_{G/H}$ and the fusion rules of such simple objects. Of course, the general computational procedure can be equally well used to determine higher-layers of categorical information regarding 1-morphisms, their composition and fusion rules, but in order to keep the main parts of the paper pedagogical and free of clutter, we relegate such discussions to appendix C.

## 2.1 2-category associated to 0-form symmetries

Let us begin with the discussion of the initial symmetry 2-category

$$\mathcal{C}_G = 2\text{-Vec}(G), \tag{3}$$

carried by a 3d QFT $\mathfrak{T}_G$ having a non-anomalous 0-form symmetry group $G$.

Throughout this paper we will use the notation

$$D_p^{(a)} : \quad p\text{-dimensional topological operators labeled by } a. \tag{4}$$

The 0-form symmetry is generated by topological codimension 1 operators, and hence $\mathfrak{T}_G$ carries topological surface operators labeled by elements of $G$, which we denote by

$$D_2^{(g)}, g \in G \tag{5}$$

These form simple objects (upto isomorphism) of $\mathcal{C}_G$ and satisfy group-like fusion rules

$$D_2^{(g)} \otimes D_2^{(h)} = D_2^{(gh)}, \qquad g, h \in G. \tag{6}$$

Each simple object $D_2^{(g)}$ carries a single simple 1-endomorphism (upto isomorphism) in $\mathcal{C}_G$ which is identified with the trivial/identity line $D_1^{(g;\text{id})}$ on the surface $D_2^{(g)}$.

$$D_1^{(g;\text{id})} : \qquad D_2^{(g)} \to D_2^{(g)} \tag{7}$$

There are no 1-morphisms between the objects $D_2^{(g)}$ and $D_2^{(g')}$ in $\mathcal{C}_G$ for $g \neq g'$. That is, the 2-category $\mathcal{C}_G$ does not involve any possible line operators between $D_2^{(g)}$ and $D_2^{(g')}$, which may exist when $\mathfrak{T}$ only admits a group $H$ of 'faithful' 0-form symmetries and we use a projection $G \to H$ to find the operators $D_2^{(g)}$. In this paper, we will assume that the 3d QFT $\mathfrak{T}_G$ has a faithful $G$ 0-form symmetry for simplicity. However, it should be noted that considerations of

---

2-category $\mathcal{C}_G = 2\text{-Vec}(G)$ was discussed in great detail in the recent paper [14] and so we do not repeat that discussion here.

[4]Non-normal subgroups can be treated using similar techniques. Explicit treatments of gaugings of non-normal subgroups $H$ can be found for $G = S_3$ and $G = D_8$ in subsequent sections.

this paper are also applicable to non-faithful $G$ 0-form symmetry cases, but require a more refined interpretation.

There is a single 2-endomorphism in $\mathcal{C}_G = 2\text{-Vec}(G)$ of the 1-morphism $D_1^{(g;\text{id})}$ which corresponds to identity local operators living on the surface $D_2^{(g)}$, and there are no 2-endomorphisms between $D_1^{(g;\text{id})}$ and $D_1^{(g';\text{id})}$ for $g \neq g'$.

**Example $G = \mathbb{Z}_2 \times \mathbb{Z}_2$.**   One of our examples in this section will be $G = \mathbb{Z}_2 \times \mathbb{Z}_2$. An example of a 3d QFT with this 0-form symmetry is pure gauge theory with gauge group $\text{PSO}(4N)$, where $G = \mathbb{Z}_2 \times \mathbb{Z}_2$ is identified with the magnetic 0-form symmetry[5] of this QFT, arising from the fact that $\text{PSO}(4N)$ admits the construction

$$\text{PSO}(4N) = \text{Spin}(4N)/\mathbb{Z}_2 \times \mathbb{Z}_2 \,, \tag{8}$$

in terms of the associated simply connected group $\text{Spin}(4N)$.

Thus, in this case, the initial symmetry 2-category is

$$\mathcal{C}_{\mathbb{Z}_2 \times \mathbb{Z}_2} = 2\text{-Vec}(\mathbb{Z}_2 \times \mathbb{Z}_2) \,. \tag{9}$$

We denote its simple objects (upto isomorphism) by

$$\text{Obj}(\mathcal{C}_{\mathbb{Z}_2 \times \mathbb{Z}_2}) = \left\{ D_2^{(\text{id})}, D_2^{(S)}, D_2^{(C)}, D_2^{(V)} \right\} \,, \tag{10}$$

corresponding to the topological surfaces generating the $G = \mathbb{Z}_2 \times \mathbb{Z}_2$ symmetry. Here $D_2^{(\text{id})}$ is the identity object and the order 2 elements are

$$D_2^{(x)} \otimes D_2^{(x)} = D_2^{(\text{id})} \,, \qquad x = S, C, V \,, \tag{11}$$

with mutual fusion rule

$$D_2^{(S)} \otimes D_2^{(C)} = D_2^{(V)} \,. \tag{12}$$

**Example $G = \mathbb{Z}_4$.**   Another example studied in this section is $G = \mathbb{Z}_4$. An example of a 3d QFT with this 0-form symmetry is pure gauge theory with gauge group $\text{PSO}(4N + 2)$, where $G = \mathbb{Z}_4$ is identified with the magnetic 0-form symmetry[6] of this QFT, arising from the fact that $\text{PSO}(4N + 2)$ admits the construction

$$\text{PSO}(4N + 2) = \text{Spin}(4N + 2)/\mathbb{Z}_4 \,, \tag{13}$$

in terms of the associated simply connected group $\text{Spin}(4N + 2)$.

The associated symmetry category is

$$\mathcal{C}_{\mathbb{Z}_4} = 2\text{-Vec}(\mathbb{Z}_4) \,, \tag{14}$$

whose simple objects (upto isomorphism) are denoted by

$$\text{Obj}(\mathcal{C}_{\mathbb{Z}_4}) = \left\{ D_2^{(\text{id})}, D_2^{(S)}, D_2^{(V)}, D_2^{(C)} \right\} \,, \tag{15}$$

with the fusion relations

$$D_2^{(S)} \otimes D_2^{(S)} = D_2^{(V)} \,, \qquad \left( D_2^{(S)} \right)^{\otimes 3} = D_2^{(C)} \,, \qquad \left( D_2^{(S)} \right)^{\otimes 4} = D_2^{(\text{id})} \,. \tag{16}$$

---

[5]There is also a $\mathbb{Z}_2$ charge-conjugation type symmetry arising from outer-automorphisms of the Lie algebra $\mathfrak{so}(4N)$, which we do not take into account at the moment. This symmetry will be accounted in section 5 where it will enhance the $\mathbb{Z}_2 \times \mathbb{Z}_2$ 0-form symmetry considered here to $D_8 = (\mathbb{Z}_2 \times \mathbb{Z}_2) \rtimes \mathbb{Z}_2$ 0-form symmetry.

[6]There is again a $\mathbb{Z}_2$ charge-conjugation type symmetry arising from outer-automorphisms of the Lie algebra $\mathfrak{so}(4N+2)$, which we do not take into account at the moment. This symmetry will be accounted in section 5 where it will enhance the $\mathbb{Z}_4$ 0-form symmetry considered here to $D_8 = \mathbb{Z}_4 \rtimes \mathbb{Z}_2$ 0-form symmetry.

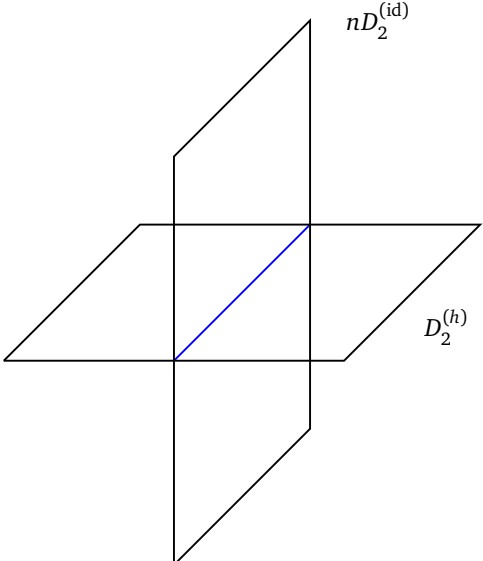

Figure 3: In order to make $nD_2^{(\mathrm{id})}$ $H$-symmetric, we need to choose line operators (shown in blue) living at the junction of $D_2^{(h)}$ for all $h \in H$ and $nD_2^{(\mathrm{id})}$.

## 2.2 Surface defects after partial gauging

Consider gauging a normal subgroup $H \triangleleft G$. The 3d QFT obtained after gauging is labeled as $\mathfrak{T}_{G/H}$. The gauging procedure converts the symmetry 2-category $\mathcal{C}_G$ carried by $\mathfrak{T}_G$ to a symmetry 2-category $\mathcal{C}_{G/H}$ carried by $\mathfrak{T}_{G/H}$. We are interested in the determination of $\mathcal{C}_{G/H}$.

**Theta defects.** From the analysis of the companion paper [1], we know that $\mathcal{C}_{G/H}$ comprises of $H$-symmetric objects and morphisms in $\mathcal{C}_G$. Let us begin by exploring various ways of making multiples of identity object $D_2^{(\mathrm{id})}$ $H$-symmetric. Consider for example making the object $nD_2^{(\mathrm{id})}$ $H$-symmetric. We need to choose junction lines between $nD_2^{(\mathrm{id})}$ and $D_2^{(h)}$ for all $h \in H$ (see figure 3) such that these junction lines can be freely rearranged (without the appearance of any extra phases) along the worldvolume of $nD_2^{(\mathrm{id})}$. By folding the $D_2^{(h)}$ surfaces away, the junction lines can be identified with lines living on the worldvolume of $nD_2^{(\mathrm{id})}$. Since there are no non-trivial associators for $D_2^{(h)}$ surfaces with $nD_2^{(\mathrm{id})}$, the folding procedure commutes with rearrangements of junctions. Hence, a choice of making $nD_2^{(\mathrm{id})}$ $H$-symmetric is a choice of lines living on $nD_2^{(\mathrm{id})}$ labeled by elements of $H$, which can be freely rearranged. Mathematically, this is a choice of a functor

$$\mathcal{S}: \mathcal{C}_1^{\Gamma^{(0)}=H} \to \mathsf{Mat}_n(\mathsf{Vec}), \tag{17}$$

where $\mathcal{C}_1^{\Gamma^{(0)}=H}$ is a non-linear 1-category discussed at the beginning of section 3.2 of the companion paper [1] capturing abstract properties of freely arrangable lines parametrized by elements of $H$, and $\mathsf{Mat}_n(\mathsf{Vec})$ is the multi-fusion 1-category formed by $n \times n$ matrices valued in the category Vec of finite-dimensional vector spaces which captures the lines living on $nD_2^{(\mathrm{id})}$. The $(i, j)$-th element of such a matrix describes a line from the $i$-th copy of $D_2^{(\mathrm{id})}$ in $nD_2^{(\mathrm{id})}$ to the $j$-th copy of $D_2^{(\mathrm{id})}$ in $nD_2^{(\mathrm{id})}$. As explained in [1], such a functor $\mathcal{S}$ describes a 2-representation of $H$. The collection of all such functors for all possible $n$ forms a fusion 2-category 2-Rep($H$). We thus find a 2-subcategory

$$\text{2-Rep}(H) \subseteq \mathcal{C}_{G/H}. \tag{18}$$

An alternative perspective is obtained by recognizing $nD_2^{(\mathrm{id})}$ as the defect obtained by stacking a 2d TQFT[7] with $n$ trivial vacua on top of $\mathfrak{T}_G$. In fact, all defects arising by stacking 2d TQFTs can be identified with $nD_2^{(\mathrm{id})}$, because all 2d TQFTs are essentially determined by their number of vacua $n$. Thus, the various ways of making $nD_2^{(\mathrm{id})}$ $H$-symmetric for various values of $n$ are parametrized by by $H$-symmetric 2d TQFTs, which as discussed in [14] is the same problem as determining functors of the form (17).

Thus, we can understand the topological surfaces of $\mathfrak{T}_{G/H}$ lying in 2-Rep$(H) \subseteq \mathcal{C}_{G/H}$ as being obtained by first stacking an $H$-symmetric TQFT on top of the spacetime occupied by $\mathfrak{T}_G$, and then gauging the combined/diagonal $H$ symmetry. In the language of [1], the topological defects in the 2-subcategory 2-Rep$(H)$ of $\mathcal{C}_{G/H}$ are **theta defects** of $\mathfrak{T}_{G/H}$.

**Other defects in $\mathcal{C}_{G/H}$.** Other defects in $\mathcal{C}_{G/H}$ arise as **twisted theta defects**. First, consider making multiples of $D_2^{(h)}$, for $h \in H$, $H$-symmetric. These lead to objects isomorphic to the objects already contained in the 2-subcategory 2-Rep$(H) \subseteq \mathcal{C}_{G/H}$ because we can convert $D_2^{(h)}$ into $D_2^{(\mathrm{id})}$ by multiplication by elements of $H$.

By same argument, we only need to study $H$-symmetrization of a single defect $D_2^{(g)}$ for a single element $g$ lying in each coset $k \in K := G/H$. Since we are dealing with a non-anomalous $G$ symmetry, there are no associators for topological defects generating $H$ in the presence of any $D_2^{(g)}$. Consequently, we obtain a copy of 2-Rep$(H)$ inside $\mathcal{C}_{G/H}$ for each element $k \in K$.

In total, we learn that the objects of $\mathcal{C}_{G/H}$ are the same as the objects of the 2-category 2-Vec$(K) \boxtimes$ 2-Rep$(H)$. However, we will see later that

$$\mathcal{C}_{G/H} \neq \text{2-Vec}(K) \boxtimes \text{2-Rep}(H). \tag{19}$$

The equality holds if and only if $G = H \times K$.

We will label the objects of $\mathcal{C}_{G/H}$ as $D_2^{(kR)}$ where $k \in K$ and $R$ a 2-representation of $H$. In fact, there exists at least one 1-morphism (none of which is an isomorphism) from $D_2^{(kR)}$ to $D_2^{(k)}$ (obtained by choosing the trivial 2-representation) for every choice of 2-representation $R$. This is often captured by saying that $D_2^{(kR)}$ and $D_2^{(k)}$ lie in the same 'Schur component'. The existence of such a 1-morphism is equivalent to the fact that $D_2^{(kR)}$ can be obtained by condensing/gauging a (possibly non-invertible) symmetry localized along the worldvolume of $D_2^{(k)}$. Thus, Schur components can be thought of as capturing defects modulo condensations. The Schur components of the 2-category $\mathcal{C}_{G/H}$ are parametrized by elements of the group $K = G/H$.

We can recognize $D_2^{(kR)}$ as a twisted theta defect whose underlying *twist* is the defect $D_2^{(k)} \in \mathcal{C}_G$ and the underlying *stack* is the 2d TQFT with $n$ trivial vacua, where $n$ is the dimension of the 2-representation $R$.

**Examples.** Returning to our main examples, for either $G = \mathbb{Z}_2 \times \mathbb{Z}_2$ or $G = \mathbb{Z}_4$, consider gauging a $H = \mathbb{Z}_2$ subgroup. For $G = \mathbb{Z}_4$ there is a unique $\mathbb{Z}_2$ subgroup generated by $V \in \mathbb{Z}_4$. On the other hand, for $G = \mathbb{Z}_2 \times \mathbb{Z}_2$ there are three possible choices of $\mathbb{Z}_2$ subgroups generated by $x \in \{S, C, V\}$, but they are all equivalent. For maintaining consistency of notation with the $G = \mathbb{Z}_4$ case, we pick the $\mathbb{Z}_2$ subgroup of $G = \mathbb{Z}_2 \times \mathbb{Z}_2$ generated by $V$.

From the above discussion, the objects of $\mathcal{C}_{\mathbb{Z}_2 \times \mathbb{Z}_2/\mathbb{Z}_2}$ and $\mathcal{C}_{\mathbb{Z}_4/\mathbb{Z}_2}$ are the same and coincide

---

[7]In this paper, for pedagogical purposes, we are not precise about the distinction between 2d TQFTs and 2d non-anomalous topological orders. See the companion paper [1] for more details.

with the objects of 2-Vec($\mathbb{Z}_2$) ⊠ 2-Rep($\mathbb{Z}_2$). However, at the level of full 2-categories

$$\mathcal{C}_{\mathbb{Z}_2 \times \mathbb{Z}_2/\mathbb{Z}_2} = \text{2-Vec}(\mathbb{Z}_2) \boxtimes \text{2-Rep}(\mathbb{Z}_2),$$
$$\mathcal{C}_{\mathbb{Z}_4/\mathbb{Z}_2} \neq \text{2-Vec}(\mathbb{Z}_2) \boxtimes \text{2-Rep}(\mathbb{Z}_2). \tag{20}$$

Let us describe the objects of $\mathcal{C}_{\mathbb{Z}_2 \times \mathbb{Z}_2/\mathbb{Z}_2}$ and $\mathcal{C}_{\mathbb{Z}_4/\mathbb{Z}_2}$ in more detail. Since the discussion is same for both categories, we refer to them together. We refer to the category $\mathcal{C}_{\mathbb{Z}_2 \times \mathbb{Z}_2}$ or $\mathcal{C}_{\mathbb{Z}_4}$ by $\mathcal{C}_G$, and $\mathcal{C}_{\mathbb{Z}_2 \times \mathbb{Z}_2/\mathbb{Z}_2}$ or $\mathcal{C}_{\mathbb{Z}_4/\mathbb{Z}_2}$ by $\mathcal{C}_{G/\mathbb{Z}_2}$.

First of all, the object $D_2^{(\text{id})}$ of $\mathcal{C}_G$ leads to a single simple object (upto isomorphism) of $\mathcal{C}_{G/\mathbb{Z}_2}$ which we refer by the same name $D_2^{(\text{id})}$. This is because there is a single 2d $\mathbb{Z}_2$-symmetric SPT phase, namely the trivial $\mathbb{Z}_2$-symmetric 2d TQFT.

The object $2D_2^{(\text{id})}$ of $\mathcal{C}_G$ also leads to a single simple object (upto isomorphism) $D_2^{(\mathbb{Z}_2)}$ of $\mathcal{C}_{G/\mathbb{Z}_2}$. The $H = \mathbb{Z}_2$ symmetry on $2D_2^{(\text{id})}$ is generated by a 1-morphism $2D_2^{(\text{id})} \to 2D_2^{(\text{id})}$, which is implemented by a $2 \times 2$ matrix of 1-morphisms $D_2^{(\text{id})} \to D_2^{(\text{id})}$, with the $(i,j)$-th entry of the matrix describing a 1-morphism from the $i$-th copy of $D_2^{(\text{id})}$ to the $j$-th copy of $D_2^{(\text{id})}$.

In this language, the matrix describing the action of $H = \mathbb{Z}_2$ symmetry on $2D_2^{(\text{id})}$ for the construction of $D_2^{(\mathbb{Z}_2)}$ is

$$\begin{pmatrix} 0 & D_1^{(\text{id})} \\ D_1^{(\text{id})} & 0 \end{pmatrix} : \quad 2D_2^{(\text{id})} \to 2D_2^{(\text{id})}, \tag{21}$$

where $D_1^{(\text{id})} : D_2^{(\text{id})} \to D_2^{(\text{id})}$ is the identity 1-endomorphism of $D_2^{(\text{id})}$ corresponding to the identity line defect in $\mathfrak{T}_G$. These objects $D_2^{(\text{id})}, D_2^{(\mathbb{Z}_2)}$ generate the 2-subcategory 2-Rep($\mathbb{Z}_2$) $\subseteq \mathcal{C}_{G/\mathbb{Z}_2}$.

Similarly, the object $D_2^{(S)}$ of $\mathcal{C}_G$ leads to a single simple object (upto isomorphism) of $\mathcal{C}_{G/\mathbb{Z}_2}$ which we refer by the same name $D_2^{(S)}$, and the object $2D_2^{(S)}$ of $\mathcal{C}_G$ also leads to a single simple object (upto isomorphism) $D_2^{(S\mathbb{Z}_2)}$ of $\mathcal{C}_{G/\mathbb{Z}_2}$. The $H = \mathbb{Z}_2$ symmetry on $2D_2^{(S)}$ is generated by

$$\begin{pmatrix} 0 & D_1^{(S;\text{id})} \\ D_1^{(S;\text{id})} & 0 \end{pmatrix} : \quad 2D_2^{(S)} \to 2D_2^{(S)}, \tag{22}$$

where $D_1^{(S;\text{id})} : D_2^{(S)} \to D_2^{(S)}$ is the identity 1-endomorphism of $D_2^{(S)}$ corresponding to the identity line defect living on $D_2^{(S)}$ in $\mathfrak{T}_G$. These objects $D_2^{(S)}, D_2^{(S\mathbb{Z}_2)}$ generate another copy of the 2-subcategory 2-Rep($\mathbb{Z}_2$) $\subseteq \mathcal{C}_{G/\mathbb{Z}_2}$.

In total we therefore have the simple objects (upto isomorphism)

$$\text{Obj}(\mathcal{C}_{\mathbb{Z}_2 \times \mathbb{Z}_2/\mathbb{Z}_2}) = \text{Obj}(\mathcal{C}_{\mathbb{Z}_4/\mathbb{Z}_2}) = \left\{ D_2^{(\text{id})}, D_2^{(\mathbb{Z}_2)}, D_2^{(S)}, D_2^{(S\mathbb{Z}_2)} \right\}. \tag{23}$$

## 2.3 Fusion of surface defects after partial gauging

The fusion of 2-representations converts the set of 2-representations into a twisted Burnside ring as discussed in detail in [14]. This controls the fusion of objects $D_2^{(R)} \in \text{2-Rep}(H) \subseteq \mathcal{C}_{G/H}$.

More generally, we have

$$D_2^{(k_1 R_1)} \otimes D_2^{(k_2 R_2)} = D_2^{\left(k_1 k_2 \left(R_1^{k_2} R_2\right)\right)}. \tag{24}$$

$R_1^{k_2}$ is a 2-representation obtained by applying the action of $k_2$ on $R_1$ (see [15] for more details) and $R_1^{k_2} R_2$ is the tensor product 2-representation of $R_1^{k_2}$ and $R_2$.

**Examples.** Let us determine the fusion rules of simple objects in our examples $G = \mathbb{Z}_2 \times \mathbb{Z}_2$, $H = \mathbb{Z}_2$ and $G = \mathbb{Z}_4, H = \mathbb{Z}_2$. Recall the objects (23).

It is easy to see that $D_2^{(\text{id})}$ is the identity object of $\mathcal{C}_{G/\mathbb{Z}_2}$. As $H = \mathbb{Z}_2$ acts trivially on $D_2^{(S)}$, the fusion follows from $\mathcal{C}_G$:

$$D_2^{(S)} \otimes D_2^{(S)} = D_2^{(\text{id})}, \tag{25}$$

which implies that $D_2^{(S)}$ generates a $\mathbb{Z}_2$ 0-form symmetry in the theory $\mathfrak{T}_{G/\mathbb{Z}_2}$.

On the other hand we have

$$D_2^{(\mathbb{Z}_2)} \otimes D_2^{(\mathbb{Z}_2)} = 2D_2^{(\mathbb{Z}_2)}. \tag{26}$$

To understand this fusion rule, we have to understand the combined $\mathbb{Z}_2$ action on the underlying defect $2D_2^{(\text{id})} \otimes 2D_2^{(\text{id})} \cong 4D_2^{(\text{id})} \in \mathcal{C}_G$ which is generated by the tensor product of the matrix of lines

$$\begin{pmatrix} 0 & D_1^{(\text{id})} \\ D_1^{(\text{id})} & 0 \end{pmatrix} \otimes \begin{pmatrix} 0 & D_1^{(\text{id})} \\ D_1^{(\text{id})} & 0 \end{pmatrix} = \begin{pmatrix} 0 & 0 & 0 & D_1^{(\text{id})} \\ 0 & 0 & D_1^{(\text{id})} & 0 \\ 0 & D_1^{(\text{id})} & 0 & 0 \\ D_1^{(\text{id})} & 0 & 0 & 0 \end{pmatrix}. \tag{27}$$

In more detail, we can label the underlying $D_2^{(\text{id})} \in \mathcal{C}_G$ objects of $D_2^{(\mathbb{Z}_2)}$ by $D_2^{(\text{id})(i)}$ for $i \in \{0, 1\}$. Then we can label the underlying $D_2^{(\text{id})} \in \mathcal{C}_{\mathfrak{T}}$ objects of $D_2^{(\mathbb{Z}_2)} \otimes D_2^{(\mathbb{Z}_2)}$ as $D_2^{(\text{id})(i,j)}$ for $i, j \in \{0, 1\}$. The combined $\mathbb{Z}_2$ acts

$$\begin{aligned} i &\to i + 1 \ (\text{mod } 2), \\ j &\to j + 1 \ (\text{mod } 2). \end{aligned} \tag{28}$$

Thus $D_2^{(\text{id})(0,0)}$ and $D_2^{(\text{id})(1,1)}$ are exchanged by $\mathbb{Z}_2$, and $D_2^{(\text{id})(0,1)}$ and $D_2^{(\text{id})(1,0)}$ are exchanged by $\mathbb{Z}_2$, leading again to the fusion rule (26).

The remaining fusion rules are[8]

$$\begin{aligned} D_2^{(S\mathbb{Z}_2)} \otimes D_2^{(\mathbb{Z}_2)} &= 2D_2^{(S\mathbb{Z}_2)}, \\ D_2^{(S\mathbb{Z}_2)} \otimes D_2^{(S\mathbb{Z}_2)} &= 2D_2^{(\mathbb{Z}_2)}, \\ D_2^{(S)} \otimes D_2^{(\mathbb{Z}_2)} &= D_2^{(S\mathbb{Z}_2)}, \\ D_2^{(S)} \otimes D_2^{(S\mathbb{Z}_2)} &= D_2^{(\mathbb{Z}_2)}. \end{aligned} \tag{29}$$

That is, the label $S$ is simply a $K = \mathbb{Z}_2$ grading on the fusion rules, which is because the conjugation action of $K = \mathbb{Z}_2$ on $H = \mathbb{Z}_2$ is trivial for both $G = \mathbb{Z}_2 \times \mathbb{Z}_2$ and $G = \mathbb{Z}_4$.

Thus, the fusion of objects of $\mathcal{C}_{\mathbb{Z}_2 \times \mathbb{Z}_2 / \mathbb{Z}_2} = 2\text{-Vec}(\mathbb{Z}_2) \boxtimes 2\text{-Rep}(\mathbb{Z}_2)$ and $\mathcal{C}_{\mathbb{Z}_4/\mathbb{Z}_2}$ are same. In addition to the fusion of objects, the 1-morphisms as well as their composition and fusion rules are also the same for $\mathcal{C}_{\mathbb{Z}_2 \times \mathbb{Z}_2 / \mathbb{Z}_2} 2\text{-Vec}(\mathbb{Z}_2) \boxtimes 2\text{-Rep}(\mathbb{Z}_2)$ and $\mathcal{C}_{\mathbb{Z}_4/\mathbb{Z}_2}$ (see appendix C.1 for details). Still, equation (20) claims that

$$\mathcal{C}_{\mathbb{Z}_4/\mathbb{Z}_2} \neq \mathcal{C}_{\mathbb{Z}_2 \times \mathbb{Z}_2 / \mathbb{Z}_2}. \tag{30}$$

What then is the difference between the two categories? The answer lies in the fact that $\mathcal{C}_{\mathbb{Z}_4/\mathbb{Z}_2}$ has some non-trivial associators, which are physically understood as the phenomenon of symmetry fractionalization, which is the subject of the next subsection.

## 2.4 Symmetry fractionalization on lines

From this subsection onward, we restrict $H$ to be an abelian group.

---

[8]Throughout this paper, when the fusion (or composition) rules are commutative, we only mention fusion rules with a single choice of order.

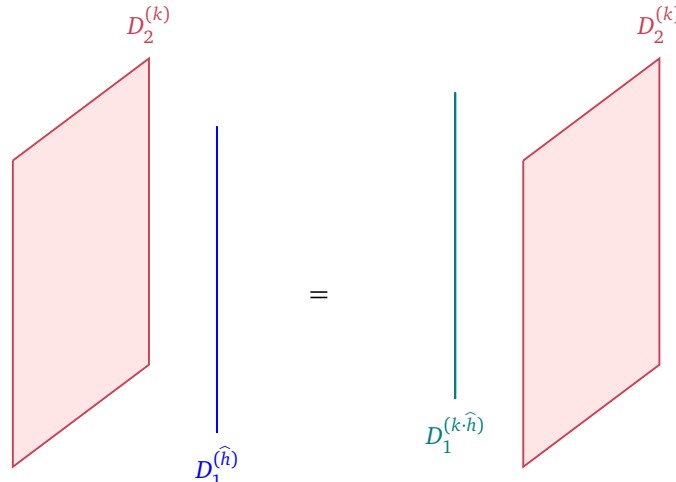

Figure 4: A 1-form symmetry generating line $D_1^{(\widehat{h})}$ (shown in blue) changes to another 1-form symmetry generating line $D_1^{(k\cdot\widehat{h})}$ (shown in teal) upon sliding it across a 0-form symmetry generating surface $D_2^{(k)}$ (shown in red). See equation (34).

**Data specifying embedding of $H$ in $G$.** To understand the distinction between $\mathcal{C}_{G/H}$ and 2-Vec$(K)\boxtimes$2-Rep$(H)$, we need to first explore the following abstract group-theoretic structure. Since $H$ is normal in $G$, there is a short exact sequence

$$1 \to H \to G \to K \equiv G/H \to 1\,. \tag{31}$$

The extension is characterized by the group cohomology (twisted by the conjugation action of $K$ on $H$) class

$$\epsilon \in H^2(K,H)\,. \tag{32}$$

As a set, we can write $G$ as pairs $(h,k)$ with $h \in H$ and $k \in K$ and product given by

$$(h,k) \times (h',k') = (h(k \circ h')\epsilon(k,k'),kk')\,, \tag{33}$$

where $(k \circ h') = kh'k^{-1}$ denotes the action of $K$ on $H$ by conjugation.

**Split 2-group.** It is well-known [33] that after gauging $H$ 0-form symmetry of $\mathfrak{T}_G$, we obtain in $\mathfrak{T}_{G/H}$ a dual 1-form symmetry with group $\widehat{H}$ which is the Pontryagin dual of $H$. If there is a non-trivial (conjugation) action of $K$ on $H$, then we have a non-trivial dual action of $K$ on $\widehat{H}$, implying that the dual 1-form symmetry $\widehat{H}$ and the residual 0-form symmetry $K$ combine to form a non-trivial (split) 2-group symmetry $\Gamma$ in which the 0-form symmetry has a non-trivial action on the 1-form symmetry.

This action is displayed in terms of topological defects generating the 0-form and 1-form symmetries in figure 4. In a categorical language, this means that we have the following non-commutativity in the fusion rules for 1-morphisms

$$D_1^{(k;\text{id})} \otimes D_1^{(\widehat{h})} = D_1^{(k\cdot\widehat{h})} \otimes D_1^{(k;\text{id})}\,. \tag{34}$$

Here $D_1^{(\widehat{h})}$ for $\widehat{h} \in \widehat{H}$ are simple 1-endomorphisms of $D_2^{(\text{id})}$ in $\mathcal{C}_{G/H}$ corresponding to topological lines generating the $\widehat{H}$ 1-form symmetry, $D_1^{(k;\text{id})}$ are identity 1-endomorphisms of objects $D_2^{(k)}$ carrying trivial 2-representation for $k \in K$ corresponding to identity lines on the surfaces $D_2^{(k)}$, and $k \cdot \widehat{h} \in \widehat{H}$ is the element obtained after the action of $k$ on $\widehat{h}$.

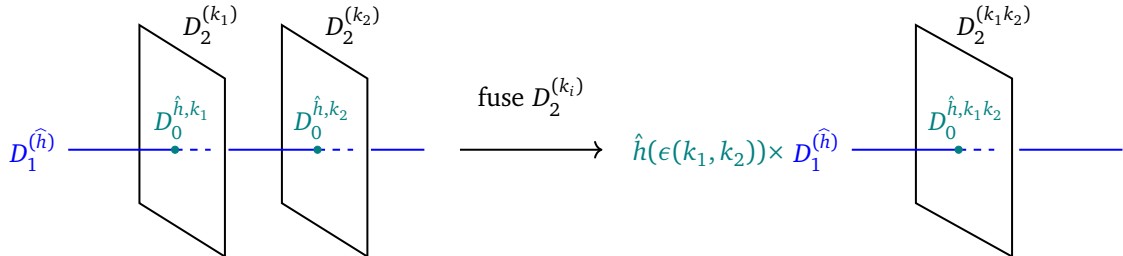

Figure 5: Symmetry Fractionalization: The junctions of topological surface operators $D_2^{(k_i)}$ generating the $K$ 0-form symmetry with a fixed line $D_1^{(\widehat{h})}$ generating the 1-form symmetry do not obey $K$ multiplication laws, but instead acquire a projectivity $\widehat{h}\big(\epsilon(k_1,k_2)\big) \in U(1)$.

The fusion rule (34) already differentiates between $\mathcal{C}_{G/H}$ and 2-Vec$(K) \boxtimes$ 2-Rep$(H)$ in special cases. However, when there is a trivial action of $K$ on $H$ as in our two examples $G = \mathbb{Z}_2 \times \mathbb{Z}_2, H = \mathbb{Z}_2$ and $G = \mathbb{Z}_4, H = \mathbb{Z}_2$, this does not provide the required distinction between the two categories. In fact we can modify the problem as follows: What is the distinction between the categories $\mathcal{C}_{G/H}$ and 2-Vec$(\Gamma)$, where $\Gamma$ is the split 2-group generated by $K$ 0-form symmetry acting on $\widehat{H}$ 1-form symmetry? When the action is trivial, we have 2-Vec$(\Gamma) = $ 2-Vec$(K) \boxtimes$ 2-Rep$(H)$ and so we reduce to our previous problem.

**'t Hooft anomaly and symmetry fractionalization.** The categories $\mathcal{C}_{G/H}$ and 2-Vec$(\Gamma)$ turn out to be equal only when we can write $G = H \rtimes K$, in which case the extension class $\epsilon$ discussed above vanishes. The extension class $\epsilon$ discussed above captures an 't Hooft anomaly for the 2-group $\Gamma$ [39] which can be expressed as

$$A^*\omega = B_2 \cup a^*\epsilon, \tag{35}$$

where $B_2$ is the background field for $H$ 1-form symmetry and $a^*$ is pullback under the background $a$ for $K$ 0-form symmetry.

The anomaly $A^*\omega$ descends from

$$\omega \in H^4(\Gamma, U(1)), \tag{36}$$

upon pull-back $A^*$ to spacetime, corresponding to a 2-group background $A$.

The element $\omega$ describes non-trivial associators (or in other words, coherence relations) in the category $\mathcal{C}_{G/H}$ distinguishing it from 2-Vec$(\Gamma)$. Using an often used notation, we can express

$$\mathcal{C}_{G/H} = \text{2-Vec}^\omega(\Gamma). \tag{37}$$

The anomaly (35) describes the fractionalization of $K$ 0-form symmetry as implemented upon topological line operators $D_1^{(\widehat{h})}$ generating $\widehat{H}$ 1-form symmetry. See figure 5.

**Derivation of symmetry fractionalization.** Let us now provide a derivation of this symmetry fractionalization phenomenon using the techniques used in this paper.

First of all, we need to understand the emergence of $\widehat{H}$ 1-form symmetry in the category $\mathcal{C}_{G/H}$. These are simple 1-endomorphisms of the identity object $D_2^{(\text{id})}$ of $\mathcal{C}_{G/H}$. Consequently, they correspond to various ways of making the identity 1-endomorphism $D_1^{(\text{id})}$ of $D_2^{(\text{id})}$ in $\mathcal{C}_G$ symmetric under $H$. Such different ways correspond to homomorphisms

$$H = \mathcal{C}_0^{\Gamma^{(0)}=H} \longrightarrow \text{End}_{\mathcal{C}_G}(D_1^{(\text{id})}) = \mathbb{C}, \tag{38}$$

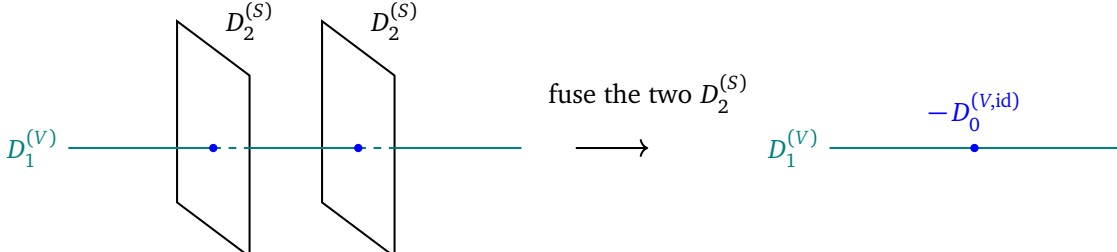

Figure 6: Symmetry Fractionalization in $\mathfrak{T}_{\mathbb{Z}_4/\mathbb{Z}_2}$: Fusing the junction of line $D_1^{(V)}$ with surface $D_2^{(S)}$ along the line $D_1^{(V)}$ produces the operator $-D_0^{(V,\text{id})}$, where $D_0^{(V,\text{id})}$ is the identity local operator on the line $D_1^{(V)}$. Thus the $\mathbb{Z}_2$ symmetry generated by $D_2^{(S)}$ is fractionalized to $\mathbb{Z}_4$ on $D_1^{(v)}$.

using the language appearing in the companion paper [1]. The non-linear 0-category $\mathcal{C}_0^{\Gamma^{(0)}}$ is the 0-form group $\Gamma^{(0)}$ itself and the vector space $\text{End}_{\mathcal{C}_G}(D_1^{(\text{id})})$ of 2-endomorphisms of the 1-morphism $D_1^{(\text{id})}$ in the 2-category $\mathcal{C}_G = 2\text{-Vec}(G)$ is simply $\mathbb{C}$. Such homomorphisms are described by elements of the Pontryagin dual group $\widehat{H}$, resulting in lines $D_1^{(\widehat{h})}$ in the category $\mathcal{C}_{G/H}$.

Now a junction local operator[9] $D_0^{(\widehat{h},k)}$ of $D_2^{(k)}$ with $D_1^{(\widehat{h})}$ in $\mathcal{C}_{G/H}$ can be uplifted to a junction of $D_2^{(1,k)}$ with $D_1^{(\text{id})}$ in $\mathcal{C}_G$, or in particular the identity operator $D_0^{(1,k;\text{id})}$ on $D_2^{(1,k)}$ in $\mathcal{C}_G$. The fusion of $D_0^{(k_1;\widehat{h})}$ with $D_0^{(k_2;\widehat{h})}$ in $\mathcal{C}_{G/H}$ as in figure 5 arises from the following fusion in $\mathcal{C}_G$

$$D_2^{(1,k_1)} \otimes D_2^{(1,k_2)} = D_2^{(\epsilon(k_1,k_2),1)} \otimes D_2^{(1,k_1 k_2)}. \tag{39}$$

Note that on the right hand side we have a surface operator $D_2^{(\epsilon(k_1,k_2),1)}$ valued purely in $H$. In the above definition of $D_1^{(\widehat{h})}$ line in terms of the data of the category $\mathcal{C}_G$, this surface operator acts by a phase $\hat{h}(\epsilon(k_1,k_2))$, leading to the fusion rule

$$D_0^{(\widehat{h},k_1)} \otimes D_0^{(\widehat{h},k_2)} = \widehat{h}(\epsilon(k_1,k_2)) D_0^{(\widehat{h},k_1 k_2)}, \tag{40}$$

in $\mathcal{C}_{G/H}$. We will provide a derivation of this from a bi-module perspective in appendix A.1.

**Examples.** The difference between trivial and non-trivial extension class is best illustrated in our examples. The relevant cohomology group is $H^2(\mathbb{Z}_2, \mathbb{Z}_2) = \mathbb{Z}_2$. The choice $G = \mathbb{Z}_2 \times \mathbb{Z}_2$, $H = \mathbb{Z}_2$ corresponds to the trivial cohomology class, while the choice $G = \mathbb{Z}_4, H = \mathbb{Z}_2$ corresponds to the non-trivial extension class $1 \neq \epsilon \in H^2(\mathbb{Z}_2, \mathbb{Z}_2) = \mathbb{Z}_2$.

There is a $\mathbb{Z}_2$ 1-form symmetry generated by an order two line $D_1^{(V)}$ in both $\mathfrak{T}_{\mathbb{Z}_2 \times \mathbb{Z}_2/\mathbb{Z}_2}$ and $\mathfrak{T}_{\mathbb{Z}_4/\mathbb{Z}_2}$. According to our above analysis, the 0-form $\mathbb{Z}_2$ symmetry of $\mathfrak{T}_{G/\mathbb{Z}_2}$ generated by the surface $D_2^{(S)}$ in $\mathcal{C}_{G/\mathbb{Z}_2}$ fractionalizes to a $\mathbb{Z}_4$ 0-form symmetry on the line $D_1^{(V)}$ for $G = \mathbb{Z}_4$. See figure 6.

## 2.5 Symmetry fractionalization on condensation surfaces

We saw in the previous subsection that the residual $K$ 0-form symmetry of $\mathcal{C}_{G/H}$ fractionalizes on lines $D_1^{(\widehat{h})}$ in $\mathcal{C}_{G/H}$ whose underlying line before $H$-gauging is the identity line $D_1^{(\text{id})}$ in $\mathcal{C}_G$.

---

[9]These junction operators are chosen to satisfy $D_0^{(\widehat{h}_1,k)} \otimes_{D_2^{(k)}} D_0^{(\widehat{h}_2,k)} = D_0^{(\widehat{h}_1 \widehat{h}_2,k)}$ where $\otimes_{D_2^{(k)}}$ denotes fusion of junction operators along the surface $D_2^{(k)}$. That is these junction operators are chosen to obey $\widehat{H}$ fusion rules for a fixed surface $D_2^{(k)}$, but as we will see they fail to obey $K$ fusion rules for a fixed line $D_1^{(\widehat{h})}$.

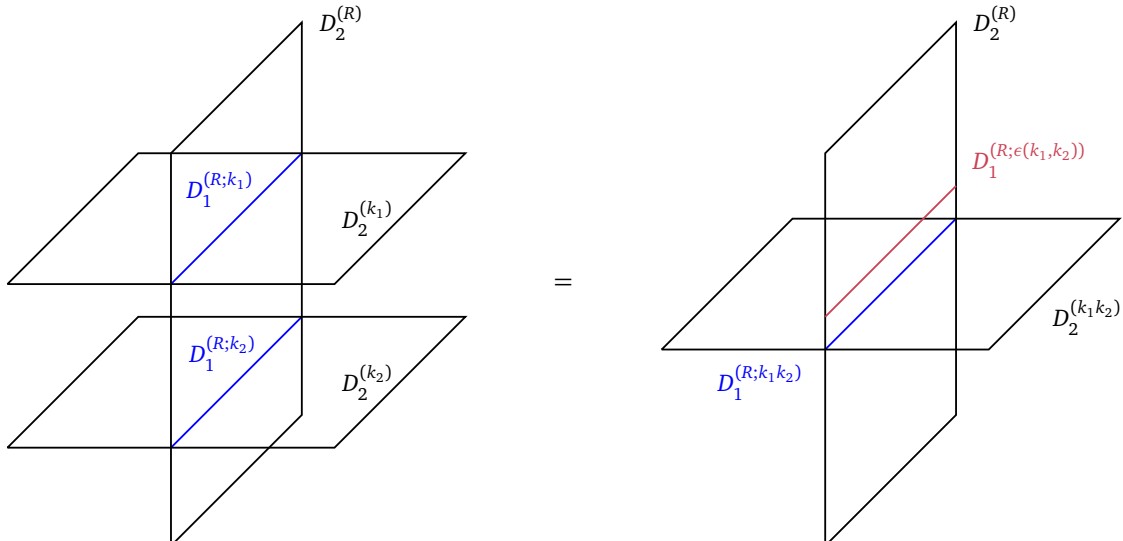

Figure 7: The figure depicts symmetry fractionalization on a surface $D_2^{(R)}$ in $\mathcal{C}_{G/H}$ specified by a 2-representation $R$ of $H$. Composing two junctions (shown in blue) of $D_2^{(R)}$ with surfaces $D_2^{(k_i)}$ generating $K$ 0-form symmetry yields an extra line (shown in red) living on $D_2^{(R)}$. This means that the bulk $K$ 0-form symmetry group fractionalizes to some larger 0-form symmetry group $K_R$ on the surface $D_2^{(R)}$.

Using similar arguments with one extra dimension added, we can see that $\mathcal{C}_{G/H}$ fractionalizes on surfaces $D_2^{(R)}$ for 2-representations $R$ in $\mathcal{C}_{G/H}$ whose underlying surfaces before $H$-gauging are multiples of the identity surface $D_2^{(\mathrm{id})}$ in $\mathcal{C}_G$.

**Derivation using gauging.** The derivation is simply a dimensional uplift of the derivation around equation (39). Label a junction line[10] between surfaces $D_2^{(R)}$ and $D_2^{(k)}$ in $\mathcal{C}_{G/H}$ as $D_1^{(R;k)}$. Then the fusion rule (39) in $\mathcal{C}_G$ implies that the fusion rule of these junctions is

$$D_1^{(R;k_1)} \otimes_{D_2^{(R)}} D_1^{(R;k_2)} = D_1^{(R;\epsilon(k_1,k_2))} \otimes_{D_2^{(R)}} D_1^{(R;k_1 k_2)}, \tag{41}$$

where $D_1^{(R;\epsilon(k_1,k_2))}$ is a line operator living on the surface $D_2^{(R)}$, or in other words a 1-endomorphism of the object $D_2^{(R)}$ in $\mathcal{C}_{G/H}$. See figure 7. The underlying line operator of $D_1^{(R;\epsilon(k_1,k_2))}$ before gauging is the line operator implementing the action of $D_2^{(\epsilon(k_1,k_2))}$ on the underlying surface $n_R D_2^{(\mathrm{id})}$ of $D_2^{(R)}$. This line operator can be made $H$-symmetric in a canonical fashion leading to the required line operator $D_1^{(R;\epsilon(k_1,k_2))}$ in $\mathcal{C}_{G/H}$.

**Derivation without using gauging.** As all the subtle information about the category $\mathcal{C}_{G/H} = 2\text{-Vec}^\omega(\Gamma)$ is encoded in $\omega$ which is equivalent to the symmetry fractionalization on lines discussed in previous subsection, it must be possible to derive the symmetry fractionalization on surfaces being discussed in this subsection as a consequence of the symmetry fractionalization on lines. That is, it should be possible to derive symmetry fractionalization on surfaces in $\mathcal{C}_{G/H}$ without relying on their construction in terms of surfaces in $\mathcal{C}_G$.

The connection between the two symmetry fractionalizations can be made by using the fact that $D_2^{(R)}$ can be constructed as a condensation surface defect by gauging a subgroup $\widehat{H}_R$

---

[10]These junction operators $D_1^{(R;k)}$ are chosen to satisfy 2-representation fusion rules for a fixed surface $D_2^{(k)}$, but as we will see they fail to obey $K$ fusion rules for a fixed surface $D_2^{(\tilde{h})}$.

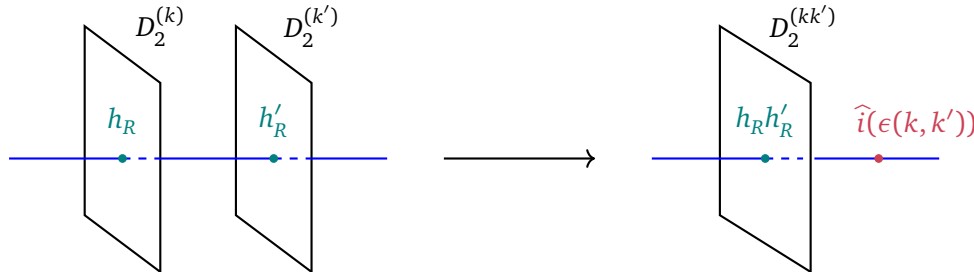

Figure 8: The blue line denotes an arbitrary line of the form $D_1^{(\widehat{h}_R)}$ for $\widehat{h}_R \in \widehat{H}_R$. A junction operator labeled by $h_R \in H_R$ (shown in teal) means that after folding the $D_1^{(\widehat{h}_R)}$ line at that junction, we are left with $h_R(\widehat{h}_R) \in U(1)$ times the identity local operator $D_0^{(k;\mathrm{id})}$ on $D_2^{(k)}$. Because of symmetry fractionalization, as we fuse $D_2^{(k)}$ and $D_2^{(k')}$, the junctions labeled by $k_R$ and $k'_R$ fuse to the junction labeled by $k_R k'_R$ times a local operator sitting on the $D_1^{(\widehat{h}_R)}$ line (shown in red) which can be identified with $\widehat{i}(\epsilon(k,k'))(\widehat{h}_R) \in U(1)$ times the identity local operator $D_0^{(\widehat{h}_R;\mathrm{id})}$ on $D_1^{(\widehat{h}_R)}$.

of the $\widehat{H}$ 1-form symmetry on a 2-dimensional surface (possibly with some discrete torsion in $H^2(\widehat{H}_R, U(1))$) in 3-dimensional spacetime occupied by $\mathfrak{T}_{G/H}$. A subset of the lines on $D_2^{(R)}$ are constructed as different ways of making the identity line $D_1^{(\mathrm{id})}$ symmetric under $\widehat{H}_R$, which is a 0-form group from the point of view of the two-dimensional surface on which $\widehat{H}_R$ condensation is occurring. By arguments similar to the ones used around (38), such lines on $D_2^{(R)}$ are parametrized by the elements of the Pontryagin dual $H_R$ of $\widehat{H}_R$. Let us call them $D_1^{(R)(h_R)}$ for $h_R \in H_R$.

Similarly, junction lines between $D_2^{(R)}$ and $D_2^{(k)}$ arise from junctions between $D_2^{(\mathrm{id})}$ and $D_2^{(k)}$, namely the identity line $D_1^{(k;\mathrm{id})}$ on $D_2^{(k)}$. The different junction lines are then again labeled by elements of $H_R$. Let us call them $D_1^{(R;k)(h_R)}$ for $h_R \in H_R$. Then computing their fusion, we find

$$D_1^{(R;k)(h_R)} \otimes_{D_2^{(R)}} D_1^{(R;k')(h'_R)} = D_1^{(R)\left(\widehat{i}\left(\epsilon(k,k')\right)\right)} \otimes_{D_2^{(R)}} D_1^{(R;kk')(h_R h'_R)}, \tag{42}$$

where $\widehat{i} : H \to H_R$ is the projection map Pontryagin dual to the inclusion map $i : \widehat{H}_R \hookrightarrow \widehat{H}$. Setting $h_R = h'_R = 1$ we recover (41).

The reason for the appearance of the extra line $D_1^{(R)\left(\widehat{i}\left(\epsilon(k_1,k_2)\right)\right)}$ in the above fusion is the symmetry fractionalization on $\widehat{H}$ lines. This is because the choice of $\widehat{H}_R$ action on $D_1^{(k;\mathrm{id})}$ defining $D_1^{(R;k)(h_R)}$ is the choice of junctions between $\widehat{H}_R$ lines and $D_1^{(k;\mathrm{id})}$. Thus, composing two $\widehat{H}_R$ actions, we obtain an extra contribution from symmetry fractionalization on $\widehat{H}_R$ lines. See figure 8.

**Example $G = \mathbb{Z}_4$, $H = \mathbb{Z}_2$.** To illustrate this symmetry fractionalization for condensation defects consider again the gauging of $H = \mathbb{Z}_2 \triangleleft \mathbb{Z}_4$. There is a single condensation surface defect $D_2^{(\mathbb{Z}_2)}$ in $\mathcal{C}_{\mathbb{Z}_4/\mathbb{Z}_2}$ obtained by gauging the $\mathbb{Z}_2$ 1-form symmetry generated by $D_1^{(V)}$ on a two-dimensional surface.

The lines living on $D_2^{(\mathbb{Z}_2)}$ generate a $\mathbb{Z}_2$ 0-form symmetry localized on $D_2^{(\mathbb{Z}_2)}$. Let us call the generator of this localized symmetry[11] as $D_1^{(\mathbb{Z}_2;-)}$. From the point of view of $\mathcal{C}_{\mathbb{Z}_4}$, this line

---

[11] Note that this line should not identified as the image of the stacking of the bulk line $D_1^{(V)}$ on top of $D_2^{(\mathbb{Z}_2)}$.

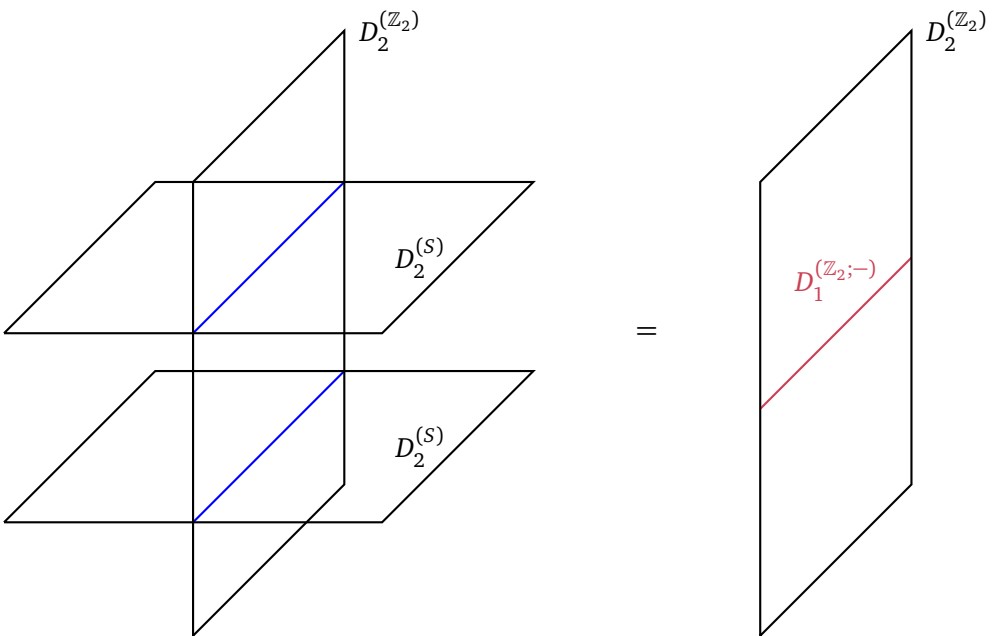

Figure 9: The figure depicts symmetry fractionalization on the condensation surface $D_2^{(\mathbb{Z}_4)}$ in $\mathcal{C}_{\mathbb{Z}_4/\mathbb{Z}_2}$. Composing two junctions (shown in blue) of $D_2^{(\mathbb{Z}_2)}$ with surfaces $D_2^{(S)}$ generating $\mathbb{Z}_2$ 0-form symmetry yields a line $D_1^{(\mathbb{Z}_2;-)}$ (shown in red) living on $D_2^{(\mathbb{Z}_2)}$. This means that the bulk $\mathbb{Z}_2$ 0-form symmetry group fractionalizes to $\mathbb{Z}_4$ 0-form symmetry group on the surface $D_2^{(\mathbb{Z}_2)}$. More precisely, the blue lines are either both $L_1^+$ or both $L_1^-$, see fusion rules (44).

arises as the $\mathbb{Z}_2$-symmetric 1-morphism

$$\begin{pmatrix} 0 & D_1^{(\mathrm{id})} \\ D_1^{(\mathrm{id})} & 0 \end{pmatrix}: \quad 2D_2^{(\mathrm{id})} \to 2D_2^{(\mathrm{id})}, \tag{43}$$

which is also the 1-morphism (21) generating the $\mathbb{Z}_2$ action converting the object $2D_2^{(\mathrm{id})}$ of $\mathcal{C}_{\mathbb{Z}_4}$ into the object $D_2^{(\mathbb{Z}_2)}$ of $\mathcal{C}_{\mathbb{Z}_4/\mathbb{Z}_2}$.

Consequently, following (41), we learn that the square of the junction of $D_2^{(\mathbb{Z}_2)}$ and $D_2^{(S)}$ along $D_2^{(\mathbb{Z}_2)}$ leaves behind the line $D_1^{(\mathbb{Z}_2;-)}$. Thus the $\mathbb{Z}_2$ 0-form symmetry of $\mathfrak{T}_{\mathbb{Z}_4/\mathbb{Z}_2}$ generated by $D_2^{(S)}$ fractionalizes to $\mathbb{Z}_4$ 0-form symmetry on the worldvolume of $D_2^{(\mathbb{Z}_2)}$ because the line $D_1^{(\mathbb{Z}_2;-)}$ has order two. See figure 9.

More precisely, there are two simple junction lines $L_1^\pm$ that can live at the junction of $D_2^{(\mathbb{Z}_2)}$ and $D_2^{(S)}$. These correspond to the two irreducible representations of the $\mathbb{Z}_2$ 1-form symmetry generated by $D_1^{(V)}$ being gauged to construct $D_2^{(\mathbb{Z}_2)}$. Applying (42), we learn the fusion rules

$$\begin{aligned} L_1^+ \otimes_{D_2^{(\mathbb{Z}_2)}} L_1^+ &= D_1^{(\mathbb{Z}_2;-)}, \\ L_1^- \otimes_{D_2^{(\mathbb{Z}_2)}} L_1^- &= D_1^{(\mathbb{Z}_2;-)}, \\ L_1^+ \otimes_{D_2^{(\mathbb{Z}_2)}} L_1^- &= D_1^{(\mathbb{Z}_2;\mathrm{id})}, \end{aligned} \tag{44}$$

which we will use in the analysis of the next section.

---

Actually, $D_1^{(V)}$ becomes the identity line $D_1^{(\mathbb{Z}_2;\mathrm{id})}$ of $D_2^{(\mathbb{Z}_2)}$ under this stacking procedure. Instead, the line $D_1^{(\mathbb{Z}_2;-)}$ is localized/trapped on $D_2^{(\mathbb{Z}_2)}$ and cannot be lifted into the bulk.

# 3 Gauging the gauged

So far we generalized the analysis in [14] to perform a partial gauging, of a normal subgroup $H \triangleleft G$ which acts as a 0-form symmetry on a 3d theory. This resulted in the category

$$\mathcal{C}_{G/H} = 2\text{-Vec}^\omega(\Gamma), \tag{45}$$

where $\omega$ is determined in terms of the associated group-cocycle $\epsilon \in H^2(K, H)$ with $K = G/H$, and $\Gamma$ is a split 2-group containing $K$ 0-form symmetry acting on $\widehat{H}$ 1-form symmetry.

In this section we discuss how to gauge the remaining $K$ 0-form symmetry. This requires developing how the gauging is realized in a setting where the symmetry is not necessarily invertible (for example, in this case there is a non-invertible 2-Rep($H$) subsymmetry). More generally – and we will consider an explicit example with $D_8$ – one can of course have multiple more steps in this gauging. Once all 0-form symmetries are gauged we expect the symmetry category to be $\mathcal{C}_{G/G} = 2\text{-Rep}(G)$. We cross-check our results against this expectation, but the main outputs of this section are the lessons we learn how to gauge invertible symmetries in 2-categories describing non-invertible symmetries more generally. Appendix A.2 provides an in depth analysis from the bi-module perspective to gauging. Here we will focus on the physical picture of symmetrizing the defects with respect to the symmetry $G/H$.

We perform an explicit analysis of $G/H$ gauging for our examples $G = \mathbb{Z}_2 \times \mathbb{Z}_2, H = \mathbb{Z}_2$ and $G = \mathbb{Z}_4, H = \mathbb{Z}_2$, and compare with the results obtained in appendix B upon performing full $G$-gauging using the procedure detailed in [14].

## 3.1 Surface defects after sequential gauging

**$K$-invariant Combinations of Surfaces.** First of all, since multiplication by $D_2^{(k)}$ relates any $D_2^{(kR)}$ to $D_2^{(R)}$, we only need to focus on making combinations of $D_2^{(R)}$ $K$-symmetric. Any other $K$-symmetric combination of objects of $\mathcal{C}_{G/H}$ isomorphic only produces objects of $\mathcal{C}_G$ isomorphic to the ones descending from $K$-symmetric combinations of $D_2^{(R)}$.

Note that we can only implement $K$ symmetry if we begin with a multiple of

$$\bigoplus_{R \in K\text{-orbit}} D_2^{(R)}, \tag{46}$$

because the action of $K$ on $H$ descends to an action of $K$ on 2-representations $R$ of $H$.

**Obstruction: Symmetry fractionalization.** There are various further obstructions in the $K$-symmetrization procedure, which are most cleanly understood in the case when the action of $K$ is trivial on a particular 2-representation $R$. For the first type of obstruction, consider making a single copy of $D_2^{(R)}$ $K$-symmetric. If $K$ 0-form symmetry is fractionalized on $D_2^{(R)}$, then it is impossible to $K$-symmetrize $D_2^{(R)}$. However, this obstruction may be cured by beginning instead with $nD_2^{(R)}$ for $n > 1$. In this case, the resulting defect in $\mathcal{C}_G$ after gauging is twisted theta with the underlying twist being $D_2^{(R)} \in \mathcal{C}_{G/H}$ and underlying stack the 2d TQFT with $n$ trivial vacua.

We can then try to implement the $K$-symmetry acting as a permutation of the $n$ copies of $D_2^{(R)}$ with different choices for the junction lines $D_1^{(R;k)(\widehat{h}_R)}$. Such a non-trivial combination of junction lines with permutation can defractionalize the 0-form symmetry back to $K$, making $nD_2^{(R)}$ $K$-symmetric. We will see soon that because of this obstruction $D_2^{(\mathbb{Z}_2)}$ in $\mathcal{C}_{\mathbb{Z}_4/\mathbb{Z}_2}$ cannot be made symmetric under $K = \mathbb{Z}_2$, but $2D_2^{(\mathbb{Z}_2)}$ can be made $K$-symmetric, which is in contrast to $\mathcal{C}_{\mathbb{Z}_2 \times \mathbb{Z}_2/\mathbb{Z}_2}$.

**Obstruction: Localized 't Hooft anomaly and projective 2-representations.** Another type of obstruction arises even when $K$ symmetry does not fractionalize on $D_2^{(R)}$. Consider for example the identity defect $D_2^{(\text{id})}$. There is a canonical junction line $D_1^{(\text{id};k)(\text{id})}$ between $D_2^{(k)}$ and $D_2^{(\text{id})}$, which upon folding of the $D_2^{(k)}$ surface reduces to the identity line $D_1^{(\text{id})}$ on $D_2^{(\text{id})}$. Making $nD_2^{(\text{id})}$ $K$-symmetric by combining $D_1^{(\text{id};k)(\text{id})}$ junctions with permutations of $n$ copies of $D_2^{(\text{id})}$ lead to the $K$-symmetric defects labeled by $n$-dimensional 2-representations of $K$.

Consider performing the same procedure with another junction line

$$D_1^{(\text{id};k)(\rho(k))} := D_1^{(\rho(k))} \otimes D_1^{(\text{id};k)(\text{id})}, \tag{47}$$

between $D_2^{(k)}$ and $D_2^{(\text{id})}$ obtained by stacking the bulk line $D_1^{(\rho(k))}$ for $\rho(k) \in \widehat{H}$ on top of $D_1^{(\text{id};k)(\text{id})}$. Here $\rho$ is a homomorphism from $K$ to $\widehat{H}$. The junctions $D_1^{(\text{id};k)(\rho(k))}$ satisfy the $K$ fusion rules

$$D_1^{(\text{id};k)(\rho(k))} \otimes_{D_2^{(\text{id})}} D_1^{(\text{id};k')(\rho(k'))} = D_1^{(\text{id};kk')(\rho(kk'))}. \tag{48}$$

However the 1-category formed by these junctions is not $\text{Vec}_K$ but rather $\text{Vec}_K^{\omega_\rho}$ where $\omega_\rho \in H^3(K, U(1))$ provides a possibly non-trivial associator between these junction lines. A representative is

$$\omega_\rho(k_1, k_2, k_3) = \rho(k_3)(\epsilon(k_1, k_2)) \in U(1), \tag{49}$$

which can be obtained as a consequence of the symmetry fractionalization of $K$ on $\widehat{H}$ lines. See figure 10 for the associator diagrams.

In other words, the junction lines $D_1^{(\text{id};k)(\rho(k))}$ generate $K$ 0-form symmetry on the two-dimensional worldvolume of $D_2^{(\text{id})}$ which is afflicted with the 't Hooft anomaly $\omega_\rho$. Thus, in order to obtain a $K$-symmetric surface defect we need to stack $D_2^{(\text{id})}$ with a 2d TQFT carrying $K$ 0-form symmetry with opposite anomaly $\omega_\rho^{-1}$, which then becomes the stack for the resulting twisted theta defect. Such 2d TQFTs are classified by *projective 2-representations*[12] of $H$ in the class $\omega_\rho$, or in other words monoidal functors of the form

$$\mathcal{C}_1^{\Gamma^{(0)}=K,\omega_\rho} \to \text{Mat}_n(\text{Vec}), \tag{50}$$

where $\mathcal{C}_1^{\Gamma^{(0)}=K,\omega_\rho}$ is a non-linear version of $\text{Vec}_K^{\omega_\rho}$ and $n$ is called the dimension of the projective 2-representation. In total, a choice of $\rho$ and a projective 2-representation $R$ of $K$ of dimension $n$ in class $\omega_\rho$ provides a way of making $nD_2^{(\text{id})}$ in $\mathcal{C}_{G/H}$ $K$-symmetric.

For future purposes, let us note that just like irreducible 2-representations of $K$ are classified by the choice of a subgroup $K'$ and an element in $H^2(K', U(1))$, the irreducible projective 2-representations of $K$ in class $\omega_\rho$ are specified by the choice of a subgroup $K'$ of $K$ on which $\omega_\rho$ trivializes and a trivialization of $\omega_\rho$ on $K'$ with two trivializations related by exact terms considered to be equivalent.

A general obstruction is a combination of the two types of obstructions discussed above. It would be very interesting to systematically understand the more general obstructions in future works.

### 3.1.1 $G = \mathbb{Z}_2 \times \mathbb{Z}_2$, $H = \mathbb{Z}_2$

Let us now consider gauging the $K = \mathbb{Z}_2$ 0-form symmetry generated by $D_2^{(S)}$ in our example $G = \mathbb{Z}_2 \times \mathbb{Z}_2, H = \mathbb{Z}_2$. The gauging process should convert the 2-category $\mathcal{C}_{\mathbb{Z}_2 \times \mathbb{Z}_2/\mathbb{Z}_2}$ into the

---

[12]Projective 2-representations in the class $\omega = 0$ coincide with the usual 2-representations.

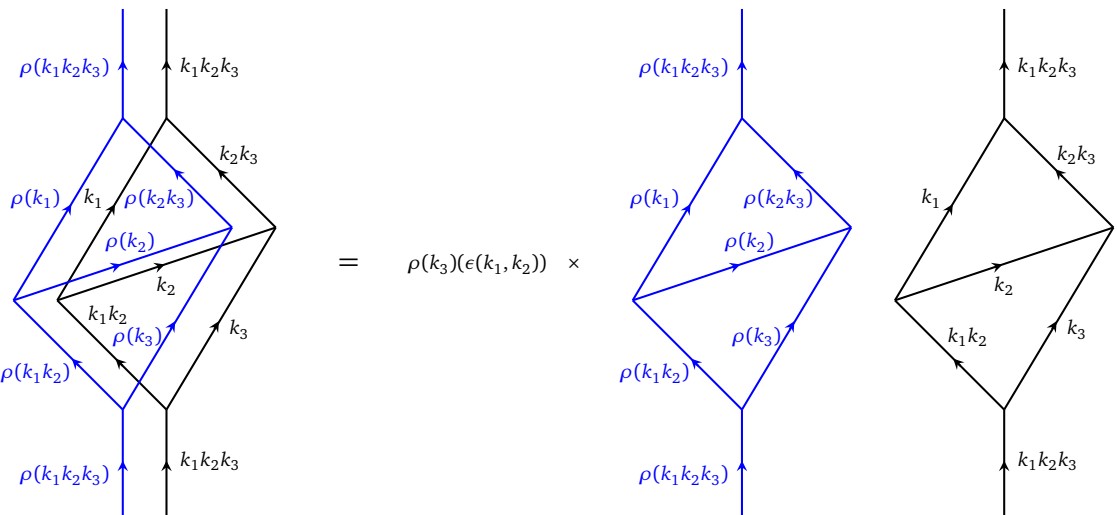

Figure 10: Left side of the figure depicts an associator comprised of $D_1^{(\mathrm{id};k)(\rho(k))}$ junction lines. They have been resolved into $D_1^{(\mathrm{id};k)(\mathrm{id})}$ junction lines (shown in black) and $D_1^{(\rho(k))}$ bulk lines (shown in blue). As we completely lift the bulk $D_1^{(\rho(k))}$ lines far away from the $D_2^{(k)}$ surfaces, the bulk line $D_1^{(\rho(k_3))}$ crosses the junction of $D_1^{(\mathrm{id};k_1)(\mathrm{id})}$, $D_1^{(\mathrm{id};k_2)(\mathrm{id})}$ and $D_1^{(\mathrm{id};k_1 k_2)(\mathrm{id})}$, or in other words $D_1^{(\rho(k_3))}$ crosses the junction of surfaces $D_2^{(k_1)}$, $D_2^{(k_2)}$ and $D_2^{(k_1 k_2)}$. Because of the mixed 't Hooft anomaly between $K$ 0-form symmetry and $\widehat{H}$ 1-form symmetry, this crossing generates a phase factor $\omega_\rho = \rho(k_3)(\epsilon(k_1, k_2))$, as shown on the right side of the figure. The purely blue and purely black associators on the right are trivial, using the fact that $K$ 0-form symmetry has no pure 't Hooft anomaly and $\widehat{H}$ 1-form symmetry has no pure 't Hooft anomaly. Thus, the associator of $D_1^{(\mathrm{id};k)(\rho(k))}$ junction lines is given by the 3-cocycle $\omega_\rho$.

2-category $\mathcal{C}_{\mathbb{Z}_2^2/\mathbb{Z}_2^2} = 2\text{-Rep}(\mathbb{Z}_2 \times \mathbb{Z}_2)$ discussed in appendix B.1. Let us see this explicitly at the level of objects.

Since the short exact sequence

$$1 \to H = \mathbb{Z}_2 \to G = \mathbb{Z}_2 \times \mathbb{Z}_2 \to K = \mathbb{Z}_2 \to 1\,, \tag{51}$$

splits, none of the obstructions discussed above are relevant, and the analysis is relatively straightforward.

First consider objects of $\mathcal{C}_{\mathbb{Z}_2^2/\mathbb{Z}_2^2}$ descending from multiples of the identity object $D_2^{(\mathrm{id})}$ of $\mathcal{C}_{\mathbb{Z}_2 \times \mathbb{Z}_2/\mathbb{Z}_2}$. Note that there are only two possible homomorphisms

$$\rho : K = \mathbb{Z}_2 \to \widehat{H} = \mathbb{Z}_2\,, \tag{52}$$

namely the trivial and the identity homomorphisms. Since the extension class $\epsilon = 0$, for both homomorphisms we have $\omega_\rho = 0$. Thus, both of them lead to a copy of 2-Rep($\mathbb{Z}_2$). Let us label the simple objects (upto isomorphism) in the 2-Rep($\mathbb{Z}_2$) $\subseteq \mathcal{C}_{\mathbb{Z}_2^2/\mathbb{Z}_2^2}$ arising from trivial homomorphism as

$$D_2^{(\mathrm{id})}, \qquad D_2^{(\mathbb{Z}_2^V)}, \tag{53}$$

where $D_2^{(\mathrm{id})}$ corresponds to the trivial 2-representation and $D_2^{(\mathbb{Z}_2^V)}$ corresponds to the non-trivial irreducible 2-representation of dimension 2. Similarly, let us label the simple objects (upto isomorphism) in the 2-Rep($\mathbb{Z}_2$) $\subseteq \mathcal{C}_{\mathbb{Z}_2^2/\mathbb{Z}_2^2}$ arising from identity homomorphism as

$$D_2^{(-)}, \qquad D_2^{(\mathbb{Z}_2^{V'})}, \tag{54}$$

where $D_2^{(-)}$ corresponds to the trivial 2-representation and $D_2^{(\mathbb{Z}_2^{V'})}$ corresponds to the non-trivial irreducible 2-representation of dimension 2.

More concretely, we can make $D_2^{(\mathrm{id})}$ symmetric under $\mathbb{Z}_2^S$ by either generating the symmetry using one of the lines $D_1^{(\mathrm{id})}$ or $D_1^{(V)}$. The resulting simple objects of $\mathcal{C}_{\mathbb{Z}_2^2/\mathbb{Z}_2^2}$ respectively are

$$D_2^{(\mathrm{id})}, \qquad D_2^{(-)}. \tag{55}$$

Likewise consider making $2D_2^{(\mathrm{id})}$ symmetric under $\mathbb{Z}_2^S$. The 1-endomorphisms of this defect are in $\mathrm{Mat}_{2\times 2}(\mathrm{Rep}(\mathbb{Z}_2))$, where $\mathrm{Rep}(\mathbb{Z}_2)$ is generated by $D_1^{(\mathrm{id})}$ and $D_1^{(V)}$. There are five choices of 1-endomorphism for generating $\mathbb{Z}_2^S$ 0-form symmetry, which give rise to the following objects in $\mathcal{C}_{\mathbb{Z}_2^2/\mathbb{Z}_2^2}$:

$$\begin{aligned}
\begin{pmatrix} D_1^{(\mathrm{id})} & 0 \\ 0 & D_1^{(\mathrm{id})} \end{pmatrix}: &\qquad D_2^{(\mathrm{id})} \oplus D_2^{(\mathrm{id})}, \\[6pt]
\begin{pmatrix} D_1^{(V)} & 0 \\ 0 & D_1^{(V)} \end{pmatrix}: &\qquad D_2^{(-)} \oplus D_2^{(-)}, \\[6pt]
\begin{pmatrix} D_1^{(\mathrm{id})} & 0 \\ 0 & D_1^{(V)} \end{pmatrix}: &\qquad D_2^{(\mathrm{id})} \oplus D_2^{(-)}, \\[6pt]
\begin{pmatrix} 0 & D_1^{(\mathrm{id})} \\ D_1^{(\mathrm{id})} & 0 \end{pmatrix}: &\qquad D_2^{(\mathbb{Z}_2^V)}, \\[6pt]
\begin{pmatrix} 0 & D_1^{(V)} \\ D_1^{(V)} & 0 \end{pmatrix}: &\qquad D_2^{(\mathbb{Z}_2^{V'})} \cong D_2^{(\mathbb{Z}_2^V)}.
\end{aligned} \tag{56}$$

To see that $D_2^{(\mathbb{Z}_2^{V'})} \cong D_2^{(\mathbb{Z}_2^V)}$, note that

$$\begin{pmatrix} D_1^{(\mathrm{id})} & 0 \\ 0 & D_1^{(V)} \end{pmatrix}, \tag{57}$$

provides a 1-morphism $D_1^{(\mathbb{Z}_2^{V'},\mathbb{Z}_2^V)}: D_2^{(\mathbb{Z}_2^{V'})} \to D_2^{(\mathbb{Z}_2^V)}$, and also a 1-morphism $D_1^{(\mathbb{Z}_2^V,\mathbb{Z}_2^{V'})}: D_2^{(\mathbb{Z}_2^V)} \to D_2^{(\mathbb{Z}_2^{V'})}$ in $\mathcal{C}_{\mathbb{Z}_2^2/\mathbb{Z}_2^2}$. The composition of these two 1-morphisms is the identity endomorphism (as the square of the above matrix is the identity matrix) of $D_2^{(\mathbb{Z}_2^V)} \in \mathcal{C}_{\mathbb{Z}_2^2/\mathbb{Z}_2^2}$. Similar to above, $D_2^{(\mathbb{Z}_2)} \in \mathcal{C}_{\mathbb{Z}_2\times\mathbb{Z}_2/\mathbb{Z}_2}$ leads to two simple objects

$$D_2^{(\mathbb{Z}_2^S)}, \; D_2^{(\mathbb{Z}_2^C)} \in \mathcal{C}_{\mathbb{Z}_2^2/\mathbb{Z}_2^2}. \tag{58}$$

The 0-form symmetry $\mathbb{Z}_2^S$ is implemented by $D_1^{(\mathbb{Z}_2;\mathrm{id})} \in \mathcal{C}_{\mathbb{Z}_2\times\mathbb{Z}_2/\mathbb{Z}_2}$ for $D_2^{(\mathbb{Z}_2^S)} \in \mathcal{C}_{\mathbb{Z}_2^2/\mathbb{Z}_2^2}$, and by $D_1^{(\mathbb{Z}_2;-)} \in \mathcal{C}_{\mathbb{Z}_2\times\mathbb{Z}_2/\mathbb{Z}_2}$ for $D_2^{(\mathbb{Z}_2^C)} \in \mathcal{C}_{\mathbb{Z}_2^2/\mathbb{Z}_2^2}$. And similar to above, $2D_2^{(\mathbb{Z}_2^V)} \in \mathcal{C}_{\mathbb{Z}_2\times\mathbb{Z}_2/\mathbb{Z}_2}$ leads to a single simple object

$$D_2^{(\mathbb{Z}_2\times\mathbb{Z}_2)} \in \mathcal{C}_{\mathbb{Z}_2^2/\mathbb{Z}_2^2}, \tag{59}$$

for which the $\mathbb{Z}_2^S$ symmetry is implemented by

$$\begin{pmatrix} 0 & D_1^{(\mathbb{Z}_2;\mathrm{id})} \\ D_1^{(\mathbb{Z}_2;\mathrm{id})} & 0 \end{pmatrix}. \tag{60}$$

In summary the surface defects in the gauged category are built upon the following simple objects

$$\mathrm{Obj}(\mathcal{C}_{\mathbb{Z}_2^2/\mathbb{Z}_2^2}) = \left\{ D_2^{(\mathrm{id})}, D_2^{(-)}, D_2^{(\mathbb{Z}_2^S)}, D_2^{(\mathbb{Z}_2^C)}, D_2^{(\mathbb{Z}_2^V)}, D_2^{(\mathbb{Z}_2\times\mathbb{Z}_2)} \right\}, \tag{61}$$

as expected from the analysis of appendix B.1.

### 3.1.2 $G = \mathbb{Z}_4$, $H = \mathbb{Z}_2$

Let us now consider the gauging of residual $\mathbb{Z}_2^S$ symmetry in the 2-category $\mathcal{C}_{\mathbb{Z}_4/\mathbb{Z}_2}$ associated to our example $G = \mathbb{Z}_4, H = \mathbb{Z}_2$. We expect the resulting category to be equivalent to $\mathcal{C}_{\mathbb{Z}_4/\mathbb{Z}_4} = 2\text{-Rep}(\mathbb{Z}_4)$ obtained by gauging the full $\mathbb{Z}_4$ 0-form symmetry in $\mathcal{C}_{\mathbb{Z}_4}$. Notice that up to the gauging of the first $\mathbb{Z}_2^V$ subgroup, the spectrum of objects and 1-morphisms has been identical irrespective of whether we start from $G = \mathbb{Z}_2 \times \mathbb{Z}_2$ or $G = \mathbb{Z}_4$. However, upon the subsequent gauging of the residual $\mathbb{Z}_2^S$ symmetry, we expect to see differences in these spectra, as in one case we should land on the symmetry 2-category $2\text{-Rep}(\mathbb{Z}_2 \times \mathbb{Z}_2)$ and in the other on $2\text{-Rep}(\mathbb{Z}_4)$, and these two 2-categories have different number of simple objects (upto isomorphism). In particular, $2\text{-Rep}(\mathbb{Z}_4)$ has less simple objects than $2\text{-Rep}(\mathbb{Z}_2 \times \mathbb{Z}_2)$, which means that some of the topological surfaces of the latter will have to be inconsistent in the former. This is due to the obstructions related to symmetry fractionalizations discussed above.

**Objects.** First consider objects of $\mathcal{C}_{\mathbb{Z}_4/\mathbb{Z}_4}$ descending from multiples of the identity object $D_2^{(\text{id})}$ of $\mathcal{C}_{\mathbb{Z}_4/\mathbb{Z}_2}$. We have to consider the trivial and identity homomorphisms

$$\rho: K = \mathbb{Z}_2 \to \widehat{H} = \mathbb{Z}_2. \tag{62}$$

Since the extension class $\epsilon$ is non-trivial, we have for trivial homomorphism $\omega_\rho = 0$, but $\omega_\rho \neq 0 \in H^3(\mathbb{Z}_2, U(1)) = \mathbb{Z}_2$ for the identity homomorphism. The trivial homomorphism leads to a copy of $2\text{-Rep}(\mathbb{Z}_2) \subseteq \mathcal{C}_{\mathbb{Z}_2^2/\mathbb{Z}_2^2}$ whose simple objects (upto isomorphism) we label as

$$D_2^{(\text{id})}, \qquad D_2^{(\mathbb{Z}_2^V)}, \tag{63}$$

where $D_2^{(\text{id})}$ corresponds to the trivial 2-representation and $D_2^{(\mathbb{Z}_2^V)}$ corresponds to the non-trivial irreducible 2-representation of dimension 2.

Now consider the simple objects (upto isomorphism) descending from the identity homomorphism $\rho$. There is only a single irreducible projective 2-representations of $\mathbb{Z}_2$ with non-trivial class $\omega_\rho$ since $\omega_\rho$ trivializes only on the trivial subgroup of $\mathbb{Z}_2$ and there is only a single choice of trivialization. Since the choice of subgroup is trivial inside $\mathbb{Z}_2$, the $\mathbb{Z}_2^S$ symmetry is spontaneously broken and the irreducible projective 2-representation has dimension 2. Let us label the corresponding simple object in $\mathcal{C}_{\mathbb{Z}_4/\mathbb{Z}_4}$ as

$$D_2^{(\mathbb{Z}_2^{V'})}. \tag{64}$$

Note that we do not obtain analogue of $D_2^{(-)} \in \mathcal{C}_{\mathbb{Z}_2^2/\mathbb{Z}_2^2}$ in $\mathcal{C}_{\mathbb{Z}_4/\mathbb{Z}_4}$. In other words, symmetry fractionalization in $\mathcal{C}_{\mathbb{Z}_4/\mathbb{Z}_2}$ has obstructed the existence of $D_2^{(-)}$ in $\mathcal{C}_{\mathbb{Z}_4/\mathbb{Z}_4}$.

Just as for the previous example, it also turns out here that

$$D_2^{(\mathbb{Z}_2^{V'})} \cong D_2^{(\mathbb{Z}_2^V)}, \tag{65}$$

though here the isomorphism is quite complicated. We relegate the explicit discussion of this isomorphism to appendix C.3.

Now let us consider making $D_2^{(\mathbb{Z}_2)}$ symmetric under $\mathbb{Z}_2^S$. This fails already at the level of lines, because as discussed above $\mathbb{Z}_2^S$ fractionalizes to $\mathbb{Z}_4$ 0-form symmetry on $D_2^{(\mathbb{Z}_2)}$. Thus, for the $\mathbb{Z}_2 \times \mathbb{Z}_2$ case, $D_2^{(\mathbb{Z}_2)} \in \mathcal{C}_{\mathbb{Z}_2 \times \mathbb{Z}_2/\mathbb{Z}_2}$ gives rise to two simple objects $D_2^{(\mathbb{Z}_2^S)}$ and $D_2^{(\mathbb{Z}_2^C)}$ in $\mathcal{C}_{\mathbb{Z}_2^2/\mathbb{Z}_2^2}$, but for the $\mathbb{Z}_4$ case we do not get even a single simple object of $\mathcal{C}_{\mathbb{Z}_4/\mathbb{Z}_4}$ from $D_2^{(\mathbb{Z}_2)} \in \mathcal{C}_{\mathbb{Z}_4/\mathbb{Z}_2}$.

On the other hand, we can successfully make $2D_2^{(\mathbb{Z}_2)}$ symmetric under $\mathbb{Z}_2^S$ by implementing the $\mathbb{Z}_2^S$ symmetry using the matrix of junction lines

$$\begin{pmatrix} 0 & L_1^+ \\ L_1^- & 0 \end{pmatrix}. \tag{66}$$

These junction lines were defined around equation (44). This leads to the simple object $D_2^{(\mathbb{Z}_4)}$ of $\mathcal{C}_{\mathbb{Z}_4/\mathbb{Z}_4}$.

In summary we obtain the simple objects

$$\mathrm{Obj}\left(\mathcal{C}_{\mathbb{Z}_4/\mathbb{Z}_4}\right) = \left\{ D_2^{(\mathrm{id})}, D_2^{(\mathbb{Z}_2^V)}, D_2^{(\mathbb{Z}_4)} \right\}, \tag{67}$$

of $\mathcal{C}_{\mathbb{Z}_4/\mathbb{Z}_4}$ starting from the 2-category $\mathcal{C}_{\mathbb{Z}_4/\mathbb{Z}_2}$ and gauging its $\mathbb{Z}_2^S$ 0-form symmetry.

## 3.2 Fusion of surface defects after sequential gauging

The fusion of surfaces in $\mathcal{C}_{G/G}$ can be computed in terms of $\mathcal{C}_{G/H}$ information by computing the fusion of underlying surfaces in $\mathcal{C}_{G/H}$ and the fusion of the junction lines implementing $K$ symmetry (which provides the junction lines implementing $K$ symmetry on the fusion of underlying surfaces).

### 3.2.1 Example: $G = \mathbb{Z}_2 \times \mathbb{Z}_2$, $H = \mathbb{Z}_2$

First of all, we clearly have

$$D_2^{(-)} \otimes D_2^{(-)} = D_2^{(\mathrm{id})}, \tag{68}$$

because the $\mathbb{Z}_2^S$ symmetry for $D_2^{(-)} \otimes D_2^{(-)}$ is implemented by $D_1^{(V)} \otimes D_1^{(V)} \cong D_1^{(\mathrm{id})}$ in $\mathcal{C}_{\mathbb{Z}_2 \times \mathbb{Z}_2/\mathbb{Z}_2}$. Next, we also have

$$D_2^{(-)} \otimes D_2^{(\mathbb{Z}_2^V)} = D_2^{(\mathbb{Z}_2^{V'})} \cong D_2^{(\mathbb{Z}_2^V)}, \tag{69}$$

because the tensor product of the junctions implementing $\mathbb{Z}_2^S$ symmetry is

$$D_1^{(V)} \otimes \begin{pmatrix} 0 & D_1^{(\mathrm{id})} \\ D_1^{(\mathrm{id})} & 0 \end{pmatrix} = \begin{pmatrix} 0 & D_1^{(V)} \\ D_1^{(V)} & 0 \end{pmatrix}. \tag{70}$$

Likewise we find

$$\begin{aligned} D_2^{(-)} \otimes D_2^{(\mathbb{Z}_2^x)} &\cong D_2^{(\mathbb{Z}_2^x)}, \qquad x = S, C, \\ D_2^{(-)} \otimes D_2^{(\mathbb{Z}_2 \times \mathbb{Z}_2)} &\cong D_2^{(\mathbb{Z}_2 \times \mathbb{Z}_2)}, \end{aligned} \tag{71}$$

because $D_1^{(V)}$ line becomes invisible when stacked on top of $D_2^{(\mathbb{Z}_2)}$ in $\mathcal{C}_{\mathbb{Z}_2 \times \mathbb{Z}_2/\mathbb{Z}_2}$. The fusion rules

$$D_2^{(\mathbb{Z}_2^V)} \otimes D_2^{(\mathbb{Z}_2^V)} \cong 2D_2^{(\mathbb{Z}_2^V)}, \tag{72}$$

follow from arguments similar to those leading to the fusion rule (26), and

$$D_2^{(\mathbb{Z}_2^x)} \otimes D_2^{(\mathbb{Z}_2^x)} \cong 2D_2^{(\mathbb{Z}_2^x)}, \qquad x = S, C, \tag{73}$$

follows from the fact that $D_1^{(\mathbb{Z}_2;i)} \otimes D_1^{(\mathbb{Z}_2;i)} \in \mathcal{C}_{\mathbb{Z}_2 \times \mathbb{Z}_2/\mathbb{Z}_2}$ is given by $D_1^{(\mathbb{Z}_2;i)} \mathrm{id}_{2\times 2}$ for $i \in \{\mathrm{id}, -\}$ (see also [14], (3.68) and (3.71) for similar arguments). The fusion rule

$$D_2^{(\mathbb{Z}_2^S)} \otimes D_2^{(\mathbb{Z}_2^C)} \cong D_2^{(\mathbb{Z}_2 \times \mathbb{Z}_2)}, \tag{74}$$

follows from the fact that $D_1^{(\mathbb{Z}_2;\text{id})} \otimes D_1^{(\mathbb{Z}_2;-)} \in \mathcal{C}_{\mathbb{Z}_2 \times \mathbb{Z}_2 / \mathbb{Z}_2}$ is given by $2 \times 2$ off-diagonal matrix with both entries $D_1^{(\mathbb{Z}_2;\text{id})}$ (see also [14], (3.69) for similar arguments). Combining the above facts, the reader can easily show the remaining fusion rules

$$D_2^{(\mathbb{Z}_2^V)} \otimes D_2^{(\mathbb{Z}_2^x)} \cong D_2^{(\mathbb{Z}_2 \times \mathbb{Z}_2)}, \qquad x = S, C, \tag{75}$$

and

$$D_2^{(\mathbb{Z}_2 \times \mathbb{Z}_2)} \otimes D_2^{(\mathbb{Z}_2 \times \mathbb{Z}_2)} \cong 4 D_2^{(\mathbb{Z}_2 \times \mathbb{Z}_2)}. \tag{76}$$

These fusion rules all match with those obtained in appendix B.1 by direct $\mathbb{Z}_2 \times \mathbb{Z}_2$ gauging of $\mathcal{C}_{\mathbb{Z}_2 \times \mathbb{Z}_2}$.

### 3.2.2 $G = \mathbb{Z}_4$, $H = \mathbb{Z}_2$

Now let us compute fusions of simple objects of $\mathcal{C}_{\mathbb{Z}_4 / \mathbb{Z}_4}$ using its construction as gauging of $\mathcal{C}_{\mathbb{Z}_4 / \mathbb{Z}_2}$. These should match the fusion rules for 2-Rep($\mathbb{Z}_4$) computed independently in appendix B.2. The fusion rule

$$D_2^{(\mathbb{Z}_2)} \otimes D_2^{(\mathbb{Z}_2)} \cong 2 D_2^{(\mathbb{Z}_2)}, \tag{77}$$

is derived in the same way as for non-identity object of 2Rep($\mathbb{Z}_2$). The fusion rule

$$D_2^{(\mathbb{Z}_2)} \otimes D_2^{(\mathbb{Z}_4)} \cong 2 D_2^{(\mathbb{Z}_4)}, \tag{78}$$

is derived by considering the endomorphisms that implement the symmetry, and by taking their tensor product

$$\begin{pmatrix} 0 & D_1^{(\text{id})} \\ D_1^{(\text{id})} & 0 \end{pmatrix} \otimes \begin{pmatrix} 0 & D_1^{(\mathbb{Z}_2^V;\text{id})} \\ D_1^{(\mathbb{Z}_2^V;\text{id})} & 0 \end{pmatrix}. \tag{79}$$

Finally, the fusion rule

$$D_2^{(\mathbb{Z}_4)} \otimes D_2^{(\mathbb{Z}_4)} \cong 4 D_2^{(\mathbb{Z}_4)}, \tag{80}$$

follows from taking the tensor product

$$\begin{pmatrix} 0 & D_1^{(\mathbb{Z}_2^V;\text{id})} \\ D_1^{(\mathbb{Z}_2^V;\text{id})} & 0 \end{pmatrix} \otimes \begin{pmatrix} 0 & D_1^{(\mathbb{Z}_2^V;\text{id})} \\ D_1^{(\mathbb{Z}_2^V;\text{id})} & 0 \end{pmatrix}, \tag{81}$$

using the fusion rule

$$D_1^{(\mathbb{Z}_2^V;\text{id})} \otimes D_1^{(\mathbb{Z}_2^V;\text{id})} = \begin{pmatrix} D_1^{(\mathbb{Z}_2^V;\text{id})} & 0 \\ 0 & D_1^{(\mathbb{Z}_2^V;\text{id})} \end{pmatrix}. \tag{82}$$

We then find that (81) evaluates to

$$\begin{pmatrix} 0 & 0 & 0 & 0 & 0 & 0 & D_1^{(\mathbb{Z}_2^V;\text{id})} & 0 \\ 0 & 0 & 0 & 0 & 0 & 0 & 0 & D_1^{(\mathbb{Z}_2^V;\text{id})} \\ 0 & 0 & 0 & 0 & D_1^{(\mathbb{Z}_2^V;\text{id})} & 0 & 0 & 0 \\ 0 & 0 & 0 & 0 & 0 & D_1^{(\mathbb{Z}_2^V;\text{id})} & 0 & 0 \\ 0 & 0 & D_1^{(\mathbb{Z}_2^V;\text{id})} & 0 & 0 & 0 & 0 & 0 \\ 0 & 0 & 0 & D_1^{(\mathbb{Z}_2^V;\text{id})} & 0 & 0 & 0 & 0 \\ D_1^{(\mathbb{Z}_2^V;\text{id})} & 0 & 0 & 0 & 0 & 0 & 0 & 0 \\ 0 & D_1^{(\mathbb{Z}_2^V;\text{id})} & 0 & 0 & 0 & 0 & 0 & 0 \end{pmatrix}, \tag{83}$$

which has four $D_2^{(\mathbb{Z}_4)}$ blocks sitting within it, justifying (80).

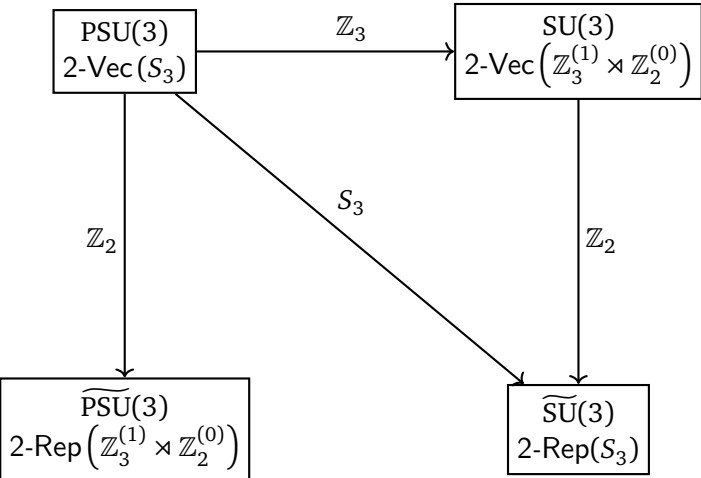

Figure 11: Categorical symmetry web for 3d gauge theories with gauge algebra $\mathfrak{su}(3)$. We label each theory by its global form of the gauge group and the associated symmetry category. Each arrow denotes a 0-form gauging and it is labelled by the subgroup of $S_3$ that is being gauged in that step.

## 4 Categorical symmetry web for $S_3$

Our general discussion was exemplified with $G = \mathbb{Z}_2 \times \mathbb{Z}_2$ and $\mathbb{Z}_4$. It is straight forward to extend these illustrative examples to other cyclic groups. In the following we will discuss two non-abelian groups: $S_3$ and $D_8$ and the resulting categorical symmetry web.

### 4.1 Setup and gauge theory

We start with the 0-form symmetry $G = S_3$, which is the smallest non-abelian group, which we express as

$$S_3 = \mathbb{Z}_3 \rtimes \mathbb{Z}_2 = \left\langle a, x \mid x^2 = a^3 = 1,\ xax = a^2 \right\rangle. \tag{84}$$

We consider 3d QFTs $\mathfrak{T}_{S_3}$ with $S_3$ 0-form symmetry and associated symmetry 2-category

$$\mathcal{C}_{S_3} = 2\text{-Vec}(S_3). \tag{85}$$

The topological surfaces, that generate the 0-form symmetry are labeled by group elements $D_2^{(g)}$, $g \in S_3$, and satisfy the fusion dictated by group multiplication. A concrete example for a gauge theory with this symmetry is pure PSU(3), which is obtained from pure SU(3) by gauging the $\mathbb{Z}_3^{(1)}$ 1-form center symmetry, which provides the $\mathbb{Z}_3^{(0)}$ subgroup of $S_3$. The non-trivial $\mathbb{Z}_2$ action comes from the outer automorphism (that acts by reflection on the Dynkin diagram). Starting from $\mathfrak{T}_{S_3} = \text{PSU}(3)$ with symmetry category $\mathcal{C}_{S_3} = 2\text{-Vec}(S_3)$ and gauging various subgroups of $S_3$, we will find the symmetry category web, with symmetry categories

$$\mathcal{C}_{S_3} = 2\text{-Vec}(S_3), \qquad \mathcal{C}_{S_3/\mathbb{Z}_3} = 2\text{-Vec}(\Gamma), \qquad \mathcal{C}_{S_3/S_3} = 2\text{-Rep}(S_3). \tag{86}$$

as illustrated in figure 11. In $\mathcal{C}_{S_3/\mathbb{Z}_3}$, the symmetry group is the 2-group $\Gamma$ formed by the 1-form symmetry $\mathbb{Z}_3$ and the 0-form symmetry $\mathbb{Z}_2$, where the latter acts on the former by outer automorphipsm $\rho$.

### 4.2 Gauging $\mathbb{Z}_3$ subgroup

We would like to determine the symmetry 2-category $\mathcal{C}_{S_3/\mathbb{Z}_3}$ of 3d QFT $\mathfrak{T}_{S_3/\mathbb{Z}_3}$ obtained by gauging $\mathbb{Z}_3$ subgroup of the $S_3$ 0-form symmetry of $\mathfrak{T}_{S_3}$. Here, we will only determine objects

and fusion of objects of the 2-category, leaving the determination of 1-morphisms and the composition and fusion rules thereof to the reader.

As described in section 2.2, the Schur components of objects in $\mathcal{C}_{S_3/\mathbb{Z}_3}$ are labelled by elements $\{1, x\} \in S_3/\mathbb{Z}_3 = \mathbb{Z}_2$. Multiples of the identity object $D_2^{(\text{id})} \in \mathcal{C}_{S_3}$ lead to a sub-2-category $2\text{Rep}(\mathbb{Z}_3)$ of $\mathcal{C}_{S_3/\mathbb{Z}_3}$, which has two simple objects (upto isomorphism)

$$D_2^{(\text{id})}, \qquad D_2^{(\mathbb{Z}_3)}. \tag{87}$$

The object $D_2^{(\mathbb{Z}_3)}$ arises from $3D_2^{(\text{id})} \in \mathcal{C}_{S_3}$, such that $a \in \mathbb{Z}_3$ acts as a cyclic permutation of the three copies of $D_2^{(\text{id})}$ in $3D_2^{(\text{id})}$. There are two different cyclic permutations $(123)$ and $(132)$ in which $a$ acts via the matrices[13]

$$D_1^{(\text{id})} P_{(123)} = \begin{pmatrix} 0 & D_1^{(\text{id})} & 0 \\ 0 & 0 & D_1^{(\text{id})} \\ D_1^{(\text{id})} & 0 & 0 \end{pmatrix}, \qquad \text{and} \qquad D_1^{(\text{id})} P_{(132)} = \begin{pmatrix} 0 & 0 & D_1^{(\text{id})} \\ D_1^{(\text{id})} & 0 & 0 \\ 0 & D_1^{(\text{id})} & 0 \end{pmatrix}. \tag{88}$$

We call the two corresponding objects of $\mathcal{C}_{S_3/\mathbb{Z}_3}$ respectively $D_2^{(\mathbb{Z}_3)}$ and $D_2^{(\mathbb{Z}_3')}$, out of which only $D_2^{(\mathbb{Z}_3)}$ features in the list (87) because $D_2^{(\mathbb{Z}_3)}$ and $D_2^{(\mathbb{Z}_3')}$ are isomorphic to each other. A 1-morphism $D_1^{(\mathbb{Z}_3', \mathbb{Z}_3; \varphi)}$ from $D_2^{(\mathbb{Z}_3')}$ to $D_2^{(\mathbb{Z}_3)}$ is a $3 \times 3$ matrix composed of $D_1^{(\text{id})} \in \mathcal{C}_{\mathfrak{T}}$ such that

$$D_1^{(\mathbb{Z}_3', \mathbb{Z}_3; \varphi)} \begin{pmatrix} 0 & D_1^{(\text{id})} & 0 \\ 0 & 0 & D_1^{(\text{id})} \\ D_1^{(\text{id})} & 0 & 0 \end{pmatrix} = \begin{pmatrix} 0 & 0 & D_1^{(\text{id})} \\ D_1^{(\text{id})} & 0 & 0 \\ 0 & D_1^{(\text{id})} & 0 \end{pmatrix} D_1^{(\mathbb{Z}_3', \mathbb{Z}_3; \varphi)}. \tag{89}$$

Some such 1-morphisms are given in terms of the transpositions

$$D_1^{(\mathbb{Z}_3', \mathbb{Z}_3; x)} = D_1^{(\text{id})} P_{(23)} = \begin{pmatrix} D_1^{(\text{id})} & 0 & 0 \\ 0 & 0 & D_1^{(\text{id})} \\ 0 & D_1^{(\text{id})} & 0 \end{pmatrix},$$

$$D_1^{(\mathbb{Z}_3', \mathbb{Z}_3; ax)} = D_1^{(\text{id})} P_{(13)} = \begin{pmatrix} 0 & 0 & D_1^{(\text{id})} \\ 0 & D_1^{(\text{id})} & 0 \\ D_1^{(\text{id})} & 0 & 0 \end{pmatrix}, \tag{90}$$

$$D_1^{(\mathbb{Z}_3', \mathbb{Z}_3; a^2 x)} = D_1^{(\text{id})} P_{(12)} = \begin{pmatrix} 0 & D_1^{(\text{id})} & 0 \\ D_1^{(\text{id})} & 0 & 0 \\ 0 & 0 & D_1^{(\text{id})} \end{pmatrix},$$

all of which are invertible and hence isomorphisms. The fusion $D_2^{(\mathbb{Z}_3)} \otimes D_2^{(\mathbb{Z}_3)}$ involves the tensor product of the $3 \times 3$ matrices in (88) and leads to

$$\begin{aligned} D_2^{(\mathbb{Z}_3)} \otimes D_2^{(\mathbb{Z}_3)} &= 3 D_2^{(\mathbb{Z}_3)}, \\ D_2^{(\mathbb{Z}_3')} \otimes D_2^{(\mathbb{Z}_3')} &= 3 D_2^{(\mathbb{Z}_3')}. \end{aligned} \tag{91}$$

The tensor product of the above isomorphisms will be important later, and can be computed to be (upto relabelling of rows and columns)

$$D_1^{(\mathbb{Z}_3', \mathbb{Z}_3; \varphi)} \otimes D_1^{(\mathbb{Z}_3', \mathbb{Z}_3; \varphi)} = D_1^{(\mathbb{Z}_3', \mathbb{Z}_3; \varphi)} \oplus \begin{pmatrix} 0 & D_1^{(\mathbb{Z}_3', \mathbb{Z}_3; \varphi)} \\ D_1^{(\mathbb{Z}_3', \mathbb{Z}_3; \varphi)} & 0 \end{pmatrix}, \tag{92}$$

---

[13]We again use the notation $P_\pi$ for the matrix representation of the permutation $\pi$.

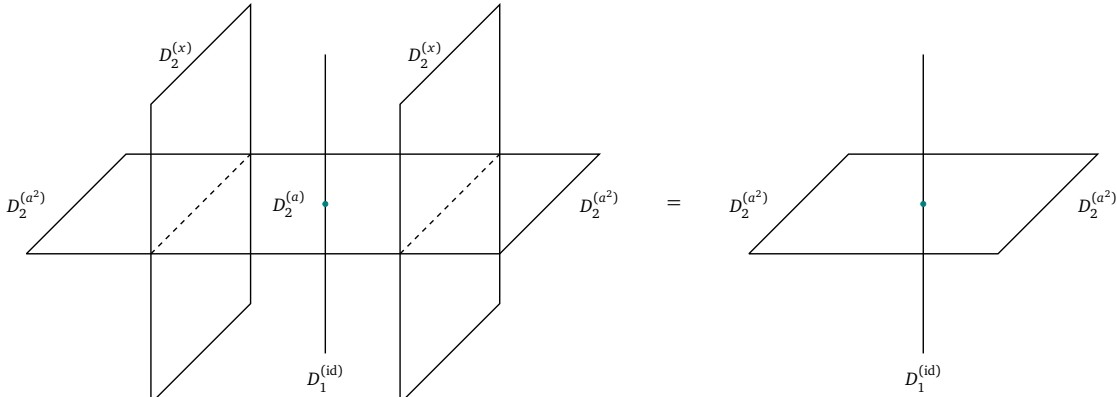

Figure 12: $D_2^{(x)}$ exchanges $D_2^{(a)}$ and $D_2^{(a^2)}$, which determine the action of the 0-form symmetry on the 1-form symmetry in the gauged theory $\mathcal{C}_{S_3/\mathbb{Z}_3}$.

for $\varphi \in \{x, ax, a^2 x\}$, as 1-morphisms from $3D_2^{(\mathbb{Z}_3')}$ to $3D_2^{(\mathbb{Z}_3)}$. Similarly $D_2^{(x)} \in \mathcal{C}_{S_3}$ leads to a single simple object (upto isomorphism), which we denote by $D_2^{(x)} \in \mathcal{C}_{S_3/\mathbb{Z}_3}$. Additionally $3D_2^{(x)} \in \mathcal{C}_{S_3}$ leads to a single simple object (upto isomorphism) denoted by $D_2^{(x\mathbb{Z}_3)}$ in $\mathcal{C}_{S_3/\mathbb{Z}_3}$. More precisely, we again naturally obtain two isomorphic objects $D_2^{(x\mathbb{Z}_3)}$ and $D_2^{(x\mathbb{Z}_3')}$ for which the combined action of $a$ from left and right is given respectively by matrices

$$
\begin{pmatrix} 0 & D_1^{(x;\mathrm{id})} & 0 \\ 0 & 0 & D_1^{(x;\mathrm{id})} \\ D_1^{(x;\mathrm{id})} & 0 & 0 \end{pmatrix}, \qquad \text{and} \qquad \begin{pmatrix} 0 & 0 & D_1^{(x;\mathrm{id})} \\ D_1^{(x;\mathrm{id})} & 0 & 0 \\ 0 & D_1^{(x;\mathrm{id})} & 0 \end{pmatrix}, \tag{93}
$$

where $D_1^{(x;\mathrm{id})} \in \mathcal{C}_{S_3}$ is the identity 1-endomorphism of $D_2^{(x)} \in \mathcal{C}_{S_3}$.

We have the fusion rules

$$
\begin{aligned}
D_2^{(x)} \otimes D_2^{(x)} &= D_2^{(\mathrm{id})}, \\
D_2^{(\mathbb{Z}_3)} \otimes D_2^{(x)} &= D_2^{(x\mathbb{Z}_3)}, \\
D_2^{(x)} \otimes D_2^{(\mathbb{Z}_3)} &= D_2^{(x\mathbb{Z}_3')} \cong D_2^{(x\mathbb{Z}_3)}, \\
D_2^{(x\mathbb{Z}_3)} \otimes D_2^{(x)} &= D_2^{(\mathbb{Z}_3)}, \\
D_2^{(x)} \otimes D_2^{(x\mathbb{Z}_3)} &= D_2^{(\mathbb{Z}_3')} \cong D_2^{(\mathbb{Z}_3)}, \\
D_2^{(\mathbb{Z}_3)} \otimes D_2^{(x\mathbb{Z}_3)} &\cong 3D_2^{(x\mathbb{Z}_3)}, \\
D_2^{(x\mathbb{Z}_3)} \otimes D_2^{(x\mathbb{Z}_3)} &\cong 3D_2^{(\mathbb{Z}_3)}.
\end{aligned} \tag{94}
$$

**Action of 0-form symmetry on 1-form symmetry.** The semi-direct product in (84) has an interesting consequence for the symmetries of $\mathfrak{T}_{S_3/\mathbb{Z}_3}$, namely the remaining $\mathbb{Z}_2$ 0-form symmetry of $\mathfrak{T}_{S_3/\mathbb{Z}_3}$ generated by $D_2^{(x)} \in \mathcal{C}_{S_3/\mathbb{Z}_3}$ acts on the $\mathbb{Z}_3$ 1-form symmetry of $\mathfrak{T}_{S_3/\mathbb{Z}_3}$ generated by simple 1-endomorphisms (upto isomorphism) of $D_2^{(\mathrm{id})}$ in $\mathcal{C}_{S_3/\mathbb{Z}_3}$

$$
\mathrm{End}_{\mathcal{C}_{S_3/\mathbb{Z}_3}}(D_2^{(\mathrm{id})}) = \left\{ D_1^{(\mathrm{id})}, D_1^{(\omega)}, D_1^{(\omega^2)} \right\}. \tag{95}
$$

To see this, let us first define $D_1^{(\omega)}$ to be the line obtained by making $D_1^{(\mathrm{id})} \in \mathcal{C}_{S_3}$ symmetric under the $\mathbb{Z}_3$ 0-form symmetry such that $a \in \mathbb{Z}_3$ acts by a third root of unity $\omega$ and $a^2$ acts by $\omega^2$. On the other hand, $D_1^{(\omega^2)}$ is the line obtained by making $D_1^{(\mathrm{id})} \in \mathcal{C}_{S_3}$ symmetric under $\mathbb{Z}_3$

0-form symmetry such that $a$ acts by $\omega^2$ and $a^2$ acts by $\omega$. Now, since the actions of $a$ and $a^2$ are exchanged by $D_2^{(x)}$ (see figure 12), we learn that

$$
\begin{aligned}
D_1^{(x;\mathrm{id})} \otimes D_1^{(\omega)} \otimes D_1^{(x;\mathrm{id})} &\cong D_1^{(\omega^2)}, \\
D_1^{(x;\mathrm{id})} \otimes D_1^{(\omega^2)} \otimes D_1^{(x;\mathrm{id})} &\cong D_1^{(\omega)},
\end{aligned}
\tag{96}
$$

which means that we have an exchange

$$
D_1^{(\omega)} \longleftrightarrow D_1^{(\omega^2)},
\tag{97}
$$

under $D_2^{(x)} \in \mathcal{C}_{S_3/\mathbb{Z}_3}$. There is a similar action of $D_2^{(x)}$ on the lines living on $D_2^{(\mathbb{Z}_3)}$. The simple 1-endomorphisms (upto isomorphism) of $D_2^{(\mathbb{Z}_3)}$ are

$$
\mathrm{End}_{\mathcal{C}_{S_3/\mathbb{Z}_3}}(D_2^{(\mathbb{Z}_3)}) = \left\{ D_1^{(\mathbb{Z}_3;\mathrm{id})}, D_1^{(\mathbb{Z}_3;a)}, D_1^{(\mathbb{Z}_3;a^2)} \right\},
\tag{98}
$$

where $D_1^{(\mathbb{Z}_3;\mathrm{id})}$ descends from the 1-endomorphism of $3D_2^{(\mathrm{id})} \in \mathcal{C}_{S_3}$ described by a diagonal $3 \times 3$ matrix with each entry being the line $D_1^{(\mathrm{id})} \in \mathcal{C}_{S_3}$. On the other hand, $D_1^{(\mathbb{Z}_3;a)}$ and $D_1^{(\mathbb{Z}_3;a^2)}$ descend from the matrix (88). Similarly, the simple 1-endomorphisms (upto isomorphism) of $D_2^{(\mathbb{Z}_3')}$ are

$$
\mathrm{End}_{\mathcal{C}_{S_3/\mathbb{Z}_3}}(D_2^{(\mathbb{Z}_3')}) = \left\{ D_1^{(\mathbb{Z}_3';\mathrm{id})}, D_1^{(\mathbb{Z}_3';a)}, D_1^{(\mathbb{Z}_3';a^2)} \right\}.
\tag{99}
$$

Since

$$
D_2^{(x)} \otimes D_2^{(\mathbb{Z}_3)} \otimes D_2^{(x)} = D_2^{(\mathbb{Z}_3')},
\tag{100}
$$

the lines of $D_2^{(\mathbb{Z}_3)}$ are mapped to the lines of $D_2^{(\mathbb{Z}_3')}$, and we need to further use an isomorphism $D_1^{(\mathbb{Z}_3',\mathbb{Z}_3;\varphi)}$ (for any $\varphi \in \{x, ax, a^2x\}$) to map the lines of $D_2^{(\mathbb{Z}_3')}$ back to the lines of $D_2^{(\mathbb{Z}_3)}$. In this way, we obtain

$$
\begin{aligned}
D_1^{(x;id)} \otimes D_1^{(\mathbb{Z}_3;a)} \otimes D_1^{(x;id)} &= D_1^{(\mathbb{Z}_3';a^2)} \cong D_1^{(\mathbb{Z}_3;a^2)}, \\
D_1^{(x;id)} \otimes D_1^{(\mathbb{Z}_3;a^2)} \otimes D_1^{(x;id)} &= D_1^{(\mathbb{Z}_3';a)} \cong D_1^{(\mathbb{Z}_3;a)}.
\end{aligned}
\tag{101}
$$

We can represent this as an action

$$
D_1^{(\mathbb{Z}_3;a)} \longleftrightarrow D_1^{(\mathbb{Z}_3;a^2)},
\tag{102}
$$

of $D_2^{(x)}$ on lines of $D_2^{(\mathbb{Z}_3)}$.

**Condensation defects and 2-group symmetry.** Due to the action (97), the $\mathbb{Z}_2$ 0-form symmetry and $\mathbb{Z}_3$ 1-form symmetry combine to form a split 2-group symmetry $\Gamma$ in the theory $\mathfrak{T}_{S_3/\mathbb{Z}_3}$. In fact, we can recognize the whole 2-category as being built completely out of the information regarding the 2-group $\Gamma$ i.e.

$$
\mathcal{C}_{S_3/\mathbb{Z}_3} \cong 2\text{-Vec}(\Gamma),
\tag{103}
$$

where the right hand side denotes the 2-category of 2-vector spaces graded by the 2-group $\Gamma$. The object $D_2^{(\mathbb{Z}_3)}$ is recovered as the condensation defect for the $\mathbb{Z}_3$ 1-form symmetry, and $D_2^{(x\mathbb{Z}_3)}$ is recovered by gauging the $\mathbb{Z}_3$ 1-form symmetry on the worldvolume of $D_2^{(x)} \in \mathcal{C}_{S_3/\mathbb{Z}_3}$.

### 4.3 Gauging $S_3$ in one go

Consider gauging the full $S_3$ 0-form symmetry of $\mathfrak{T}_{S_3}$ to pass to the theory $\mathfrak{T}_{S_3/S_3}$. The symmetry 2-category $\mathcal{C}_{S_3/S_3}$ associated to $\mathfrak{T}_{S_3/S_3}$ is

$$\mathcal{C}_{S_3/S_3} = 2\text{-Rep}(S_3), \tag{104}$$

which was described in detail in [14]. The simple objects (upto isomorphism) of $\mathcal{C}_{S_3/S_3}$ are

$$\text{Obj}(\mathcal{C}_{S_3/S_3}) = \left\{ D_2^{(\text{id})}, D_2^{(\mathbb{Z}_2)}, D_2^{(\mathbb{Z}_3)}, D_2^{(S_3)} \right\}, \tag{105}$$

with fusion rules

$$\begin{aligned}
D_2^{(\mathbb{Z}_2)} \otimes D_2^{(\mathbb{Z}_2)} &\cong 2 D_2^{(\mathbb{Z}_2)}, \\
D_2^{(\mathbb{Z}_2)} \otimes D_2^{(\mathbb{Z}_3)} &\cong D_2^{(S_3)}, \\
D_2^{(\mathbb{Z}_2)} \otimes D_2^{(S_3)} &\cong 2 D_2^{(S_3)}, \\
D_2^{(\mathbb{Z}_3)} \otimes D_2^{(\mathbb{Z}_3)} &\cong D_2^{(\mathbb{Z}_3)} \oplus D_2^{(S_3)}, \\
D_2^{(\mathbb{Z}_3)} \otimes D_2^{(S_3)} &\cong 3 D_2^{(S_3)}, \\
D_2^{(S_3)} \otimes D_2^{(S_3)} &\cong 6 D_2^{(S_3)}.
\end{aligned} \tag{106}$$

We will use this as a point of comparision in the following step-wise gauging.

### 4.4 Gauging $S_3$ sequentially

We can also construct $\mathfrak{T}_{S_3/S_3}$ by gauging the $\mathbb{Z}_2$ 0-form symmetry generated by $D_2^{(x)} \in \mathcal{C}_{S_3/\mathbb{Z}_3}$ of the theory $\mathfrak{T}_{S_3/\mathbb{Z}_3}$. We would like to recover the $\mathcal{C}_{S_3/S_3}$ fusion rules in (106) by implementing this $\mathbb{Z}_2$ gauging on the symmetry 2-category $\mathcal{C}_{S_3/\mathbb{Z}_3}$.

From multiples of $D_2^{(\text{id})} \in \mathcal{C}_{S_3/\mathbb{Z}_3}$, we obtain a sub-2-category $2\text{Rep}(\mathbb{Z}_2)$ of $\mathcal{C}_{S_3/S_3}$ described by objects

$$D_2^{(\text{id})}, \ D_2^{(\mathbb{Z}_2)} \in \mathcal{C}_{S_3/S_3}. \tag{107}$$

Now consider making $D_2^{(\mathbb{Z}_3)} \in \mathcal{C}_{S_3/\mathbb{Z}_3}$ symmetric under $\mathbb{Z}_2$ 0-form symmetry. Because of the fusion rule (100), a way of making $D_2^{(\mathbb{Z}_3)} \in \mathcal{C}_{S_3/\mathbb{Z}_3}$ symmetric under $\mathbb{Z}_2$ 0-form symmetry requires a choice of a morphism $M_{up} : D_2^{(\mathbb{Z}_3')} \to D_2^{(\mathbb{Z}_3)} \in \mathcal{C}_{S_3/\mathbb{Z}_3}$ and a morphism $M_p : D_2^{(\mathbb{Z}_3)} \to D_2^{(\mathbb{Z}_3')} \in \mathcal{C}_{S_3/\mathbb{Z}_3}$ such that we have

$$\begin{aligned}
M_{up} \circ M_p &= D_1^{(\mathbb{Z}_3';\text{id})}, \\
M_p \circ M_{up} &= D_1^{(\mathbb{Z}_3;\text{id})}.
\end{aligned} \tag{108}$$

We can thus choose $M_{up} = D_1^{(\mathbb{Z}_3',\mathbb{Z}_3;\varphi)}$ for any $\varphi \in \{x, ax, a^2 x\}$ defined above. The different choices of $\varphi$ lead to isomorphic objects in $\mathcal{C}_{S_3/S_3}$. For concreteness, let us pick $M_{up} = D_1^{(\mathbb{Z}_3',\mathbb{Z}_3;x)}$ and call the resulting object as $D_2^{(\mathbb{Z}_3)}$ in $\mathcal{C}_{S_3/S_3}$. Finally, the object $D_2^{(S_3)}$ in $\mathcal{C}_{S_3/S_3}$ arises by making $2 D_2^{(\mathbb{Z}_3)} \in \mathcal{C}_{S_3/\mathbb{Z}_3}$ symmetric under $\mathbb{Z}_2$ 0-form symmetry by implementing it using the isomorphism

$$\begin{pmatrix} 0 & D_1^{(\mathbb{Z}_3',\mathbb{Z}_3;x)} \\ D_1^{(\mathbb{Z}_3',\mathbb{Z}_3;x)} & 0 \end{pmatrix} : 2 D_2^{(\mathbb{Z}_3')} \to 2 D_2^{(\mathbb{Z}_3)} \in \mathcal{C}_{S_3/\mathbb{Z}_3}. \tag{109}$$

Now let us recover the fusion rules (106). The first rule $D_2^{(\mathbb{Z}_2)} \otimes D_2^{(\mathbb{Z}_2)} \cong 2 D_2^{(\mathbb{Z}_2)}$ is simply part of the 2-Rep($\mathbb{Z}_2$) data that we discussed above. Computing $D_2^{(\mathbb{Z}_2)} \otimes D_2^{(\mathbb{Z}_3)}$ involves computing

the matrix tensor product

$$\begin{pmatrix} 0 & D_1^{(\text{id})} \\ D_1^{(\text{id})} & 0 \end{pmatrix} \otimes D_1^{(\mathbb{Z}_3',\mathbb{Z}_3;x)} = \begin{pmatrix} 0 & D_1^{(\mathbb{Z}_3',\mathbb{Z}_3;x)} \\ D_1^{(\mathbb{Z}_3',\mathbb{Z}_3;x)} & 0 \end{pmatrix}, \tag{110}$$

leading to $D_2^{(\mathbb{Z}_2)} \otimes D_2^{(\mathbb{Z}_3)} \cong D_2^{(S_3)}$. Similarly, computing

$$\begin{pmatrix} 0 & D_1^{(\text{id})} \\ D_1^{(\text{id})} & 0 \end{pmatrix} \otimes \begin{pmatrix} 0 & D_1^{(\mathbb{Z}_3',\mathbb{Z}_3;x)} \\ D_1^{(\mathbb{Z}_3',\mathbb{Z}_3;x)} & 0 \end{pmatrix}, \tag{111}$$

leads to $D_2^{(\mathbb{Z}_2)} \otimes D_2^{(S_3)} \cong 2D_2^{(S_3)}$. To compute $D_2^{(\mathbb{Z}_3)} \otimes D_2^{(\mathbb{Z}_3)}$, we have to compute the fusion $D_1^{(\mathbb{Z}_3',\mathbb{Z}_3;x)} \otimes D_1^{(\mathbb{Z}_3',\mathbb{Z}_3;x)}$ which we computed earlier and found to be given by (92), thus leading to the fusion rule

$$D_2^{(\mathbb{Z}_3)} \otimes D_2^{(\mathbb{Z}_3)} \cong D_2^{(\mathbb{Z}_3)} \oplus D_2^{(S_3)}. \tag{112}$$

For $D_2^{(\mathbb{Z}_3)} \otimes D_2^{(S_3)}$, we compute

$$D_1^{(\mathbb{Z}_3',\mathbb{Z}_3;x)} \otimes \begin{pmatrix} 0 & D_1^{(\mathbb{Z}_3',\mathbb{Z}_3;x)} \\ D_1^{(\mathbb{Z}_3',\mathbb{Z}_3;x)} & 0 \end{pmatrix} = \bigoplus_{j=1}^{3} \begin{pmatrix} 0 & D_1^{(\mathbb{Z}_3',\mathbb{Z}_3;x)} \\ D_1^{(\mathbb{Z}_3',\mathbb{Z}_3;x)} & 0 \end{pmatrix}, \tag{113}$$

leading to the fusion rule

$$D_2^{(\mathbb{Z}_3)} \otimes D_2^{(S_3)} \cong 3D_2^{(S_3)}. \tag{114}$$

Finally, computing

$$\begin{pmatrix} 0 & D_1^{(\mathbb{Z}_3',\mathbb{Z}_3;x)} \\ D_1^{(\mathbb{Z}_3',\mathbb{Z}_3;x)} & 0 \end{pmatrix} \otimes \begin{pmatrix} 0 & D_1^{(\mathbb{Z}_3',\mathbb{Z}_3;x)} \\ D_1^{(\mathbb{Z}_3',\mathbb{Z}_3;x)} & 0 \end{pmatrix}, \tag{115}$$

it is straightforward to show that $D_2^{(S_3)} \otimes D_2^{(S_3)} \cong 6D_2^{(S_3)}$. We have therefore reproduced the 2-Rep($S_3$) fusion rules displayed in (106).

## 4.5 Gauging the 2-group symmetry

Consider gauging the 2-group symmetry $\Gamma$ of the theory $\mathfrak{T}_{S_3/\mathbb{Z}_3}$. This is equivalent to first ungauging $\mathbb{Z}_3$ and then gauging $\mathbb{Z}_2$, so we reach the theory $\mathfrak{T}_{S_3/\mathbb{Z}_2}$ which is obtained from $\mathfrak{T}_{S_3}$ by gauging one of the $\mathbb{Z}_2$ subgroups of the $S_3$ 0-form symmetry of $\mathfrak{T}_{S_3}$. In other words, we have

$$\mathfrak{T}_{S_3/\mathbb{Z}_3/\Gamma} \cong \mathfrak{T}_{S_3/\mathbb{Z}_2}, \tag{116}$$

and hence the symmetry 2-category $\mathcal{C}_{S_3/\mathbb{Z}_2}$ associated to $\mathfrak{T}_{S_3/\mathbb{Z}_2}$ is

$$\mathcal{C}_{S_3/\mathbb{Z}_2} \cong 2\text{-Rep}(\Gamma), \tag{117}$$

where the right hand side denotes the 2-category formed by 2-representations of the 2-group $\Gamma$. This 2-category can be easily computed using the description presented in [14] and it has three simple objects

$$\text{Obj}(2\text{-Rep}(\Gamma)) = \left\{ D_2^{(\text{id})}, D_2^{(\mathbb{Z}_2)}, D_2^{(\omega\omega^2)} \right\}, \tag{118}$$

with fusion rules

$$\begin{aligned} D_2^{(\mathbb{Z}_2)} \otimes D_2^{(\mathbb{Z}_2)} &\cong 2D_2^{(\mathbb{Z}_2)}, \\ D_2^{(\mathbb{Z}_2)} \otimes D_2^{(\omega\omega^2)} &\cong 2D_2^{(\omega\omega^2)}, \\ D_2^{(\omega\omega^2)} \otimes D_2^{(\omega\omega^2)} &\cong D_2^{(\mathbb{Z}_2)} \oplus D_2^{(\omega\omega^2)}. \end{aligned} \tag{119}$$

## 4.6 Gauging the $\mathbb{Z}_2$ subgroup

Now consider instead directly gauging $\mathbb{Z}_2$ subgroup of $S_3$ to transition from $\mathcal{C}_{S_3}$ to $\mathcal{C}_{S_3/\mathbb{Z}_2}$. Without loss of generality, we choose the $\mathbb{Z}_2$ subgroup generated by $x \in S_3$. Multiples of $D_2^{(\mathrm{id})} \in \mathcal{C}_{S_3}$ give rise to two simple objects

$$D_2^{(\mathrm{id})}, D_2^{(\mathbb{Z}_2)} \in \mathrm{Obj}(\mathcal{C}_{S_3/\mathbb{Z}_2}). \tag{120}$$

Since we have

$$D_2^{(x)} \otimes D_2^{(a)} \otimes D_2^{(x)} = D_2^{(a^2)}, \tag{121}$$

and there are no 1-morphisms between $D_2^{(a)}$ and $D_2^{(a^2)}$, it is not possible to make either $D_2^{(a)}$ or $D_2^{(a^2)}$ individually symmetric under $\mathbb{Z}_2^x$, but we can consider making their sum $D_2^{(a)} \oplus D_2^{(a^2)}$ symmetric under $\mathbb{Z}_2^x$. This requires choosing an isomorphism $D_2^{(a^2)} \oplus D_2^{(a)} \to D_2^{(a)} \oplus D_2^{(a^2)}$, which can be taken to be given by

$$\begin{pmatrix} 0 & D_1^{(a^2;\mathrm{id})} \\ D_1^{(a;\mathrm{id})} & 0 \end{pmatrix}, \tag{122}$$

where the first row corresponds to $D_2^{(a^2)}$, the second row corresponds to $D_2^{(a)}$, the first column corresponds to $D_2^{(a)}$, and the second column corresponds to $D_2^{(a^2)}$. This gives rise to the object

$$D_2^{(\omega\omega^2)} \in \mathrm{Obj}(\mathcal{C}_{S_3/\mathbb{Z}_2}). \tag{123}$$

Let us now re-derive the fusion rules (119). The first rule $D_2^{(\mathbb{Z}_2)} \otimes D_2^{(\mathbb{Z}_2)} \cong 2D_2^{(\mathbb{Z}_2)}$ is part of the 2Rep($\mathbb{Z}_2$) sub-2-category of $\mathcal{C}_{S_3/\mathbb{Z}_2}$. To compute $D_2^{(\mathbb{Z}_2)} \otimes D_2^{(\omega\omega^2)}$, we need to compute the tensor product

$$\begin{pmatrix} 0 & D_1^{(\mathrm{id})} \\ D_1^{(\mathrm{id})} & 0 \end{pmatrix} \otimes \begin{pmatrix} 0 & D_1^{(a^2;\mathrm{id})} \\ D_1^{(a;\mathrm{id})} & 0 \end{pmatrix} = \begin{pmatrix} 0 & 0 & 0 & D_1^{(a^2;\mathrm{id})} \\ 0 & 0 & D_1^{(a;\mathrm{id})} & 0 \\ 0 & D_1^{(a^2;\mathrm{id})} & 0 & 0 \\ D_1^{(a;\mathrm{id})} & 0 & 0 & 0 \end{pmatrix}, \tag{124}$$

where, for the matrix on the right hand side, the first and third rows correspond to $D_2^{(a^2)}$, the second and fourth rows correspond to $D_2^{(a)}$, the first and third columns correspond to $D_2^{(a)}$, and the second and fourth columns correspond to $D_2^{(a^2)}$. Since the right hand side matrix decomposes as a direct sum of (122) with itself, we recover the fusion rule

$$D_2^{(\mathbb{Z}_2)} \otimes D_2^{(\omega\omega^2)} \cong 2D_2^{(\omega\omega^2)}. \tag{125}$$

For the fusion $D_2^{(\omega\omega^2)} \otimes D_2^{(\omega\omega^2)}$, compute the tensor product

$$\begin{pmatrix} 0 & D_1^{(a^2;\mathrm{id})} \\ D_1^{(a;\mathrm{id})} & 0 \end{pmatrix} \otimes \begin{pmatrix} 0 & D_1^{(a^2;\mathrm{id})} \\ D_1^{(a;\mathrm{id})} & 0 \end{pmatrix} = \begin{pmatrix} 0 & 0 & 0 & D_1^{(a;\mathrm{id})} \\ 0 & 0 & D_1^{(\mathrm{id})} & 0 \\ 0 & D_1^{(\mathrm{id})} & 0 & 0 \\ D_1^{(a^2;\mathrm{id})} & 0 & 0 & 0 \end{pmatrix}, \tag{126}$$

where, for the matrix on the right hand side, the first row corresponds to $D_2^{(a)}$, the second and third rows correspond to $D_2^{(\mathrm{id})}$, the fourth row corresponds to $D_2^{(a^2)}$, the first column corresponds to $D_2^{(a^2)}$, the second and third columns correspond to $D_2^{(\mathrm{id})}$, and the fourth column corresponds to $D_2^{(a)}$. From the direct sum decomposition of this matrix, we learn that

$$D_2^{(\omega\omega^2)} \otimes D_2^{(\omega\omega^2)} \cong D_2^{(\mathbb{Z}_2)} \oplus D_2^{(\omega\omega^2)}. \tag{127}$$

Therefore, we have reproduced the fusion rules of 2-Rep($\Gamma$) displayed in (119) by directly gauging $\mathbb{Z}_2$ in 2-Vec($S_3$).

# 5 The $D_8$-categorical symmetry web for 3d orthogonal gauge theories

## 5.1 Web of 3d orthogonal gauge theories

A rich class of categorical symmetries arises for gauge theories with orthogonal gauge groups, with gauge algebra $\mathfrak{so}(2N)$. We will now construct the full web of categorical symmetries and most importantly discuss each of the gauging steps. We will see that although the webs for $\mathfrak{so}(4N)$ and $\mathfrak{so}(4N+2)$ are same, the gauge groups labeling the same nodes in the two webs are of different types.

We start with the theory which has purely a 0-form symmetry, which in this case is $D_8$, and is realized in the PSO($4N$) and PSO($4N+2$) gauge theories. Here the group $D_8$ will be presented as follows

$$D_8 = \mathbb{Z}_4 \rtimes \mathbb{Z}_2 = \left\langle a, x \ \middle| \ x^2 = a^4 = 1, xax = a^3 \right\rangle. \tag{128}$$

The symmetry category $\mathcal{C}_{D_8}$ attached to PSO($4N$) and PSO($4N+2$) gauge theories has surface defects $D_2^{(g)}$, $g \in D_8$, and is given by

$$\mathcal{C}_{D_8} = 2\text{-Vec}(D_8). \tag{129}$$

All other gauge theories with gauge groups (including both connected and disconnected ones) having gauge algebras $\mathfrak{so}(4N)$ and $\mathfrak{so}(4N+2)$ can be obtained by gauging a 0-form symmetry subgroup of $D_8$. The resulting symmetry category webs are shown in figures 1 and 13. We will discuss both the partial gaugings of subgroups starting with 2-Vec($D_8$), but also the subsequent gauging of the categories by the remaining subgroups.

Let us specify precisely how these symmetries are realized in the 3d gauge theories: perhaps the most familiar theory is the Spin($4N$) gauge theory, which has 0-form symmetry $\mathbb{Z}_2^x$, that acts as outer automorphism, and 1-form symmetry $\mathbb{Z}_2 \times \mathbb{Z}_2$ that acts (on Wilson lines) as the center[14] of Spin($4N$). Gauging one of the two $\mathbb{Z}_2$ 1-form symmetries leads to two equivalent theories with gauge groups called Ss($4N$) or Sc($4N$), while gauging the diagonal $\mathbb{Z}_2$ subgroup of the 1-form symmetry gives the SO($4N$) theory. Gauging the full 1-form symmetry results in the theory PSO($4N$), which is our starting point for the web. Gauging instead the 0-form symmetry starting with Spin($4N$) results in Pin$^+$($4N$), which is also the theory obtained by gauging the full $D_8$ 0-form symmetry of the PSO($4N$) theory. The remaining gaugings are shown in figure 1.

Analogously, Spin($4N+2$) gauge theory has outer-automorphism 0-form symmetry $\mathbb{Z}_2^x$ and center 1-form symmetry $\mathbb{Z}_4$. Gauging $\mathbb{Z}_2$ subgroup of the 1-form symmetry gives the SO($4N+2$) theory. Gauging the full 1-form symmetry results in the theory PSO($4N+2$), which is our starting point for the web. Gauging instead the 0-form symmetry starting with Spin($4N+2$) results in Pin$^+$($4N+2$), which is also the theory obtained by gauging the full $D_8$ 0-form symmetry of the PSO($4N+2$) theory. The remaining gaugings are shown in figure 13.

In figures 1 and 13 we have indicated the 0-form symmetry groups that we gauge in each of the steps. Starting with PSO($4N$) and PSO($4N+2$) these are subgroups of $D_8$, however

---

[14]The two $\mathbb{Z}_2$ in the center are distinguished by the irreducible spinor representation (namely spinor or cospinor) they act on.

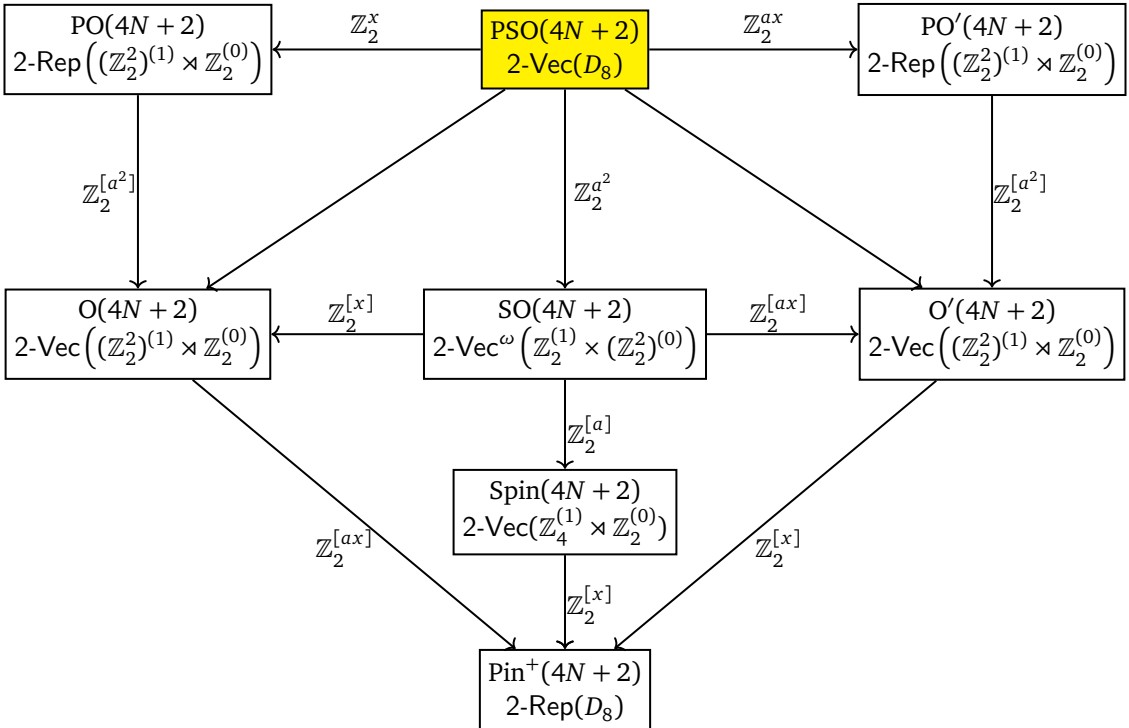

Figure 13: Categorical symmetry web for 3d gauge theories with gauge algebra $\mathfrak{so}(4N+2)$. We label each theory by its global gauge group, and the 2-category which is its symmetry category. The arrows denote gaugings of 0-form symmetries and each arrow is labelled by the subgroup of $D_8$ that is gauged along that arrow. We discuss each of these steps in the text. Note that each arrow corresponds to a 0-form symmetry gauging. The notation for the groups $\mathbb{Z}_2^{[g]}$ is explained in the text around (130).

in subsequent gaugings the generators are uniquely fixed only up to left multiplication by the elements of the subgroup that was gauged in the previous step. In this case we label the subgroup by $[g]$. For example, after gauging $\mathbb{Z}_2^{a^2}$ the remaining group is $D_8/\mathbb{Z}_2^{a^2}$ and we can gauge e.g. the group generated by $ax$, which is equivalent to $a^3x$ etc. We then denote the corresponding subgroup by

$$\mathbb{Z}_2^{[ax]} = \{g\,ax;\ g\in\mathbb{Z}_2^{a^2}\}. \tag{130}$$

To determine which left coset we consider simply requires tracing back the gauging steps to $PSO(4N)$.[15]

Note that the webs shown in figures 1 and 13 are symmetric under reflection across the vertical axis, which can be understood as the action of outer-automorphism of $D_8$. This relates different theories that have same symmetries.[16]

---

[15]We refrain from specifying which groups we use for the coset action. They can be determined using the figure from the groups that appear in the gauging process prior to the one under consideration.

[16]The inner automorphisms of $D_8$ have been modded out from the webs as this relates theories appearing in the webs to isomorphic theories. For example, if we gauge $\mathbb{Z}_2^{a^3x}$ (related to $\mathbb{Z}_2^{ax}$ by conjugation) 0-form symmetry of $PSO(4N)$, we obtain the $Sc(4N)$ gauge theory which is isomorphic to the $Ss(4N)$ gauge theory obtained by gauging $\mathbb{Z}_2^{ax}$ 0-form symmetry of $PSO(4N)$.

## 5.2 $\mathrm{PSO}(2N) \to \mathrm{SO}(2N)$

In this subsection, we gauge the normal $\mathbb{Z}_2$ subgroup of $D_8$ generated by the element $a^2$ and determine the resulting 2-category $\mathcal{C}_{D_8/\mathbb{Z}_2^{a^2}}$. The starting point is the category

$$\boxed{\mathcal{C}_{D_8} = 2\text{-Vec}(D_8)\,,} \tag{131}$$

and we will show that after gauging $\mathbb{Z}_2^{a^2}$ we obtain

$$\boxed{\mathcal{C}_{D_8/\mathbb{Z}_2^{a^2}} = 2\text{-Vec}^\omega\left(\mathbb{Z}_2^{(0)} \times \mathbb{Z}_2^{(0)} \times \mathbb{Z}_2^{(1)}\right),} \tag{132}$$

which carries a 2-group with $\mathbb{Z}_2^{(0)} \times \mathbb{Z}_2^{(0)}$ 0-form symmetry and $\mathbb{Z}_2^{(1)}$ 1-form symmetry, with trivial action and trivial Postnikov class, but with a non-trivial anomaly $\omega$ (which is a 4-cocycle on the 2-group $H^4(B\Gamma, U(1))$). The anomaly implies that there is fractionalization of $\mathbb{Z}_2^{(0)} \times \mathbb{Z}_2^{(0)}$ 0-form symmetry on the line operator generating $\mathbb{Z}_2^{(1)}$ 1-form symmetry and also on the condensation surface defect associated to $\mathbb{Z}_2^{(1)}$ 1-form symmetry.

At the level of 3d orthogonal gauge theories, $\mathbb{Z}_2^{a^2}$ gauging corresponds to the transitions $\mathrm{PSO}(2N) \to \mathrm{SO}(2N)$ – see figures 1 and 13.

**Objects and 1-morphisms.** Different Schur components of simple objects in $\mathcal{C}_{D_8/\mathbb{Z}_2^{a^2}}$ are labelled by elements $g \in \{1, a, x, ax\} = \mathbb{Z}_2 \times \mathbb{Z}_2 = D_8/\mathbb{Z}_2^{a^2}$. In each Schur component, we have a copy of 2-Rep($\mathbb{Z}_2$). Thus, the simple objects (upto isomorphism) of $\mathcal{C}_{D_8/\mathbb{Z}_2^{a^2}}$ can be labeled as

$$\mathrm{Obj}(\mathcal{C}_{D_8/\mathbb{Z}_2^{a^2}}) = \left\{ D_2^{(g)}, D_2^{(g\mathbb{Z}_2)} \,\middle|\, g \in \{1, a, x, ax\} = \mathbb{Z}_2 \times \mathbb{Z}_2 = D_8/\mathbb{Z}_2^{a^2} \right\}. \tag{133}$$

These are twisted theta defects with underlying twist being $D_2^{(g)} \in \mathcal{C}_{D_8}$. For $g = \mathrm{id}$, we have the ordinary theta defects of [14].

The fusions of objects follows 2-Rep($\mathbb{Z}_2$) fusion rules graded by the Schur component group $\mathbb{Z}_2 \times \mathbb{Z}_2$. That is,

$$
\begin{aligned}
D_2^{(g)} \otimes D_2^{(h)} &\cong D_2^{(gh)}\,, \\
D_2^{(g\mathbb{Z}_2)} \otimes D_2^{(h)} &\cong D_2^{(gh\mathbb{Z}_2)}\,, \\
D_2^{(g\mathbb{Z}_2)} \otimes D_2^{(h\mathbb{Z}_2)} &\cong 2D_2^{(gh\mathbb{Z}_2)}.
\end{aligned}
\tag{134}
$$

The 1-morphisms of $\mathcal{C}_{D_8/\mathbb{Z}_2^{a^2}}$ are also 1-morphisms of 2-Rep($\mathbb{Z}_2$) graded by $\mathbb{Z}_2 \times \mathbb{Z}_2$. In particular, we have simple 1-endomorphisms (upto isomorphism)

$$D_1^{(g;\mathrm{id})}\,, \qquad D_1^{(g;a^2)}\,, \tag{135}$$

of $D_2^{(g)}$. Here $D_1^{(g,\mathrm{id})}$ is the identity 1-endomorphism (under composition) and $D_1^{(g,a^2)}$ is a non-identity 1-endomorphism of order two. We also have simple 1-endomorphisms (upto isomorphism)

$$D_1^{(g\mathbb{Z}_2,\mathrm{id})}\,, \qquad D_1^{(g\mathbb{Z}_2;-)}\,, \tag{136}$$

of $D_2^{(g\mathbb{Z}_2)}$. Here $D_1^{(g\mathbb{Z}_2;\mathrm{id})}$ is the identity 1-endomorphism (under composition) and $D_1^{(g\mathbb{Z}_2;-)}$ is a non-identity 1-endomorphism of order two.

In summary, at the level of objects and 1-morphisms we seem to obtain that $\mathcal{C}_{D_8/\mathbb{Z}_2^{a^2}}$ equals

$$2\text{-Vec}(\mathbb{Z}_2 \times \mathbb{Z}_2) \boxtimes 2\text{-Rep}(\mathbb{Z}_2) = 2\text{-Vec}\left(\mathbb{Z}_2^{(0)} \times \mathbb{Z}_2^{(0)} \times \mathbb{Z}_2^{(1)}\right). \tag{137}$$

We will now show that in $\mathcal{C}_{D_8/\mathbb{Z}_2^{a^2}}$ there is in fact an additional non-trivial cocycle $\omega$ due to symmetry fractionalization.

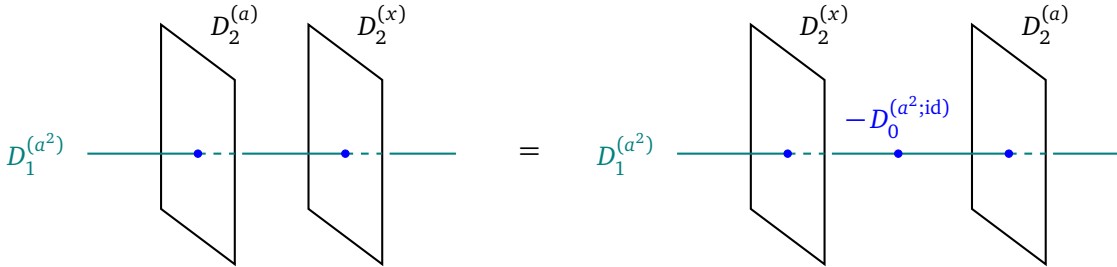

Figure 14: The $\mathbb{Z}_2 \times \mathbb{Z}_2$ symmetry generated by $D_2^{(x)}$ and $D_2^{(a)}$ fractionalizes on the line $D_1^{(a^2)}$. As a consequence, commuting $D_2^{(x)}$ past $D_2^{(a)}$ on $D_1^{(a^2)}$, leaves behind the local operator $-D_0^{(a^2;\text{id})}$.

**Mixed anomaly and symmetry fractionalization.** Given that the extension

$$1 \to \mathbb{Z}_2^{a^2} \to D_8 \to \mathbb{Z}_2 \times \mathbb{Z}_2 \to 1\,, \tag{138}$$

is non-split, characterized by a non-trivial extension class $\epsilon \in H^2(\mathbb{Z}_2 \times \mathbb{Z}_2, \mathbb{Z}_2)$, we expect the theory $\mathfrak{T}_{D_8/\mathbb{Z}_2^{a^2}}$ to have a mixed 't Hooft anomaly $\omega$ given by (35).

This mixed anomaly is associated to symmetry fractionalization of $\mathbb{Z}_2 \times \mathbb{Z}_2$ 0-form symmetry generated by $D_2^{(g)} \in \mathcal{C}_{D_8/\mathbb{Z}_2^{a^2}}$ on the bulk line $D_1^{(a^2)}$. In fact, $\mathbb{Z}_2 \times \mathbb{Z}_2$ 0-form symmetry fractionalizes to $D_8$ 0-form symmetry on $D_1^{(a^2)}$.

First of all, the $\mathbb{Z}_2$ subgroup of $\mathbb{Z}_2 \times \mathbb{Z}_2$ generated by $D_2^{(a)}$ fractionalizes to $\mathbb{Z}_4$ on $D_1^{(a^2)}$ by the same arguments as encountered previously in the study of $G = \mathbb{Z}_4, H = \mathbb{Z}_2$ example. See figure 6. On the other hand, $\mathbb{Z}_2$ subgroup of $\mathbb{Z}_2 \times \mathbb{Z}_2$ generated by $D_2^{(x)}$ cannot fractionalize because $D_2^{(x)}$ generates a $\mathbb{Z}_2$ symmetry in $\mathcal{C}_{D_8}$. Similarly, the $\mathbb{Z}_2$ subgroup generated by $D_2^{(ax)}$ cannot fractionalize for the same reason.

However, this means that $D_2^{(a)}$ and $D_2^{(x)}$ must not commute on the line $D_1^{(a^2)}$. To see this directly, consider a configuration in $\mathcal{C}_{D_8/\mathbb{Z}_2^{a^2}}$ where $D_1^{(a^2)}$ intersects a $D_2^{(a)}$ and a $D_2^{(x)}$ surface. In $\mathcal{C}_{D_8}$, this corresponds to a configuration where $D_1^{(\text{id})}$ intersects $D_2^{(a)}$ and $D_2^{(x)}$. We can commute the two surfaces while they still intersect $D_1^{(a^2)}$. If we perform this operation in $\mathcal{C}_{D_8}$, we find a configuration where $D_1^{(\text{id})}$ intersects $D_2^{(x)}$ and $D_2^{(a^3)}$. Writing $D_2^{(a^3)}$ as $D_2^{(a^2)} \otimes D_2^{(a)}$ and using the definition of $D_1^{(a^2)} \in \mathcal{C}_{D_8/\mathbb{Z}_2^{a^2}}$, we learn that this commutation process produces a non-trivial sign, due to the appearance of a non-trivial local operator $-D_0^{(a^2;\text{id})}$ (namely minus times the identity operator) living on the line $D_1^{(a^2)}$. See figure 14.

The $\mathbb{Z}_2 \times \mathbb{Z}_2$ 0-form symmetry also fractionalizes to $D_8$ symmetry on the worldvolume of the condensation defect $D_2^{(\mathbb{Z}_2)}$ in $\mathcal{C}_{D_8/\mathbb{Z}_2^{a^2}}$ with the line $D_1^{(\mathbb{Z}_2;-)}$ living on $D_2^{(\mathbb{Z}_2)}$ providing the central $\mathbb{Z}_2$. Call a line living at the junction of $D_2^{(\mathbb{Z}_2)}$ and $D_2^{(g)}$ as $D_1^{(\mathbb{Z}_2;g)}$. Then we have

$$
\begin{aligned}
D_1^{(\mathbb{Z}_2;a)} \otimes_{D_2^{(\mathbb{Z}_2)}} D_1^{(\mathbb{Z}_2;a)} &= D_1^{(\mathbb{Z}_2;-)}\,, \\
D_1^{(\mathbb{Z}_2;x)} \otimes_{D_2^{(\mathbb{Z}_2)}} D_1^{(\mathbb{Z}_2;x)} &= D_1^{(\mathbb{Z}_2;\text{id})}\,, \\
D_1^{(\mathbb{Z}_2;ax)} \otimes_{D_2^{(\mathbb{Z}_2)}} D_1^{(\mathbb{Z}_2;ax)} &= D_1^{(\mathbb{Z}_2;\text{id})}\,, \\
D_1^{(\mathbb{Z}_2;a)} \otimes_{D_2^{(\mathbb{Z}_2)}} D_1^{(\mathbb{Z}_2;x)} &= D_1^{(\mathbb{Z}_2;-)} \otimes_{D_2^{(\mathbb{Z}_2)}} D_1^{(\mathbb{Z}_2;x)} \otimes_{D_2^{(\mathbb{Z}_2)}} D_1^{(\mathbb{Z}_2;a)}\,,
\end{aligned}
\tag{139}
$$

following from the general arguments of section 2.5, where $D_1^{(\mathbb{Z}_2;\mathrm{id})}$ and $D_1^{(\mathbb{Z}_2;-)}$ are lines living on $D_2^{(\mathbb{Z}_2)}$ discussed above.

## 5.3 $\mathrm{SO}(4N) \to \mathrm{O}'(4N)$ and $\mathrm{SO}(4N+2) \to \mathrm{Spin}(4N+2)$

We now discuss the gauging of $\mathbb{Z}_2^a$ 0-form symmetry generated by $D_2^{(a)}$ in $\mathcal{C}_{D_8/\mathbb{Z}_2^{a2}}$. At the level of 3d orthogonal gauge theories, this corresponds to the transitions $\mathrm{SO}(4N) \to \mathrm{O}'(4N)$ and $\mathrm{SO}(4N+2) \to \mathrm{Spin}(4N+2)$ – see figures 1 and 13.

Alternatively, we could have gauged $\mathbb{Z}_4^a$ 0-form symmetry generated by $D_2^{(a)}$ in $\mathcal{C}_{D_8}$, thus passing to the 2-category $\mathcal{C}_{D_8/\mathbb{Z}_4}$. Thus the above $\mathbb{Z}_2^a$ gauging procedure should implement the transition

$$\mathcal{C}_{D_8/\mathbb{Z}_2^{a2}} \to \mathcal{C}_{D_8/\mathbb{Z}_4}, \tag{140}$$

which we will find to be

$$\boxed{\mathcal{C}_{D_8/\mathbb{Z}_4} = 2\text{-Vec}\left(\mathbb{Z}_4^{(1)} \rtimes \mathbb{Z}_2^{(0)}\right).} \tag{141}$$

This 2-category contains a split 2-group symmetry comprised of a $\mathbb{Z}_2^{(0)}$ 0-form symmetry and a $\mathbb{Z}_4^{(1)}$ 1-form symmetry with an outer-automorphism action of $\mathbb{Z}_2^{(0)}$ on $\mathbb{Z}_4^{(1)}$ and trivial Postnikov class.

This is well-known for the case in which the gauged theory $\mathfrak{T}_{D_8/\mathbb{Z}_4}$ is the $\mathrm{Spin}(4N+2)$ gauge theory. This has a center $\mathbb{Z}_4$ 1-form symmetry acted upon by $\mathbb{Z}_2$ outer-automorphism 0-form symmetry.

**Objects and 1-morphisms.** Upon gauging $\mathbb{Z}_2^a$, the following pairs of simple objects

$$\left\{ D_2^{(\mathrm{id})} \sim D_2^{(a)}, \quad D_2^{(x)} \sim D_2^{(ax)}, \quad D_2^{(\mathbb{Z}_2)} \sim D_2^{(a\mathbb{Z}_2)}, \quad D_2^{(x\mathbb{Z}_2)} \sim D_2^{(ax\mathbb{Z}_2)} \right\}, \tag{142}$$

in $\mathcal{C}_{D_8/\mathbb{Z}_2^{a2}}$ are related by left (or equivalently right) action of $D_2^{(a)}$. Therefore, we expect to get two Schur components of simple objects labelled by $g \in \{\mathrm{id}, x\} \in D_8/\mathbb{Z}_4 \cong \mathbb{Z}_2$.

The rest of the analysis is completely parallel to that of section 3.1.2 and we obtain the following simple objects (upto isomorphism) of $\mathcal{C}_{D_8/\mathbb{Z}_4}$

$$\mathcal{C}_{D_8/\mathbb{Z}_4}^{\mathrm{Ob}} = \left\{ D_2^{(g)}, D_2^{(g\mathbb{Z}_2)}, D_2^{(g\mathbb{Z}_4)} \big| g \in \{1, x\} = D_8/\mathbb{Z}_4 = \mathbb{Z}_2 \right\}. \tag{143}$$

These are (twisted) theta defects with underlying twist $D_2^{(g)} \in \mathcal{C}_{D_8/\mathbb{Z}_2^{a2}}$.

The fusion rules are determined similarly to the fusion rules in section 3.2.2 and are found to be fusion rules for 2-Rep($\mathbb{Z}_4$) graded by the group of Schur components $\mathbb{Z}_2$

$$\begin{aligned}
D_2^{(g)} \otimes D_2^{(h)} &= D_2^{(gh)}, \\
D_2^{(g\mathbb{Z}_2)} \otimes D_2^{(h)} &= D_2^{(gh\mathbb{Z}_2)}, \\
D_2^{(g\mathbb{Z}_4)} \otimes D_2^{(h)} &= D_2^{(gh\mathbb{Z}_4)}, \\
D_2^{(g\mathbb{Z}_2)} \otimes D_2^{(h\mathbb{Z}_2)} &= 2D_2^{(gh\mathbb{Z}_2)}, \\
D_2^{(g\mathbb{Z}_2)} \otimes D_2^{(h\mathbb{Z}_4)} &= 2D_2^{(gh\mathbb{Z}_4)}, \\
D_2^{(g\mathbb{Z}_4)} \otimes D_2^{(h\mathbb{Z}_4)} &= 4D_2^{(gh\mathbb{Z}_4)},
\end{aligned} \tag{144}$$

with $g, h \in \{\mathrm{id}, x\}$.

The story continues in a similar fashion for 1-morphisms as well, where the analysis is just a $\mathbb{Z}_2$ graded version of the analysis of appendix B.2 obtaining 1-morphisms of $\mathcal{C}_{\mathbb{Z}_4/\mathbb{Z}_4}$ by studying $\mathbb{Z}_2$ symmetric 1-morphisms of $\mathcal{C}_{\mathbb{Z}_4/\mathbb{Z}_2}$. In particular, we label the 1-endomorphisms of $D_2^{(\mathrm{id})}$ in $\mathcal{C}_{D_8/\mathbb{Z}_4}$ as

$$\left\{ D_1^{(\mathrm{id})}, \quad D_1^{(\omega)}, \quad D_1^{(\omega^2)}, \quad D_1^{(\omega^3)} \right\}, \tag{145}$$

which form a $\mathbb{Z}_4$ 1-form symmetry group of $\mathfrak{T}_{D_8/\mathbb{Z}_4}$

In summary, at the level of objects and 1-morphisms we seem to obtain that $\mathcal{C}_{D_8/\mathbb{Z}_4}$ equals

$$2\text{-Vec}(\mathbb{Z}_2) \boxtimes 2\text{-Rep}(\mathbb{Z}_4) = 2\text{-Vec}\left( \mathbb{Z}_2^{(0)} \times \mathbb{Z}_4^{(1)} \right). \tag{146}$$

We will now show that in $\mathcal{C}_{D_8/\mathbb{Z}_4}$ there is in fact additionally an action of $\mathbb{Z}_2^{(0)}$ on $\mathbb{Z}_4^{(1)}$.

**2-group action.** Let us now determine the action of the $\mathbb{Z}_2$ 0-form symmetry generated by $D_2^{(x)}$ on the set of bulk lines in $\mathcal{C}_{D_8/\mathbb{Z}_4}$.

First consider $D_1^{(\omega^2)}$, which comes from the making the identity simple 1-endomorphism $D_1^{(\mathrm{id})}$ symmetric under $\mathbb{Z}_2^a$. In particular, we define $D_1^{(\omega^2)}$ such that $a \in \mathbb{Z}_2^a$ acts by $\omega^2 = e^{\pi i}$. Since such an action is its own inverse,

$$D_1^{(x;\mathrm{id})} \otimes D_1^{(\omega^2)} \otimes D_1^{(x;\mathrm{id})} = D_1^{(\omega^2)}, \tag{147}$$

implying no exchange of $D_1^{(\omega^2)}$ under $D_2^{(x)} \in \mathcal{C}_{D_8/\mathbb{Z}_4}$.

Now consider $D_1^{(\omega)}$ and $D_1^{(\omega^3)}$, which descend from making $D_1^{(a^2)}$ symmetric under $\mathbb{Z}_2^a$. Importantly, recall that $\mathbb{Z}_2^a$ fractionalizes on $D_1^{(a^2)}$, such that it acts as a $\mathbb{Z}_4$. Let us define $D_1^{(\omega)}$ such that $a$ acts by $\omega = e^{\pi i/2}$ and $a^3$ acts by $\omega^3 = e^{-\pi i/2}$. Analogously, we define $D_1^{(\omega^3)}$ such that $a$ acts by $\omega^3$ and $a^3$ acts by $\omega$. Since the actions of $a$ and $a^3$ are exchanged by $D_2^{(x)}$, we learn that

$$D_2^{(x)}: \qquad D_1^{(\omega)} \longleftrightarrow D_1^{(\omega^3)}. \tag{148}$$

Due to this action, the $\mathbb{Z}_2^{(0)}$ 0-form symmetry generated by $D_2^{(x)}$ and the $\mathbb{Z}_4^{(1)}$ 1-form symmetry generated by $D_1^{(\omega)}$ of $\mathfrak{T}_{D_8/\mathbb{Z}_4}$ combine to form a split 2-group

$$\Gamma = \mathbb{Z}_4^{(1)} \rtimes \mathbb{Z}_2^{(0)}. \tag{149}$$

Hence we find

$$\mathcal{C}_{D_8/\mathbb{Z}_4} \cong 2\text{Vec}_\Gamma. \tag{150}$$

## 5.4  $\mathbf{PSO}(4N) \to \mathbf{O}'(4N)$ and $\mathbf{PSO}(4N+2) \to \mathbf{Spin}(4N+2)$

In this subsection, we perform the direct gauging of $\mathbb{Z}_4$ starting from $\mathcal{C}_{D_8}$ thus implementing the transition

$$\mathcal{C}_{D_8} \to \mathcal{C}_{D_8/\mathbb{Z}_4}. \tag{151}$$

At the level of 3d orthogonal gauge theories, this corresponds to the transitions $\mathrm{PSO}(4N) \to \mathrm{O}'(4N)$ and $\mathrm{PSO}(4N+2) \to \mathrm{Spin}(4N+2)$ – see figures 1 and 13.

**Objects.** Firstly, the Schur components of objects in $\mathcal{C}_{D_8/\mathbb{Z}_4}$ are labelled by $g \in \{\text{id}, x\} = D_8/\mathbb{Z}_4 = \mathbb{Z}_2$. To determine the objects in the gauged category we consider $nD_2^{(g)}$.

For $n = 1$, we obtain $D_2^{(g)} \in \mathcal{C}_{D_8/\mathbb{Z}_4}$ which descend from $D_2^{(g)} \in \mathcal{C}_{D_8}$, with the $\mathbb{Z}_4$ symmetry implemented via the identity line $D_1^{(g;\text{id})}$. Next, we obtain two dimensional simple objects which descend from $2D_2^{(g)} \in \mathcal{C}_{D_8}$, such that the $\mathbb{Z}_4$ symmetry is implemented by the permutation matrix

$$(12): \quad D_1^{(g;\text{id})} P_{(12)} = \begin{pmatrix} 0 & D_1^{(g;\text{id})} \\ D_1^{(g;\text{id})} & 0 \end{pmatrix} \in \text{Mat}_2(\text{End}_{\mathcal{C}_{D_8}}(D_2^{(g)})). \tag{152}$$

We will use the notation $P_\pi$ for the matrix that implements the permutation $\pi$.

Physically this object corresponds to a TQFT with the $\mathbb{Z}_4$ symmetry spontaneously broken down to its $\mathbb{Z}_2$ subgroup, such that each vacua is invariant under $a^2 \in \mathbb{Z}_4$, while the two vacua are exchanged by the action of $D_2^{(a)}$ and $D_2^{(a^3)}$. We denote this object by $D_2^{(g\mathbb{Z}_2)} \in \mathcal{C}_{D_8/\mathbb{Z}_4}$.

For $n = 3$ there are no new objects, since there are no length 3 orbits of $\mathbb{Z}_4$. For $n = 4$, we get objects for which the action of $a$ can be represented by morphisms

$$D_2^{(\mathbb{Z}_4)}: \qquad D_1^{(g;\text{id})} P_{(1234)}, \qquad D_2^{(\mathbb{Z}_4')}: \qquad D_1^{(g;\text{id})} P_{(1432)}. \tag{153}$$

However these two objects are isomorphic as there exist invertible lines in $4D_2^{(\text{id})} \in \mathcal{C}_{D_8}$ which intertwine between the above two options. We will therefore only add $D_2^{(g\mathbb{Z}_4)}$ to the list of simple objects. Physically, this corresponds to a $\mathbb{Z}_4$ symmetric TQFT in which the $\mathbb{Z}_4$ symmetry is spontaneously broken. Therefore it has four vacua that are permuted cyclically by the $\mathbb{Z}_4$ generator. We denote this object as $D_2^{(\mathbb{Z}_4)} \in \mathcal{C}_{D_8/\mathbb{Z}_4}$. There are no additional indecomposable objects. To summarize the set of simple objects is

$$\text{Obj}(\mathcal{C}_{D_8/\mathbb{Z}_4}) = \left\{ D_2^{(\text{id})}, \quad D_2^{(\mathbb{Z}_2)}, \quad D_2^{(\mathbb{Z}_4)}, \quad D_2^{(x)}, \quad D_2^{(x\mathbb{Z}_2)}, \quad D_2^{(x\mathbb{Z}_4)} \right\}. \tag{154}$$

**Fusion rules.** By lifting the objects in $\mathcal{C}_{D_8/\mathbb{Z}_4}$ to the parent symmetry category $\mathcal{C}_{D_8}$, one finds that the object $D_2^{(\text{id})} \in \mathcal{C}_{D_8/\mathbb{Z}_4}$ is the identity under fusion, while $D_2^{(x)} \in \mathcal{C}_{D_8/\mathbb{Z}_4}$ generates a $\mathbb{Z}_2$ 0-form symmetry. Therefore we have

$$\begin{aligned} D_2^{(\text{id})} \otimes D_2^{(x)} &= D_2^{(x)}, \\ D_2^{(x)} \otimes D_2^{(x)} &= D_2^{(\text{id})}. \end{aligned} \tag{155}$$

Similarly, the fusion product for $D_2^{(g\mathbb{Z}_2)} \otimes D_2^{(g\mathbb{Z}_2)}$ can be obtained as in the cases studied previously by taking the tensor product of 1-endomorphisms in (152) and splitting them into orbits under the $\mathbb{Z}_4$ action. Doing so we find

$$D_2^{(g\mathbb{Z}_2)} \otimes D_2^{(h\mathbb{Z}_2)} = 2D_2^{(gh\mathbb{Z}_2)}, \tag{156}$$

where $g, h \in D_8/\mathbb{Z}_4 = \mathbb{Z}_2$. Next, to compute $D_2^{(g\mathbb{Z}_2)} \otimes D_2^{(h\mathbb{Z}_4)}$, we recall that $D_2^{(g\mathbb{Z}_2)}$ and $D_2^{(h\mathbb{Z}_4)}$ descend from $2D_2^{(\text{id})}$ and $4D_2^{(\text{id})}$ respectively in $\mathcal{C}_{D_8}$. Therefore in $\mathcal{C}_{D_8}$, we may denote objects in the fusion outcome by the tuples $(i, j)$ where $i \in \{0, 1\}$ and $j \in \{0, 1, 2, 3\}$. The action of $a \in \mathbb{Z}_4$ is given by the cyclic permutation

$$(i, j) \longrightarrow (i + 1 \bmod 2, j + 1 \bmod 4), \tag{157}$$

therefore we again obtain two orbits $[(0,0)]$ and $[(1,0)]$ implying the fusion rule

$$D_2^{(g\mathbb{Z}_2)} \otimes D_2^{(h\mathbb{Z}_4)} = 2D_2^{(gh\mathbb{Z}_4)}. \tag{158}$$

Similarly, to compute the fusion rule for $D_2^{(g\mathbb{Z}_4)} \otimes D_2^{(h\mathbb{Z}_4)}$ we lift to $\mathcal{C}_{D_8}$, where we may denote objects in the fusion outcome by the tuples $(i,j)$ where $i,j \in \{0,1,2,3\}$. Under the action of $a \in \mathbb{Z}_4$,

$$(i,j) \longrightarrow (i+1 \bmod 4, j+1 \bmod 4), \tag{159}$$

therefore we obtain four orbits $[(0,0)]$, $[(1,0)]$, $[(2,0)]$ and $[(3,0)]$ implying the fusion rule

$$D_2^{(g\mathbb{Z}_4)} \otimes D_2^{(h\mathbb{Z}_4)} = 4D_2^{(gh\mathbb{Z}_4)}. \tag{160}$$

**1-morphisms.** The 1-endomorphisms of $D_2^{(g)}$ in $\mathcal{C}_{D_8/\mathbb{Z}_4}$ are related to the different ways in which the 1-endomorphisms of $D_2^{(g)} \in \mathcal{C}_{D_8}$ can be made $\mathbb{Z}_4$-symmetric. The 1-category of 1-endomorphisms of $D_2^{(g)} \in \mathcal{C}_{D_8}$ is Vec generated by the identity line $D_1^{(g;\mathrm{id})}$. Since the permutation action of $\mathbb{Z}_4$ on this line is trivial, the ways in which it can be made $\mathbb{Z}_4$ symmetric corresponds to assigning a representation of $\mathbb{Z}_4$. Thus we realize that in $\mathcal{C}_{D_8/\mathbb{Z}_4}$

$$\mathrm{End}_{\mathcal{C}_{D_8/\mathbb{Z}_4}}\left(D_2^{(g)}\right) = \left\{D_1^{(g;\mathrm{id})}, \quad D_1^{(g;a)}, \quad D_1^{(g;a^2)}, \quad D_1^{(g;a^3)}\right\} \cong \mathrm{Rep}(\mathbb{Z}_4). \tag{161}$$

Similarly, the 1-endomorphisms of $D_2^{(g\mathbb{Z}_2)} \in \mathcal{C}_{D_8/\mathbb{Z}_4}$ descend from $\mathbb{Z}_4$ symmetric 1-endomorphisms of $2D_2^{(g)}$ in $\mathcal{C}_{D_8}$.

The set of 1-endomorphsims of $2D_2^{(\mathrm{id})}$ is $\mathrm{Mat}_2(\mathrm{Vec})$. The two rows and two columns of these $2 \times 2$ matrices are exchanged under the action of $a \in \mathbb{Z}_4$. Notably the $\mathbb{Z}_2$ subgroup generated by $a^2$ acts trivially, therefore, the 1-endomorphisms can carry representations of $\mathbb{Z}_2 \subset \mathbb{Z}_4$. Therefore we obtain the following 1-endomorphisms

$$\mathrm{End}_{\mathcal{C}_{D_8/\mathbb{Z}_4}}(D_2^{(g\mathbb{Z}_2)}) = \left\{D_1^{(g\mathbb{Z}_2;\mathrm{id})}, \quad D_1^{(g\mathbb{Z}_2;-)}, \quad D_1^{(g\mathbb{Z}_2;a^2)}, \quad D_1^{(g\mathbb{Z}_2;a^2-)}\right\} \cong \mathrm{Vec}_{\mathbb{Z}_2} \boxtimes \mathrm{Rep}(\mathbb{Z}_2), \tag{162}$$

where each of the 1-endomorphisms descend from the following matrices in $\mathrm{End}_{\mathcal{C}_{D_8}}(2D_2^{(g)})$

$$D_1^{(g\mathbb{Z}_2;\mathrm{id})}\Big|_{\mathcal{C}_{D_8}} = D_1^{(g;\mathrm{id})}P_{(1)(2)}, \quad D_1^{(g\mathbb{Z}_2;-)}\Big|_{\mathcal{C}_{D_8}} = D_1^{(g;\mathrm{id})}P_{(12)},$$
$$D_1^{(g\mathbb{Z}_2;a^2)}\Big|_{\mathcal{C}_{D_8}} = D_1^{(g;a^2)}P_{(1)(2)}, \quad D_1^{(g\mathbb{Z}_2;a^2-)}\Big|_{\mathcal{C}_{D_8}} = D_1^{(g;a^2)}P_{(12)}. \tag{163}$$

The 1-endomorphisms of $D_2^{(\mathbb{Z}_4)} \in \mathcal{C}_{D_8/\mathbb{Z}_4}$ descend from $\mathbb{Z}_4$ symmetric 1-endomorphisms of $4D_2^{(\mathrm{id})}$ in $\mathcal{C}_{D_8}$. The set of 1-endomorphisms of $4D_2^{(\mathrm{id})} \in \mathcal{C}_{D_8}$ is $\mathrm{Mat}_4(\mathrm{Vec})$. The four rows and four columns of these $4 \times 4$ matrices are cyclically permuted the action of $a \in \mathbb{Z}_4$, therefore we obtain the following set of simple 1-morphisms

$$\mathrm{End}_{\mathcal{C}_{D_8/\mathbb{Z}_4}}(D_2^{(g\mathbb{Z}_4)}) = \left\{D_1^{(g\mathbb{Z}_4;\mathrm{id})}, \quad D_1^{(g\mathbb{Z}_4;a)}, \quad D_1^{(g\mathbb{Z}_4;a^2)}, \quad D_1^{(g\mathbb{Z}_4;a^3)}\right\} \cong \mathrm{Vec}_{\mathbb{Z}_4}. \tag{164}$$

The labels $\mathrm{id}, a, a^2, a^3$ correspond to the matrices

$$D_1^{(\mathbb{Z}_4;\mathrm{id})}\Big|_{\mathcal{C}_{D_8}} = D_1^{(g;\mathrm{id})}P_{(1)(2)(3)(4)}, \qquad D_1^{(\mathbb{Z}_4;a)}\Big|_{\mathcal{C}_{D_8}} = D_1^{(g;\mathrm{id})}P_{(1234)},$$
$$D_1^{(\mathbb{Z}_4;a^2)}\Big|_{\mathcal{C}_{D_8}} = D_1^{(g;\mathrm{id})}P_{(13)(24)}, \qquad D_1^{(\mathbb{Z}_4;a^3)}\Big|_{\mathcal{C}_{D_8}} = D_1^{(g;\mathrm{id})}P_{(1432)}. \tag{165}$$

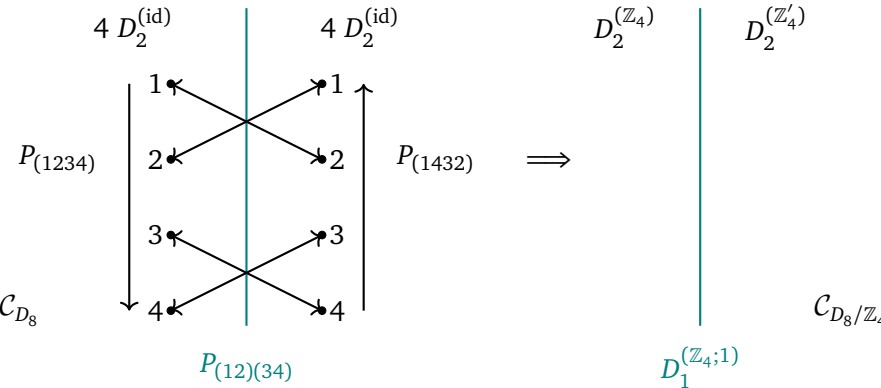

Figure 15: Action of the permutation (12)(34) on the vacua of the theory (left) and the associated defect in the gauged theory.

Finally let us discuss more carefully the isomorphism lines between $D_2^{(g\mathbb{Z}_4)}$ and $D_2^{(g\mathbb{Z}_4')}$ defined using the two choices of $\mathbb{Z}_4$ action implemented by $D_1^{(g\mathbb{Z}_4;a)}$ and $D_1^{(g\mathbb{Z}_4;a^3)}$ respectively (see (153) and (165)). These will play an important role in a subsequent gauging from $\mathcal{C}_{D_8/\mathbb{Z}_4}$ to $\mathcal{C}_{D_8/D_8}$. A choice of isomorphism is given by the line

$$D_1^{(g\mathbb{Z}_4;1)} = D_1^{(g;\text{id})} P_{(12)(34)}, \tag{166}$$

as it intertwines the two above mentioned $\mathbb{Z}_4$ actions, i.e.,

$$D_1^{(g\mathbb{Z}_4;1)} \circ D_1^{(g\mathbb{Z}_4;a)} = D_1^{(g\mathbb{Z}_4;a^3)} \circ D_1^{(g\mathbb{Z}_4;1)}, \tag{167}$$

where $\circ$ denotes composition of lines, explicitly computed using matrix product. In particular if we label the four copies of $D_2^{(\text{id})}$ in $D_2^{(g\mathbb{Z}_4)}$ lifted to $\mathcal{C}_{D_8}$ as $i \in \{1,2,3,4\}$, then $D_1^{(g\mathbb{Z}_4;1)}$ maps copies $\{1,2,3,4\}$ in $D_2^{(g\mathbb{Z}_4)}\big|_{\mathcal{C}_{D_8}}$ to copies $\{2,1,4,3\}$ in $D_2^{(g\mathbb{Z}_4')}\big|_{\mathcal{C}_{D_8}}$, see figure 15. Other choices of isomorphisms can be implemented using

$$\begin{aligned}
D_1^{(g\mathbb{Z}_4;2)} &= D_1^{(g\mathbb{Z}_4;1)} \circ D_1^{(g\mathbb{Z}_4;a)}, \\
D_1^{(g\mathbb{Z}_4;3)} &= D_1^{(g\mathbb{Z}_4;1)} \circ D_1^{(g\mathbb{Z}_4;a^2)}, \\
D_1^{(g\mathbb{Z}_4;4)} &= D_1^{(g\mathbb{Z}_4;1)} \circ D_1^{(g\mathbb{Z}_4;a^3)}.
\end{aligned} \tag{168}$$

**Fusion of 1-morphisms.** The fusion of 1-morphisms is computed as follows. First we take the tensor product by lifting it to $\mathcal{C}_{D_8}$. Subsequently in order to identify the $\mathcal{C}_{D_8/\mathbb{Z}_4}$ morphisms, we decompose the result into orbits of $\mathbb{Z}_4$. For the defect that is realized by a permutation matrix $P_\pi$, we can then identify the action of $\pi$ on the $\mathbb{Z}_4$-orbits.

Consider first $D_1^{(g\mathbb{Z}_4;1)}$ as in (166). Note that $D_1^{(g\mathbb{Z}_4;1)} \otimes D_1^{(g\mathbb{Z}_4;1)}$ is a map from $D_2^{(g\mathbb{Z}_4)} \otimes D_2^{(g\mathbb{Z}_4)}$ to $D_2^{(g\mathbb{Z}_4')} \otimes D_2^{(g\mathbb{Z}_4')}$, see figure 16. In the pre-gauged category $D_1^{(g\mathbb{Z}_4;1)} \otimes D_1^{(g\mathbb{Z}_4;1)}$ is an automorphism of 16 copies of $D_2^{(\text{id})}$ labelled by $(i,j)$ with $i,j \in \{1,2,3,4\}$.

We first decompose the 16 copies of $D_2^{(\text{id})}$ in orbits of the $\mathbb{Z}_4$ action, so to identify the corresponding condensation surfaces in $\mathcal{C}_{D_8/\mathbb{Z}_4}$. We then identify the permutation action $P_{(12)(34)}$ among the copies $(i,j)$, which allows us to decompose $D_1^{(g\mathbb{Z}_4;1)} \otimes D_1^{(g\mathbb{Z}_4;1)}$ into 1-morphisms

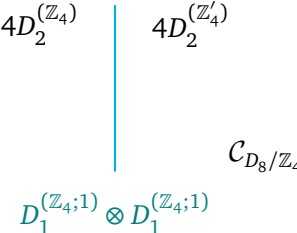

$$4D_2^{(\mathbb{Z}_4)} \qquad\Big|\qquad 4D_2^{(\mathbb{Z}_4')}$$

$$\mathcal{C}_{D_8/\mathbb{Z}_4}$$

$$D_1^{(\mathbb{Z}_4;1)} \otimes D_1^{(\mathbb{Z}_4;1)}$$

Figure 16: The fusion of two line defects $D_1^{(\mathbb{Z}_4;1)}$ is a morphism between four copies of the condensation defect $D_2^{(\mathbb{Z}_4)}$.

$D_{1\,i,j}^{(g\mathbb{Z}_4;1)}$ between the $i$-th copy of $D_2^{(\mathbb{Z}_4)}$ in $4D_2^{(\mathbb{Z}_4)}$ to the $j$-th copy of $D_2^{(\mathbb{Z}_4')}$ in $4D_2^{(\mathbb{Z}_4')}$. Doing so, $D_1^{(g\mathbb{Z}_4;1)} \otimes D_1^{(g\mathbb{Z}_4;1)}$ can be decomposed into two 4d and one 8d orbit:

$$D_1^{(g\mathbb{Z}_4;1)} \otimes D_1^{(g\mathbb{Z}_4;1)}\Big|_{\mathcal{C}_{D_8}}^{\text{Orb-1}} = \left\{ (1,1) \leftrightarrow (2,2), (3,3) \leftrightarrow (4,4) \right\},$$

$$D_1^{(g\mathbb{Z}_4;1)} \otimes D_1^{(g\mathbb{Z}_4;1)}\Big|_{\mathcal{C}_v}^{\text{Orb-2}} = \left\{ (1,3) \leftrightarrow (2,4), (3,1) \leftrightarrow (4,2) \right\},$$

$$D_1^{(g\mathbb{Z}_4;1)} \otimes D_1^{(g\mathbb{Z}_4;1)}\Big|_{\mathcal{C}_{D_8}}^{\text{Orb-3}} = \left\{ (1,2) \leftrightarrow (2,1), (2,3) \leftrightarrow (1,4), (3,4) \leftrightarrow (4,3), (4,1) \leftrightarrow (3,2) \right\}.$$

$$(169)$$

Here the double arrows identify the permutation action $P_{(12)(34)}$.

From this, the fusion rules are easily read off:

$$D_1^{(g\mathbb{Z}_4;1)} \otimes D_1^{(g\mathbb{Z}_4;1)} = D_1^{(g\mathbb{Z}_4;1)} \oplus D_1^{(g\mathbb{Z}_4;1)} \oplus \begin{pmatrix} 0 & D_1^{(g\mathbb{Z}_4;1)} \\ D_1^{(g\mathbb{Z}_4;1)} & 0 \end{pmatrix}. \qquad (170)$$

Similarly, we may compute the fusion product of $D_1^{(g\mathbb{Z}_4;2)}$ with itself: $D_1^{(g\mathbb{Z}_4;2)}$ lifts to $\mathrm{End}_{\mathcal{C}_{D_8}}(4D_2^{(\mathrm{id})})$ as the matrix

$$D_1^{(g\mathbb{Z}_4;2)}\Big|_{\mathcal{C}_{D_8}} = D_1^{(g;\mathrm{id})} P_{(13)(2)(4)}. \qquad (171)$$

Then computing the tensor product of (171) with itself and decomposing into orbits under the $\mathbb{Z}_4$ action, we again find two 4d orbits and one 8d one:

$$D_1^{(g\mathbb{Z}_4;2)} \otimes D_1^{(g\mathbb{Z}_4;2)}\Big|_{\mathcal{C}_{D_8}}^{\text{Orb-1}} = \left\{ (1,1) \leftrightarrow (3,3), (2,2), (4,4) \right\},$$

$$D_1^{(g\mathbb{Z}_4;1)} \otimes D_1^{(g\mathbb{Z}_4;1)}\Big|_{\mathcal{C}_{D_8}}^{\text{Orb-2}} = \left\{ (1,3) \leftrightarrow (3,1), (2,4) \leftrightarrow (4,2) \right\},$$

$$D_1^{(g\mathbb{Z}_4;2)} \otimes D_1^{(g\mathbb{Z}_4;2)}\Big|_{\mathcal{C}_{D_8}}^{\text{Orb-3}} = \left\{ (1,2) \leftrightarrow (3,2), (2,3) \leftrightarrow (2,1), (3,4) \leftrightarrow (1,4), (4,1) \leftrightarrow (4,3) \right\}.$$

$$(172)$$

Therefore we obtain a similar fusion rule as for $D_1^{(g\mathbb{Z}_4;2)}$, i.e.,

$$D_1^{(g\mathbb{Z}_4;2)} \otimes D_1^{(g\mathbb{Z}_4;2)} = D_1^{(g\mathbb{Z}_4;2)} \oplus D_1^{(g\mathbb{Z}_4;2)} \oplus \begin{pmatrix} 0 & D_1^{(g\mathbb{Z}_4;2)} \\ D_1^{(g\mathbb{Z}_4;2)} & 0 \end{pmatrix}. \qquad (173)$$

## 5.5 $O'(4N) \to Pin^+(4N)$ and $Spin(4N+2) \to Pin^+(4N+2)$

In this subsection, we perform the gauging of $\mathbb{Z}_2^x$ 0-form symmetry in $\mathcal{C}_{D_8/\mathbb{Z}_4}$. This implements the transition from $\mathcal{C}_{D_8/\mathbb{Z}_4}$ to

$$\boxed{\mathcal{C}_{D_8/D_8} = 2\text{-Rep}(D_8).} \tag{174}$$

The structure of 2-Rep($D_8$) can be independently computed using the methods of [14], providing a consistency check on the method used here.

At the level of 3d orthogonal gauge theories, this corresponds to the transitions $O'(4N) \to Pin^+(4N)$ and $Spin(4N+2) \to Pin^+(4N+2)$ – see figures 1 and 13.

**Objects.** First consider making $D_2^{(\text{id})} \in \mathcal{C}_{D_8/\mathbb{Z}_4}$ symmetric under $\mathbb{Z}_2^x$. Choose a junction line $D_1^{(\text{id};x)(\text{id})}$ between $D_2^{(\text{id})}$ and $D_2^{(x)}$ which has the property that upon folding $D_2^{(x)}$ surface away, the junction line $D_1^{(\text{id};x)(\text{id})}$ is converted into the line $D_1^{(\text{id})}$ living on $D_2^{(\text{id})}$. This junction line satisfies

$$D_1^{(\text{id};x)(\text{id})} \otimes_{D_2^{(\text{id})}} D_1^{(\text{id};x)(\text{id})} = D_1^{(\text{id})}, \tag{175}$$

and hence can be used to make $D_2^{(\text{id})} \in \mathcal{C}_{D_8/\mathbb{Z}_4}$ symmetric under $\mathbb{Z}_2^x$, leading to the identity object

$$D_2^{(\text{id})} \in \mathcal{C}_{D_8/D_8}. \tag{176}$$

There are other junction lines between $D_2^{(\text{id})}$ and $D_2^{(x)}$ in $\mathcal{C}_{D_8/\mathbb{Z}_4}$

$$D_1^{(\text{id};x)(y)} := D_1^{(y)} \otimes_{D_2^{(\text{id})}} D_1^{(\text{id};x)(\text{id})}, \tag{177}$$

for $y \in \{\text{id}, \omega, \omega^2, \omega^3\}$ obtained by stacking the bulk line $D_1^{(y)}$ on top of $D_1^{(\text{id};x)(\text{id})}$. Note that we have

$$\begin{aligned}
D_1^{(\text{id};x)(y)} \otimes_{D_2^{(\text{id})}} D_1^{(\text{id};x)(y)} &= D_1^{(y)} \otimes_{D_2^{(\text{id})}} D_1^{(\text{id};x)(\text{id})} \otimes_{D_2^{(\text{id})}} D_1^{(y)} \otimes_{D_2^{(\text{id})}} D_1^{(\text{id};x)(\text{id})} \\
&= D_1^{(y)} \otimes_{D_2^{(\text{id})}} D_1^{(y^{-1})} \otimes_{D_2^{(\text{id})}} D_1^{(\text{id};x)(\text{id})} \otimes_{D_2^{(\text{id})}} D_1^{(\text{id};x)(\text{id})} = D_1^{(\text{id})},
\end{aligned} \tag{178}$$

where $D_1^{(y)}$ is converted into $D_1^{(y^{-1})}$ upon crossing through $D_1^{(\text{id};x)(\text{id})}$ due to the 2-group action. Thus each junction line $D_1^{(\text{id};x)(y)}$ can be used to make $D_2^{(\text{id})}$ symmetric under $\mathbb{Z}_2^x$ leading to objects

$$D_2^{(\text{id},y)} \in \mathcal{C}_{D_8/D_8}. \tag{179}$$

However, we claim that

$$D_2^{(\text{id},\omega^2)} \cong D_2^{(\text{id})}. \tag{180}$$

To see this, note that $D_1^{(z)} \in \mathcal{C}_{D_8/\mathbb{Z}_4}$ for $z \in \{\omega, \omega^3\}$ descend to 1-morphisms

$$\begin{aligned}
D_1^{(\text{id};\omega^2)(z)} &: \ D_2^{(\text{id})} \to D_2^{(\text{id},\omega^2)}, \\
D_1^{(\omega^2;\text{id})(z)} &: \ D_2^{(\text{id},\omega^2)} \to D_2^{(\text{id})},
\end{aligned} \tag{181}$$

as $D_1^{(z)} \in \mathcal{C}_{D_8/\mathbb{Z}_4}$ intertwines the $\mathbb{Z}_2^x$ actions implemented by $D_1^{(\text{id};x)(\text{id})}$ and $D_1^{(\text{id};x)(\omega^2)}$. These 1-morphisms are actually isomorphisms because

$$D_1^{(\text{id};\omega^2)(z)} \circ D_1^{(\omega^2;\text{id})(z^{-1})} = D_1^{(\text{id})} \in \mathcal{C}_{D_8/D_8}. \tag{182}$$

Similarly, we have

$$D_2^{(\text{id},\omega)} \cong D_2^{(\text{id},\omega^3)}. \tag{183}$$

Thus, we obtain from $D_2^{(\text{id})} \in \mathcal{C}_{D_8/\mathbb{Z}_4}$ two simple objects (upto isomorphism)

$$D_2^{(\text{id})}, \qquad D_2^{(\text{id},\omega)}, \tag{184}$$

of $\mathcal{C}_{D_8/D_8}$.

Now let us consider the object $2D_2^{(\text{id})} \in \mathcal{C}_{D_8/\mathbb{Z}_4}$. This can be made symmetric under $\mathbb{Z}_2^x$ using either of the following matrices[17] in $\text{Mat}_2(\text{Rep}(\mathbb{Z}_4))$

$$\begin{pmatrix} 0 & D_1^{(\text{id})} \\ D_1^{(\text{id})} & 0 \end{pmatrix}, \quad \begin{pmatrix} 0 & D_1^{(\omega^2)} \\ D_1^{(\omega^2)} & 0 \end{pmatrix}, \quad \begin{pmatrix} 0 & D_1^{(\omega)} \\ D_1^{(\omega)} & 0 \end{pmatrix}, \quad \begin{pmatrix} 0 & D_1^{(\omega^3)} \\ D_1^{(\omega^3)} & 0 \end{pmatrix}. \tag{185}$$

However, it can be shown that all these choices give rise to isomorphic objects in $\mathcal{C}_{D_8/D_8}$. Then up to isomorphism we obtain a single simple object

$$D_2^{(\mathbb{Z}_2)} \in \mathcal{C}_{D_8/D_8}. \tag{186}$$

Now let us consider the simple object $D_2^{(\mathbb{Z}_2)} \in \mathcal{C}_{D_8/\mathbb{Z}_4}$. To understand the possible ways of making $D_2^{(\mathbb{Z}_2)}$ symmetric under $\mathbb{Z}_2^x$, recall that its 1-endomorphisms

$$\text{End}_{\mathcal{C}_{D_8/\mathbb{Z}_4}}(D_2^{(\mathbb{Z}_2)}) = \left\{ D_1^{(\mathbb{Z}_2;\text{id})}, \quad D_1^{(\mathbb{Z}_2;-)}, \quad D_1^{(\mathbb{Z}_2;a^2)}, \quad D_1^{(\mathbb{Z}_2;a^2-)} \right\}, \tag{187}$$

form the fusion 1-category $\text{Vec}(\mathbb{Z}_2) \boxtimes \text{Rep}(\mathbb{Z}_2)$ which is invariant under $\mathbb{Z}_2^x$ action.

Therefore, we can make $D_2^{(\mathbb{Z}_2)} \in \mathcal{C}_{D_8/\mathbb{Z}_4}$ symmetric under $\mathbb{Z}_2^x$ using any of its 1-endomorphisms. Correspondingly, we get 4 simple objects

$$D_2^{(\mathbb{Z}_2')}, \quad D_2^{(\mathbb{Z}_2',\epsilon)}, \quad D_2^{(\mathbb{Z}_2'')}, \quad D_2^{(\mathbb{Z}_2'',\delta)}, \tag{188}$$

in $\mathcal{C}_{D_8/D_8}$, where $\mathbb{Z}_2^x$ is implemented via $D_1^{(\mathbb{Z}_2;\text{id})}$, $D_1^{(\mathbb{Z}_2;a^2)}$, $D_1^{(\mathbb{Z}_2;-)}$ and $D_1^{(\mathbb{Z}_2;a^2-)}$ respectively.

Next, let us consider the object $2D_2^{(\mathbb{Z}_2)} \in \mathcal{C}_{D_8/\mathbb{Z}_4}$ whose 1-endomorphisms form the category $\text{Mat}_2(\text{Vec}_{\mathbb{Z}_2} \boxtimes \text{Rep}(\mathbb{Z}_2))$. Since all the simple 1-endomorphisms are $\mathbb{Z}_2^x$ invariant order two lines under composition, it follows that $2D_2^{(\mathbb{Z}_2)} \in \mathcal{C}_{\mathfrak{T}/\mathbb{Z}_4}$ can be made $\mathbb{Z}_2^x$ symmetric using any of the following matrices

$$\begin{pmatrix} 0 & D_1^{(\mathbb{Z}_2;i)} \\ D_1^{(\mathbb{Z}_2;i)} & 0 \end{pmatrix}, \quad i = \text{id}, -, a^2, a^2 - . \tag{189}$$

It can be shown that up to isomorphism, all these choices give rise to a single simple object

$$D_2^{(\mathbb{Z}_4)} \in \mathcal{C}_{D_8/D_8}. \tag{190}$$

Finally, let us consider simple objects in $\mathcal{C}_{D_8/D_8}$ descending from $D_2^{(\mathbb{Z}_4)} \in \mathcal{C}_{D_8/\mathbb{Z}_4}$. Recall that we have two isomorphic objects $D_2^{(\mathbb{Z}_4')}$ and $D_2^{(\mathbb{Z}_4)}$ in $\mathcal{C}_{D_8/\mathbb{Z}_4}$, see equation (153), corresponding to two different choices of $\mathbb{Z}_4$ action on $4D_2^{(\text{id})}$ in $\mathcal{C}_{D_8}$. These two isomorphic objects are related by $\mathbb{Z}_2^x$ action, i.e. we have the fusion

$$D_2^{(x)} \otimes D_2^{(\mathbb{Z}_4)} \otimes D_2^{(x)} = D_2^{(\mathbb{Z}_4')}. \tag{191}$$

---

[17]We do not discuss the diagonal implementations of $\mathbb{Z}_2^x$ on $2D_2^{(\text{id})} \in \mathcal{C}_{D_8/\mathbb{Z}_4}$ as these lead to decomposable (not simple) objects in $\mathcal{C}_{D_8/D_8}$.

Table 2: Fusion of simple objects in 2-Rep($D_8$). In the table we have labelled each simple object $D_2^{(a)}$ by the corresponding $(a)$.

| | (id) | (id,$\mu$) | ($\mathbb{Z}_2$) | ($\mathbb{Z}_2'$) | ($\mathbb{Z}_2',\epsilon$) | ($\mathbb{Z}_2''$) | ($\mathbb{Z}_2'',\delta$) | ($\mathbb{Z}_4$) | ($\mathbb{Z}_4'$) | ($\mathbb{Z}_4''$) | ($D_8$) |
|---|---|---|---|---|---|---|---|---|---|---|---|
| (id) | (id) | (id,$\mu$) | ($\mathbb{Z}_2$) | ($\mathbb{Z}_2'$) | ($\mathbb{Z}_2',\epsilon$) | ($\mathbb{Z}_2''$) | ($\mathbb{Z}_2'',\delta$) | ($\mathbb{Z}_4$) | ($\mathbb{Z}_4'$) | ($\mathbb{Z}_4''$) | ($D_8$) |
| (id,$\mu$) | (id,$\mu$) | (id) | ($\mathbb{Z}_2$) | ($\mathbb{Z}_2'$) | ($\mathbb{Z}_2',\epsilon$) | ($\mathbb{Z}_2''$) | ($\mathbb{Z}_2'',\delta$) | ($\mathbb{Z}_4$) | ($\mathbb{Z}_4'$) | ($\mathbb{Z}_4''$) | ($D_8$) |
| ($\mathbb{Z}_2$) | ($\mathbb{Z}_2$) | ($\mathbb{Z}_2$) | $2(\mathbb{Z}_2)$ | ($\mathbb{Z}_4$) | ($\mathbb{Z}_4$) | ($\mathbb{Z}_4$) | ($\mathbb{Z}_4$) | $2(\mathbb{Z}_4)$ | ($D_8$) | ($D_8$) | $2(D_8)$ |
| ($\mathbb{Z}_2'$) | ($\mathbb{Z}_2'$) | ($\mathbb{Z}_2'$) | ($\mathbb{Z}_4$) | $2(\mathbb{Z}_2')$ | $2(\mathbb{Z}_2',\epsilon)$ | ($\mathbb{Z}_4$) | ($\mathbb{Z}_4$) | $2(\mathbb{Z}_4)$ | $2(\mathbb{Z}_4')$ | ($D_8$) | $2(D_8)$ |
| ($\mathbb{Z}_2',\epsilon$) | ($\mathbb{Z}_2',\epsilon$) | ($\mathbb{Z}_2',\epsilon$) | ($\mathbb{Z}_4$) | $2(\mathbb{Z}_2',\epsilon)$ | $2(\mathbb{Z}_2')$ | ($\mathbb{Z}_4$) | ($\mathbb{Z}_4$) | $2(\mathbb{Z}_4)$ | $2(\mathbb{Z}_4')$ | ($D_8$) | $2(D_8)$ |
| ($\mathbb{Z}_2''$) | ($\mathbb{Z}_2''$) | ($\mathbb{Z}_2''$) | ($\mathbb{Z}_4$) | ($\mathbb{Z}_4$) | ($\mathbb{Z}_4$) | $2(\mathbb{Z}_2'')$ | $2(\mathbb{Z}_2'',\delta)$ | $2(\mathbb{Z}_4)$ | ($D_8$) | $2(\mathbb{Z}_4'')$ | $2(D_8)$ |
| ($\mathbb{Z}_2'',\delta$) | ($\mathbb{Z}_2'',\delta$) | ($\mathbb{Z}_2'',\delta$) | ($\mathbb{Z}_4$) | ($\mathbb{Z}_4$) | ($\mathbb{Z}_4$) | $2(\mathbb{Z}_2'',\delta)$ | $2(\mathbb{Z}_2'')$ | $2(\mathbb{Z}_4)$ | ($D_8$) | $2(\mathbb{Z}_4'')$ | $2(D_8)$ |
| ($\mathbb{Z}_4$) | ($\mathbb{Z}_4$) | ($\mathbb{Z}_4$) | $2(\mathbb{Z}_4)$ | $2(\mathbb{Z}_4)$ | $2(\mathbb{Z}_4)$ | $2(\mathbb{Z}_4)$ | $2(\mathbb{Z}_4)$ | $4(\mathbb{Z}_4)$ | $2(D_8)$ | $2(D_8)$ | $4(D_8)$ |
| ($\mathbb{Z}_4'$) | ($\mathbb{Z}_4'$) | ($\mathbb{Z}_4'$) | ($D_8$) | $2(\mathbb{Z}_4')$ | $2(\mathbb{Z}_4')$ | ($D_8$) | ($D_8$) | $2(D_8)$ | $2(\mathbb{Z}_4')\oplus(D_8)$ | $2(D_8)$ | $4(D_8)$ |
| ($\mathbb{Z}_4''$) | ($\mathbb{Z}_4''$) | ($\mathbb{Z}_4''$) | ($D_8$) | ($D_8$) | ($D_8$) | $2(\mathbb{Z}_4'')$ | $2(\mathbb{Z}_4'')$ | $2(D_8)$ | $2(D_8)$ | $2(\mathbb{Z}_4'')\oplus(D_8)$ | $4(D_8)$ |
| ($D_8$) | ($D_8$) | ($D_8$) | $2(D_8)$ | $2(D_8)$ | $2(D_8)$ | $2(D_8)$ | $2(D_8)$ | $4(D_8)$ | $4(D_8)$ | $4(D_8)$ | $8(D_8)$ |

Because of this, making $D_2^{(\mathbb{Z}_4)}$ symmetric under $\mathbb{Z}_2^x$ requires prescribing a 1-morphism $M_{up} : D_2^{(\mathbb{Z}_4')} \to D_2^{(\mathbb{Z}_4)}$ and $M_p : D_2^{(\mathbb{Z}_4)} \to D_2^{(\mathbb{Z}_4')}$ such that their compositions are $M_{up} \circ M_p = D_1^{(\mathbb{Z}_4';\mathrm{id})}$ and $M_p \circ M_{up} = D_1^{(\mathbb{Z}_4;\mathrm{id})}$. We can choose as such 1-morphisms any of the lines

$$D_1^{(\mathbb{Z}_4;i)}, \quad i = 1, 2, 3, 4, \tag{192}$$

discussed in (166) and (168). Among these four lines, $D_1^{(\mathbb{Z}_4;1)}$ and $D_1^{(\mathbb{Z}_4;3)}$ lead to isomorphic objects in $\mathcal{C}_{D_8/D_8}$, and similarly for $D_1^{(\mathbb{Z}_4;2)}$ and $D_1^{(\mathbb{Z}_4;4)}$, so that we get two isomorphism classes of simple objects in $\mathcal{C}_{D_8/D_8}$. Let us denote the first simple object by $D_2^{(\mathbb{Z}_4')}$, and pick $D_1^{(\mathbb{Z}_4;1)}$ as the corresponding 1-morphism for concretness. The second object is denoted by $D_2^{(\mathbb{Z}_4'')}$ with 1-morphism given by $D_1^{(\mathbb{Z}_4;2)}$.

There is one more simple object up to isomorphisms in $\mathcal{C}_{D_8/D_8}$ that comes from $2D_2^{(\mathbb{Z}_4)} \in \mathcal{C}_{D_8/\mathbb{Z}_4}$. This can be made $\mathbb{Z}_2^x$ symmetric using either of the following options

$$\begin{pmatrix} 0 & D_1^{(\mathbb{Z}_4;1)} \\ D_1^{(\mathbb{Z}_4;1)} & 0 \end{pmatrix}, \qquad \begin{pmatrix} 0 & D_1^{(\mathbb{Z}_4;2)} \\ D_1^{(\mathbb{Z}_4;2)} & 0 \end{pmatrix}, \tag{193}$$

which up to isomorphisms, give a single simple object in $\mathcal{C}_{D_8/D_8}$, denoted by $D_2^{(D_8)}$.

To summarize, we have obtained

$$\mathrm{Obj}(\mathcal{C}_{\mathfrak{T}/D_8}) = \left\{ D_2^{(\mathrm{id})},\ D_2^{(\mathrm{id}),\mu},\ D_2^{(\mathbb{Z}_2)},\ D_2^{(\mathbb{Z}_2')},\ D_2^{(\mathbb{Z}_2'),\epsilon},\ D_2^{(\mathbb{Z}_2'')},\ D_2^{(\mathbb{Z}_2''),\delta},\ D_2^{(\mathbb{Z}_4)},\ D_2^{(\mathbb{Z}_4')},\ D_2^{(\mathbb{Z}_4'')},\ D_2^{(D_8)} \right\}. \tag{194}$$

This reproduces exactly the set of simple objects of 2-Rep($D_8$).

**Fusion rules.** Since there are several simple objects in $\mathcal{C}_{D_8/D_8}$, we refrain from presenting the computations of all of the simple objects. Instead we only detail those fusions that exemplify some subtle computational features. The fusion rules for all objects are collected in table 2.

Let us begin by describing the fusion rules between the two dimensional simple objects $D_2^{(\mathbb{Z}_2')}$, $D_2^{(\mathbb{Z}_2'),\epsilon}$, $D_2^{(\mathbb{Z}_2'')}$, $D_2^{(\mathbb{Z}_2''),\delta}$ which descend from $D_2^{(\mathbb{Z}_2)}$ in $\mathcal{C}_{D_8/\mathbb{Z}_4}$. Recall that the 1-endomorphisms of $D_2^{(\mathbb{Z}_2)}$ in $\mathcal{C}_{D_8/\mathbb{Z}_4}$ which implement the corresponding $\mathbb{Z}_2^x$ symmetry in these objects are $D_1^{(\mathbb{Z}_2;\mathrm{id})}$, $D_1^{(\mathbb{Z}_2;a^2)}$, $D_1^{(\mathbb{Z}_2;-)}$ and $D_1^{(\mathbb{Z}_2;a^2-)}$ respectively. Therefore, to compute the fusion of two such objects, we first need to compute the fusion of these 1-endomorphisms. For instance, consider the following self-fusions

$$\begin{aligned} D_1^{(\mathbb{Z}_2;\mathrm{id})} \otimes D_1^{(\mathbb{Z}_2;\mathrm{id})} &= 2D_1^{(\mathbb{Z}_2;\mathrm{id})}, \\ D_1^{(\mathbb{Z}_2;a^2)} \otimes D_1^{(\mathbb{Z}_2;a^2)} &= 2D_1^{(\mathbb{Z}_2;\mathrm{id})}, \end{aligned} \tag{195}$$

which imply the fusion rules

$$
\begin{aligned}
D_2^{(\mathbb{Z}_2')} \otimes D_2^{(\mathbb{Z}_2')} &= 2D_2^{(\mathbb{Z}_2')}, \\
D_2^{(\mathbb{Z}_2';\epsilon)} \otimes D_2^{(\mathbb{Z}_2';\epsilon)} &= 2D_2^{(\mathbb{Z}_2')}.
\end{aligned}
\tag{196}
$$

Notice that while $D_2^{(\mathbb{Z}_2')}$ has a self-fusion reminiscent of the condensation defect in 2-Rep$(\mathbb{Z}_2)$, the simple object $D_2^{(\mathbb{Z}_2';\epsilon)}$ is different. This has to do with the fact that $D_2^{(\mathbb{Z}_2')}$ and $D_2^{(\mathbb{Z}_2';\epsilon)}$, as $D_8$ equivariant 2d TQFTs correspond to the case where $D_8$ is spontaneously broken down to the $\mathbb{Z}_2 \times \mathbb{Z}_2$ subgroup generated by $\{a^2, x\} \in D_8$. Each vacua in either of the two simple objects then realizes an SPT labelled an element in $H^2 = (\mathbb{Z}_2 \times \mathbb{Z}_2, U(1)) \cong \mathbb{Z}_2$. The defect labelled by $\epsilon$ corresponds to the non-trivial SPT and therefore has an additional $\mathbb{Z}_2$ structure in its fusion rule.

Next, let us consider fusions between different two dimensional simple objects. Some of the corresponding fusions between the 1-endomorphisms of $D_2^{(\mathbb{Z}_2)} \in \mathcal{C}_{D_8/\mathbb{Z}_4}$ are

$$
\begin{aligned}
D_1^{(\mathbb{Z}_2;\mathrm{id})} \otimes D_1^{(\mathbb{Z}_2;a^2)} &= 2D_1^{(\mathbb{Z}_2;a^2)}, \\
D_1^{(\mathbb{Z}_2;a^2)} \otimes D_1^{(\mathbb{Z}_2;a^2-)} &= \begin{pmatrix} 0 & D_1^{(\mathbb{Z}_2;\mathrm{id})} \\ D_1^{(\mathbb{Z}_2;\mathrm{id})} & 0 \end{pmatrix}.
\end{aligned}
\tag{197}
$$

Thus, we obtain the following fusion rules for simple objects in $\mathcal{C}_{D_8/D_8}$

$$
\begin{aligned}
D_2^{(\mathbb{Z}_2')} \otimes D_2^{(\mathbb{Z}_2';\epsilon)} &= 2D_2^{(\mathbb{Z}_2';\epsilon)}, \\
D_2^{(\mathbb{Z}_2';\epsilon)} \otimes D_2^{(\mathbb{Z}_2'';\delta)} &= D_2^{(\mathbb{Z}_4)}.
\end{aligned}
\tag{198}
$$

Here we may interpret the simple objects labelled by $\mathbb{Z}_2'$ and $\mathbb{Z}_2''$ as $D_8$ equivariant 2d TQFTs that preserve the $\mathbb{Z}_2 \times \mathbb{Z}_2$ subgroups generated by $\{a^2, x\}$ and $\{xa, a^2\}$ respectively. Therefore the fusion of objects labelled by $\mathbb{Z}_2'$ produces another object labelled by $\mathbb{Z}_2'$, i.e., one that breaks $D_8$ down to the same subgroup $\langle a^2, x \rangle \cong \mathbb{Z}_2 \times \mathbb{Z}_2$. Instead when we fuse objects labelled by $\mathbb{Z}_2'$ with those labelled $\mathbb{Z}_2''$, the fusion product is the $D_2^{(\mathbb{Z}_4)}$ defect which only preserves the intersection $\{1, a^2\} \cong \mathbb{Z}_2$.

Next, let us consider the fusion of the two dimensional simple objects with the four dimensional objects $D_2^{(\mathbb{Z}_4)}$, $D_2^{(\mathbb{Z}_4')}$ and $D_2^{(\mathbb{Z}_4)}$. For instance,

$$
D_2^{(\mathbb{Z}_2',\epsilon)} \otimes D_2^{(\mathbb{Z}_4)} = 2D_2^{(\mathbb{Z}_4)},
\tag{199}
$$

which follows from the fusion of 1-morphisms implementing the $\mathbb{Z}_2^x$ symmetry

$$
\begin{aligned}
D_1^{(\mathbb{Z}_2;a^2)} \otimes \begin{pmatrix} 0 & D_1^{(\mathbb{Z}_2;\mathrm{id})} \\ D_1^{(\mathbb{Z}_2;\mathrm{id})} & 0 \end{pmatrix} &= \begin{pmatrix} 0 & D_1^{(\mathbb{Z}_2;a^2)} \\ D_1^{(\mathbb{Z}_2;a^2)} & 0 \end{pmatrix} \oplus \begin{pmatrix} 0 & D_1^{(\mathbb{Z}_2;a^2)} \\ D_1^{(\mathbb{Z}_2;a^2)} & 0 \end{pmatrix} \\
&\cong \begin{pmatrix} 0 & D_1^{(\mathbb{Z}_2;\mathrm{id})} \\ D_1^{(\mathbb{Z}_2;\mathrm{id})} & 0 \end{pmatrix} \oplus \begin{pmatrix} 0 & D_1^{(\mathbb{Z}_2;\mathrm{id})} \\ D_1^{(\mathbb{Z}_2;\mathrm{id})} & 0 \end{pmatrix},
\end{aligned}
\tag{200}
$$

where in the last line we have used an isomorphism between the $\mathbb{Z}_2^x$ actions. Next, let us compute the fusion between $D_2^{(\mathbb{Z}_2',\epsilon)}$ and $D_2^{(\mathbb{Z}_4')}$. Recall that the defect $D_2^{(\mathbb{Z}_4')}$ descends from $D_2^{(\mathbb{Z}_4)} \in \mathcal{C}_{D_8/\mathbb{Z}_4}$ on which the $\mathbb{Z}_2$ symmetry is implemented by the line $D_1^{(\mathbb{Z}_4;1)}$ in (166). Therefore we need to compute

$$
D_1^{(\mathbb{Z}_2;a^2)} \otimes D_1^{(\mathbb{Z}_4;1)} = D_1^{(\mathbb{Z}_4;1)} \oplus D_1^{(\mathbb{Z}_4;1)},
\tag{201}
$$

which implies the fusion rule

$$D_2^{(\mathbb{Z}_2';\epsilon)} \otimes D_2^{(\mathbb{Z}_4')} = 2D_2^{(\mathbb{Z}_4')}. \tag{202}$$

Let us instead compute the fusion rule between $D_2^{(\mathbb{Z}_2'')}$ and $D_2^{(\mathbb{Z}_4')}$. Following the same procedure we now find

$$D_1^{(\mathbb{Z}_2;-)} \otimes D_1^{(\mathbb{Z}_4;1)} = \begin{pmatrix} 0 & D_1^{(\mathbb{Z}_4;1)} \\ D_1^{(\mathbb{Z}_4;1)} & 0 \end{pmatrix}, \tag{203}$$

which implies the fusion rule

$$D_2^{(\mathbb{Z}_2'';\epsilon)} \otimes D_2^{(\mathbb{Z}_4')} = D_2^{(D_8)}. \tag{204}$$

Note that the fusion rules (204) and (204) are consistent with the interpretation of the higher-dimensional defects as corresponding to $D_8$ equivariant 2d TQFTs that break the $D_8$ symmetry to different subgroups. In particular the defects labelled by $\mathbb{Z}_2'$, $\mathbb{Z}_2''$ and $\mathbb{Z}_4'$ correspond to TQFTs where $D_8$ is spontaneously broken to $\langle x, a^2 \rangle \cong \mathbb{Z}_2^2$, $\langle ax, a^2 \rangle \cong \mathbb{Z}_2^2$ and $\langle x \rangle \cong \mathbb{Z}_2$ respectively. Therefore the fusion outcome of (202) is an eight dimensional composite defect where each vacua preserves the intersection $\langle x \rangle \cong \mathbb{Z}_2$. In contrast, the fusion outcome in (204) is an eight dimensional composite defect where each vacua preserves no symmetry since $\langle x \rangle \cap \langle ax, a^2 \rangle$ is trivial.

Next, we compute the fusion rules among the four dimensional defects. Consider the fusion rule $D_2^{(\mathbb{Z}_4')} \otimes D_2^{(\mathbb{Z}_4')}$. To understand the fusion of surfaces in $\mathcal{C}_{D_8/D_8}$, we need to compute the fusion of the 1-morphism lines $D_1^{(\mathbb{Z}_4;1)}$. In (170) we computed this to be

$$D_1^{(\mathbb{Z}_4;1)} \otimes D_1^{(\mathbb{Z}_4;1)} = D_1^{(\mathbb{Z}_4;1)} \oplus D_1^{(\mathbb{Z}_4;1)} \oplus \begin{pmatrix} 0 & D_1^{(\mathbb{Z}_4;1)} \\ D_1^{(\mathbb{Z}_4;1)} & 0 \end{pmatrix}. \tag{205}$$

Therefore, we read-off the fusion rule

$$D_2^{(\mathbb{Z}_4')} \otimes D_2^{(\mathbb{Z}_4')} = 2D_2^{(\mathbb{Z}_4')} \oplus D_2^{(D_8)}. \tag{206}$$

Similarly, from (173) we can read off the fusion rule for $D_2^{(\mathbb{Z}_4'')}$

$$D_2^{(\mathbb{Z}_4'')} \otimes D_2^{(\mathbb{Z}_4'')} = 2D_2^{(\mathbb{Z}_4'')} \oplus D_2^{(D_8)}. \tag{207}$$

## 5.6 $\mathbf{PSO(4N) \to PO/Ss(4N)}$ and $\mathbf{PSO(4N+2) \to PO/PO'(4N+2)}$

So far we considered the gaugings originating from $\mathcal{C}_{D_8} = 2\text{-Vec}(D_8)$ by first gauging the normal subgroups $\mathbb{Z}_2^{a^2}$ or $\mathbb{Z}_4^a$ and then subsequent subgroups. We can instead consider beginning with gauging of non-normal subgroups. For $D_8$ these are all $\mathbb{Z}_2$ and are generated in turn by

$$x, \ xa, \ xa^2, \ xa^3. \tag{208}$$

As these are related by automorphisms of $D_8$, we can consider gauging a particular one namely $\mathbb{Z}_2^x$ and the results for gauging any other non-normal $\mathbb{Z}_2$ will be same.

However, as $x$ and $ax$ are not related by inner-automorphisms, $\mathfrak{T}_{D_8/\mathbb{Z}_2^x}$ and $\mathfrak{T}_{D_8/\mathbb{Z}_2^{ax}}$ can be distinct QFTs with associated 2-categories being same

$$\mathcal{C}_{D_8/\mathbb{Z}_2^x} \cong \mathcal{C}_{D_8/\mathbb{Z}_2^{ax}}. \tag{209}$$

At the level of 3d orthogonal gauge theories, $\mathbb{Z}_2^x$ gauging corresponds to the transitions $PSO(2N) \to PO(2N)$ while $\mathbb{Z}_2^{ax}$ gauging corresponds to the transitions $PSO(4N) \to Ss(4N)$ and $PSO(4N+2) \to PO'(4N+2)$ – see figures 1 and 13.

We find that the resulting category is

$$\mathcal{C}_{D_8/\mathbb{Z}_2^x} \cong 2\text{-Rep}\left(\left(\mathbb{Z}_2^{(1)} \times \mathbb{Z}_2^{(1)}\right) \rtimes \mathbb{Z}_2^{(0)}\right). \tag{210}$$

namely the 2-representation 2-category of a 2-group with $\mathbb{Z}_2^{(1)} \times \mathbb{Z}_2^{(1)}$ 1-form symmetry and $\mathbb{Z}_2^{(0)}$ 0-form symmetry acting on it by exchanging the two $\mathbb{Z}_2^{(1)}$ factors, and trivial Postnikov class.

**Objects.** Let us first consider the objects $D_2^{(g)} \in \mathcal{C}_{D_8}$ with $g \in \{\text{id}, a^2\}$ which are not acted upon by $D_2^{(x)}$ (by conjugation). $D_2^{(g)} \in \mathcal{C}_{D_8}$ give rise to the simple objects

$$D_2^{(\text{id})}, \quad D_2^{(a^2)} \in \mathcal{C}_{D_8/\mathbb{Z}_2^x}, \tag{211}$$

which descend from the trivial way of making $D_2^{(g)} \in \mathcal{C}_{D_8}$ symmetric under $Z_2^x$ with $x$ action implemented via the identity lines $D_1^{(g;\text{id})}$. Similarly, $2D_2^{(g)} \in \mathcal{C}_{D_8}$ give rise to another pair of simple objects denoted by

$$D_2^{(\mathbb{Z}_2)}, \quad D_2^{(a^2\mathbb{Z}_2)} \in \mathcal{C}_{D_8/\mathbb{Z}_2^x}, \tag{212}$$

which descend from making $2D_2^{(g)} \in \mathcal{C}_{D_8}$ symmetric under $\mathbb{Z}_2^x$ such that $x \in \mathbb{Z}_2^x$ is implemented via the off-diagonal matrix

$$\begin{pmatrix} 0 & D_1^{(g;\text{id})} \\ D_1^{(g;\text{id})} & 0 \end{pmatrix}. \tag{213}$$

Next let us consider the elements $\{a, a^3\} \subset D_8/\mathbb{Z}_2^x$ which are swapped under conjugation by $x$. This implies that there is no way of making $D_2^{(a)}$ or $D_2^{(a^3)}$ individually symmetric under $\mathbb{Z}_2^x$, however their sum $D_2^{(a)} \oplus D_2^{(a^3)}$ can be made $\mathbb{Z}_2^x$ symmetric by implementing $x \in \mathbb{Z}_2^x$ via

$$\begin{pmatrix} 0 & D_1^{(a^3;\text{id})} \\ D_1^{(a;\text{id})} & 0 \end{pmatrix}, \tag{214}$$

in $\mathcal{C}_{D_8}$, which maps $D_2^{(a)} \oplus D_2^{(a^3)} \to D_2^{(a^3)} \oplus D_2^{(a)}$. We denote the simple object in $\mathcal{C}_{D_8/\mathbb{Z}_2^x}$ which descends from $D_2^{(a)} \oplus D_2^{(a^3)} \in \mathcal{C}_{D_8}$ with $x$ implemented via (214) as

$$D_2^{(a,a^3)} \in \mathcal{C}_{D_8/\mathbb{Z}_2^x}. \tag{215}$$

To summarize, the simple objects in $\mathcal{C}_{D_8/\mathbb{Z}_2^x}$ are

$$\text{Obj}(\mathcal{C}_{D_8/\mathbb{Z}_2^x}) = \left\{ D_2^{(\text{id})}, D_2^{(\mathbb{Z}_2)}, D_2^{(a^2)}, D_2^{(a^2\mathbb{Z}_2)}, D_2^{(a,a^3)} \right\}. \tag{216}$$

**Fusion rules.** The objects descending from $D_2^{(g)}$ and $2D_2^{(g)}$ for $g \in \{\text{id}, a^2\}$ in $\mathcal{C}_{D_8}$ form $2\text{-Rep}(\mathbb{Z}_2) \boxtimes 2\text{-Vec}(\mathbb{Z}_2)$ and therefore satisfy the fusion rules

$$\begin{aligned} D_2^{(g)} \otimes D_2^{(h)} &\cong D_2^{(gh)}, \\ D_2^{(g)} \otimes D_2^{(h\mathbb{Z}_2)} &\cong D_2^{(gh\mathbb{Z}_2)}, \\ D_2^{(g\mathbb{Z}_2)} \otimes D_2^{(h\mathbb{Z}_2)} &\cong 2D_2^{(gh\mathbb{Z}_2)}, \end{aligned} \tag{217}$$

where $a^4 = \text{id}$. The fusion rules with the object $D_2^{(a,a^3)}$ can be computed by lifting to $\mathcal{C}_{D_8}$, performing the fusions and then organizing the fusion outcomes into indecomposable $\mathbb{Z}_2^x$ orbits. Firstly, since $D_2^{(\text{id})}$ is the identity object in $\mathcal{C}_{D_8/\mathbb{Z}_2^x}$, we get

$$D_2^{(a,a^3)} \otimes D_2^{(\text{id})} = D_2^{(a,a^3)}. \tag{218}$$

To compute the fusion $D_2^{(a,a^3)} \otimes D_2^{(\mathbb{Z}_2)}$ in $\mathcal{C}_{D_8/\mathbb{Z}_2^x}$, we need to compute the tensor product of the objects lifted to $\mathcal{C}_{D_8}$ which is

$$\left[ D_2^{(a,a^3)} \otimes D_2^{(\mathbb{Z}_2)} \right]\Big|_{\mathcal{C}_{D_8}} = (D_2^{(a)} \oplus D_2^{(a^3)}) \otimes 2D_2^{(\text{id})} = 2(D_2^{(a)} \oplus D_2^{(a^3)}). \tag{219}$$

Furthermore the $\mathbb{Z}_2$ action on the fusion outcome in $\mathcal{C}$ is given by the tensor product of (214) and (213)

$$
\begin{pmatrix} 0 & D_1^{(a^3;\text{id})} \\ D_1^{(a;\text{id})} & 0 \end{pmatrix} \otimes \begin{pmatrix} 0 & D_1^{(\text{id})} \\ D_1^{(\text{id})} & 0 \end{pmatrix} = \begin{pmatrix} 0 & 0 & 0 & D_1^{(a^3;\text{id})} \\ 0 & 0 & D_1^{(a^3;\text{id})} & 0 \\ 0 & D_1^{(a;\text{id})} & 0 & 0 \\ D_1^{(a;\text{id})} & 0 & 0 & 0 \end{pmatrix}, \tag{220}
$$

which decomposes into a direct sum of two orbits under the $\mathbb{Z}_2^x$ action, therefore we conclude

$$D_2^{(a,a^3)} \otimes D_2^{(\mathbb{Z}_2)} \cong 2D_2^{(a,a^3)}. \tag{221}$$

Analogously, the following fusion rules can also be derived

$$
\begin{aligned}
D_2^{(a,a^3)} \otimes D_2^{(a^2)} &\cong D_2^{(a,a^3)}, \\
D_2^{(a,a^3)} \otimes D_2^{(a^2\mathbb{Z}_2)} &\cong 2D_2^{(a,a^3)}.
\end{aligned} \tag{222}
$$

Finally we compute the fusion rules $D_2^{(a,a^3)} \otimes D_2^{(a,a^3)}$. Again we first lift to $\mathcal{C}_{D_8}$ and at the level of objects,

$$\left[ D_2^{(a,a^3)} \otimes D_2^{(a,a^3)} \right]\Big|_{\mathcal{C}_{D_8}} = (D_2^{(a)} \oplus D_2^{(a^3)}) \otimes (D_2^{(a)} \oplus D_2^{(a^3)}) = 2D_2^{(\text{id})} \oplus 2D_2^{(a^2)}. \tag{223}$$

Next, we consider the $\mathbb{Z}_2$ action on the fusion outcome in $\mathcal{C}_{D_8}$, which is given by the tensor product

$$
\begin{pmatrix} 0 & D_1^{(a^3;\text{id})} \\ D_1^{(a;\text{id})} & 0 \end{pmatrix} \otimes \begin{pmatrix} 0 & D_1^{(a^3;\text{id})} \\ D_1^{(a;\text{id})} & 0 \end{pmatrix} = \begin{pmatrix} 0 & 0 & 0 & D_1^{(a^2;\text{id})} \\ 0 & 0 & D_1^{(\text{id})} & 0 \\ 0 & D_1^{(\text{id})} & 0 & 0 \\ D_1^{(a^2;\text{id})} & 0 & 0 & 0 \end{pmatrix}, \tag{224}
$$

which correspondingly also decomposes into two orbits under $\mathbb{Z}_2^x$. Therefore we read off the fusion rule

$$D_2^{(a,a^3)} \otimes D_2^{(a,a^3)} = D_2^{(\mathbb{Z}_2)} \oplus D_2^{(a^2\mathbb{Z}_2)}. \tag{225}$$

We see that these are the fusion rules of the 2-category (210) by comparing with the fusion rules determined in [14] (section 3.5).

## 5.7 $\text{PSO}(4N) \rightarrow \text{O}/\text{Spin}(4N)$ and $\text{PSO}(4N+2) \rightarrow \text{O}/\text{O}'(4N+2)$

In this subsection, we consider gauging $\mathbb{Z}_2 \times \mathbb{Z}_2$ 0-form symmetry of $D_8$, where one of the $\mathbb{Z}_2$ is generated by one of the following elements

$$x, \ xa, \ xa^2, \ xa^3, \tag{226}$$

and the other $\mathbb{Z}_2$ is generated by $a^2$. All the different choices for $\mathbb{Z}_2 \times \mathbb{Z}_2$ thus obtained are related by automorphisms, so lead to isomorphic gaugings. As in the previous subsection, if two gaugings are only related by outer-automorphisms then the resulting theories after gauging may be different theories with same associated 2-categories.

For concreteness, we pick one $\mathbb{Z}_2$ generated by $ax$ and the other $\mathbb{Z}_2$ generated by $a^2$, calling the 2-category obtained after gauging as $\mathcal{C}_{D_8/\mathbb{Z}_2 \times \mathbb{Z}_2}$, which we determine to be

$$\boxed{\mathcal{C}_{D_8/\mathbb{Z}_2 \times \mathbb{Z}_2} = 2\text{-Vec}\left( (\mathbb{Z}_2^{(1)} \times \mathbb{Z}_2^{(1)}) \rtimes \mathbb{Z}_2^{(0)} \right),} \tag{227}$$

which comprises of a 2-group with $\mathbb{Z}_2^{(1)} \times \mathbb{Z}_2^{(1)}$ 1-form symmetry and $\mathbb{Z}_2^{(0)}$ 0-form symmetry acting on it by exchanging the two $\mathbb{Z}_2^{(1)}$ factors, and trivial Postnikov class. Here, we see again the 2-group whose 2-representations captured $\mathcal{C}_{D_8/\mathbb{Z}_2^x}$. This is no coincidence of course, as $\mathcal{C}_{D_8/\mathbb{Z}_2^x}$ can be obtained by gauging the full 2-group symmetry of $\mathcal{C}_{D_8/\mathbb{Z}_2^{ax} \times \mathbb{Z}_2^{a^2}}$.

At the level of 3d orthogonal gauge theories, $\mathbb{Z}_2^x \times \mathbb{Z}_2^{a^2}$ gauging corresponds to the transitions $\text{PSO}(2N) \rightarrow \text{O}(2N)$ while $\mathbb{Z}_2^{ax} \times \mathbb{Z}_2^{a^2}$ gauging corresponds to the transitions $\text{PSO}(4N) \rightarrow \text{Spin}(4N)$ and $\text{PSO}(4N+2) \rightarrow \text{O}'(4N+2)$ – see figures 1 and 13.

**Objects.** The Schur components of objects in $\mathcal{C}_{D_8/\mathbb{Z}_2 \times \mathbb{Z}_2}$ are labelled by $g \in \{\text{id}, x\} \cong \mathbb{Z}_2 = D_8/(\mathbb{Z}_2 \times \mathbb{Z}_2)$, such that the Schur component labelled by $g$ descends from different ways of making $nD_2^{(g)} \in \mathcal{C}_{D_8}$ symmetric under $\mathbb{Z}_2 \times \mathbb{Z}_2$. As we will see, the identity Schur component corresponds to the 2-category 2-Rep$(\mathbb{Z}_2 \times \mathbb{Z}_2)$. Firstly, a single copy $D_2^{(g)} \in \mathcal{C}_{D_8}$ is made $\mathbb{Z}_2 \times \mathbb{Z}_2$ symmetric by defining a monoidal functor

$$\text{Vec}_{\mathbb{Z}_2 \times \mathbb{Z}_2} \rightarrow \text{Vec}, \tag{228}$$

which are classified by classes in $H^2(\mathbb{Z}_2 \times \mathbb{Z}_2, U(1)) \cong \mathbb{Z}_2$. Therefore, we get two simple objects in $\mathcal{C}_{D_8/\mathbb{Z}_2 \times \mathbb{Z}_2}$ from $D_2^{(g)} \in \mathcal{C}_{D_8}$. We denote these as

$$D_2^{(g)}, \qquad D_2^{(g-)}, \tag{229}$$

corresponding to the trivial and non-trivial element in $H^2(\mathbb{Z}_2 \times \mathbb{Z}_2, U(1))$. Next, there are three indecomposable ways in which $2D_2^{(g)}$ can be made $\mathbb{Z}_2 \times \mathbb{Z}_2$ symmetric by providing monoidal functors

$$\text{Vec}_{\mathbb{Z}_2 \times \mathbb{Z}_2} \rightarrow \text{Mat}_2(\text{Vec}). \tag{230}$$

These descend to three simple objects in $\mathcal{C}_{D_8/\mathbb{Z}_2 \times \mathbb{Z}_2}$, which we denote by

$$D_2^{(g\mathbb{Z}_2^{xa})}, \qquad D_2^{(g\mathbb{Z}_2^{a^2})}, \qquad D_2^{(g\mathbb{Z}_2^{xa^3})}. \tag{231}$$

More precisely, $D_2^{(g\mathbb{Z}_2^\varphi)}$ with $\varphi \in \{xa, a^2, xa^3\}$, descends from $2D_2^g \in \mathcal{C}_{D_8}$ with the elements id, $\varphi$ implemented by the diagonal matrix

$$\begin{pmatrix} D_1^{(g;\text{id})} & 0 \\ 0 & D_1^{(g;\text{id})} \end{pmatrix}, \tag{232}$$

while the remaining two symmetry operations in $\mathbb{Z}_2 \times \mathbb{Z}_2$ implemented via the off-diagonal matrix

$$\begin{pmatrix} 0 & D_1^{(g;\text{id})} \\ D_1^{(g;\text{id})} & 0 \end{pmatrix}. \tag{233}$$

There are no new indecomposable objects that descend from $3D_2^{(g)} \in \mathcal{C}_{D_8}$ since there are no indecomposable order three orbits of $\mathbb{Z}_2 \times \mathbb{Z}_2$. We get a final simple object from $4D_2^{(g)} \in \mathcal{C}_{D_8}$ on which the elements $xa, xa^3, a^2 \in \mathbb{Z}_2 \times \mathbb{Z}_2$ act via the matrices $D_1^{(g;\text{id})} P_{(12)} \otimes 1_2$, $D_1^{(g;\text{id})} 1_2 \otimes P_{(12)}$ and $D_1^{(g;\text{id})} \sigma^x \otimes P_{(12)}$. We denote this simple object as $D_2^{(g\mathbb{Z}_2 \times \mathbb{Z}_2)}$. To summarize, there are twelve simple objects in $\mathcal{C}_{D_8/\mathbb{Z}_2 \times \mathbb{Z}_2}$[18]

$$\text{Obj}(\mathcal{C}_{D_8/\mathbb{Z}_2 \times \mathbb{Z}_2}) = \left\{ D_2^{(g)}, D_2^{(g-)}, D_2^{(g\mathbb{Z}_2^{xa})}, D_2^{(g\mathbb{Z}_2^{a^2})}, D_2^{(g\mathbb{Z}_2^{xa^3})}, D_2^{(g\mathbb{Z}_2 \times \mathbb{Z}_2)} \ \middle| \ g \in \{\text{id}, x\} \right\}. \tag{234}$$

**Fusion rules.** The fusion rules of the identity Schur component are completely analogous to the $\mathbb{Z}_2 \times \mathbb{Z}_2$ case described in section 2. Meanwhile, the Schur component descending from $D_2^{(x)} \in \mathcal{C}_{D_8}$ provides a $\mathbb{Z}_2$ grading to the fusion rules since

$$D_2^{(x)} \otimes D_2^{(x)} = D_2^{(\text{id})} \in \mathcal{C}_{D_8}. \tag{235}$$

More precisely, the simple objects $D_2^{(g)}$ and $D_2^{(g-)}$ are invertible objects with fusion rules

$$\begin{aligned} D_2^{(g)} \otimes D_2^{(h)} &\cong D_2^{(gh)}, \\ D_2^{(g)} \otimes D_2^{(h-)} &\cong D_2^{(gh-)}, \\ D_2^{(g-)} \otimes D_2^{(h-)} &\cong D_2^{(gh)}, \end{aligned} \tag{236}$$

where $g, h \in \{\text{id}, x\}$, $x^2 = \text{id}$. The fusions between the 2-dimensional and 4-dimensional objects with the invertible objects are

$$\begin{aligned} D_2^{(g)} \otimes D_2^{(h\mathbb{Z}_2^{\varphi})} &\cong D_2^{(gh\mathbb{Z}_2^{\varphi})}, \\ D_2^{(g-)} \otimes D_2^{(h\mathbb{Z}_2^{\varphi})} &\cong D_2^{(gh\mathbb{Z}_2^{\varphi})}, \\ D_2^{(g)} \otimes D_2^{(h\mathbb{Z}_2 \times \mathbb{Z}_2)} &\cong D_2^{(gh\mathbb{Z}_2 \times \mathbb{Z}_2)}, \\ D_2^{(g-)} \otimes D_2^{(h\mathbb{Z}_2 \times \mathbb{Z}_2)} &\cong D_2^{(gh\mathbb{Z}_2 \times \mathbb{Z}_2)}. \end{aligned} \tag{237}$$

The fusions between the 2-dimensional objects is

$$D_2^{(g\mathbb{Z}_2^{\varphi})} \otimes D_2^{(h\mathbb{Z}_2^{\varphi'})} \cong \begin{cases} 2D_2^{(gh\mathbb{Z}_2^{\varphi})}, & \text{if} \quad \varphi = \varphi', \\ D_2^{(gh\mathbb{Z}_2 \times \mathbb{Z}_2)}, & \text{if} \quad \varphi \neq \varphi'. \end{cases} \tag{238}$$

The remaining fusion rules are

$$\begin{aligned} D_2^{(g\mathbb{Z}_2^{\varphi})} \otimes D_2^{(h\mathbb{Z}_2 \times \mathbb{Z}_2)} &\cong 2D_2^{(gh\mathbb{Z}_2 \times \mathbb{Z}_2)}, \\ D_2^{(g\mathbb{Z}_2 \times \mathbb{Z}_2)} \otimes D_2^{(h\mathbb{Z}_2 \times \mathbb{Z}_2)} &\cong 4D_2^{(gh\mathbb{Z}_2 \times \mathbb{Z}_2)}. \end{aligned} \tag{239}$$

---

[18]In what follows, we denote $D_2^{\text{id}-} = D_2^-$ in $\mathcal{C}_{D_8/\mathbb{Z}_2 \times \mathbb{Z}_2}$.

**Morphisms.** Let us now consider the 1-endomorphisms of each of the objects in $\mathcal{C}_{D_8/\mathbb{Z}_2\times\mathbb{Z}_2}$. Firstly, the 1-endomorphisms for the objects $D_2^{(g)}$ and $D_2^{(g-)}$ in $\mathcal{C}_{D_8/\mathbb{Z}_2\times\mathbb{Z}_2}$ correspond to the ways in which the 1-endomorphisms of $D_2^{(g)} \in \mathcal{C}_{D_8}$ can be made $\mathbb{Z}_2 \times \mathbb{Z}_2$ symmetric. Recall that the 1-category of lines on $D_2^{(g)} \in \mathcal{C}_{D_8}$ is Vec generated by the single line denoted by $D_1^{(g;\mathrm{id})} \in \mathcal{C}_{D_8}$. Therefore, our task boils down to enumerating the ways in which $\mathbb{Z}_2 \times \mathbb{Z}_2$ symmetry can be implemented on $D_1^{(g;\mathrm{id})}$. Noticing that the endomorphism space of $D_1^{(g;\mathrm{id})}$ is itself one-dimensional, these different ways simply correspond to the one-dimensional irreducible representations of $\mathbb{Z}_2 \times \mathbb{Z}_2$. Therefore we find

$$
\begin{aligned}
\mathrm{End}_{\mathcal{C}_{D_8/\mathbb{Z}_2\times\mathbb{Z}_2}}(D_2^{(g)}) &= \left\{ D_1^{(g,\mathrm{id})}, D_1^{(g,xa)}, D_1^{(g,a^2)}, D_1^{(g,xa^3)} \right\} \cong \mathrm{Rep}(\mathbb{Z}_2 \times \mathbb{Z}_2), \\
\mathrm{End}_{\mathcal{C}_{D_8/\mathbb{Z}_2\times\mathbb{Z}_2}}(D_2^{(g-)}) &= \left\{ D_1^{(g-,\mathrm{id})}, D_1^{(g-,xa)}, D_1^{(g-,a^2)}, D_1^{(g-,xa^3)} \right\} \cong \mathrm{Rep}(\mathbb{Z}_2 \times \mathbb{Z}_2),
\end{aligned}
\tag{240}
$$

where the line $D_1^{(g,\mathrm{id})}$ and $D_1^{(g-,\mathrm{id})}$ are the trivial $\mathbb{Z}_2 \times \mathbb{Z}_2$ representation lines on $D_2^{(g)}$ and $D_2^{(g-)}$ in $\mathcal{C}_{D_8/\mathbb{Z}_2\times\mathbb{Z}_2}$ respectively. Similarly $D_1^{(g,\varphi)}$ and $D_1^{(g,\varphi)}$ are the lines on $D_2^{(g)}$ and $D_2^{(g-)}$ in $\mathcal{C}_{D_8/\mathbb{Z}_2\times\mathbb{Z}_2}$ respectively which carry the trivial representation of $\varphi \in \mathbb{Z}_2 \times \mathbb{Z}_2$ and the sign representation of the remaining order two elements in $\mathbb{Z}_2 \times \mathbb{Z}_2$.

Next, let us consider the endomorphsims of the two dimensional objects in $D_2^{(g\mathbb{Z}_2^\varphi)} \in \mathcal{C}_{D_8/\mathbb{Z}_2\times\mathbb{Z}_2}$ which descend from $2D_2^{(g)} \in \mathcal{C}_{D_8}$. The 1-endomorphisms of $2D_2^{(g)} \in \mathcal{C}_{D_8}$ are $\mathrm{Mat}_2(\mathrm{Vec})$ which we need to make symmetric under $\mathbb{Z}_2 \times \mathbb{Z}_2$. Recall that the action of $\mathbb{Z}_2 \times \mathbb{Z}_2$ was chosen such that $\varphi$ acted diagonally, therefore we may now dress the 1-endomorphisms of $2D_2^{(g)} \in \mathcal{C}_{D_8}$ with representations of $\mathbb{Z}_2^\varphi$. Doing so we obtain

$$
\begin{aligned}
\mathrm{End}_{\mathcal{C}_{D_8/\mathbb{Z}_2\times\mathbb{Z}_2}}(D_2^{(g\mathbb{Z}_2^\varphi)}) &= \left\{ D_1^{(g\mathbb{Z}_2^\varphi;\mathrm{id}+)}, D_1^{(g\mathbb{Z}_2^\varphi;\mathrm{id}-)}, D_1^{(g\mathbb{Z}_2^\varphi;-+)}, D_1^{(g\mathbb{Z}_2^\varphi;--)} \right\} \\
&\cong \mathrm{Rep}(\mathbb{Z}_2^\varphi) \boxtimes \mathrm{Vec}_{\mathbb{Z}_2\times\mathbb{Z}_2/\mathbb{Z}_2^\varphi}.
\end{aligned}
\tag{241}
$$

Here $D_1^{(g\mathbb{Z}_2^\varphi;\mathrm{id}s)}$ and $D_1^{(g\mathbb{Z}_2^\varphi;-s)}$ descend from the endomorphism corresponding to the diagonal and off-diagonal $2 \times 2$ matrices in $\mathrm{Mat}_2(\mathrm{Vec})$ respectively, with each entry carrying the $s \in \pm$ irreducible representation of $\mathbb{Z}_2^\varphi$.

Let us finally decribe the 1-endomorphsisms of the object $D_2^{(g\mathbb{Z}_2\times\mathbb{Z}_2)}$. Now we need to look for $\mathbb{Z}_2 \times \mathbb{Z}_2$ symmetric 1-morphisms in $\mathrm{Mat}_4(\mathrm{Vec}) = \mathrm{End}_{\mathcal{C}_{D_8}}(4D_2^{(g)})$. Let us label the entries of such $4 \times 4$ matrices as $(i,j);(k,l)$, where $i,j,k,l \in \{0,1\}$ and the tuples $(i,j)$ and $(k,l)$ label the rows and columns respectively. Then the $\mathbb{Z}_2 \times \mathbb{Z}_2$ symmetry acts as

$$
\begin{aligned}
\mathrm{id} &: (i,j);(k,l) \longrightarrow (i,j);(k,l), \\
xa &: (i,j);(k,l) \longrightarrow (i+1 \bmod 2, j);(k+1 \bmod 2, l), \\
xa^3 &: (i,j);(k,l) \longrightarrow (i, j+1 \bmod 2);(k, l+1 \bmod 2), \\
a^2 &: (i,j);(k,l) \longrightarrow (i+1 \bmod 2, j+1 \bmod 2);(k+1 \bmod 2, l+1 \bmod 2).
\end{aligned}
\tag{242}
$$

We need to look for $\mathrm{Mat}_4(\mathrm{Vec}) = \mathrm{End}_{\mathcal{C}_{D_8}}(4D_2^{(g)})$ that commute with this action. It can be checked that matrices satisfying this condition are also

$$
D_1^{(g;\mathrm{id})}\mathbf{1}_4, \quad D_1^{(g;\mathrm{id})}\sigma^x \otimes \mathbf{1}_2, \quad D_1^{(g;\mathrm{id})}\mathbf{1}_2 \otimes \sigma^x, \quad D_1^{(g;\mathrm{id})}\sigma^x \otimes \sigma^x.
\tag{243}
$$

These matrices form a permutation representation of $\mathbb{Z}_2 \times \mathbb{Z}_2$ and we label the corresponding 1-endomorphisms as

$$
\begin{aligned}
\mathrm{End}_{\mathcal{C}_{D_8/\mathbb{Z}_2\times\mathbb{Z}_2}} &= \left\{ D_1^{(g\mathbb{Z}_2\times\mathbb{Z}_2;\mathrm{id})}, D_1^{(g\mathbb{Z}_2\times\mathbb{Z}_2;xa)}, D_1^{(g\mathbb{Z}_2\times\mathbb{Z}_2;xa^3)}, D_1^{(g\mathbb{Z}_2\times\mathbb{Z}_2;a^2)} \right\} \\
&\cong \mathrm{Vec}_{\mathbb{Z}_2\times\mathbb{Z}_2}.
\end{aligned}
\tag{244}
$$

Similarly, the remaining morphisms between objects and their fusion/composition rules can be computed.

**Outer automorphism action and 2-group structure.** At the level of objects and 1-morphisms and their fusion composition rules, the symmetry category $\mathcal{C}_{D_8/\mathbb{Z}_2 \times \mathbb{Z}_2}$ is isomorphic to

$$2\text{-Rep}(\mathbb{Z}_2 \times \mathbb{Z}_2) \boxtimes 2\text{-Vec}(\mathbb{Z}_2). \tag{245}$$

However this would also be the case had we started from $\mathbb{Z}_2^3$ and gauged a normal $\mathbb{Z}_2 \times \mathbb{Z}_2$ subgroup. The fact that we started form 2-Vec($D_8$) shows up in the action of the 2-Vec($\mathbb{Z}_2$) symmetry on the 2-Rep($\mathbb{Z}_2 \times \mathbb{Z}_2$) subcategory. Notably, $\mathcal{C}_{D_8/\mathbb{Z}_2 \times \mathbb{Z}_2}$ has a $\mathbb{Z}_2$ 0-form symmetry generated by $D_2^{(x)}$. This defect descends from $D_2^{(x)} \in \mathcal{C}_{D_8}$ which had the following action

$$D_2^{(x)} : D_2^{(xa)} \longleftrightarrow D_2^{(xa^3)} \in \mathcal{C}_{D_8}. \tag{246}$$

This action descends to an action on various objects and morphisms of $\mathcal{C}_{D_8/\mathbb{Z}_2 \times \mathbb{Z}_2}$. First at the level of objects clearly

$$D_2^{(x)} : \quad D_2^{(g\mathbb{Z}_2^{xa})} \longleftrightarrow D_2^{(g\mathbb{Z}_2^{xa^3})} \in \mathcal{C}_{D_8/\mathbb{Z}_2 \times \mathbb{Z}_2}, \tag{247}$$

since the action of $xa$ and $xa^3$ is swapped under $x$. Along with the object, naturally the 1-endomorphisms are also swapped under the $D_2^{(x)}$ action. It can be checked that the remaining simple objects, being symmetric under swapping $xa$ and $xa^3$ are consequently invariant under the action of $D_2^{(x)}$. The 1-endomorphisms of these objects however still incur a non-trivial action of $D_2^{(x)}$. For reasons similar to above, we find that $D_2^{(x)}$ acts on the 1-endomorphisms of $D_2^{(g)} \in \mathcal{C}_{D_8/\mathbb{Z}_2 \times \mathbb{Z}_2}$ as

$$D_2^{(x)} : \quad D_1^{(g;xa)} \longleftrightarrow D_1^{(g;xa^3)} \quad \in \text{End}_{\mathcal{C}_{D_8/\mathbb{Z}_2 \times \mathbb{Z}_2}}(D_2^{(g)}). \tag{248}$$

Similarly

$$D_2^{(x)} : \quad D_1^{(g-;xa)} \longleftrightarrow D_1^{(g-;xa^3)} \quad \in \text{End}_{\mathcal{C}_{D_8/\mathbb{Z}_2 \times \mathbb{Z}_2}}(D_2^{(g-)}). \tag{249}$$

The simple object $D_2^{(g\mathbb{Z}_2^{a^2})}$ along with its 1-endomorphisms are invariant under the $D_2^x$ action. Finally, the endomorphisms of $D_2^{(g\mathbb{Z}_2 \times \mathbb{Z}_2)} \in \mathcal{C}_{D_8/\mathbb{Z}_2 \times \mathbb{Z}_2}$ transform as

$$D_2^{(x)} : \quad D_1^{(g\mathbb{Z}_2 \times \mathbb{Z}_2;xa)} \longleftrightarrow D_1^{(g\mathbb{Z}_2 \times \mathbb{Z}_2;xa^3)} \in \text{End}_{\mathcal{C}_{D_8/\mathbb{Z}_2 \times \mathbb{Z}_2}}(D_2^{(g\mathbb{Z}_2 \times \mathbb{Z}_2)}), \tag{250}$$

due to the fact that under $D_2^{(x)}$, the tensor decomposed blocks in (243) are swapped.

## 5.8 $SO(2N) \rightarrow O(2N)$

In this subsection we consider gauging the subgroup $\mathbb{Z}_2^x = \{1, x\}$ in the symmetry category $\mathcal{C}_{D_8/\mathbb{Z}_2^{a^2}}$. This implements the transition from $\mathcal{C}_{D_8/\mathbb{Z}_2^{a^2}}$ to

$$\boxed{\mathcal{C}_{D_8/\mathbb{Z}_2 \times \mathbb{Z}_2} = 2\text{-Vec}((\mathbb{Z}_2^2)^{(1)} \rtimes \mathbb{Z}_2),} \tag{251}$$

and corresponds to going from the $SO(2N)$ theory to the $O(2N)$ theory, see figures 1 and 13.

**Objects.**   Upon gauging $\mathbb{Z}_2^x$, the following pairs of simple objects are identified

$$\left\{ D_2^{(\text{id})} \sim D_2^{(x)}, \quad D_2^{(a)} \sim D_2^{(ax)}, \quad D_2^{(\mathbb{Z}_2)} \sim D_2^{(x\mathbb{Z}_2)}, \quad D_2^{(a\mathbb{Z}_2)} \sim D_2^{(ax\mathbb{Z}_2)} \right\}, \tag{252}$$

so that we get two Schur components of simple objects labelled by $g \in \{\text{id}, a\} \in D_8/(\mathbb{Z}_2 \times \mathbb{Z}_2) \cong \mathbb{Z}_2$. There are two objects which descend from $D_2^{(g)} \in \mathcal{C}_{D_8/\mathbb{Z}_2^{a^2}}$ with the $\mathbb{Z}_2^x$ symmetry implemented by $D_1^{(g;\text{id})}$ and $D_1^{(g;a^2)}$ respectively. We denote these as

$$D_2^{(g)}, \ D_2^{(g-)} \in \text{Obj}(\mathcal{C}_{D_8/\mathbb{Z}_2 \times \mathbb{Z}_2}), \tag{253}$$

which correspond to the objects (229). There is also a simple object which descends from $2D_2^{(g)} \in \mathcal{C}_{D_8/\mathbb{Z}_2^{a^2}}$ with the $\mathbb{Z}_2^x$ symmetry implemented via the off diagonal matrix

$$\begin{pmatrix} 0 & D_1^{(g;\text{id})} \\ D_1^{(g;\text{id})} & 0 \end{pmatrix} \in \text{Mat}_2(\text{Vec}). \tag{254}$$

We denote this object by $D_2^{(g\mathbb{Z}_2^x)}$ (and corresponds to the object denoted $D_2^{(g\mathbb{Z}_2^{a^2})}$ in (231)). Next we can make $D_2^{(g\mathbb{Z}_2)} \in \mathcal{C}_{D_8/\mathbb{Z}_2^{a^2}}$ symmetric under $\mathbb{Z}_2^x$ in two ways, i.e., by implementing $\mathbb{Z}_2^x$ via $D_1^{(g\mathbb{Z}_2,\text{id})}$ or $D_1^{(g\mathbb{Z}_2,-)}$ respectively. We denote these two objects by

$$D_2^{(g\mathbb{Z}_2^{a^2})}, D_2^{(g\mathbb{Z}_2^{xa^2})} \in \mathcal{C}_{D_8/\mathbb{Z}_2^{a^2}}, \tag{255}$$

and correspond to $D_2^{(g\mathbb{Z}_2^{xa})}$ and $D_2^{(g\mathbb{Z}_2^{xa^3})}$ in (231). Finally we have another object descending from $2D_2^{(g\mathbb{Z}_2)} \in \mathcal{C}_{D_8/\mathbb{Z}_2^{a^2}}$, which can be made $\mathbb{Z}_2^x$ symmetric using

$$\begin{pmatrix} 0 & D_1^{(g\mathbb{Z}_2;\text{id})} \\ D_1^{(g\mathbb{Z}_2;\text{id})} & 0 \end{pmatrix} \in \text{Mat}_2(\text{Vec}). \tag{256}$$

We denote this object by $D_2^{(g\mathbb{Z}_2 \times \mathbb{Z}_2)}$. To summarize, we have recovered the set of simple objects in O($2N$),

$$\text{Obj}(\mathcal{C}_{D_8/\mathbb{Z}_2 \times \mathbb{Z}_2}) = \left\{ D_2^{(g)}, \quad D_2^{(g-)}, \quad D_2^{(g\mathbb{Z}_2^x)}, \quad D_2^{(g\mathbb{Z}_2^{a^2})}, \quad D_2^{(g\mathbb{Z}_2^{xa^2})}, \quad D_2^{(g\mathbb{Z}_2 \times \mathbb{Z}_2)} \ \middle| \ g \in \{\text{id}, a\} \right\}. \tag{257}$$

which is consistent with (234).

**Fusion rules.**   By noting the following fusion rules in $\mathcal{C}_{D_8/\mathbb{Z}_2^{a^2}}$

$$\begin{aligned} D_2^{(g)} \otimes D_2^{(h)} &\cong D_2^{(gh)}, \\ D_1^{(g;\text{id})} \otimes D_1^{(h;\text{id})} &\cong D_1^{(gh;\text{id})}, \\ D_1^{(g;\text{id})} \otimes D_1^{(h;a^2)} &\cong D_1^{(gh;a^2)}, \\ D_1^{(g;a^2)} \otimes D_1^{(h;a^2)} &\cong D_1^{(gh;\text{id})}. \end{aligned} \tag{258}$$

where $g, h \in \{\text{id}, a\}$ and $a^2 = \text{id}$. We can immediately read-off

$$\begin{aligned} D_2^{(g)} \otimes D_2^{(h)} &\cong D_2^{(gh)}, \\ D_2^{(g)} \otimes D_2^{(h-)} &\cong D_2^{(gh-)}, \\ D_2^{(g-)} \otimes D_2^{(h-)} &\cong D_2^{(gh)}, \end{aligned} \tag{259}$$

in $\mathcal{C}_{D_8/\mathbb{Z}_2 \times \mathbb{Z}_2}$. Next, we compute the fusion rules between the defects with dimension 2, i.e, $D_2^{(g\mathbb{Z}_2^x)}, D_2^{(g\mathbb{Z}_2^{a^2})}, D_2^{(g\mathbb{Z}_2^{xa^2})}$ and the invertible defects. Firstly, we get

$$
\begin{aligned}
D_2^{(g)} \otimes D_2^{(h\mathbb{Z}_2^x)} &\cong D_2^{(gh\mathbb{Z}_2^x)}, \\
D_2^{(g-)} \otimes D_2^{(h\mathbb{Z}_2^x)} &\cong D_2^{(gh\mathbb{Z}_2^x)},
\end{aligned}
\tag{260}
$$

which is obtained straightforwardly by computing the tensor product of the 1-morphisms implementing the $\mathbb{Z}_2^x$ symmetry on the corresponding objects lifted to $\mathcal{C}_{D_8/\mathbb{Z}_2 \times \mathbb{Z}_2}$. Next, we obtain

$$
\begin{aligned}
D_2^{(g)} \otimes D_2^{(h\mathbb{Z}_2^{a^2})} &\cong D_2^{(gh\mathbb{Z}_2^{a^2})}, \\
D_2^{(g-)} \otimes D_2^{(h\mathbb{Z}_2^{a^2})} &\cong D_2^{(gh\mathbb{Z}_2^{a^2})}, \\
D_2^{(g)} \otimes D_2^{(h\mathbb{Z}_2^{xa^2})} &\cong D_2^{(gh\mathbb{Z}_2^{xa^2})}, \\
D_2^{(g-)} \otimes D_2^{(h\mathbb{Z}_2^{xa^2})} &\cong D_2^{(gh\mathbb{Z}_2^{xa^2})},
\end{aligned}
\tag{261}
$$

which follows from the fusion rules

$$
\begin{aligned}
D_2^{(g)} \otimes D_2^{(h\mathbb{Z}_2)} &\cong D_2^{(gh\mathbb{Z}_2)}, \\
D_1^{(g;\varphi)} \otimes D_1^{(h\mathbb{Z}_2;\theta)} &\cong D_1^{(gh\mathbb{Z}_2;\theta)},
\end{aligned}
\tag{262}
$$

with $\varphi \in \{\text{id}, a^2\}$ and $\theta \in \{\text{id}, -\}$. The fusion rules among the two dimensional defects are

$$
D_2^{(g\mathbb{Z}_2^\varphi)} \otimes D_2^{(h\mathbb{Z}_2^{\varphi'})} \cong \begin{cases} 2D_2^{(gh\mathbb{Z}_2^\varphi)}, & \text{if} \quad \varphi = \varphi', \\ D_2^{(gh\mathbb{Z}_2 \times \mathbb{Z}_2)}, & \text{if} \quad \varphi \neq \varphi', \end{cases}
\tag{263}
$$

where $\varphi, \varphi' \in \{x, a^2, xa^2\}$. As usual, these fusion rules follow from the tensor product of 1-morphism implementing $\mathbb{Z}_2^x$ in $\mathcal{C}_{D_8/\mathbb{Z}_2^{a^2}}$. The relevant fusions are

$$
\begin{aligned}
D_1^{(g\mathbb{Z}_2;\text{id})} \otimes D_1^{(h\mathbb{Z}_2;\text{id})} &\cong D_1^{(gh\mathbb{Z}_2;\text{id})} \oplus D_1^{(gh\mathbb{Z}_2;\text{id})}, \\
D_1^{(g\mathbb{Z}_2;-)} \otimes D_1^{(h\mathbb{Z}_2;-)} &\cong D_1^{(gh\mathbb{Z}_2;-)} \oplus D_1^{(gh\mathbb{Z}_2;-)}, \\
D_1^{(g\mathbb{Z}_2;-)} \otimes D_1^{(h\mathbb{Z}_2;\text{id})} &\cong \begin{pmatrix} 0 & D_1^{(gh\mathbb{Z}_2;-)} \\ D_1^{(gh\mathbb{Z}_2;-)} & 0 \end{pmatrix}.
\end{aligned}
\tag{264}
$$

Similarly, the remaining fusions (239) can be computed using the same methods. At the level of the objects and fusion rules, we have recovered the category

$$
2\text{-Rep}(\mathbb{Z}_2 \times \mathbb{Z}_2) \boxtimes 2\text{-Vec}(\mathbb{Z}_2).
\tag{265}
$$

**Outer automorphism action and 2-group structure.** Additionally there is an outer automorphism action of the $\mathbb{Z}_2$ 0-form symmetry generator $D_2^{(a)}$ on the objects in $2\text{-Rep}(\mathbb{Z}_2 \times \mathbb{Z}_2)$. Let us recover the outer automorphism action derived in section 5.2. Consider first, the line-like function $D_1^{(x,g\mathbb{Z}_2)}$ between the $D_2^{(x)}$ defect and the $D_2^{(g\mathbb{Z}_2)}$ defect in $\mathcal{C}_{D_8/\mathbb{Z}_2^{a^2}}$. Due to the higher-symmetry fractionalization on the $D_2^{(g\mathbb{Z}_2)}$ defect, the $D_2^{(a)}$ defect acts on this line as

$$
D_1^{(a,g\mathbb{Z}_2)} \circ D_1^{(x,g\mathbb{Z}_2)} \circ D_1^{(a,g\mathbb{Z}_2)} \cong D_1^{(g\mathbb{Z}_2,-)} \circ D_1^{(x,g\mathbb{Z}_2)}.
\tag{266}
$$

This is equivalent to the statement that $D_1^{(a,g\mathbb{Z}_2)}$ is the junction line between two surfaces where the $\mathbb{Z}_2^x$ action is related by composition with $D_1^{(g\mathbb{Z}_2;-)}$. As discussed above, these two surfaces are precisely the pre-images of $D_2^{(g\mathbb{Z}_2^{a^2})}$ and $D_2^{(g\mathbb{Z}_2^{xa^2})}$ respectively in $\mathcal{C}_{D_8/\mathbb{Z}_2^{a^2}}$. This implies the action

$$D_2^{(a)}: \qquad D_2^{(g\mathbb{Z}_2^{a^2})} \longleftrightarrow D_2^{(g\mathbb{Z}_2^{xa^2})} \in \mathcal{C}_{D_8/\mathbb{Z}_2 \times \mathbb{Z}_2}. \tag{267}$$

Finally, let us derive the action of $D_2^{(a)}$ on the 1-endomorphisms of $D_2^{(g)}$ in $\mathcal{C}_{D_8/\mathbb{Z}_2 \times \mathbb{Z}_2}$. There are four simple 1-endomorphisms

$$\mathrm{End}_{\mathcal{C}_{D_8/\mathbb{Z}_2 \times \mathbb{Z}_2}}(D_2^{(\mathrm{id})}) = \left\{ D_1^{(g;\mathrm{id})}, \quad D_1^{(g;a^2)}, \quad D_1^{(g;x)}, \quad D_1^{(g;xa^2)} \right\} \cong \mathsf{Rep}(\mathbb{Z}_2 \times \mathbb{Z}_2). \tag{268}$$

In particular, the lines $D_1^{(g;\mathrm{id})}$ and $D_1^{(g;x)}$ descend from the line $D_1^{(g;\mathrm{id})}$ in $\mathcal{C}_{D_8/\mathbb{Z}_2^{a^2}}$, with the $\mathbb{Z}_2^x$ implemented via the even and odd 2-morphism $D_0^{(g;\mathrm{id})}$ and $-D_0^{(g;\mathrm{id})}$ respectively. Similarly the lines $D_1^{(g;a^2)}$ and $D_1^{(g;xa^2)}$ descend from the line $D_1^{(g;a^2)}$ in $\mathcal{C}_{D_8/\mathbb{Z}_2^{a^2}}$, with the $\mathbb{Z}_2^x$ implemented via the even and odd 2-morphism $D_0^{(g;a^2;\mathrm{id})}$ and $-D_0^{(g;a^2,\mathrm{id})}$ respectively.

Let us now consider the setup where the line $D_1^{(g;a^2)}$ in $\mathcal{C}_{D_8/\mathbb{Z}_2^{a^2}}$ pierces the surface $D_2^{(x)}$. Let us denote the junction by $\mathcal{O}_{(g,x)}$. As mentioned, $\mathcal{O}_{(g,x)} = \pm D_0^{(g;a^2,\mathrm{id})}$ and depending on the choice of sign we get $D_1^{(g;a^2)}$ or $D_1^{(g;xa^2)}$ in $\mathcal{C}_{D_8/\mathbb{Z}_2 \times \mathbb{Z}_2}$. Due to the symmetry fractionalization in the SO(2N) category, the surface $D_2^{(a)}$ acts on $\mathcal{O}_{(g,x)}$ by a change of sign. Consequently after gauging $\mathbb{Z}_2^x$, $D_2^{(a)}$ has the action

$$D_2^{(a)}: \qquad D_1^{(g;a^2)} \longleftrightarrow D_1^{(g;xa^2)} \in \mathcal{C}_{D_8/\mathbb{Z}_2 \times \mathbb{Z}_2}. \tag{269}$$

This concludes our journey through the categorical $D_8$-web.

# 6 Generalization to $d > 3$ dimensions

Generalizing the methods of this paper and the companion paper [1], one can consider studying symmetry webs comprising of $(d-1)$-categories associated to orthogonal gauge theories in general $d$ dimensions.

## 6.1 Maximal symmetry category associated to higher-groups

Before we describe the $(d-1)$-categories associated to orthogonal gauge theories, let us discuss a few general features first. Consider a $d$-dimensional QFT $\mathfrak{T}$ with a faithfully acting $\Gamma$ higher-group symmetry, comprising of various component $p$-form symmetry groups $\Gamma^{(p)}$. By definition, this means that $\mathfrak{T}$ contains topological invertible operators of codimension-$(p+1)$ labeled by elements of $\Gamma^{(p)}$ such that there does not exist a topological codimension-$(p+2)$ operator sitting at the end of a topological codimension-$(p+1)$ operator labeled by a non-identity element of $\Gamma^{(p)}$. The fusion of these codimension-$(p+1)$ operators is controlled by the group multiplication in $\Gamma^{(p)}$, and the operators of different codimensions interact with each other via acting on each other or operators of higher-codimension arising when performing topological moves upon operators of lower-codimension. These operators along with these properties describe a non-$\mathbb{C}$-linear $(d-1)$-category

$$\mathcal{C}_{d-1}^{\Gamma}, \tag{270}$$

associated to the higher-group $\Gamma$. Since we have topological local operators in $\mathfrak{T}$ given by $\mathbb{C}$, we can extend $\mathcal{C}_{d-1}^{\Gamma}$ to a $\mathbb{C}$-linear $(d-1)$-category

$$\text{(d}-1\text{)-Vec}^0(\Gamma), \tag{271}$$

which contains topological local operators valued in $\mathbb{C}$. If additionally the $\Gamma$ higher-group symmetry carries a 't Hooft anomaly specified by an element $\omega \in H^{d+1}\big(B\Gamma, U(1)\big)$ where $B\Gamma$ denotes the classifying space for the higher-group $\Gamma$, then it can be incorporated as associators or coherence relations on top of the $(d-1)$-category $(d-1)$-Vec$^0(\Gamma)$, giving rise to a new $(d-1)$-category that we label as

$$\text{(d}-1\text{)-Vec}^{0,\omega}(\Gamma). \tag{272}$$

Now, we can perform various operations on topological operators in $(d-1)$-Vec$^0(\Gamma)$ to produce new topological operators that are not contained in $(d-1)$-Vec$^0(\Gamma)$. The first operation is to include all possible condensation defects of various codimensions that can be constructed using operators in $(d-1)$-Vec$^0(\Gamma)$. This operation at the level of $(d-1)$-category is known as Karoubi completion. After including condensation defects, that is after Karoubi completing $(d-1)$-Vec$^0(\Gamma)$, we obtain a larger $(d-1)$-category

$$\text{(d}-1\text{)-Vec}(\Gamma), \tag{273}$$

which can be mathematically understood as the $(d-1)$-category formed by $(d-1)$-vector spaces graded by higher-group $\Gamma$. In other words, the presence of faithfully acting higher-group $\Gamma$ symmetry in $\mathfrak{T}$ means that $\mathfrak{T}$ contains all the topological operators contained in the fusion[19] $(d-1)$-category $(d-1)$-Vec$(\Gamma)$. Incorporating an 't Hooft anomaly $\omega$, we instead obtain the $(d-1)$-category

$$\text{(d}-1\text{)-Vec}^{\omega}(\Gamma). \tag{274}$$

We can actually enlarge $(d-1)$-Vec$(\Gamma)$ even further by stacking $(d-p-1)$-dimensional TQFTs on the $(d-p-1)$-dimensional topological defects generating $\Gamma$ before performing condensations. After including TQFTs in this fashion, we obtain a $(d-1)$-category

$$\mathcal{T}_{d-1}(\Gamma), \tag{275}$$

which can be understood as $\Gamma$-graded version of $(d-1)$-category $\mathcal{T}_{d-1}$ of $(d-1)$-dimensional TQFTs. We label its anomalous version as

$$\mathcal{T}_{d-1}^{\omega}(\Gamma). \tag{276}$$

In total, the presence of faithfully acting higher-group $\Gamma$ symmetry in $\mathfrak{T}$ means that $\mathfrak{T}$ contains all the topological operators contained in the $(d-1)$-category $\mathcal{T}_{d-1}(\Gamma)$, which we refer to as the *maximal* symmetry $(d-1)$-category associated to the higher group $\Gamma$.

## 6.2 Maximal symmetry categories for orthogonal gauge theories

Begin with pure Spin$(2N)$ gauge theory in $d$ spacetime dimensions. This theory contains a split 2-group symmetry $\Gamma^{(1)} \rtimes \Gamma^{(0)}$ comprising of a $\Gamma^{(1)} = \mathbb{Z}_2^2$ or $\Gamma^{(1)} = \mathbb{Z}_4$ center 1-form symmetry acted upon by $\Gamma^{(0)} = \mathbb{Z}_2$ outer-automorphism 0-form symmetry. From our discussion in the previous subsection, we learn that pure Spin$(2N)$ gauge theory in $d$ spacetime dimensions carries a symmetry $(d-1)$-category

$$\boxed{\mathcal{C}_{\text{Spin}(2N)} = \mathcal{T}_{d-1}\left(\Gamma^{(1)} \rtimes \mathbb{Z}_2^{(0)}\right),} \tag{277}$$

---

[19]Note that Karoubi completion is an essential condition in the definition of fusion higher-categories [49]. Thus $(d-1)$-Vec$^0(\Gamma)$ is not a fusion $(d-1)$-category, but $(d-1)$-Vec$(\Gamma)$ is a fusion $(d-1)$-category.

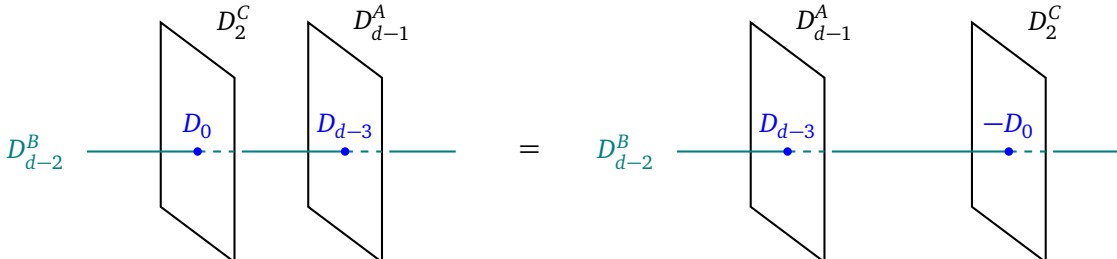

Figure 17: Symmetry Fractionalization in general dimensions: the string diagram for the anomaly (cocycle) $A_1 \cup B_2 \cup C_{d-2} \subset \omega_{2N}$. Denoting the $p$-dimensional topological defects associated to the symmetries with background field $B_{d-p}$ as $D_p^B$, the figure shows the string diagram that encodes the non-trivial mixed anomaly. The intersection of $D_{d-2}$ with $D_{d-1}$ corresponds to a codimension 1 operator in $D_{d-2}$, and gives rise to a 0-form symmetry generator on that defect. The statement of the anomaly is, that the local operator $D_0$ that sits at the intersection of $D_{d-2}$ and $D_2$ is charged under this 0-form symmetry.

arising from the presence of $\Gamma^{(1)} \rtimes \mathbb{Z}_2^{(0)}$ 2-group symmetry.

Gauging the full $\Gamma^{(1)} \rtimes \mathbb{Z}_2^{(0)}$ 2-group symmetry takes us to the pure $d$-dimensional PO($2N$) gauge theory. From the arguments of the companion paper [1], we can describe the associated symmetry $(d-1)$-category by

$$\mathcal{C}_{\text{PO}(2N)} = (\mathsf{d}-1)\text{-Rep}_{\mathcal{T}_{d-1}}\left(\Gamma^{(1)} \rtimes \mathbb{Z}_2^{(0)}\right), \tag{278}$$

which is the $(d-1)$-category formed by $(d-1)$-representations of the 2-group $\Gamma^{(1)} \rtimes \mathbb{Z}_2^{(0)}$ valued in the target $(d-1)$-category $\mathcal{T}_{d-1}$ of $(d-1)$-dimensional TQFTs.

Gauging only the $\Gamma^{(1)}$ 1-form symmetry of pure Spin($2N$) gauge theory takes us to the pure $d$-dimensional PSO($2N$) gauge theory, which has to carry now a dual $(d-2)$-group symmetry $\Gamma^{(d-3)} \rtimes \mathbb{Z}_2^{(0)}$ comprising of a $(d-3)$-form symmetry group $\Gamma^{(d-3)} = \widehat{\Gamma^{(1)}}$ acted upon by the residual $\mathbb{Z}_2^{(0)}$ 0-form symmetry. From our discussion in the previous subsection, we learn that the associated symmetry $(d-1)$-category is

$$\mathcal{C}_{\text{PSO}(2N)} = \mathcal{T}_{d-1}\left(\Gamma^{(d-3)} \rtimes \mathbb{Z}_2^{(0)}\right). \tag{279}$$

Gauging the full $\Gamma^{(d-3)} \rtimes \mathbb{Z}_2^{(0)}$ $(d-2)$-group symmetry takes us to the pure $d$-dimensional Pin$^+$($2N$) gauge theory. Thus the associated symmetry $(d-1)$-category is

$$\mathcal{C}_{\text{Pin}^+(2N)} = (\mathsf{d}-1)\text{-Rep}_{\mathcal{T}_{d-1}}\left(\Gamma^{(d-3)} \rtimes \mathbb{Z}_2^{(0)}\right), \tag{280}$$

which is the $(d-1)$-category formed by $(d-1)$-representations of the $(d-2)$-group $\Gamma^{(d-3)} \rtimes \mathbb{Z}_2^{(0)}$ valued in the target $(d-1)$-category $\mathcal{T}_{d-1}$ of $(d-1)$-dimensional TQFTs.

Gauging the $\mathbb{Z}_2$ subgroup of $\Gamma^{(1)}$ 1-form symmetry not acted upon by $\mathbb{Z}_2^{(0)}$ 0-form symmetry of pure Spin($2N$) gauge theory takes us to the pure $d$-dimensional SO($2N$) gauge theory, which has to carry now a dual $(d-2)$-group symmetry $\mathbb{Z}_2^{(d-3)} \times \mathbb{Z}_2^{(1)} \times \mathbb{Z}_2^{(0)}$ decomposing into a direct product of $\mathbb{Z}_2^{(d-3)}$ $(d-3)$-form symmetry group, $\mathbb{Z}_2^{(1)}$ 1-form symmetry group and $\mathbb{Z}_2^{(0)}$ 0-form symmetry group. Additionally these higher-form symmetries carry a mixed 't Hooft anomaly which differs for the SO($4N$) and the SO($4N + 2$) theories. For SO($4N$), the anomaly is

$$A^* \omega_{4N} = A_1 \cup B_2 \cup C_{d-2}, \tag{281}$$

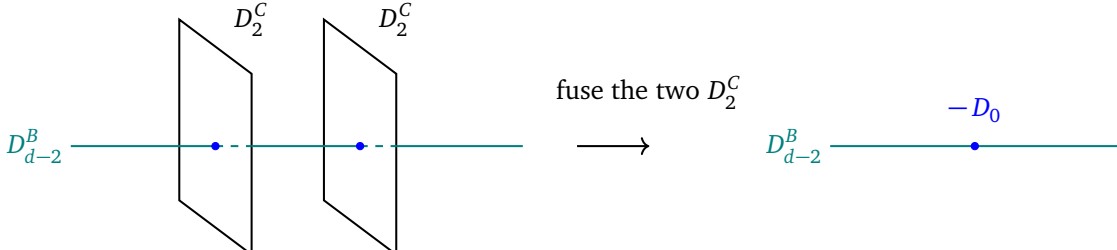

Figure 18: Symmetry Fractionalization in general dimensions: the string diagram for the anomaly (cocyle) $B_2 \cup \mathrm{Bock}(C_{d-2})$. Fusing the two surfaces $D_2^C$ along $D_{d-2}^B$ produces a local operator $-D_0$ on $D_{d-2}^B$.

where $A_1$ is the background gauge field for $\mathbb{Z}_2^{(0)}$, $B_2$ is the background gauge field for $\mathbb{Z}_2^{(1)}$ and $C_{d-2}$ is the background gauge field for $\mathbb{Z}_2^{(d-3)}$, while for SO($4N+2$), the anomaly is

$$A^* \omega_{4N+2} = A_1 \cup B_2 \cup C_{d-2} + \mathrm{Bock}(B_2) \cup C_{d-2}. \tag{282}$$

The first term in these anomalies corresponds to the string diagrams shown in figure 17. For the other term appearing in (282), the string diagram is given by figure (18).

Thus, from our discussion in the previous subsection, we learn that the associated symmetry $(d-1)$-category is

$$\boxed{\mathcal{C}_{\mathrm{SO}(2N)} = \mathcal{T}_{d-1}^{\omega_{2N}}\left(\mathbb{Z}_2^{(d-3)} \times \mathbb{Z}_2^{(1)} \times \mathbb{Z}_2^{(0)}\right).} \tag{283}$$

The 't Hooft anomaly discussed above can be derived by expressing the 2-group symmetry of the Spin($2N$) gauge theory in terms of background gauge fields for $\Gamma^{(1)}$ and $\mathbb{Z}_2^{(0)}$. For Spin($4N$), these backgrounds satisfy

$$\delta C_2 = A_1 \cup B_2, \tag{284}$$

while for Spin($4N+2$), these backgrounds satisfy

$$\delta C_2 = A_1 \cup B_2 + \mathrm{Bock}(B_2), \tag{285}$$

where $A_1$ is the background gauge field for $\mathbb{Z}_2^{(0)}$ 0-form symmetry, $C_2$ is the background gauge field for the $\mathbb{Z}_2$ subgroup of $\Gamma^{(1)}$ 1-form symmetry not acted upon by $\mathbb{Z}_2^{(0)}$ and $B_2$ is the background gauge field for the remaining $\Gamma^{(1)}/\mathbb{Z}_2$ 1-form symmetry. As explained in [50], upon gauging the $\mathbb{Z}_2$ 1-form symmetry associated to $C_2$, we obtain the expressions for the 't Hooft anomalies described above.

Gauging the $\mathbb{Z}_2^{(0)}$ 0-form symmetry of pure SO($2N$) gauge theory takes us to the pure $d$-dimensional O($2N$) gauge theory, which has to carry now a dual $(d-1)$-group symmetry $\mathbb{Z}_2^{(d-2)} \times \mathbb{Z}_2^{(d-3)} \times \mathbb{Z}_2^{(1)}$ decomposing into a direct product of $\mathbb{Z}_2^{(d-2)}$ $(d-2)$-form symmetry group, $\mathbb{Z}_2^{(d-3)}$ $(d-3)$-form symmetry group and $\mathbb{Z}_2^{(1)}$ 1-form symmetry group. There is a non-trivial Postnikov class such that the background fields satisfy

$$\delta A_{d-1} = B_2 \cup C_{d-2}, \tag{286}$$

where $A_{d-1}$ is the background gauge field for $\mathbb{Z}_2^{(d-2)}$, $B_2$ is the background gauge field for $\mathbb{Z}_2^{(1)}$ and $C_{d-2}$ is the background gauge field for $\mathbb{Z}_2^{(d-3)}$. This comes from dualizing the first term on the right hand side in the anomalies (281) and (282). The second term on the right hand side

in the anomaly (282) descends to an additional mixed 't Hooft anomaly between the 1-form and $(d-3)$-form symmetries of the O$(4N+2)$ gauge theory given by

$$\omega = \text{Bock}(B_2) \cup C_{d-2}, \tag{287}$$

which is in $H^{d+1}(B\Gamma, U(1))$. Thus, the associated symmetry $(d-1)$-category for O$(4N)$ is

$$\mathcal{C}_{\text{O}(4N)} = \mathcal{T}_{d-1}\left(\mathbb{Z}_2^{(d-2)} \times \mathbb{Z}_2^{(d-3)} \times \mathbb{Z}_2^{(1)}; \Theta = B_2 \cup C_{d-2}\right), \tag{288}$$

where the expression in the bracket denotes the $(d-1)$-group with the Postnikov class $\Theta$. On the other hand, the associated symmetry $(d-1)$-category for O$(4N+2)$ is

$$\mathcal{C}_{\text{O}(4N+2)} = \mathcal{T}_{d-1}^{\text{Bock}(B_2) \cup C_{d-2}}\left(\mathbb{Z}_2^{(d-2)} \times \mathbb{Z}_2^{(d-3)} \times \mathbb{Z}_2^{(1)}; \Theta = B_2 \cup C_{d-2}\right), \tag{289}$$

where the superscript denotes an additional anomaly on the $(d-1)$-group.

Finally, note that the $(d-1)$-group symmetry of the O$(4N)$ gauge theory can be gauged but the gauging of the $(d-1)$-group symmetry of the O$(4N+2)$ gauge theory is obstructed by the 't Hooft anomaly $\omega$ described above. After performing this gauging operation on O$(4N)$, we obtain the Ss$(4N)$ gauge theory, whose associated symmetry $(d-1)$-category can then be expressed as

$$\mathcal{C}_{\text{Ss}(4N)} = (\text{d}-1)\text{-Rep}_{\mathcal{T}_{d-1}}\left(\mathbb{Z}_2^{(d-2)} \times \mathbb{Z}_2^{(d-3)} \times \mathbb{Z}_2^{(1)}; \Theta = B_2 \cup C_{d-2}\right). \tag{290}$$

Figure 19 and 20 illustrate the webs for $d$-dimensional $\mathfrak{so}(4N)$ and $\mathfrak{so}(4N+2)$ theories respectively.

# 7 Conclusions and Outlook

In this paper we developed the gauging of 0-form symmetries in the presence of categorical symmetries, which act on QFTs in 3d.

The overarching mathematical structure that captures all aspects of such symmetry categories are fusion 2-categories. In terms of the unified perspective on non-invertible symmetries detailed in the companion paper [1], the present paper analyses *(twisted) theta defects* in fusion 2-categories, which incorporate stacking of $G$-symmetric TQFTs prior to gauging. Initial aspects of this proposal were developed in [14], which was then fully generalized in [1].

Concretely, in this paper we developed the 0-form symmetry gauging, starting with subgroups of finite groups $G$, and then subsequent partial gaugings. Along the way we encountered a multitude of interesting effects, which we fleshed out in the context of concrete gaugings. The starting point are categories 2-Vec$(G)$, which is the symmetry category (or subcategory) of a theory with $G$ 0-form symmetry. The multitude of symmetry categories that can be obtained by gauging a subset of the 0-form symmetry span the categorical symmetry web. We provided a detailed analysis for the groups $G = \mathbb{Z}_4, \mathbb{Z}_2 \times \mathbb{Z}_2, S_3, D_8$, which all correspond to symmetries of 3d gauge theories. The richest class of examples are those of $\mathfrak{so}(2N)$ theories, that form a $D_8$-categorical symmetry web.

In the process of spinning the categorical webs, we observe multitudes of interesting new effects. First and foremost the symmetry fractionalization, which is closely tied to the presence of 't Hooft anomalies in the gauge theory description, but which we formulate in terms of the topological defects that comprise the symmetry category. Gauging in the presence of non-invertible defects turns out to be rather subtle, as is duly demonstrated in the symmetry webs detailed in this paper, most notably the $D_8$-web.

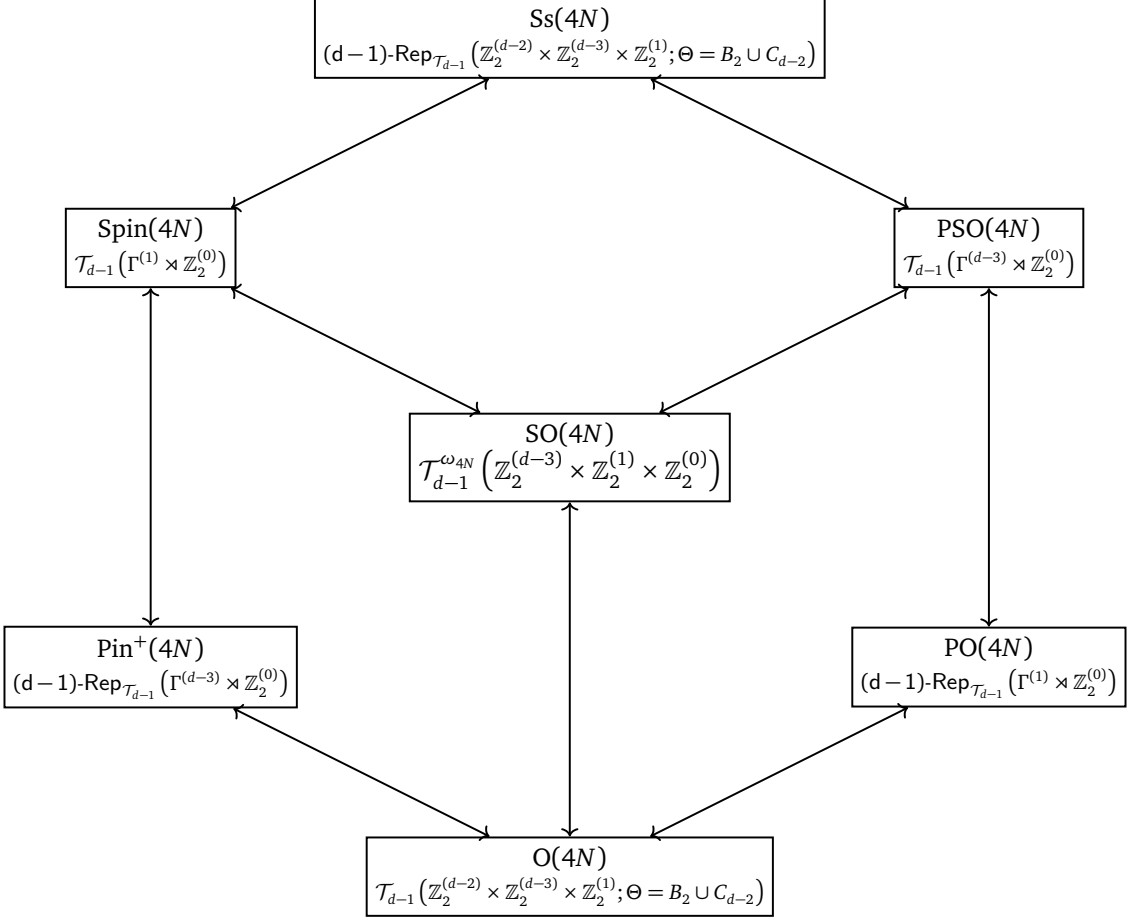

Figure 19: Categorical symmetry web in $d$ dimensions for $\mathfrak{so}(4N)$. The arrows indicate the gauging of simple subgroups of the invertible part of the category.

This paper develops only one aspect of the general classification program in [1]. Concretely, even in the realm of gauging 0-form symmetries, an important question remains, namely the development of a good computational tool to gauge, when there are no surfaces left invariant by a non-trivial subgroup of the 0-form group that is being gauged. An example of such a gauging in the presence of non-invertibles is shown in figure 1 in the gauging from $Ss(4N)$ to $\mathrm{Spin}(4N)$, with the surface being $D_2^{(a,a^3)}$ of $\mathcal{C}_{D_8/\mathbb{Z}_2^x}$. There are numerous generalizations to be explored following the proposal in [1], including the study of twisted theta defects in higher-form and higher-group gaugings, and higher-dimensional extensions, as initiated in section 6.

## Acknowledgments

We thank Thibault Décoppet, Clement Delcamp and Jingxiang Wu for discussions.

**Funding information** Part of this work has been carried out while two of the authors (LB, SSN) were at the Aspen Center for Physics, which is supported by National Science Foundation grant PHY-1607611. This work is supported by the European Union's Horizon 2020 Framework through the ERC grants 682608 (LB, SSN) and 787185 (LB). SSN acknowledges support through the Simons Foundation Collaboration on "Special Holonomy in Geometry, Analysis, and Physics", Award ID: 724073, Schäfer-Nameki. AT is supported by the Swedish Research Council (VR) through grants number 2019-04736 and 2020-00214.

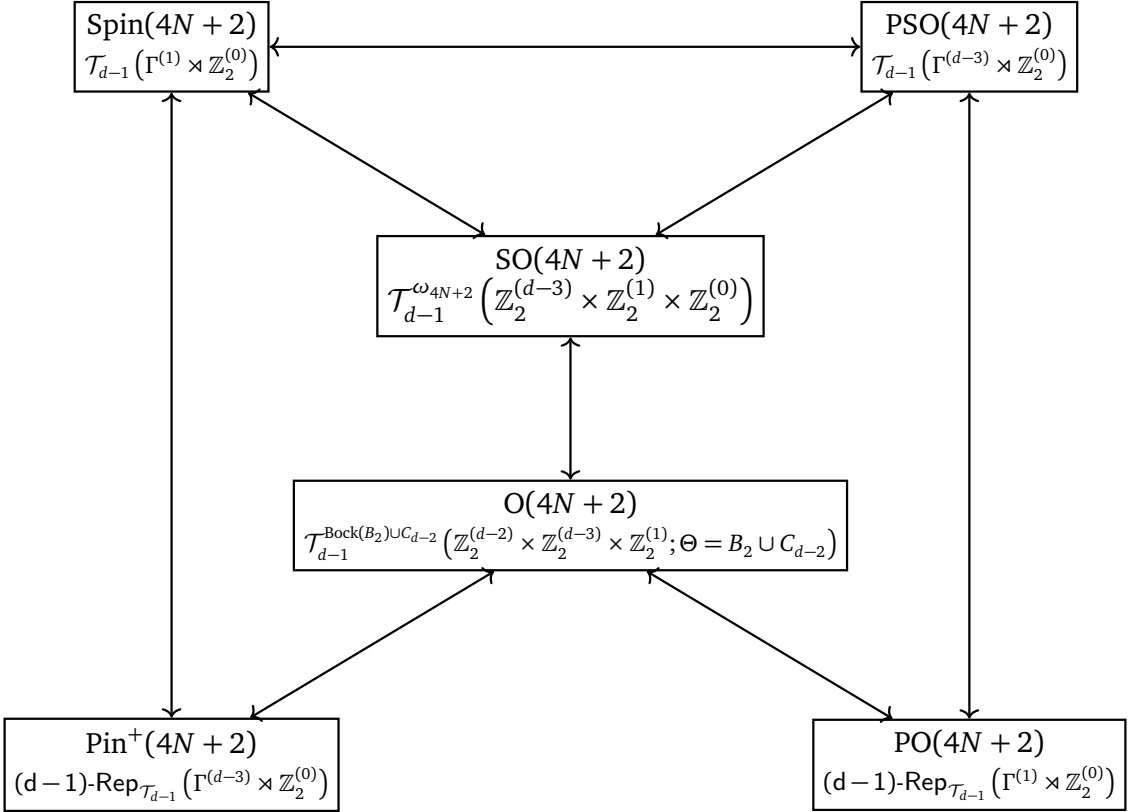

Figure 20: Categorical symmetry web in $d$ dimensions for $\mathfrak{so}(4N + 2)$. The arrows indicate the gauging of simple subgroups of the invertible part of the category.

## A   Bimodules and 2-representations

### A.1   Gauging a normal subgroup $H \subseteq G$

In this section we review gauging a normal subgroup $H \lhd G$ following the discussion in [15]. We assume $H$ to be a finite Abelian group. The fact that $H$ is normal in $G$ means that there is a short exact sequence

$$1 \to H \to G \to K \equiv G/H \to 1 \,. \tag{A.1}$$

The extension is characterized by the group cohomology class

$$\epsilon \in H^2(K, H) \,. \tag{A.2}$$

As a set, we can write $G$ as pairs $(h, k)$ with $h \in H$ and $k \in K$ with the product given by

$$(h, k) \times (h', k') = (hh' \epsilon(k, k'), kk') \,, \tag{A.3}$$

where we assume there is no action of $K$ on $H$ by conjugation in $G$.

The gauging of $H$ corresponds to choosing the algebra-object

$$A_H = \bigoplus_{h \in H} D_2^{(h)} \,. \tag{A.4}$$

What we want to determine first are the topological surfaces of the theory $\mathfrak{T}/H$, i.e. objects in the symmetry category $\mathcal{C}_{\mathfrak{T}/H}$. We can define a topological surface in $\mathfrak{T}/H$ as a general surface in $\mathfrak{T}$

$$D_2 = \bigoplus_{g \in G} n_g D_2^{(g)} \,, \tag{A.5}$$

satisfying consistency conditions that guarantee it remains topological in the presence of the networks of algebra objects. This means the algebra object should be able to end on $D_2$ from the left, from the right, and that the left and right actions commute.

Concretely, instructions on how $A_H$ can end on $D_2$ from the left are implemented via a 1-morphism

$$l \in \mathrm{Hom}\,(A_H \otimes D_2, D_2)\,. \tag{A.6}$$

Similarly, instructions of how to end the algebra from the right are given by

$$r \in \mathrm{Hom}\,(D_2 \otimes A_H, D_2)\,. \tag{A.7}$$

To work out the bimodules explicitly, it is more convenient to work with the components of the above 1-morphisms

$$\begin{aligned}
l_{h,g}: \quad & D_2^{(h)} \otimes n_g D_2^{(g)} \to n_{hg} D_2^{(hg)}\,, \\
r_{g,h}: \quad & n_g D_2^{(g)} \otimes D_2^{(h)} \to n_{gh} D_2^{(gh)}\,,
\end{aligned} \tag{A.8}$$

involving a specific $D_2^{(h)}$ inside $A_H$ and $n_g D_2^{(g)}$ inside $D_2$ in (A.5). The 1-morphisms must satisfy compatibility conditions which, as we are in a 2-category, are not equality but hold up to 2-isomorphisms. The first one involves the unit axiom of $A_H$ and tells us that there must be 2-isomorphisms

$$\begin{aligned}
u_g^l: \qquad & D_1^{(n_g D_2^{(g)};\mathrm{id})} \Rightarrow l_{\mathrm{id},g}\,, \\
u_g^r: \qquad & D_1^{(n_g D_2^{(g)};\mathrm{id})} \Rightarrow r_{g,\mathrm{id}}\,,
\end{aligned} \tag{A.9}$$

where $D_1^{(n_g D_2^{(g)};\mathrm{id})}$ denotes the identity 1-morphism on $n_g D_2^{(g)}$. Concretely, this means that ending $D_2^{(\mathrm{id})}$ on $n_g D_2^{(g)}$ should be equivalent to doing nothing. The second condition involves the product axiom of $A_H$ and tells us that there must be 2-isomorphisms

$$\begin{aligned}
\phi_{g;h,h'}^l: \qquad & l_{hh',g} \Rightarrow l_{h,h'g} \otimes l_{h',g}\,, \\
\phi_{g;h,h'}^r: \qquad & r_{g,hh'} \Rightarrow r_{gh,h'} \otimes r_{g,h}\,.
\end{aligned} \tag{A.10}$$

These 2-isomorphisms implement the equivalence of ending $D_2^{(h)}$ and then $D_2^{(h')}$ on $n_g D_2^{(g)}$ or ending $D_2^{(hh')}$ directly (see figure 21). These conditions turn $D_2$ into a left $A_H$-module and a right $A_H$-module separately.

If we finally impose that we can also commute the left and right $A_H$ actions, this turns $D_2$ into a $(A_H, A_H)$-bimodule. This means that we should have a 2-isomorphism (see figure 22)

$$\phi_{g;hh'}^{lr}: \qquad l_{h,gh'} \otimes r_{g,h'} \Rightarrow r_{hg,h'} \otimes l_{h,g}\,. \tag{A.11}$$

The 2-isomorphisms themselves have to satisfy consistency conditions. The first one is related to the unit axiom and states

$$\phi_{g;h,\mathrm{id}}^l = D_0^{(l_{h,g};\mathrm{id})} \otimes u_g^l\,, \quad \phi_{g;\mathrm{id},h}^l = u_{hg}^l \otimes D_0^{(l_{h,g};\mathrm{id})}\,, \tag{A.12}$$

where $D_0^{(l_{h,g};\mathrm{id})}$ denotes the identity local operator on the 1-morphism $l_{h,g}$. One can write down similar conditions for the corresponding right 2-isomorphisms. There is also an associativity condition that states that

$$\phi_{g;h_1 h_2,h_3}^l \circ (\phi_{h_3 g;h_1,h_2}^l \otimes D_0^{(l_{h_3,g};\mathrm{id})}) = \phi_{g;h_1,h_2 h_3}^l \circ (D_0^{(l_{h_1,h_2 h_3 g};\mathrm{id})} \otimes \phi_{g;h_2,h_3}^l)\,. \tag{A.13}$$

One can write down a similar associativity condition for the corresponding right 2-isomorphisms.

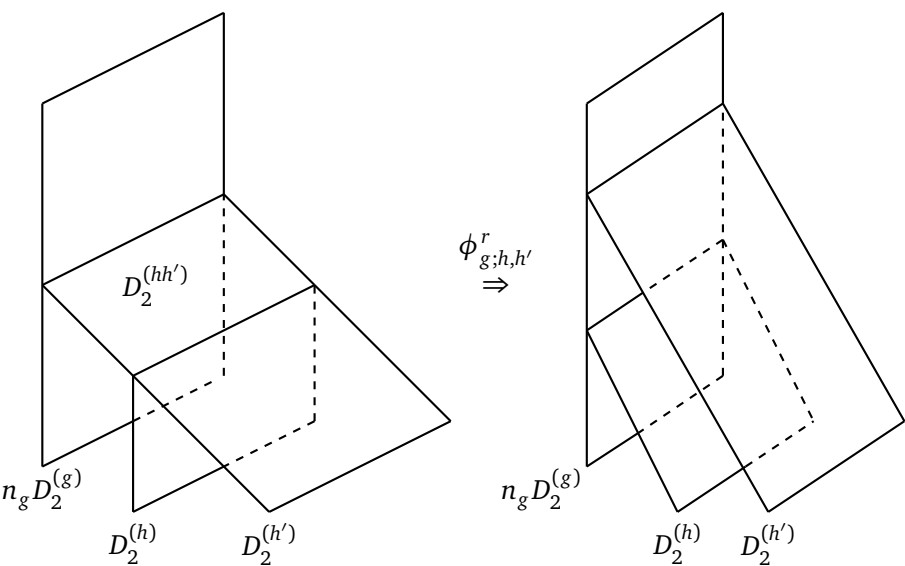

Figure 21: Right-module consistency condition: the 2-isomorphism $\phi^r_{g;h,h'}$ implements the equivalence of ending $D_2^h$ and $D_2^{h'}$ or $D_2^{hh'}$ on $n_g D_2^{(g)}$. An analogous picture can be drawn for $\phi^l_{g;h,h'}$.

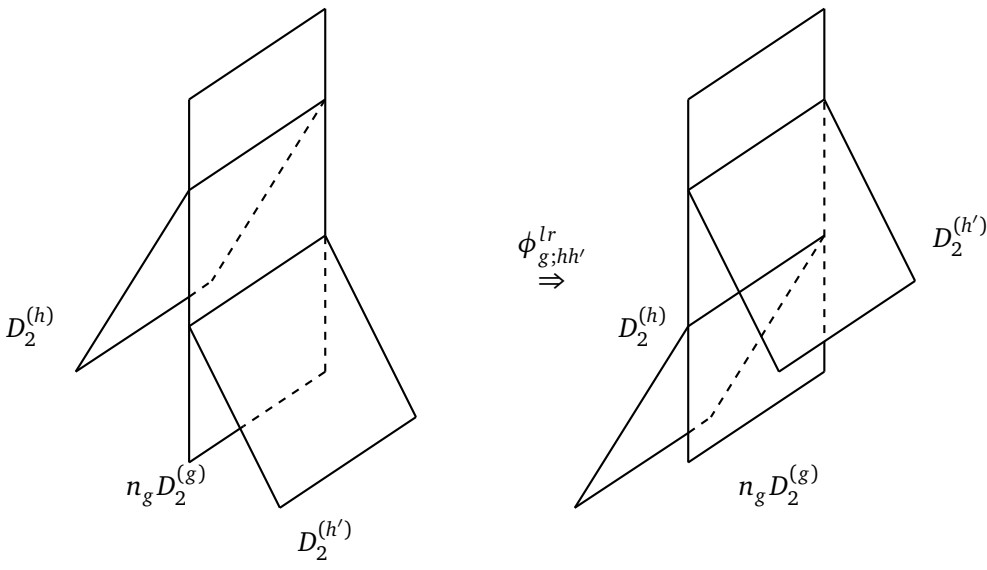

Figure 22: Compatibility condition between the left and right actions on $n_g D_2^{(g)}$, whose commutativity is up to the 2-isomorphism $\phi^{lr}_{g;hh'}$.

Let us now show how to solve the above conditions. First of all, considering (A.10) for $hh' = \mathrm{id}$ gives us a 2-isomorphism

$$l_{\mathrm{id},g} \Rightarrow l_{h^{-1},hg} \otimes l_{h,g}\,, \tag{A.14}$$

which we can compose with $u_g^l$ to obtain a 2-isomorphism

$$l_{h^{-1},hg} \otimes l_{h,g} \Rightarrow D_1^{(n_g D_2^{(g)};\mathrm{id})}\,. \tag{A.15}$$

This implies that $l_{h,g}$ is weakly invertible (i.e. up to 2-isomorphisms), with $(l_{h,g})^{-1} := l_{h^{-1},hg}$. Similarly, from (A.10) we also obtain that $r_{g,h}$ is weakly invertible with $(r_{g,h})^{-1} := r_{gh,h^{-1}}$. At this point it is convenient to write an element $g \in G$ as a pair $(h,k)$ and consider again (A.10) in the special case $g = (\mathrm{id}, k)$. In the following we will use an abuse of notation for which $(\mathrm{id}, k) \equiv k$. This gives a 2-isomorphism from

$$l_{hh',k} \Rightarrow l_{h,h'k} \otimes l_{h',k}\,, \tag{A.16}$$

which implies that the general component $l_{h,g}$ can be completely determined in terms of $l_{h,k}$ via

$$l_{h,h'k} \Rightarrow l_{hh',k} \otimes (l_{h',k})^{-1}\,. \tag{A.17}$$

We can do an analogous discussion for the right 1-morphisms, that can be completely determined in terms of $r_{k,h}$. To summarize, we can restrict to 1-morphism components

$$\begin{aligned}
l_{h,k}: & \quad D_2^{(h,\mathrm{id})} \otimes n_k D_2^{(\mathrm{id},k)} \to n_{(h,k)} D_2^{(h,k)}\,, \\
r_{k,h}: & \quad n_k D_2^{(\mathrm{id},k)} \otimes D_2^{(h,\mathrm{id})} \to n_{(h,k)} D_2^{(h,k)}\,.
\end{aligned} \tag{A.18}$$

In particular, this allows us to identify

$$n_{(h,k)} D_2^{(h,k)} \cong n_k D_2^{(k)}\,, \quad \forall\, h \in H\,, \tag{A.19}$$

with $n_{(h,k)} \equiv n_{(\mathrm{id},k)} := n_k \; \forall h \in H$. We can now form the combination

$$\rho_{h,k} := (r_{k,h})^{-1} \circ l_{h,k} : n_k D_2^{(k)} \to n_k D_2^{(k)}\,, \tag{A.20}$$

which concretely represents a topological junction line created by the $D_2^{(h)}$ surface going through $n_k D_2^{(k)}$, see figure 23.

We can now rewrite the conditions (A.9) and (A.10) in terms of the combination 1-morphism $\rho_{h,k}$

$$\begin{aligned}
u_k: & \quad D_1^{(n_k D_2^{(k)};\mathrm{id})} \Rightarrow \rho_{\mathrm{id},k}\,, \\
\phi_{h,h';k}: & \quad \rho_{hh',k} \Rightarrow \rho_{h,k} \circ \rho_{h',k}\,.
\end{aligned} \tag{A.21}$$

In terms of $\rho_{h,k}$, we can reformulate the conditions (A.12) and (A.13) as

$$\begin{aligned}
\phi_{h,\mathrm{id};k} &= D_0^{(\rho_{h,k};\mathrm{id})} \otimes u_k\,, \\
\phi_{\mathrm{id},h;k} &= u_k \otimes D_0^{(\rho_{h,k};\mathrm{id})}\,,
\end{aligned} \tag{A.22}$$

and

$$\phi_{h_1 h_2,h_3;k} \circ (\phi_{h_1,h_2;k} \otimes D_0^{(\rho_{h_3,k};\mathrm{id})}) = \phi_{h_1,h_2 h_3;k} \circ (D_0^{(\rho_{h_1,k};\mathrm{id})} \otimes \phi_{h_2,h_3;k})\,, \tag{A.23}$$

respectively.

To summarize, we find that surfaces in $\mathcal{T}/H$ are labelled by the following data

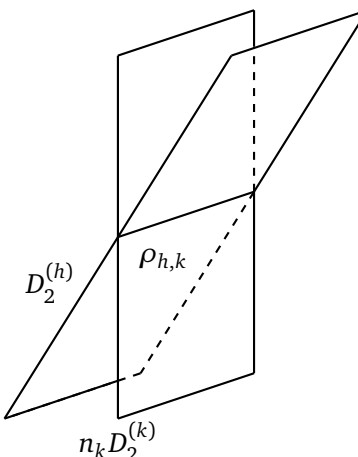

Figure 23: The 1-morphism $\rho_{h,k}$ represents the line singled out by the $D_2^{(h)}$ surface crossing $n_k D_2^{(k)}$.

- An object $n_k D_2^{(k)}$ specified by an integer $n_k$ for every $k \in K$;

- a 1-morphisms $\rho_{h,k} \in \mathrm{Hom}(D_2^{(h)} \otimes n_k D_2^{(k)}, n_k D_2^{(k)} \otimes D_2^{(h)})$;

- 2-isomorphisms $u_k$ and $\phi_{h,h';k}$.

As discussed in [15, 51] the 2-isomorphisms are completely determined by a 2-cocycle $c$ in

$$c \in H^2(H, U(1)^{n_k}).\tag{A.24}$$

This is the data defining a symmetry category which at the level of objects is

$$2\text{-Vec}(K) \boxtimes 2\text{-Rep}(H).\tag{A.25}$$

In particular, we can label simple objects by

- an element $k \in K$;

- a subgroup $J \subseteq H$;

- a 2-cocycle $c \in H^2(J, U(1))$.

We denote the corresponding topological surfaces by

$$D_2^{(kH/J,c)} \in \mathcal{C}_{\mathfrak{T}/H}.\tag{A.26}$$

**1-morphisms.** Now let us consider how to determine 1-morphisms. Similarly to what we discussed for objects, 1-morphisms in $\mathcal{C}_{\mathfrak{T}/H}$ are given by 1-morphisms in $\mathcal{C}_{\mathfrak{T}}$ that are compatible with the presence of networks of $A_H$.

For simplicity, let focus on determining 1-endomorphisms of the identity simple object $D_2^{(\mathrm{id})}$. These descend from the simple 1-endomorphism of $D_2^{(\mathrm{id})}$ in $\mathcal{C}_{\mathfrak{T}}$, which is simply $D_1^{(\mathrm{id})}$. In order for $D_1^{(\mathrm{id})}$ to remain topological in the presence of $A_H$, we need to impose consistency conditions which can be expressed in terms of the local operators sitting at the junction of $D_1^{(\mathrm{id})}$ and the 1-morphism $\rho_h$ we defined in (A.20) (here we are suppressing the index $k$ since we

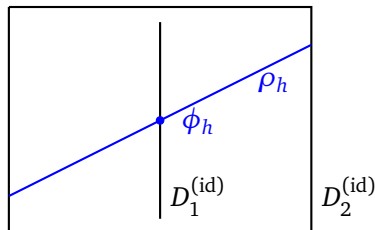

Figure 24: 1-morphisms in $\mathcal{C}_{\mathfrak{T}/H}$ must be compatible with the presence of the algebra object. In this picture one should think of a $D_2^{(h)}$ surface going through the page along $\rho_h$.

are looking at $k = \text{id}$). We recall that this is the junction line which is singled out by a surface $D_2^{(h)}$ inside $A_H$ crossing $D_2^{(\text{id})}$. Let us denote this local operator by

$$\phi_h \in \text{Hom}(\rho_h \otimes D_1^{(\text{id})}, D_1^{(\text{id})} \otimes \rho_h), \tag{A.27}$$

as shown in figure 24.

Now consider two surfaces, $D_2^{(h_1)}$ and $D_2^{(h_1)}$, crossing $D_2^{(\text{id})}$. This will single out two local operators $\phi_{h_1}$ and $\phi_{h_2}$ sitting at the junction of $D_1^{(\text{id})}$ with $\rho_{h_1}$ and $\rho_{h_2}$ respectively. From the fusion of surfaces, which follows the $H$ multiplication,

$$D_2^{(h_1)} \otimes D_2^{(h_2)} = D_2^{(h_1 h_2)}, \tag{A.28}$$

we learn that the junction operators must satisfy

$$\phi_{h_1 h_2} = \phi_{h_1} \circ \phi_{h_2}. \tag{A.29}$$

In summary, we have learned that a 1-endomorphism of $D_2^{(\text{id})}$ is given by $D_1^{(\text{id})}$ with a choice of phase $\phi_h$ satisfying (A.29). Then we have recovered that 1-endomorphisms of $D_2^{(\text{id})}$ form the fusion category $\text{Rep}(H)$. Let us denote them by $D_1^{(\chi)}$, with $\chi \in \widehat{H}$.

This can be easily generalized to 1-morphisms between $D_2^{(\text{id})}$ and the SPT defect $D_2^{(\text{id},c)}$, with $c \in H^2(H, U(1))$ in $\mathcal{C}_{\mathfrak{T}/H}$. In this instance, (A.29) is relaxed to

$$\phi_{h_1 h_2} = c(h_1, h_2) \phi_{h_1} \circ \phi_{h_2}, \tag{A.30}$$

due to an anomaly inflow mechanism (see [15]). This implies that these morphisms form the fusion category $\text{Rep}^c(H)$ of projective representations of $H$.

**Non-trivial extension.** This is not the end of the story if the subgroup $H$ we are gauging comes from a non-trivial extension (A.1). In particular, consider a determined 1-endomorphism $D_1^{(\chi)}$ of $D_2^{(\text{id})}$, which we defined as $D_1^{(\text{id})}$ in $\mathcal{C}_{\mathfrak{T}}$ with a choice of phase $\phi_h$ coming from its intersection with $\rho_h$. Now consider two surfaces *not* inside the algebra object $A_H$, say $D_2^{(k_1)}$ and $D_2^{(k_2)}$, with $k_1, k_2 \in K$, crossing $D_1^{(\chi)}$. Their fusion in $\mathcal{C}_{\mathfrak{T}}$ is given by

$$D_2^{(k_1)} \otimes D_2^{(k_2)} = D_2^{(\epsilon(k_1, k_2), k_1 k_2)}. \tag{A.31}$$

In particular then fusing $D_2^{(k_1)}$ and $D_2^{(k_2)}$ produces an object $D_2^{(\tilde{h})}$ with $\tilde{h} := \epsilon(k_1, k_2) \in H$ inside the algebra $A_H$. If we perform this fusion of surfaces in 2-Vec$(K)$ along $D_1^{(\chi)}$, we then obtain a possibly non-trivial (depending on the $\phi_h$ entering the definition of $D_1^{(\chi)}$) local operator $\phi_{\tilde{h}}$ living on the line (see figure 25). This phenomenon represents a fractionalization of the $K$

**Figure 25:** The 0-form symmetry $K$ fractionalizes on the line generating the 1-form symmetry $\widehat{H}$ dual to $H$. This manifests itself in a presence of a phase $\phi_{\tilde{h}}$ upon fusing two surfaces in 2Vec$K$ along the line.

0-form symmetry on the 1-endomorphisms of $D_2^{(\mathrm{id})}$, namely the lines generating the 1-form symmetry dual to $H$.

Let us elucidate the above discussion in the concrete example of gauging $\mathbb{Z}_2^V$ in $\mathbb{Z}_4$ that we discussed in the main text (see 2.4). The first step is determining the 1-endomorphisms of $D_2^{(\mathrm{id})}$ in $\mathcal{C}_{\mathfrak{T}/\mathbb{Z}_2^V}$. For this we need to consider local operators $\phi_V$ at the junction of $D_1^{(\mathrm{id})}$ and $\rho_V$. These have to satisfy $(\phi_V)^2 = 1$. The non-trivial 1-endomorphism of $D_2^{(\mathrm{id})}$, which we denote by $D_1^{(V)}$ in $\mathcal{C}_{\mathfrak{T}/\mathbb{Z}_2^V}$, is such that $\phi_V = -1$. Due to the non-trivial extension

$$1 \to \mathbb{Z}_2^V \to \mathbb{Z}_4 \to \mathbb{Z}_2^S \to 1 \,, \tag{A.32}$$

however, we also have to worry about configurations where we fuse two $D_2^{(S)}$ surfaces crossing $D_1^{(V)}$. In particular, since $D_2^{(S)} \otimes D_2^{(S)} = D_2^{(V)}$, if we use our definition of $D_1^{(V)}$ as $D_1^{(\mathrm{id})}$ in $\mathcal{C}_{\mathfrak{T}}$ with the choice $\phi_V = -1$, we learn that this process produces a non-trivial $-1$ phase. This represents the fractionalization of $\mathbb{Z}_2^S$ on $D_1^{(V)}$ that we discussed in the main text.

## A.2 Gauging the remaining $G/H$ subgroup

We now consider gauging the remaining 0-form symmetry $K = G/H$ in $\mathfrak{T}/H := \mathfrak{T}'$. First of all, recall that a generic object in the symmetry category of $\mathfrak{T}/H$ can be written as

$$D_2 = \bigoplus_{(k,H/J)} m_{(k,H/J)} D_2^{(k,H/J)} \,, \tag{A.33}$$

where $k \in K$ is associated to a simple object in 2-Vec$(K)$ while $H/J$ specifies a simple object in 2-Rep$(H)$ (here we are suppressing the additional $c$ data for simplicity).

The gauging of $K$ in $\mathcal{C}_{\mathfrak{T}/H}$ is implemented via the algebra object

$$A_K = \bigoplus_{k \in K} D_2^{(k)} \,. \tag{A.34}$$

A topological surface in $\mathfrak{T}'/K$ is then defined as usual as generic surface (A.33) compatible with the presence of the algebra objects. We first define left and right 1-morphisms specifiying how the algebra object can end on $D_2$ from left and right respectively. These are given in terms of components

$$\begin{aligned} l_{k,(p,H/J)} : \quad & D_2^{(k)} \otimes m_{(p,H/J)} D_2^{(p,H/J)} \to m_{(kp,H/J)} D_2^{(kp,H/J)} \,, \\ r_{(p,H/J),k} : \quad & m_{(p,H/J)} D_2^{(p,H/J)} \otimes D_2^{(k)} \to m_{(pk,H/J)} D_2^{(pk,H/J)} \,. \end{aligned} \tag{A.35}$$

This left and right 1-morphisms must satisfy conditions analogous to the ones described in the previous subsection. In particular, we have 2-isomorphisms implementing the left and right

product conditions

$$l_{kk',(p,H/J)} \Rightarrow l_{k,(k'p,H/J)} \otimes l_{k',(p,H/J)}, \tag{A.36}$$
$$r_{(p,H/J),kk'} \Rightarrow r_{(pk,H/J),k'} \otimes r_{(p,H/J),k},$$

as well as the commutativity condition

$$l_{k,(pk',H/J)} \otimes r_{(p,H/J),k'} \Rightarrow r_{(kp,H/J),k'} \otimes l_{k,(p,H/J)}. \tag{A.37}$$

Again from (A.36) it follows that $l$ and $r$ are weakly invertible. Moreover, considering (A.36) for the special pairs $(\mathrm{id}, H/J) := H/J$, corresponding to purely 2-representations of $H$, we obtain that we can determine the most generic components in terms of 1-morphism

$$l_{k,H/J} : D_2^{(k)} \otimes m_{(\mathrm{id},H/J)} D_2^{(\mathrm{id},H/J)} \to m_{(k,H/J)} D_2^{(k,H/J)}, \tag{A.38}$$
$$r_{H/J,k} : m_{(\mathrm{id},H/J)} D_2^{(\mathrm{id},H/J)} \otimes D_2^{(k)} \to m_{(k,H/J)} D_2^{(k,H/J)}.$$

This provides an identification

$$m_{(k,H/J)} D_2^{(k,H/J)} = m_{(\mathrm{id},H/J)} D_2^{(\mathrm{id},H/J)} := m_{H/J} D_2^{(H/J)}, \quad \forall\, k \in K. \tag{A.39}$$

We can now define as before the combination

$$\rho_{k,H/J} = (r_{H/J,k})^{-1} \otimes l_{k,H/J} \in \mathrm{Hom}(D_2^{(k)} \otimes m_{H/J} D_2^{(H/J)}, m_{H/J} D_2^{(H/J)} \otimes D_2^{(k)}), \tag{A.40}$$

in terms of which the condition (A.37) now tells us that there should be a 2-isomorphism

$$\phi_{k,k';H/J} : \quad \rho_{kk',H/J} \Rightarrow \rho_{k,H/J} \circ \rho_{k',H/J}. \tag{A.41}$$

Similarly, the 2-isomorphism conditions can be rewritten in terms of $\rho_{k,H/J}$. In particular, the associativity condition reads

$$\phi_{k_1 k_2, k_3; H/J} \circ (\phi_{k_1, k_2; H/J} \otimes D_0^{(\rho_{k_3; H/J}; \mathrm{id})}) = \phi_{k_1, k_2 k_3; H/J} \circ (D_0^{(\rho_{k_1, H/J}; \mathrm{id})} \otimes \phi_{k_2, k_3; H/J}). \tag{A.42}$$

Notice that differently to the previous subsection, here in general we will have more topological surfaces descending from a single $m_{H/J} D_2^{H/J}$. This is due to the fact that we have multiple choices for the 1-morphism $\rho_{k,H/J}$, coming from the fact that the symmetry category $\mathcal{C}_{\mathfrak{T}/H}$ has non-trivial lines descending from the gauging of $H$. An important comment is that if there is a symmetry fractionalization in $\mathcal{C}_{\mathfrak{T}/H}$ coming from a non-trivial group extension, this could render some topological surface $m_{H/J} D_2^{(H/J)}$ with a particular choice of $\rho_{k,H/J}$ inconsistent. This could happen because the bimodule condition fails already at the level of 1-morphism, meaning that there is no 2-isomorphism such that (A.41) is satisfied, or more subtly at the level of 2-isomorphisms, meaning that (A.42) fails. We have seen examples of this phenomenon discussing the gauging of $\mathbb{Z}_2^S$ in $\mathcal{C}_{\mathfrak{T}/\mathbb{Z}_2^V}$ in section 3, where in particular $D_2^{(\mathbb{Z}_2^V)}$ does not give rise to an object in $\mathcal{C}_{\mathfrak{T}/\mathbb{Z}_4}$ because the condition (A.41) is not satisfied.

# B Gauging full $G$: Symmetry category 2-Rep($G$)

In this appendix we apply the methods of this paper and of [14], for $G = \mathbb{Z}_2 \times \mathbb{Z}_2$ and $G = \mathbb{Z}_4$, to compute the 2-category $\mathcal{C}_{G/G}$ by performing a direct gauging of the full $G$ 0-form symmetry inside $\mathcal{C}_G$. The answer for general $G$ is

$$\mathcal{C}_{G/G} = 2\text{-Rep}(G), \tag{B.1}$$

and we describe the structure of the 2-category 2-Rep($G$) for $G = \mathbb{Z}_2 \times \mathbb{Z}_2$ and $G = \mathbb{Z}_4$ in great detail. The purpose of this appendix is to provide a consistency check for the computation of $\mathcal{C}_{G/G}$ performed using sequential gauging of $\mathbb{Z}_2$ inside $\mathcal{C}_{G/\mathbb{Z}_2}$ in the main text.

## B.1 $G = \mathbb{Z}_2 \times \mathbb{Z}_2$

The starting point is $\mathcal{C}_{\mathbb{Z}_2 \times \mathbb{Z}_2} = 2\text{-Vec}(\mathbb{Z}_2 \times \mathbb{Z}_2)$.

**Objects.** The objects (upto isomorphism) of $\mathcal{C}_{\mathbb{Z}_2^2/\mathbb{Z}_2^2}$ all descend from multiples of $D_2^{(\mathrm{id})} \in \mathcal{C}_{\mathbb{Z}_2 \times \mathbb{Z}_2}$, because any simple object of $\mathcal{C}_{\mathbb{Z}_2 \times \mathbb{Z}_2}$ is related to $D_2^{(\mathrm{id})} \in \mathcal{C}_{\mathbb{Z}_2 \times \mathbb{Z}_2}$ by multiplication action of $\mathbb{Z}_2 \times \mathbb{Z}_2$.

The object $D_2^{(\mathrm{id})} \in \mathcal{C}_{\mathbb{Z}_2 \times \mathbb{Z}_2}$ can be made $\mathbb{Z}_2 \times \mathbb{Z}_2$ symmetric by providing a monoidal functor

$$\mathsf{Vec}(\mathbb{Z}_2 \times \mathbb{Z}_2) \to \mathsf{Vec}. \tag{B.2}$$

Such functors are classified by elements of

$$H^2(\mathbb{Z}_2 \times \mathbb{Z}_2, U(1)) \cong \mathbb{Z}_2, \tag{B.3}$$

leading to two objects

$$D_2^{(\mathrm{id})}, \qquad D_2^{(-)}, \tag{B.4}$$

in $\mathcal{C}_{\mathbb{Z}_2^2/\mathbb{Z}_2^2}$. The object $D_2^{(\mathrm{id})} \in \mathcal{C}_{\mathbb{Z}_2^2/\mathbb{Z}_2^2}$ is obtained by making $D_2^{(\mathrm{id})} \in \mathcal{C}_{\mathbb{Z}_2 \times \mathbb{Z}_2}$ symmetric under $\mathbb{Z}_2 \times \mathbb{Z}_2$ in a trivial way: Each element of $\mathbb{Z}_2^2$ is implemented by the identity line $D_1^{(\mathrm{id})}$ on $D_2^{(\mathrm{id})}$ and the junction operators between $\mathbb{Z}_2^2$ lines are identity local operators $D_0^{(\mathrm{id})}$ between identity lines. On the other hand, the object $D_2^{(-)} \in \mathcal{C}_{\mathbb{Z}_2^2/\mathbb{Z}_2^2}$ is obtained by making $D_2^{(\mathrm{id})} \in \mathcal{C}_{\mathbb{Z}_2 \times \mathbb{Z}_2}$ symmetric under $\mathbb{Z}_2 \times \mathbb{Z}_2$ in a non-trivial way: Each element of $\mathbb{Z}_2^2$ is still implemented by the identity line $D_1^{(\mathrm{id})}$ on $D_2^{(\mathrm{id})}$ but the junction operators between $\mathbb{Z}_2^2$ lines are now $\pm D_0^{(\mathrm{id})}$, where the sign depends on which elements of $\mathbb{Z}_2^2$ are being considered. In the language of [14], $D_2^{(-)}$ is obtained by stacking a 2d $\mathbb{Z}_2 \times \mathbb{Z}_2$ SPT phase on top of $D_2^{(\mathrm{id})} \in \mathcal{C}_{\mathbb{Z}_2 \times \mathbb{Z}_2}$ and then gauging $\mathbb{Z}_2 \times \mathbb{Z}_2$.

The object $2D_2^{(\mathrm{id})} \in \mathcal{C}_{\mathbb{Z}_2 \times \mathbb{Z}_2}$ can be made $\mathbb{Z}_2 \times \mathbb{Z}_2$ symmetric in three different irreducible ways, leading to objects

$$D_2^{(\mathbb{Z}_2^S)}, \qquad D_2^{(\mathbb{Z}_2^C)}, \qquad D_2^{(\mathbb{Z}_2^V)}, \tag{B.5}$$

in $\mathcal{C}_{\mathbb{Z}_2^2/\mathbb{Z}_2^2}$. For $D_2^{(\mathbb{Z}_2^x)}$ with $x \in \{S, C, V\}$, the $\mathbb{Z}_2^x$ symmetry is implemented by diagonal matrix with both entries $D_1^{(\mathrm{id})}$, and the other two $\mathbb{Z}_2$ symmetries are implemented by an off-diagonal matrix with both entries $D_1^{(\mathrm{id})}$. In the language of [14], we have that $\mathbb{Z}_2^x$ is spontaneously preserved/unbroken while the other two $\mathbb{Z}_2$ are spontaneously broken.

The object $4D_2^{(\mathrm{id})} \in \mathcal{C}_{\mathbb{Z}_2 \times \mathbb{Z}_2}$ can be made $\mathbb{Z}_2 \times \mathbb{Z}_2$ symmetric in an irreducible way, leading to object

$$D_2^{(\mathbb{Z}_2 \times \mathbb{Z}_2)}, \tag{B.6}$$

in $\mathcal{C}_{\mathbb{Z}_2^2/\mathbb{Z}_2^2}$. Let us label the four copies of $D_2^{(\mathrm{id})} \in \mathcal{C}_{\mathbb{Z}_2 \times \mathbb{Z}_2}$ inside $4D_2^{(\mathrm{id})} \in \mathcal{C}_{\mathbb{Z}_2 \times \mathbb{Z}_2}$ as $D_2^{(\mathrm{id})(i)}$ with $i \in \{1, S, C, V\}$. Then the line implementing $\mathbb{Z}_2^j$ for $j \in \{1, S, C, V\}$ sends $D_2^{(\mathrm{id})(i)}$ to $D_2^{(\mathrm{id})(ij)}$ and carries $D_1^{(\mathrm{id})}$ in these entries.

In total, the set of simple objects (upto isomorphism) of $\mathcal{C}_{\mathbb{Z}_2^2/\mathbb{Z}_2^2}$ is

$$\mathcal{C}_{\mathbb{Z}_2^2/\mathbb{Z}_2^2}^{\mathrm{ob}} = \left\{ D_2^{(\mathrm{id})}, \quad D_2^{(-)}, \quad D_2^{(\mathbb{Z}_2^S)}, \quad D_2^{(\mathbb{Z}_2^C)}, \quad D_2^{(\mathbb{Z}_2^V)}, \quad D_2^{(\mathbb{Z}_2 \times \mathbb{Z}_2)} \right\}. \tag{B.7}$$

The surface $D_2^{(\mathrm{id})} \in \mathcal{C}_{\mathbb{Z}_2^2/\mathbb{Z}_2^2}$ is the identity object. The fusions of objects in $\{D_2^{(\mathrm{id})}, D_2^{(-)}\}$ follow the group multiplication in (B.3) leading to

$$D_2^{(-)} \otimes D_2^{(-)} \cong D_2^{(\mathrm{id})}. \tag{B.8}$$

We must have

$$D_2^{(-)} \otimes D_2^{(\mathbb{Z}_2^x)} \cong D_2^{(\mathbb{Z}_2^x)},$$
$$D_2^{(-)} \otimes D_2^{(\mathbb{Z}_2 \times \mathbb{Z}_2)} \cong D_2^{(\mathbb{Z}_2 \times \mathbb{Z}_2)},$$ 
(B.9)

for all $x \in \{S, C, V\}$, since tensoring with $D_2^{(-)}$ does not change the quantum dimension (in other words the dimension of 2-representation), as the quantum dimension of $D_2^{(-)}$ is 1, and the tensoring procedure also does not modify which $\mathbb{Z}_2$ symmetries are spontaneously broken or preserved. We obtain

$$D_2^{(\mathbb{Z}_2^x)} \otimes D_2^{(\mathbb{Z}_2^x)} \cong 2D_2^{(\mathbb{Z}_2^x)},$$ 
(B.10)

for all $x \in \{S, C, V\}$ by following arguments similar to those leading to (26). On the other hand, we have

$$D_2^{(\mathbb{Z}_2^x)} \otimes D_2^{(\mathbb{Z}_2^y)} \cong D_2^{(\mathbb{Z}_2 \times \mathbb{Z}_2)},$$ 
(B.11)

for all $x \neq y \in \{S, C, V\}$. It is easy to see that every $\mathbb{Z}_2$ in $\mathbb{Z}_2 \times \mathbb{Z}_2$ is spontaneously broken after taking the above tensor product. Finally, we must have

$$D_2^{(\mathbb{Z}_2 \times \mathbb{Z}_2)} \otimes D_2^{(\mathbb{Z}_2 \times \mathbb{Z}_2)} \cong 4D_2^{(\mathbb{Z}_2 \times \mathbb{Z}_2)},$$ 
(B.12)

as the dimension of the tensor product 2-representation is 16, with no spontaneous preservation of any $\mathbb{Z}_2$ for any of the 16 underlying vacua.

**1-morphisms.** The 1-endomorphisms of $D_2^{(\mathrm{id})} \in \mathcal{C}_{\mathbb{Z}_2^2/\mathbb{Z}_2^2}$ form the fusion category $\mathrm{Rep}(\mathbb{Z}_2 \times \mathbb{Z}_2)$ under composition. The set of simple 1-endomorphisms (upto isomorphism) of $D_2^{(\mathrm{id})} \in \mathcal{C}_{\mathbb{Z}_2^2/\mathbb{Z}_2^2}$ is

$$\mathcal{C}_{\mathbb{Z}_2^2/\mathbb{Z}_2^2}^{D_2^{(\mathrm{id})}} = \left\{ D_1^{(\mathrm{id})}, \quad D_1^{(S)}, \quad D_1^{(C)}, \quad D_1^{(V)} \right\},$$ 
(B.13)

where $D_1^{(\mathrm{id})} \in \mathcal{C}_{\mathbb{Z}_2^2/\mathbb{Z}_2^2}$ corresponds to trivial representation of $\mathbb{Z}_2 \times \mathbb{Z}_2$ and $D_1^{(x)} \in \mathcal{C}_{\mathbb{Z}_2^2/\mathbb{Z}_2^2}$ for $x \in \{S, C, V\}$ corresponds to irreducible representation of $\mathbb{Z}_2 \times \mathbb{Z}_2$ in which $\mathbb{Z}_2^x$ acts by a non-trivial sign, while the other two $\mathbb{Z}_2$ act trivially.

The 1-morphisms from $D_2^{(\mathrm{id})} \in \mathcal{C}_{\mathbb{Z}_2^2/\mathbb{Z}_2^2}$ to $D_2^{(-)} \in \mathcal{C}_{\mathbb{Z}_2^2/\mathbb{Z}_2^2}$ arise from 1-endomorphisms of $D_2^{(\mathrm{id})} \in \mathcal{C}_{\mathbb{Z}_2 \times \mathbb{Z}_2}$, which are given by vector spaces. Such a vector space is made $\mathbb{Z}_2 \times \mathbb{Z}_2$ symmetric by converting it into a projective representation $R$ for $\mathbb{Z}_2 \times \mathbb{Z}_2$ lying in the non-trivial class in (B.3). This is because $D_2^{(-)} \in \mathcal{C}_{\mathbb{Z}_2^2/\mathbb{Z}_2^2}$ and $D_2^{(\mathrm{id})} \in \mathcal{C}_{\mathbb{Z}_2^2/\mathbb{Z}_2^2}$ differ by the non-trivial 2d $\mathbb{Z}_2 \times \mathbb{Z}_2$ SPT phase and so an interface between them should carry an anomaly for the $\mathbb{Z}_2 \times \mathbb{Z}_2$ symmetry, which translates to the fact that the vector space underlying the interface must form a projective representation of $\mathbb{Z}_2 \times \mathbb{Z}_2$. There is a single such irreducible projective representation whose underlying vector space has dimension 2, and thus the set of simple 1-morphisms (upto isomorphism) from $D_2^{(\mathrm{id})} \in \mathcal{C}_{\mathbb{Z}_2^2/\mathbb{Z}_2^2}$ to $D_2^{(-)} \in \mathcal{C}_{\mathbb{Z}_2^2/\mathbb{Z}_2^2}$

$$\mathcal{C}_{\mathbb{Z}_2^2/\mathbb{Z}_2^2}^{D_2^{(\mathrm{id})}, D_2^{(-)}} = \left\{ D_1^{(\mathrm{id},-)} \right\},$$ 
(B.14)

comprises of a single element $D_1^{(\mathrm{id},-)}$ whose quantum dimension is 2. Similarly, the set of simple 1-morphisms (upto isomorphism) from $D_2^{(-)} \in \mathcal{C}_{\mathbb{Z}_2^2/\mathbb{Z}_2^2}$ to $D_2^{(\mathrm{id})} \in \mathcal{C}_{\mathbb{Z}_2^2/\mathbb{Z}_2^2}$

$$\mathcal{C}_{\mathbb{Z}_2^2/\mathbb{Z}_2^2}^{D_2^{(-)}, D_2^{(\mathrm{id})}} = \left\{ D_1^{(-,\mathrm{id})} \right\},$$ 
(B.15)

comprises of a single element $D_1^{(-,\mathrm{id})}$ corresponding again to the non-trivial irreducible projective representation of $\mathbb{Z}_2 \times \mathbb{Z}_2$.

On the other hand, the 1-endomorphisms of $D_2^{(-)} \in \mathcal{C}_{\mathbb{Z}_2^2/\mathbb{Z}_2^2}$ correspond to non-projective usual representations of $\mathbb{Z}_2 \times \mathbb{Z}_2$ because, just as in going from $D_2^{(\mathrm{id})}$ to $D_2^{(\mathrm{id})}$, there is no non-trivial SPT phase involved in going from $D_2^{(-)}$ to $D_2^{(-)}$. Thus, we have the set of simple 1-endomorphisms (upto isomorphism) of $D_2^{(-)} \in \mathcal{C}_{\mathbb{Z}_2^2/\mathbb{Z}_2^2}$ as

$$\mathcal{C}_{\mathbb{Z}_2^2/\mathbb{Z}_2^2}^{D_2^{(-)}} = \left\{ D_1^{(-;\mathrm{id})}, \quad D_1^{(-;S)}, \quad D_1^{(-;C)}, \quad D_1^{(-;V)} \right\}, \tag{B.16}$$

where the labeling follows the same rules as for elements of the set $\mathcal{C}_{\mathbb{Z}_2^2/\mathbb{Z}_2^2}^{D_2^{(\mathrm{id})}}$.

The composition of the 1-morphisms in $\mathcal{C}_{\mathbb{Z}_2^2/\mathbb{Z}_2^2}^{D_2^{(\mathrm{id})},D_2^{(-)}}$ or $\mathcal{C}_{\mathbb{Z}_2^2/\mathbb{Z}_2^2}^{D_2^{(-)},D_2^{(\mathrm{id})}}$ with the 1-morphisms in $\mathcal{C}_{\mathbb{Z}_2^2/\mathbb{Z}_2^2}^{D_2^{(\mathrm{id})}}$ or $\mathcal{C}_{\mathbb{Z}_2^2/\mathbb{Z}_2^2}^{D_2^{(-)}}$ is given by tensoring the projective representation with irreducible representations of $\mathbb{Z}_2 \times \mathbb{Z}_2$. Since the irreducible representations have dimension 1, the dimension of the projective representation remains unchanged under the tensor product operation, and hence we must have

$$
\begin{aligned}
D_1^{(x)} \circ D_1^{(\mathrm{id},-)} &\cong D_1^{(\mathrm{id},-)} \circ D_1^{(-;x)} \cong D_1^{(\mathrm{id},-)}, \\
D_1^{(-,\mathrm{id})} \circ D_1^{(x)} &\cong D_1^{(-;x)} \circ D_1^{(-,\mathrm{id})} \cong D_1^{(-,\mathrm{id})},
\end{aligned}
\tag{B.17}
$$

for all $x \in \{\mathrm{id}, S, C, V\}$.

Consider now $D_1^{(\mathrm{id},-)} \circ D_1^{(-,\mathrm{id})}$, which involves taking the tensor product of the irreducible projective representation with itself. The tensor product representation is a usual representation of dimension four, and hence decomposes into four irreducible representations. We want to determine what these irreducible representations are. A projective representation of $\mathbb{Z}_2 \times \mathbb{Z}_2$ is specified by matrices $M_S$, $M_C$ and $M_V$ corresponding to elements $S$, $C$ and $V$ in $\mathbb{Z}_2 \times \mathbb{Z}_2$ which satisfy

$$M_S^2 = M_C^2 = -M_V^2 = 1, \qquad M_S M_C = -M_C M_S = M_V, \tag{B.18}$$

For the irreducible projective representation $R$, we can take

$$M_S = \sigma_x, \qquad M_C = \sigma_y, \qquad M_V = i\sigma_z, \tag{B.19}$$

where $\sigma_i$ are the well-known $2 \times 2$ Pauli matrices. Let us label the basis vectors of $\mathbb{C}^2$ on which the above Pauli matrices act as $v_0$ and $v_1$. Then, $v_i \otimes v_j$ form a basis for $R \otimes R$ on which the action of $\mathbb{Z}_2^S$ is

$$v_0 \otimes v_0 \longleftrightarrow v_1 \otimes v_1, \qquad v_0 \otimes v_1 \longleftrightarrow v_1 \otimes v_0, \tag{B.20}$$

and the action of $\mathbb{Z}_2^C$ is

$$v_0 \otimes v_0 \longleftrightarrow -v_1 \otimes v_1, \qquad v_0 \otimes v_1 \longleftrightarrow v_1 \otimes v_0. \tag{B.21}$$

Thus vectors

$$
\begin{aligned}
v_{+,-} &:= v_0 \otimes v_0 + v_1 \otimes v_1, \\
v_{-,+} &:= v_0 \otimes v_0 - v_1 \otimes v_1, \\
v_{+,+} &:= v_0 \otimes v_1 + v_1 \otimes v_0, \\
v_{-,-} &:= v_0 \otimes v_1 - v_1 \otimes v_0,
\end{aligned}
\tag{B.22}
$$

individually span irreducible representations of $\mathbb{Z}_2 \times \mathbb{Z}_2$ with vector $v_{i,j}$ spanning the irreducible representation in which $\mathbb{Z}_2^S$ acts with sign $i$ and $\mathbb{Z}_2^C$ acts with sign $j$. Thus, we obtain that

$$D_1^{(\mathrm{id},-)} \circ D_1^{(-,\mathrm{id})} \cong D_1^{(\mathrm{id})} \oplus D_1^{(S)} \oplus D_1^{(C)} \oplus D_1^{(V)}. \tag{B.23}$$

By the same argument we also obtain that

$$D_1^{(-,\mathrm{id})} \circ D_1^{(\mathrm{id},-)} \cong D_1^{(-;\mathrm{id})} \oplus D_1^{(-;S)} \oplus D_1^{(-;C)} \oplus D_1^{(-;V)}. \tag{B.24}$$

The morphisms for $D_2^{(\mathbb{Z}_2^x)} \in \mathcal{C}_{\mathbb{Z}_2^2/\mathbb{Z}_2^2}$ to $D_2^{(\mathrm{id})} \in \mathcal{C}_{\mathbb{Z}_2^2/\mathbb{Z}_2^2}$ are given by $2 \times 1$ matrices valued in $\mathrm{Rep}(\mathbb{Z}_2^x)$ with both entries of the matrix being the same object of $\mathrm{Rep}(\mathbb{Z}_2^x)$. Thus, the set of simple 1-morphisms (upto isomorphism) from $D_2^{(\mathbb{Z}_2^x)} \in \mathcal{C}_{\mathbb{Z}_2^2/\mathbb{Z}_2^2}$ to $D_2^{(\mathrm{id})} \in \mathcal{C}_{\mathbb{Z}_2^2/\mathbb{Z}_2^2}$ is

$$\mathcal{C}_{\mathbb{Z}_2^2/\mathbb{Z}_2^2}^{D_2^{(\mathbb{Z}_2^x)}, D_2^{(\mathrm{id})}} = \left\{ D_1^{(\mathbb{Z}_2^x, \mathrm{id}; +)}, \quad D_1^{(\mathbb{Z}_2^x, \mathrm{id}; -)} \right\}, \tag{B.25}$$

where $D_1^{(\mathbb{Z}_2^x, \mathrm{id}; s)}$ is the 1-morphism descending from choosing the irreducible representation of $\mathbb{Z}_2^x$ given by the sign $s$. Similarly, the set of simple 1-morphisms (upto isomorphism) from $D_2^{(\mathrm{id})} \in \mathcal{C}_{\mathbb{Z}_2^2/\mathbb{Z}_2^2}$ to $D_2^{(\mathbb{Z}_2^x)} \in \mathcal{C}_{\mathbb{Z}_2^2/\mathbb{Z}_2^2}$ is

$$\mathcal{C}_{\mathbb{Z}_2^2/\mathbb{Z}_2^2}^{D_2^{(\mathrm{id})}, D_2^{(\mathbb{Z}_2^x)}} = \left\{ D_1^{(\mathrm{id}, \mathbb{Z}_2^x; +)}, \quad D_1^{(\mathrm{id}, \mathbb{Z}_2^x; -)} \right\}, \tag{B.26}$$

as such 1-morphisms are furnished by $1 \times 2$ matrices valued in $\mathrm{Rep}(\mathbb{Z}_2^x)$ with both entries of the matrix being the same object of $\mathrm{Rep}(\mathbb{Z}_2^x)$.

Similarly, the set of simple 1-morphisms (upto isomorphism) from $D_2^{(\mathbb{Z}_2^x)} \in \mathcal{C}_{\mathbb{Z}_2^2/\mathbb{Z}_2^2}$ to $D_2^{(-)} \in \mathcal{C}_{\mathbb{Z}_2^2/\mathbb{Z}_2^2}$ is

$$\mathcal{C}_{\mathbb{Z}_2^2/\mathbb{Z}_2^2}^{D_2^{(\mathbb{Z}_2^x)}, D_2^{(-)}} = \left\{ D_1^{(\mathbb{Z}_2^x, -; +)}, \quad D_1^{(\mathbb{Z}_2^x, -; -)} \right\}, \tag{B.27}$$

and the set of simple 1-morphisms (upto isomorphism) from $D_2^{(-)} \in \mathcal{C}_{\mathbb{Z}_2^2/\mathbb{Z}_2^2}$ to $D_2^{(\mathbb{Z}_2^x)} \in \mathcal{C}_{\mathbb{Z}_2^2/\mathbb{Z}_2^2}$ is

$$\mathcal{C}_{\mathbb{Z}_2^2/\mathbb{Z}_2^2}^{D_2^{(-)}, D_2^{(\mathbb{Z}_2^x)}} = \left\{ D_1^{(-, \mathbb{Z}_2^x; +)}, \quad D_1^{(-, \mathbb{Z}_2^x; -)} \right\}. \tag{B.28}$$

The set of simple 1-endomorphisms (upto isomorphism) of $D_2^{(\mathbb{Z}_2^x)} \in \mathcal{C}_{\mathbb{Z}_2^2/\mathbb{Z}_2^2}$ is

$$\mathcal{C}_{\mathbb{Z}_2^2/\mathbb{Z}_2^2}^{D_2^{(\mathbb{Z}_2^x)}} = \left\{ D_1^{(\mathbb{Z}_2^x; \mathrm{id}+)}, \quad D_1^{(\mathbb{Z}_2^x; \mathrm{id}-)}, \quad D_1^{(\mathbb{Z}_2^x; -+)}, \quad D_1^{(\mathbb{Z}_2^x; --)} \right\}, \tag{B.29}$$

where $D_1^{(\mathbb{Z}_2^x; \mathrm{id}s)}$ arises from the diagonal $2 \times 2$ matrix with both entries having irrep of $\mathbb{Z}_2^x$ given by sign $s$, and $D_1^{(\mathbb{Z}_2^x; -s)}$ arises from the off-diagonal $2 \times 2$ matrix with both entries having irrep of $\mathbb{Z}_2^x$ given by sign $s$.

The set of simple 1-morphisms (upto isomorphism) from $D_2^{(\mathbb{Z}_2^x)} \in \mathcal{C}_{\mathbb{Z}_2^2/\mathbb{Z}_2^2}$ to $D_2^{(\mathbb{Z}_2^y)} \in \mathcal{C}_{\mathbb{Z}_2^2/\mathbb{Z}_2^2}$ for $x \neq y$ is

$$\mathcal{C}_{\mathbb{Z}_2^2/\mathbb{Z}_2^2}^{D_2^{(\mathbb{Z}_2^x)}, D_2^{(\mathbb{Z}_2^y)}} = \left\{ D_1^{(\mathbb{Z}_2^x, \mathbb{Z}_2^y)} \right\}, \tag{B.30}$$

where $D_1^{(\mathbb{Z}_2^x, \mathbb{Z}_2^y)}$ corresponds to a $2 \times 2$ matrix with all entries of the matrix being $\mathbb{C}$.

Additionally, there is one simple 1-morphism (upto isomorphism) from $D_2^{(\mathbb{Z}_2 \times \mathbb{Z}_2)}$ to $D_2^{(x)}$ for $x \in \{\mathrm{id}, -\}$, one simple 1-morphism (upto isomorphism) from $D_2^{(x)}$ to $D_2^{(\mathbb{Z}_2 \times \mathbb{Z}_2)}$ for $x \in \{\mathrm{id}, -\}$, two simple 1-morphisms (upto isomorphism) from $D_2^{(\mathbb{Z}_2 \times \mathbb{Z}_2)}$ to $D_2^{(\mathbb{Z}_2^x)}$, two simple 1-morphisms (upto isomorphism) from $D_2^{(\mathbb{Z}_2^x)}$ to $D_2^{(\mathbb{Z}_2 \times \mathbb{Z}_2)}$, and four simple 1-endomorphisms (upto isomorphism) of $D_2^{(\mathbb{Z}_2 \times \mathbb{Z}_2)}$.

**Condensation defects.** As described in detail in [14], all the non-trivial simple objects of $\mathcal{C}_{\mathbb{Z}_2^2/\mathbb{Z}_2^2}$ have to be condensation defects. Following their analysis, we find that the surface defects corresponding to these simple objects can be produced as follows:

- $D_2^{(\mathbb{Z}_2^x)} \in \mathcal{C}_{\mathbb{Z}_2^2/\mathbb{Z}_2^2}$ is obtained by gauging the $\mathbb{Z}_2^x$ 1-form symmetry generated by $D_1^{(x)} \in \mathcal{C}_{\mathbb{Z}_2^2/\mathbb{Z}_2^2}$ along a surface in spacetime (occupied by identity defect $D_2^{(\mathrm{id})}$).

- $D_2^{(\mathbb{Z}_2 \times \mathbb{Z}_2)} \in \mathcal{C}_{\mathbb{Z}_2^2/\mathbb{Z}_2^2}$ is obtained by gauging the full $\mathbb{Z}_2 \times \mathbb{Z}_2$ 1-form symmetry along a surface in spacetime (occupied by identity defect $D_2^{(\mathrm{id})}$).

- $D_2^{(-)} \in \mathcal{C}_{\mathbb{Z}_2^2/\mathbb{Z}_2^2}$ is obtained by gauging the full $\mathbb{Z}_2 \times \mathbb{Z}_2$ 1-form symmetry along with a theta angle specified by the non-trivial 2d $\mathbb{Z}_2 \times \mathbb{Z}_2$ SPT phase along a surface in spacetime (occupied by identity defect $D_2^{(\mathrm{id})}$).

## B.2 $G = \mathbb{Z}_4$

We consider here the gauging of $\mathbb{Z}_4$ starting from a theory with symmetry category $\mathcal{C}_{\mathbb{Z}_4} = 2\text{-Vec}(\mathbb{Z}_4)$ whose objects are given in (15). We will show that

$$\mathcal{C}_{\mathbb{Z}_4/\mathbb{Z}_4} = 2\text{-Rep}(\mathbb{Z}_4). \tag{B.31}$$

**Objects.** The objects in $\mathcal{C}_{\mathbb{Z}_4/\mathbb{Z}_4}$ descend from multiples of the identity object i.e., $nD_2^{(\mathrm{id})} \in \mathcal{C}_{\mathbb{Z}_4}$ for some some integer $n$. As described in section 2, for a given $n$, this boils down to prescribing a monoidal functor

$$\text{Vec}(\mathbb{Z}_4) \to \text{Mat}_n(\text{Vec}). \tag{B.32}$$

For $n = 1$, there is a unique such monoidal functor upto isomorphism. This physically corresponds to making a 2d TQFT with a single vacua $\mathbb{Z}_4$ invariant which can be done in a unique way since there are no 2d $\mathbb{Z}_4$ SPTs (since $H^2(\mathbb{Z}_4, U(1))$ is trivial). We will denote this object in $\mathcal{C}_{\mathbb{Z}_4/\mathbb{Z}_4}$ as $D_2^{(\mathrm{id})}$. Next, let us consider the case $n = 2$, therefore we need to prescribe a monoidal functor from $\text{Vec}(\mathbb{Z}_4)$ to $\text{Mat}_2(\text{Vec})$. There is one indecomposable way to do this, such that $S, C \in \mathbb{Z}_4$ map to the $2 \times 2$ matrix with non-zero entries $D_1^{(\mathrm{id})}$ on off-diagonals, while $\mathrm{id}, V \in \mathbb{Z}_4$ map to the diagonal $2 \times 2$ matrix with $D_1^{(\mathrm{id})}$ on the diagonal. Physically this object corresponds to a TQFT with the $\mathbb{Z}_4$ symmetry spontaneously broken down to its $\mathbb{Z}_2$ subgroup, such that each vacua is invariant under $V$, while the two vacua are exchanged by the action of $D_2^{(S)}$ and $D_2^{(C)}$. We denote this object as $D_2^{(\mathbb{Z}_2^V)} \in \mathcal{C}_{\mathbb{Z}_4/\mathbb{Z}_4}$. Moving on, we do not get any new simple objects for $n = 3$ since there are no 3-dimensional orbits of $\mathbb{Z}_4$. For $n = 4$, we get a single (upto isomorphism) indecomposable object, for which the action of $S$ can be represented as

$$\begin{pmatrix} 0 & D_1^{(\mathrm{id})} & 0 & 0 \\ 0 & 0 & D_1^{(\mathrm{id})} & 0 \\ 0 & 0 & 0 & D_1^{(\mathrm{id})} \\ D_1^{(\mathrm{id})} & 0 & 0 & 0 \end{pmatrix}. \tag{B.33}$$

Physically, this corresponds to a $\mathbb{Z}_4$ symmetric TQFT in which the $\mathbb{Z}_4$ symmetry is spontaneously broken. Therefore it has four vacua that are permuted cyclically by the $\mathbb{Z}_4$ generator. We denote this object as $D_2^{(\mathbb{Z}_4^S)} \in \mathcal{C}_{\mathbb{Z}_4/\mathbb{Z}_4}$. There are no additional indecomposable objects. To summarize the set of indecomposable objects is

$$\left\{ D_2^{(\mathrm{id})}, \quad D_2^{(\mathbb{Z}_2^V)}, \quad D_2^{(\mathbb{Z}_4^S)} \right\}. \tag{B.34}$$

**Fusion of objects.** The object $D_2^{(\text{id})}$ is the identity under fusion. The fusion product for $D_2^{(\mathbb{Z}_2^V)} \otimes D_2^{(\mathbb{Z}_2^V)}$ can be obtained by recalling that $D_2^{(\mathbb{Z}_2^V)}$ lifts to $2D_2^{(\text{id})}$ in $\mathcal{C}_{\mathbb{Z}_4}$. Hence the objects in the fusion product can be labelled by elements in the tuple $(i, j)$ where $i, j \in \{0, 1\}$. These transform under $S \in \mathbb{Z}_4$ as

$$(i, j) \longrightarrow (i + 1 \bmod 2, j + 1 \bmod 2), \tag{B.35}$$

which splits into the two indecomposable $\mathbb{Z}_4$ orbits labelled by orbit representatives $[(0, 0)]$ and $[(1, 0)]$. Each orbit descends to a copy of $D_2^{(\mathbb{Z}_2^V)} \in \mathcal{C}_{\mathbb{Z}_4/\mathbb{Z}_4}$ implying the fusion rule

$$D_2^{(\mathbb{Z}_2^V)} \otimes D_2^{(\mathbb{Z}_2^V)} = 2D_2^{(\mathbb{Z}_2^V)}. \tag{B.36}$$

Similarly, to compute $D_2^{(\mathbb{Z}_2^V)} \otimes D_2^{(\mathbb{Z}_4^S)}$, we recall that $D_2^{(\mathbb{Z}_4^S)}$ descends from $4D_2^{(\text{id})}$ in $\mathcal{C}_{\mathbb{Z}_4}$, therefore in the pre-gauged category, we may denote objects in the fusion outcome by the tuples $(i, j)$ where $i \in \{0, 1\}$ and $j \in \{0, 1, 2, 3\}$. Under the action of $S \in \mathbb{Z}_4$,

$$(i, j) \longrightarrow (i + 1 \bmod 2, j + 1 \bmod 4), \tag{B.37}$$

therefore we again obtain two orbits $[(0, 0)]$ and $[(1, 0)]$ implying the fusion rule

$$\begin{aligned} D_2^{(\mathbb{Z}_2^V)} \otimes D_2^{(\mathbb{Z}_4^S)} &= 2D_2^{(\mathbb{Z}_4^S)}, \\ D_2^{(\mathbb{Z}_4^S)} \otimes D_2^{(\mathbb{Z}_2^V)} &= 2D_2^{(\mathbb{Z}_4^S)}. \end{aligned} \tag{B.38}$$

Similarly, the fusion rule $D_2^{(\mathbb{Z}_4^S)} \otimes D_2^{(\mathbb{Z}_4^S)}$ follows exactly the same way, giving

$$D_2^{(\mathbb{Z}_4^S)} \otimes D_2^{(\mathbb{Z}_4^S)} = 4D_2^{(\mathbb{Z}_4^S)}. \tag{B.39}$$

**Morphisms.** First we focus on the 1-endomorphisms of each of the three objects. The distinct 1-endomorphisms of $D_2^{(\text{id})}$ in $\mathcal{C}_{\mathbb{Z}_4/\mathbb{Z}_4}$ are related to the distint ways in which the 1-endomorphisms of $D_2^{(\text{id})} \in \mathcal{C}_{\mathbb{Z}_4}$ can be made $\mathbb{Z}_4$-symmetric. The 1-category of 1-endomorphisms of $D_2^{(\text{id})} \in \mathcal{C}_{\mathbb{Z}_4}$ is $\mathsf{Vec}$ generated by the identity line $D_1^{\text{id}}$. Since the permutation action of $\mathbb{Z}_4$ on this line is trivial, the ways in which it can be made $\mathbb{Z}_4$ symmetric corresponds to assigning a representation of $\mathbb{Z}_4$. Thus we realize that

$$\mathrm{End}_{\mathcal{C}_{\mathbb{Z}_4/\mathbb{Z}_4}} = \left\{ D_1^{(\alpha)} \right\} \cong \mathrm{Rep}(\mathbb{Z}_4). \tag{B.40}$$

The 1-endomorphisms of $D_2^{(\mathbb{Z}_2^V)} \in \mathcal{C}_{\mathbb{Z}_4/\mathbb{Z}_4}$ descend from $\mathbb{Z}_4$ symmetric 1-endomorphisms of $2D_2^{(\text{id})}$ in $\mathcal{C}_{\mathbb{Z}_4}$. The set of 1-endomorphsims of $2D_2^{(\text{id})}$ is $\mathsf{Mat}_2(\mathsf{Vec})$. The two rows and two columns of these $2 \times 2$ matrices are exchanged under the action of $S \in \mathbb{Z}_4$. Notably the $\mathbb{Z}_2$ subgroup generated by $V$ acts trivially, therefore, the 1-endomorphisms can carry representations of $\mathbb{Z}_2^V$. Therefore we obtain the following 1-endomorphisms

$$\mathrm{End}_{\mathcal{C}_{\mathbb{Z}_4/\mathbb{Z}_4}}(D_2^{(\mathbb{Z}_2^V)}) = \left\{ D_1^{(\mathbb{Z}_2^V; \text{id})}, \ D_1^{(\mathbb{Z}_2^V; -)}, \ D_1^{(\mathbb{Z}_2^V; V)}, \ D_1^{(\mathbb{Z}_2^V; V-)} \right\} \cong \mathsf{Vec}(\mathbb{Z}_2) \boxtimes \mathrm{Rep}(\mathbb{Z}_2^V). \tag{B.41}$$

The labels 'id' and 'V' correspond to the diagonal $2 \times 2$ matrices with a trivial and non-trivial representations of $\mathbb{Z}_2^V$. Meanwhile the labels '$-$' and 'V$-$' correspond to the off-diagonal $2 \times 2$ matrices with a trivial and non-trivial representations of $\mathbb{Z}_2^V$ respectively.

The 1-endomorphisms of $D_2^{(\mathbb{Z}_4^S)} \in \mathcal{C}_{\mathbb{Z}_4/\mathbb{Z}_4}$ descend from $\mathbb{Z}_4$ symmetric 1-endomorphisms of $4D_2^{(\text{id})}$ in $\mathcal{C}_{\mathbb{Z}_4}$. The set of 1-endomorphsims of $4D_2^{(\text{id})}$ is $\mathsf{Mat}_4(\mathsf{Vec})$. The four rows and four

columns of these $4 \times 4$ matrices are cyclically permuted the action of $S \in \mathbb{Z}_4$, therefore we obtain the following set of simple 1-morphisms

$$\text{End}_{\mathcal{C}_{\mathbb{Z}_4/\mathbb{Z}_4}}(D_2^{(\mathbb{Z}_4^S)}) = \left\{ D_1^{(\mathbb{Z}_4^S;\text{id})}, \quad D_1^{(\mathbb{Z}_4^S;S)}, \quad D_1^{(\mathbb{Z}_4^S;V)}, \quad D_1^{(\mathbb{Z}_4^S;C)} \right\}. \tag{B.42}$$

The labels id, $S, V, C$ correspond to the matrices

$$\begin{aligned} D_1^{(\mathbb{Z}_4^S;\text{id})}\Big|_{\mathcal{C}_{\mathbb{Z}_4}} &= D_1^{(\text{id})} P_{(1)(2)(3)(4)}, \quad D_1^{(\mathbb{Z}_4^S;S)}\Big|_{\mathcal{C}_{\mathbb{Z}_4}} = D_1^{(\text{id})} P_{(1234)}, \\ D_1^{(\mathbb{Z}_4^S;V)}\Big|_{\mathcal{C}_{\mathbb{Z}_4}} &= D_1^{(\text{id})} P_{(13)(24)}, \quad D_1^{(\mathbb{Z}_4^S;C)}\Big|_{\mathcal{C}_{\mathbb{Z}_4}} = D_1^{(\text{id})} P_{(1432)}. \end{aligned} \tag{B.43}$$

Here we indicated the morphisms again in terms of permutation matrices $P_\pi$. The morphisms from $D_2^{(\mathbb{Z}_2^V)}$ to $D_2^{(\text{id})}$ in $\mathcal{C}_{\mathbb{Z}_4/\mathbb{Z}_4}$ correspond to the ways 1-morphisms from $2D_2^{(\text{id})}$ to $D_2^{(\text{id})}$ in $\mathcal{C}_{\mathbb{Z}_4}$ can be made $\mathbb{Z}_4$ symmetric. These are elements in $\text{Mat}_{2\times1}(\text{Vec})$. Since, the rows of the $2 \times 1$ matrices are exchanged under $S \in \mathbb{Z}_4$, while the matrices are invariant under $\mathbb{Z}_2^V \subset \mathbb{Z}_4$, there are two indecomposable 1-morphisms

$$\mathcal{C}_{\mathbb{Z}_4/\mathbb{Z}_4}^{D_2^{(\mathbb{Z}_2^V)}, D_2^{(\text{id})}} = \left\{ D_1^{(\mathbb{Z}_2^V,\text{id});(\text{id})}, \quad D_1^{(\mathbb{Z}_2^V,\text{id});(V)} \right\}, \tag{B.44}$$

which correspond to the $2 \times 1$ matrices with both entries $D_1^{(\text{id})}$ with the superscripts 'id' and '$V$' denote trivial and non-trivial $\mathbb{Z}_2^V$ representations respectively. Similarly, the indecomposable 1-morphisms from $D_2^{(\text{id})}$ to $D_2^{(\mathbb{Z}_2^V)} \in \mathcal{C}_{\mathbb{Z}_4/\mathbb{Z}_4}$ are

$$\mathcal{C}_{\mathbb{Z}_4/\mathbb{Z}_4}^{D_2^{(\text{id})}, D_2^{(\mathbb{Z}_2^V)}} = \left\{ D_1^{(\text{id},\mathbb{Z}_2^V);(\text{id})}, \quad D_1^{(\text{id},\mathbb{Z}_2^V);(V)} \right\}, \tag{B.45}$$

where both the 1-morphism lines correspond to the $1 \times 2$ matrices with both entries $D_1^{(\text{id})}$ in $\mathcal{C}_{\mathbb{Z}_4}$.

The morphisms from $D_2^{(\mathbb{Z}_4^S)} \in \mathcal{C}_{\mathbb{Z}_4/\mathbb{Z}_4}$ to $D_2^{(\text{id})} \in \mathcal{C}_{\mathbb{Z}_4/\mathbb{Z}_4}$ correspond to the ways 1-morphsims from $4D_2^{(\text{id})}$ to $D_2^{(\text{id})} \in \mathcal{C}_{\mathbb{Z}_4}$ can be made $\mathbb{Z}_4$ symmetric. These correspond to $4 \times 1$ matrices valued in Vec with the $\mathbb{Z}_4$ action cyclically permuting the four rows. There is consequently a single indecomposable 1-morphism denoted by

$$\mathcal{C}_{\mathbb{Z}_4/\mathbb{Z}_4}^{D_2^{(\mathbb{Z}_4^S)}, D_2^{(\text{id})}} = \left\{ D_1^{(\mathbb{Z}_4^S,\text{id})} \right\}. \tag{B.46}$$

which corresponds to the $4 \times 1$ matrix with all entries $D_1^{(\text{id})}$ in $\mathcal{C}_{\mathbb{Z}_4}$. Similarly there is also a single indecomposable morphism from $D_2^{(\text{id})}$ to $D_2^{(\mathbb{Z}_4^S)}$ in $\mathcal{C}_{\mathbb{Z}_4/\mathbb{Z}_4}$

$$\mathcal{C}_{\mathbb{Z}_4/\mathbb{Z}_4}^{D_2^{(\text{id})}, D_2^{(\mathbb{Z}_4^S)}} = \left\{ D_1^{(\text{id},\mathbb{Z}_4^S)} \right\}, \tag{B.47}$$

which descends from the $1 \times 4$ matrix with all entries $D_1^{(\text{id})}$ in $\mathcal{C}_{\mathbb{Z}_4}$.

Finally, for similar reasons, there is also a unique indecomposable 1-morphism each from $D_2^{(\mathbb{Z}_4^S)}$ to $D_2^{(\mathbb{Z}_2^V)}$ and vice versa. These correspond to $4 \times 2$ and $2 \times 4$ matrices with all entries $D_1^{(\text{id})}$ in the pre-gauged category. As we have seen, there are morphisms between every pair of indecomposable objects in 2-Rep($\mathbb{Z}_4$), implying that there is a single Schur component of objects. This is in fact a general feature for 2-Rep($G$) for any finite group $G$.

**Composition of morphisms.** As in the previous examples, the composition of morphisms is given by combining $\mathbb{Z}_4$ actions. This is obtained by taking the matrix product of the morphisms lifted to $\mathcal{C}_{\mathbb{Z}_4}$, as well as, the tensor product of representations of subgroups of $\mathbb{Z}_4$, wherever applicable. The composition of 1-endomorphisms of $D_2^{(\mathrm{id})}$ correspond to taking a matrix product of $1 \times 1$ matrices, which is trivial. Additionally, one needs to take a tensor product of $\mathbb{Z}_4$ representations which gives

$$D_1^\alpha \circ D_1^\beta = D_1^{\alpha \cdot \beta}, \tag{B.48}$$

where $\alpha, \beta \in \mathsf{Rep}(\mathbb{Z}_4)$. Since $\mathsf{Rep}(\mathbb{Z}_4) \cong \mathbb{Z}_4$, we label $\alpha, \beta \in \{0, 1, 2, 3\}$ with $\alpha \cdot \beta \equiv \alpha + \beta \bmod 4$.

The composition of the 1-endomorphisms of $D_2^{(\mathbb{Z}_2^V)}$ are given by taking a matrix product of the $2 \times 2$ endomorphism matrices of $2D_2^{(\mathrm{id})} \in \mathcal{C}_{\mathbb{Z}_4}$, and subsequently taking a tensor product of representations of $\mathbb{Z}_2^V$. Doing so, one finds the following (commutative) composition rules that have a $\mathbb{Z}_2 \times \mathbb{Z}_2$ structure

$$
\begin{aligned}
D_1^{(\mathbb{Z}_2^V, V)} \circ D_1^{(\mathbb{Z}_2^V, V)} &= D_1^{(\mathbb{Z}_2^V, \mathrm{id})}, \\
D_1^{(\mathbb{Z}_2^V, V)} \circ D_1^{(\mathbb{Z}_2^V, V-)} &= D_1^{(\mathbb{Z}_2^V, -)}, \\
D_1^{(\mathbb{Z}_2^V, V)} \circ D_1^{(\mathbb{Z}_2^V, -)} &= D_1^{(\mathbb{Z}_2^V, V-)}, \\
D_1^{(\mathbb{Z}_2^V, V-)} \circ D_1^{(\mathbb{Z}_2^V, V-)} &= D_1^{(\mathbb{Z}_2^V, \mathrm{id})}, \\
D_1^{(\mathbb{Z}_2^V, V-)} \circ D_1^{(\mathbb{Z}_2^V, -)} &= D_1^{(\mathbb{Z}_2^V, V)}, \\
D_1^{(\mathbb{Z}_2^V, -)} \circ D_1^{(\mathbb{Z}_2^V, -)} &= D_1^{(\mathbb{Z}_2^V, \mathrm{id})}.
\end{aligned}
\tag{B.49}
$$

The composition rules between the 1-endomorphisms of $D_2^{(\mathbb{Z}_4^S)}$ in (B.42) can be conveniently obtained by using the matrix product of 1-endomorphsims of $4D_2^{(\mathrm{id})} \in \mathcal{C}_{\mathbb{Z}_4}$ given in (B.43). Since these $4 \times 4$ matrices are a permutation representation of $\mathbb{Z}_4$, the resulting composition rules have a $\mathbb{Z}_4$ group structure and the 1-category of 1-endomorphisms can be identified as $\mathsf{Vec}(\mathbb{Z}_4)$. More precisely, $D_1^{(\mathbb{Z}_4^S; \mathrm{id})}$ is the identity line in $\mathsf{Vec}(\mathbb{Z}_4)$, $D_1^{(\mathbb{Z}_4^S; S)}$ and $D_1^{(\mathbb{Z}_4^S; C)}$ are the order 4 lines and $D_1^{(\mathbb{Z}_4^S; V)}$ is the order 2 line.

Next, consider $D_1^{(\mathrm{id}, \mathbb{Z}_2^V); (\mathrm{id})} \circ D_1^{(\mathbb{Z}_2^V, \mathrm{id}); (\mathrm{id})}$ which lifts to a matrix product of a $1 \times 2$ with a $2 \times 1$ matrix in $\mathcal{C}_{\mathbb{Z}_4}$. Both entries of both the matrices are $D_1^{(\mathrm{id})}$ and the columns and rows (respectively) of the matrices are swapped under the action of $S \in \mathbb{Z}_4$. The resulting $1 \times 1$ matrix is $D_1^{(\mathrm{id})} \oplus D_1^{(\mathrm{id})}$, for which the two $D_1^{(\mathrm{id})}$ components are swapped under $S$. Since the action of $V \in \mathbb{Z}_4$ leaves the two $D_1^{(\mathrm{id})}$ components fixed, we may further decompose the outcome after composing the morphisms into a direct sum of representations $0 \oplus 2 \in \mathsf{Rep}(\mathbb{Z}_4)$. Therefore we read off the composition rule

$$D_1^{(\mathrm{id}, \mathbb{Z}_2^V); (\mathrm{id})} \circ D_1^{(\mathbb{Z}_2^V, \mathrm{id}); (\mathrm{id})} = D_1^0 \oplus D_1^2. \tag{B.50}$$

Using analogous arguments and identifying the representation $V \sim 2 \in \mathsf{Rep}(\mathbb{Z}_4)$, we obtain the following rules

$$
\begin{aligned}
D_1^{(\mathrm{id}, \mathbb{Z}_2^V); (V)} \circ D_1^{(\mathbb{Z}_2^V, \mathrm{id}); (\mathrm{id})} &= D_1^0 \oplus D_1^2, \\
D_1^{(\mathrm{id}, \mathbb{Z}_2^V); (V)} \circ D_1^{(\mathbb{Z}_2^V, \mathrm{id}); (V)} &= D_1^0 \oplus D_1^2, \\
D_1^{(\mathrm{id}, \mathbb{Z}_2^V); (\mathrm{id})} \circ D_1^{(\mathbb{Z}_2^V, \mathrm{id}); (V)} &= D_1^0 \oplus D_1^2.
\end{aligned}
\tag{B.51}
$$

The composition rules for $D_1^{(\mathbb{Z}_2^V,\mathrm{id});(\mathrm{id})} \circ D_1^{(\mathrm{id},\mathbb{Z}_2^V);(\mathrm{id})}$ can be similarly obtained by lifting to $\mathcal{C}_{\mathbb{Z}_4}$ and taking a matrix product in the reverse order, i.e., of $2 \times 1$ with a $1 \times 2$ matrix

$$
\begin{aligned}
\left\{ D_1^{(\mathbb{Z}_2^V,\mathrm{id});(\mathrm{id})} \circ D_1^{(\mathrm{id},\mathbb{Z}_2^V);(\mathrm{id})} \right\}\Big|_{\mathcal{C}_{\mathbb{Z}_4}} &= \begin{pmatrix} D_1^{(\mathrm{id})} \\ D_1^{(\mathrm{id})} \end{pmatrix} \circ \begin{pmatrix} D_1^{(\mathrm{id})} & D_1^{(\mathrm{id})} \end{pmatrix} \\
&= \begin{pmatrix} D_1^{(\mathrm{id})} & 0 \\ 0 & D_1^{(\mathrm{id})} \end{pmatrix} \oplus \begin{pmatrix} 0 & D_1^{(\mathrm{id})} \\ D_1^{(\mathrm{id})} & 0 \end{pmatrix} \\
&= D_1^{(\mathbb{Z}_2^V;\mathrm{id})} \oplus D_1^{(\mathbb{Z}_2^V;-)}.
\end{aligned}
\tag{B.52}
$$

Similarly, one finds the following composition rules for 1-morphisms that carry non-trivial $\mathbb{Z}_2^V$ representations

$$
\begin{aligned}
D_1^{(\mathbb{Z}_2^V,\mathrm{id});(\mathrm{id})} \circ D_1^{(\mathrm{id},\mathbb{Z}_2^V);(V)} &= D_1^{(\mathbb{Z}_2^V;V)} \oplus D_1^{(\mathbb{Z}_2^V;V-)}, \\
D_1^{(\mathbb{Z}_2^V,\mathrm{id});(V)} \circ D_1^{(\mathrm{id},\mathbb{Z}_2^V);(\mathrm{id})} &= D_1^{(\mathbb{Z}_2^V;V)} \oplus D_1^{(\mathbb{Z}_2^V;V-)}, \\
D_1^{(\mathbb{Z}_2^V,\mathrm{id});(V)} \circ D_1^{(\mathrm{id},\mathbb{Z}_2^V);(V)} &= D_1^{(\mathbb{Z}_2^V;V)} \oplus D_1^{(\mathbb{Z}_2^V;V-)}.
\end{aligned}
\tag{B.53}
$$

Next we compute the composition of the 1-morphisms between $D_2^{(\mathbb{Z}_2^V)}$ and $D_2^{(\mathbb{Z}_4^S)}$. These have the form

$$
\begin{aligned}
\left\{ D_1^{(\mathbb{Z}_2^V,\mathbb{Z}_4^S)} \circ D_1^{(\mathbb{Z}_4^S,\mathbb{Z}_2^V)} \right\}\Big|_{\mathcal{C}_{\mathbb{Z}_4}} &= \begin{pmatrix} D_1^{(\mathrm{id})} & D_1^{(\mathrm{id})} & D_1^{(\mathrm{id})} & D_1^{(\mathrm{id})} \\ D_1^{(\mathrm{id})} & D_1^{(\mathrm{id})} & D_1^{(\mathrm{id})} & D_1^{(\mathrm{id})} \end{pmatrix} \cdot \begin{pmatrix} D_1^{(\mathrm{id})} & D_1^{(\mathrm{id})} \\ D_1^{(\mathrm{id})} & D_1^{(\mathrm{id})} \\ D_1^{(\mathrm{id})} & D_1^{(\mathrm{id})} \\ D_1^{(\mathrm{id})} & D_1^{(\mathrm{id})} \end{pmatrix} \\
&= \begin{pmatrix} 4D_1^{(\mathrm{id})} & 0 \\ 0 & 4D_1^{(\mathrm{id})} \end{pmatrix} \oplus \begin{pmatrix} 0 & 4D_1^{(\mathrm{id})} \\ 4D_1^{(\mathrm{id})} & 0 \end{pmatrix} \\
&= \left\{ 2D_1^{(\mathbb{Z}_2^V,\mathrm{id})} \oplus 2D_1^{(\mathbb{Z}_2^V,V)} \oplus 2D_1^{(\mathbb{Z}_2^V,-)} \oplus 2D_1^{(\mathbb{Z}_2^V,V-)} \right\}\Big|_{\mathcal{C}_{\mathbb{Z}_4}}.
\end{aligned}
\tag{B.54}
$$

Therefore we read-off the fusion rule

$$
D_1^{(\mathbb{Z}_2^V,\mathbb{Z}_4^S)} \circ D_1^{(\mathbb{Z}_4^S,\mathbb{Z}_2^V)} = 2D_1^{(\mathbb{Z}_2^V,\mathrm{id})} \oplus 2D_1^{(\mathbb{Z}_2^V,V)} \oplus 2D_1^{(\mathbb{Z}_2^V,-)} \oplus 2D_1^{(\mathbb{Z}_2^V,V-)}.
\tag{B.55}
$$

Similarly, we get

$$
\begin{aligned}
\left\{ D_1^{(\mathbb{Z}_2^V,\mathbb{Z}_2^V)} \circ D_1^{(\mathbb{Z}_4^S,\mathbb{Z}_4^S)} \right\}\Big|_{\mathcal{C}_{\mathbb{Z}_4}} &= \begin{pmatrix} D_1^{(\mathrm{id})} & D_1^{(\mathrm{id})} \\ D_1^{(\mathrm{id})} & D_1^{(\mathrm{id})} \\ D_1^{(\mathrm{id})} & D_1^{(\mathrm{id})} \\ D_1^{(\mathrm{id})} & D_1^{(\mathrm{id})} \end{pmatrix} \cdot \begin{pmatrix} D_1^{(\mathrm{id})} & D_1^{(\mathrm{id})} & D_1^{(\mathrm{id})} & D_1^{(\mathrm{id})} \\ D_1^{(\mathrm{id})} & D_1^{(\mathrm{id})} & D_1^{(\mathrm{id})} & D_1^{(\mathrm{id})} \end{pmatrix} \\
&= \left\{ 2D_1^{(\mathbb{Z}_4^S,\mathrm{id})} \oplus 2D_1^{(\mathbb{Z}_4^S,S)} \oplus 2D_1^{(\mathbb{Z}_4^S,V)} \oplus 2D_1^{(\mathbb{Z}_4^S,C)} \right\}\Big|_{\mathcal{C}_{\mathbb{Z}_4}}.
\end{aligned}
\tag{B.56}
$$

Therefore we read off the composition rule

$$
D_1^{(\mathbb{Z}_2^V,\mathbb{Z}_2^V)} \circ D_1^{(\mathbb{Z}_4^S,\mathbb{Z}_4^S)} = 2D_1^{(\mathbb{Z}_4^S,\mathrm{id})} \oplus 2D_1^{(\mathbb{Z}_4^S,S)} \oplus 2D_1^{(\mathbb{Z}_4^S,V)} \oplus 2D_1^{(\mathbb{Z}_4^S,C)}.
\tag{B.57}
$$

# C 1-morphisms in categorical symmetry webs

In this section, we apply the methods of this paper to compute 1-morphisms in partial and sequential gaugings for the $G = \mathbb{Z}_2 \times \mathbb{Z}_2$ and $G = \mathbb{Z}_4$ examples.

## C.1 1-morphisms in the $G = \mathbb{Z}_2 \times \mathbb{Z}_2$ web

In what follows, we denote by $D_2^{(\mathbb{Z}_2^V)}$ the object of $\mathcal{C}_{\mathbb{Z}_2 \times \mathbb{Z}_2 / \mathbb{Z}_2}$ that was denoted by $D_2^{(\mathbb{Z}_2)}$ in the main text. Similarly, we denote by $D_2^{(S\mathbb{Z}_2^V)}$ the object of $\mathcal{C}_{\mathbb{Z}_2 \times \mathbb{Z}_2 / \mathbb{Z}_2}$ that was denoted by $D_2^{(S\mathbb{Z}_2)}$ in the main text.

### C.1.1 1-morphisms for $\mathcal{C}_{\mathbb{Z}_2 \times \mathbb{Z}_2} \to \mathcal{C}_{\mathbb{Z}_2 \times \mathbb{Z}_2 / \mathbb{Z}_2}$

Let us begin by determining 1-endomorphisms of $D_2^{(\mathrm{id})} \in \mathcal{C}_{\mathbb{Z}_2^2 / \mathbb{Z}_2^V}$. These descend from different ways of making the 1-endomorphisms of $D_2^{(\mathrm{id})} \in \mathcal{C}_{\mathbb{Z}_2^2}$ symmetric under $\mathbb{Z}_2^V$. As already discussed above, the 1-endomorphism category of $D_2^{(\mathrm{id})} \in \mathcal{C}_{\mathbb{Z}_2^2}$ is Vec, generated by identity line $D_1^{(\mathrm{id})}$ of $\mathbb{Z}_2^2$. That is, a 1-endomorphism of $D_2^{(\mathrm{id})} \in \mathcal{C}_{\mathbb{Z}_2^2}$ is a vector space $W$ and a way of making it $\mathbb{Z}_2^V$ symmetric is provided by a representation $R$ of $\mathbb{Z}_2^V$ whose underlying vector space is $W$. Thus, 1-endomorphisms of $D_2^{(\mathrm{id})} \in \mathcal{C}_{\mathbb{Z}_2^2 / \mathbb{Z}_2^V}$ are given by representations of $\mathbb{Z}_2^V$ and form the category $\mathrm{Rep}(\mathbb{Z}_2^V)$. This category has two simple objects, both of which correspond to 1-dimensional representations, which means that both of them descend from $D_1^{(\mathrm{id})} \in \mathcal{C}_{\mathbb{Z}_2^2}$ without non-trivial multiplicity. Indeed, different ways of making $D_1^{(\mathrm{id})} \in \mathcal{C}_{\mathbb{Z}_2^2}$ symmetric under $\mathbb{Z}_2^V$ correspond to homomorphisms

$$\mathsf{Vec}(\mathbb{Z}_2^V) \to \mathsf{Vec} = \mathrm{End}(D_1^{(\mathrm{id})}), \tag{C.1}$$

where the right hand side describes the 2-endomorphisms or topological local operators living on $D_1^{(\mathrm{id})} \in \mathcal{C}_{\mathbb{Z}_2^2}$ which are generated by the identity local operator $D_0^{(\mathrm{id})}$ of $\mathbb{Z}_2^2$. These homomorphisms precisely correspond to irreducible representations of $\mathbb{Z}_2^V$.

Thus the set of simple 1-endomorphisms (upto isomorphism) of $D_2^{(\mathrm{id})} \in \mathcal{C}_{\mathbb{Z}_2^2 / \mathbb{Z}_2^V}$ is

$$\mathrm{End}_{\mathcal{C}_{\mathbb{Z}_2^2 / \mathbb{Z}_2^V}}(D_2^{(\mathrm{id})}) = \left\{ D_1^{(\mathrm{id})}, \quad D_1^{(V)} \right\}, \tag{C.2}$$

where $D_1^{(\mathrm{id})} \in \mathcal{C}_{\mathbb{Z}_2^2 / \mathbb{Z}_2^V}$ corresponds to trivial representation of $\mathbb{Z}_2^V$ and $D_1^{(-)} \in \mathcal{C}_{\mathbb{Z}_2^2 / \mathbb{Z}_2^V}$ corresponds to non-trivial irreducible representation of $\mathbb{Z}_2^V$.

The 1-endomorphisms of $D_2^{(\mathbb{Z}_2^V)} \in \mathcal{C}_{\mathbb{Z}_2^2 / \mathbb{Z}_2^V}$ descend from $\mathbb{Z}_2^V$ symmetric 1-endomorphisms of $2D_2^{(\mathrm{id})} \in \mathcal{C}_{\mathbb{Z}_2^2}$. As discussed above, the 1-endomorphisms of $2D_2^{(\mathrm{id})} \in \mathcal{C}_{\mathbb{Z}_2^2}$ form fusion 1-category $\mathsf{Mat}_2(\mathsf{Vec})$. The group $\mathbb{Z}_2^V$ acts on $\mathsf{Mat}_2(\mathsf{Vec})$ by simultaneously exchanging the two rows and two columns of these $2 \times 2$ matrices. Thus, the set of simple 1-endomorphisms (upto isomorphism) of $D_2^{(\mathbb{Z}_2^V)} \in \mathcal{C}_{\mathbb{Z}_2^2 / \mathbb{Z}_2^V}$ is

$$\mathcal{C}_{\mathbb{Z}_2^2 / \mathbb{Z}_2^V}^{D_2^{(\mathbb{Z}_2^V)}} = \left\{ D_1^{(\mathbb{Z}_2^V ; \mathrm{id})}, \quad D_1^{(\mathbb{Z}_2^V ; -)} \right\}, \tag{C.3}$$

where $D_1^{(\mathbb{Z}_2^V ; \mathrm{id})}$ corresponds to the diagonal matrix with both entries $D_1^{(\mathrm{id})}$ and $D_1^{(\mathbb{Z}_2^V ; -)}$ corresponds to the off-diagonal matrix with both entries $D_1^{(\mathrm{id})}$.

The set of simple 1-endomorphisms (upto isomorphism) of $D_2^{(S)} \in \mathcal{C}_{\mathbb{Z}_2^2/\mathbb{Z}_2^V}$ is determined analogously to the case of $D_2^{(\text{id})} \in \mathcal{C}_{\mathbb{Z}_2^2/\mathbb{Z}_2^V}$, and again corresponds to $\text{Rep}(\mathbb{Z}_2^V)$. We denote the simple 1-endomorphisms as

$$\mathcal{C}_{\mathbb{Z}_2^2/\mathbb{Z}_2^V}^{D_2^{(S)}} = \left\{ D_1^{(S;\text{id})}, \quad D_1^{(S;V)} \right\}. \tag{C.4}$$

Finally, the set of simple 1-endomorphisms (upto isomorphism) of $D_2^{(S\mathbb{Z}_2^V)} \in \mathcal{C}_{\mathbb{Z}_2^2/\mathbb{Z}_2^V}$ is

$$\mathcal{C}_{\mathbb{Z}_2^2/\mathbb{Z}_2^V}^{D_2^{(S\mathbb{Z}_2^V)}} = \left\{ D_1^{(S\mathbb{Z}_2^V;\text{id})}, \quad D_1^{(S\mathbb{Z}_2^V;-)} \right\}, \tag{C.5}$$

and the computation is completely analogous to the case of $D_2^{(\mathbb{Z}_2^V)} \in \mathcal{C}_{\mathbb{Z}_2^2/\mathbb{Z}_2^V}$.

The 1-morphisms from $D_2^{(\mathbb{Z}_2^V)} \in \mathcal{C}_{\mathbb{Z}_2^2/\mathbb{Z}_2^V}$ to $D_2^{(\text{id})} \in \mathcal{C}_{\mathbb{Z}_2^2/\mathbb{Z}_2^V}$ descend from 1-morphisms from $2D_2^{(\text{id})} \in \mathcal{C}_{\mathbb{Z}_2^2}$ to $D_2^{(\text{id})} \in \mathcal{C}_{\mathbb{Z}_2^2}$ which form the category $\text{Mat}_{2\times1}(\text{Vec})$ of $2 \times 1$ matrices valued in vector spaces. The $\mathbb{Z}_2^V$ action exchanges the two rows and thus the set of simple 1-morphisms (upto isomorphism) from $D_2^{(\mathbb{Z}_2^V)} \in \mathcal{C}_{\mathbb{Z}_2^2/\mathbb{Z}_2^V}$ to $D_2^{(\text{id})} \in \mathcal{C}_{\mathbb{Z}_2^2/\mathbb{Z}_2^V}$

$$\mathcal{C}_{\mathbb{Z}_2^2/\mathbb{Z}_2^V}^{D_2^{(\mathbb{Z}_2^V)}, D_2^{(\text{id})}} = \left\{ D_1^{(\mathbb{Z}_2^V,\text{id})} \right\}, \tag{C.6}$$

comprises of a single element $D_1^{(\mathbb{Z}_2^V,\text{id})}$ which corresponds to the $2 \times 1$ matrix with both entries $D_1^{(\text{id})}$. Similarly, the set of simple 1-morphisms (upto isomorphism) from $D_2^{(\text{id})} \in \mathcal{C}_{\mathbb{Z}_2^2/\mathbb{Z}_2^V}$ to $D_2^{(\mathbb{Z}_2^V)} \in \mathcal{C}_{\mathbb{Z}_2^2/\mathbb{Z}_2^V}$

$$\mathcal{C}_{\mathbb{Z}_2^2/\mathbb{Z}_2^V}^{D_2^{(\text{id})}, D_2^{(\mathbb{Z}_2^V)}} = \left\{ D_1^{(\text{id},\mathbb{Z}_2^V)} \right\}, \tag{C.7}$$

comprises of a single element $D_1^{(\text{id},\mathbb{Z}_2^V)}$ which corresponds to a $1 \times 2$ matrix in $\text{Mat}_{1\times2}(\text{Vec})$ with both entries $D_1^{(\text{id})}$, as the $\mathbb{Z}_2^V$ action exchanges the two columns.

In exactly the same way as above, we have that the set of simple 1-morphisms (upto isomorphism) from $D_2^{(S\mathbb{Z}_2^V)} \in \mathcal{C}_{\mathbb{Z}_2^2/\mathbb{Z}_2^V}$ to $D_2^{(S)} \in \mathcal{C}_{\mathbb{Z}_2^2/\mathbb{Z}_2^V}$

$$\mathcal{C}_{\mathbb{Z}_2^2/\mathbb{Z}_2^V}^{D_2^{(S\mathbb{Z}_2^V)}, D_2^{(S)}} = \left\{ D_1^{(S\mathbb{Z}_2^V,S)} \right\}, \tag{C.8}$$

comprises of a single element $D_1^{(S\mathbb{Z}_2^V,S)}$ corresponding to the $2 \times 1$ matrix in $\text{Mat}_{2\times1}(\text{Vec})$ with both entries $D_1^{(\text{id})}$, and that the set of simple 1-morphisms (upto isomorphism) from $D_2^{(S)} \in \mathcal{C}_{\mathbb{Z}_2^2/\mathbb{Z}_2^V}$ to $D_2^{(S\mathbb{Z}_2^V)} \in \mathcal{C}_{\mathbb{Z}_2^2/\mathbb{Z}_2^V}$

$$\mathcal{C}_{\mathbb{Z}_2^2/\mathbb{Z}_2^V}^{D_2^{(S)}, D_2^{(S\mathbb{Z}_2^V)}} = \left\{ D_1^{(S,S\mathbb{Z}_2^V)} \right\}, \tag{C.9}$$

comprises of a single element $D_1^{(S,S\mathbb{Z}_2^V)}$ corresponding to the $1 \times 2$ matrix in $\text{Mat}_{1\times2}(\text{Vec})$ with both entries $D_1^{(\text{id})}$.

Finally there are no 1-morphisms between the remaining pairs of objects in $\mathcal{C}_{\mathbb{Z}_2^2/\mathbb{Z}_2^V}$ as there are no 1-morphisms between $mD_2^{(\text{id})} \in \mathcal{C}_{\mathbb{Z}_2^2}$ and $nD_2^{(S)} \in \mathcal{C}_{\mathbb{Z}_2^2}$.

**Composition of 1-morphisms.** 1-morphisms are composed by combining the $\mathbb{Z}_2^V$ actions. Let us consider the composition of 1-endomorphisms of $D_2^{(\text{id})} \in \mathcal{C}_{\mathbb{Z}_2^2/\mathbb{Z}_2^V}$. These endomorphisms are labeled by $\mathbb{Z}_2^V$ representations, and the composition corresponds to taking tensor product of $\mathbb{Z}_2^V$ representations. At the level of simple objects, we learn that $D_1^{(\text{id})}$ is identity under composition and

$$D_1^{(V)} \circ D_1^{(V)} \cong D_1^{(\text{id})}, \tag{C.10}$$

that is $D_1^{(V)}$ squares to $D_1^{(\text{id})}$. Similarly, the 1-endomorphisms of $D_2^{(S)} \in \mathcal{C}_{\mathbb{Z}_2^2/\mathbb{Z}_2^V}$ are composed by tensoring $\mathbb{Z}_2^V$ representations, and the simple 1-endomorphisms form fusion 1-category $\text{Rep}(\mathbb{Z}_2^V)$ under composition with $D_1^{(S;\text{id})}$ being identity and

$$D_1^{(S;V)} \circ D_1^{(S;V)} \cong D_1^{(S;\text{id})}, \tag{C.11}$$

that is $D_1^{(S;V)}$ squaring to $D_1^{(S;\text{id})}$.

The composition of 1-endomorphisms of $D_2^{(\mathbb{Z}_2^V)} \in \mathcal{C}_{\mathbb{Z}_2^2/\mathbb{Z}_2^V}$ is obtained by multiplying matrices in $\text{Mat}_2(\text{Vec})$. This implies that $D_1^{(\mathbb{Z}_2^V;\text{id})}$ is identity 1-endomorphism under composition and

$$D_1^{(\mathbb{Z}_2^V;-)} \circ D_1^{(\mathbb{Z}_2^V;-)} \cong D_1^{(\mathbb{Z}_2^V;\text{id})}, \tag{C.12}$$

that is $D_1^{(\mathbb{Z}_2^V;-)}$ squares to $D_1^{(\mathbb{Z}_2^V;\text{id})}$. The composition of 1-endomorphisms of $D_2^{(S\mathbb{Z}_2^V)} \in \mathcal{C}_{\mathbb{Z}_2^2/\mathbb{Z}_2^V}$ follows the same procedure. The identity is $D_1^{(S\mathbb{Z}_2^V;\text{id})}$ and

$$D_1^{(S\mathbb{Z}_2^V;-)} \circ D_1^{(S\mathbb{Z}_2^V;-)} \cong D_1^{(S\mathbb{Z}_2^V;\text{id})}, \tag{C.13}$$

that is $D_1^{(S\mathbb{Z}_2^V;-)}$ squares to $D_1^{(S\mathbb{Z}_2^V;\text{id})}$.

The composition of 1-endomorphisms in $\mathcal{C}_{\mathbb{Z}_2^2/\mathbb{Z}_2^V}^{D_2^{(\mathbb{Z}_2^V)}}$ with 1-morphisms in $\mathcal{C}_{\mathbb{Z}_2^2/\mathbb{Z}_2^V}^{D_2^{(\mathbb{Z}_2^V)},D_2^{(\text{id})}}$ and $\mathcal{C}_{\mathbb{Z}_2^2/\mathbb{Z}_2^V}^{D_2^{(\text{id})},D_2^{(\mathbb{Z}_2^V)}}$ is simply matrix multiplication, leading to

$$\begin{aligned} D_1^{(\mathbb{Z}_2^V;x)} \circ D_1^{(\mathbb{Z}_2^V,\text{id})} &\cong D_1^{(\mathbb{Z}_2^V,\text{id})}, \\ D_1^{(\text{id},\mathbb{Z}_2^V)} \circ D_1^{(\mathbb{Z}_2^V;x)} &\cong D_1^{(\text{id},\mathbb{Z}_2^V)}, \end{aligned} \tag{C.14}$$

for $x \in \{\text{id}, -\}$. Similarly, we have

$$\begin{aligned} D_1^{(S\mathbb{Z}_2^V;x)} \circ D_1^{(S\mathbb{Z}_2^V,\text{id})} &\cong D_1^{(S\mathbb{Z}_2^V,\text{id})}, \\ D_1^{(\text{id},S\mathbb{Z}_2^V)} \circ D_1^{(S\mathbb{Z}_2^V;x)} &\cong D_1^{(\text{id},S\mathbb{Z}_2^V)}, \end{aligned} \tag{C.15}$$

for $x \in \{\text{id}, -\}$.

Finally, consider $D_1^{(\text{id},\mathbb{Z}_2^V)} \circ D_1^{(\mathbb{Z}_2^V,\text{id})}$. Matrix multiplication gives rise to $1 \times 1$ matrix with the only entry being $D_1^{(\text{id})} \oplus D_1^{(\text{id})}$. The $\mathbb{Z}_2^V$ action flips the two $D_1^{(\text{id})}$ factors in $D_1^{(\text{id})} \oplus D_1^{(\text{id})}$ making it into a two-dimensional representation of $\mathbb{Z}_2^V$ which decomposes into trivial plus non-trivial irreducible representations, which means that we have

$$D_1^{(\text{id},\mathbb{Z}_2^V)} \circ D_1^{(\mathbb{Z}_2^V,\text{id})} \cong D_1^{(\text{id})} \oplus D_1^{(V)}. \tag{C.16}$$

Similarly, for $D_1^{(\mathbb{Z}_2^V,\text{id})} \circ D_1^{(\text{id},\mathbb{Z}_2^V)}$, the matrix multiplication gives rise to $2 \times 2$ matrix with every entry being $D_1^{(\text{id})}$, which means that we have

$$D_1^{(\mathbb{Z}_2^V,\text{id})} \circ D_1^{(\text{id},\mathbb{Z}_2^V)} \cong D_1^{(\mathbb{Z}_2^V;\text{id})} \oplus D_1^{(\mathbb{Z}_2^V;-)}. \tag{C.17}$$

### C.1.2    1-morphisms for $\mathcal{C}_{\mathbb{Z}_2 \times \mathbb{Z}_2 / \mathbb{Z}_2} \to \mathcal{C}_{\mathbb{Z}_2^2 / \mathbb{Z}_2^2}$

We now consider the 1-morphisms after gauging – in a step-wise fashion – the full $\mathbb{Z}_2 \times \mathbb{Z}_2$ 0-form symmetry. This is to be compared with the 1-morphisms we determined in appendix B.1

Consider first the 1-endomorphisms of $D_2^{(\mathrm{id})} \in \mathcal{C}_{\mathbb{Z}_2^2 / \mathbb{Z}_2^2}$. These descend from $\mathbb{Z}_2^S$ symmetric 1-endomorphisms of $D_2^{(\mathrm{id})} \in \mathcal{C}_{\mathbb{Z}_2^2 / \mathbb{Z}_2^V}$. We can dress $D_1^{(\mathrm{id})} \in \mathcal{C}_{\mathbb{Z}_2^2 / \mathbb{Z}_2^V}$ by irreducible representations of $\mathbb{Z}_2^S$ to obtain two simple 1-endomorphisms

$$D_1^{(\mathrm{id})}, \qquad D_1^{(V)}, \tag{C.18}$$

of $D_2^{(\mathrm{id})} \in \mathcal{C}_{\mathbb{Z}_2^2 / \mathbb{Z}_2^2}$, where $D_1^{(\mathrm{id})} \in \mathcal{C}_{\mathbb{Z}_2^2 / \mathbb{Z}_2^2}$ corresponds to the trivial representation of $\mathbb{Z}_2^S$ and $D_1^{(V)} \in \mathcal{C}_{\mathbb{Z}_2^2 / \mathbb{Z}_2^2}$ corresponds to the non-trivial irreducible representation of $\mathbb{Z}_2^S$. Similarly, we can dress $D_1^{(V)} \in \mathcal{C}_{\mathbb{Z}_2^2 / \mathbb{Z}_2^V}$ by irreducible representations of $\mathbb{Z}_2^S$ to obtain two simple 1-endomorphisms

$$D_1^{(S)}, \qquad D_1^{(C)}, \tag{C.19}$$

of $D_2^{(\mathrm{id})} \in \mathcal{C}_{\mathbb{Z}_2^2 / \mathbb{Z}_2^2}$, where $D_1^{(S)} \in \mathcal{C}_{\mathbb{Z}_2^2 / \mathbb{Z}_2^2}$ corresponds to the trivial representation of $\mathbb{Z}_2^S$ and $D_1^{(C)} \in \mathcal{C}_{\mathbb{Z}_2^2 / \mathbb{Z}_2^2}$ corresponds to the non-trivial irreducible representation of $\mathbb{Z}_2^S$. Thus,

$$\mathcal{C}_{\mathbb{Z}_2^2 / \mathbb{Z}_2^2}^{D_2^{(\mathrm{id})}} = \left\{ D_1^{(\mathrm{id})}, \quad D_1^{(S)}, \quad D_1^{(C)}, \quad D_1^{(V)} \right\}, \tag{C.20}$$

is the full set of simple 1-endomorphisms (upto isomorphism) of $D_2^{(\mathrm{id})} \in \mathcal{C}_{\mathbb{Z}_2^2 / \mathbb{Z}_2^2}$.

Similarly,

$$\mathcal{C}_{\mathbb{Z}_2^2 / \mathbb{Z}_2^2}^{D_2^{(-)}} = \left\{ D_1^{(-;\mathrm{id})}, \quad D_1^{(-;S)}, \quad D_1^{(-;C)}, \quad D_1^{(-;V)} \right\}, \tag{C.21}$$

is the set of simple 1-endomorphisms (upto isomorphism) of $D_2^{(-)} \in \mathcal{C}_{\mathbb{Z}_2^2 / \mathbb{Z}_2^2}$ (where we have used analogous notation to the one used for elements of $\mathcal{C}_{\mathbb{Z}_2^2 / \mathbb{Z}_2^2}^{D_2^{(\mathrm{id})}}$).

The set of simple 1-morphisms (upto isomorphism) from $D_2^{(\mathrm{id})} \in \mathcal{C}_{\mathbb{Z}_2^2 / \mathbb{Z}_2^2}$ to $D_2^{(-)} \in \mathcal{C}_{\mathbb{Z}_2^2 / \mathbb{Z}_2^2}$

$$\mathcal{C}_{\mathbb{Z}_2^2 / \mathbb{Z}_2^2}^{D_2^{(\mathrm{id})}, D_2^{(-)}} = \left\{ D_1^{(\mathrm{id}, -)} \right\}, \tag{C.22}$$

comprises of a single element $D_1^{(\mathrm{id}, -)}$, whose underlying 1-morphism is $D_1^{(\mathrm{id})} \oplus D_1^{(V)} \in \mathcal{C}_{\mathbb{Z}_2^2 / \mathbb{Z}_2^V}$. Similarly, the set of simple 1-morphisms (upto isomorphism) from $D_2^{(-)} \in \mathcal{C}_{\mathbb{Z}_2^2 / \mathbb{Z}_2^2}$ to $D_2^{(\mathrm{id})} \in \mathcal{C}_{\mathbb{Z}_2^2 / \mathbb{Z}_2^2}$

$$\mathcal{C}_{\mathbb{Z}_2^2 / \mathbb{Z}_2^2}^{D_2^{(-)}, D_2^{(\mathrm{id})}} = \left\{ D_1^{(-, \mathrm{id})} \right\}, \tag{C.23}$$

comprises of a single element $D_1^{(-, \mathrm{id})}$, whose underlying 1-morphism is again $D_1^{(\mathrm{id})} \oplus D_1^{(V)} \in \mathcal{C}_{\mathbb{Z}_2^2 / \mathbb{Z}_2^V}$.

The set of simple 1-morphisms (upto isomorphism) from $D_2^{(\mathbb{Z}_2^V)} \in \mathcal{C}_{\mathbb{Z}_2^2 / \mathbb{Z}_2^2}$ to $D_2^{(\mathrm{id})} \in \mathcal{C}_{\mathbb{Z}_2^2 / \mathbb{Z}_2^2}$ is

$$\mathcal{C}_{\mathbb{Z}_2^2 / \mathbb{Z}_2^2}^{D_2^{(\mathbb{Z}_2^V)}, D_2^{(\mathrm{id})}} = \left\{ D_1^{(\mathbb{Z}_2^V, \mathrm{id}; +)}, \quad D_1^{(\mathbb{Z}_2^V, \mathrm{id}; -)} \right\}, \tag{C.24}$$

where $D_1^{(\mathbb{Z}_2^V, \mathrm{id}; +)}$ descends from 1-morphism $2 D_2^{(\mathrm{id})} \to D_2^{(\mathrm{id})} \in \mathcal{C}_{\mathbb{Z}_2^2 / \mathbb{Z}_2^V}$ described by a $2 \times 1$ matrix with both entries given by $D_1^{(\mathrm{id})} \in \mathcal{C}_{\mathbb{Z}_2^2 / \mathbb{Z}_2^V}$, and $D_1^{(\mathbb{Z}_2^V, \mathrm{id}; -)}$ is described by a $2 \times 1$ matrix with both

entries given by $D_1^{(V)} \in \mathcal{C}_{\mathbb{Z}_2^2/\mathbb{Z}_2^V}$. Similarly, the set of simple 1-morphisms (upto isomorphism) from $D_2^{(\mathrm{id})} \in \mathcal{C}_{\mathbb{Z}_2^2/\mathbb{Z}_2^2}$ to $D_2^{(\mathbb{Z}_2^V)} \in \mathcal{C}_{\mathbb{Z}_2^2/\mathbb{Z}_2^2}$ is

$$\mathcal{C}_{\mathbb{Z}_2^2/\mathbb{Z}_2^2}^{D_2^{(\mathrm{id})}, D_2^{(\mathbb{Z}_2^V)}} = \left\{ D_1^{(\mathrm{id}, \mathbb{Z}_2^V; +)}, \quad D_1^{(\mathrm{id}, \mathbb{Z}_2^V; -)} \right\}, \tag{C.25}$$

where $D_1^{(\mathrm{id}, \mathbb{Z}_2^V; +)}$ is described by a $1 \times 2$ matrix with both entries given by $D_1^{(\mathrm{id})} \in \mathcal{C}_{\mathbb{Z}_2^2/\mathbb{Z}_2^V}$, and $D_1^{(\mathrm{id}, \mathbb{Z}_2^V; -)}$ is described by a $1 \times 2$ matrix with both entries given by $D_1^{(V)} \in \mathcal{C}_{\mathbb{Z}_2^2/\mathbb{Z}_2^V}$.

The set of simple 1-morphisms (upto isomorphism) from $D_2^{(\mathbb{Z}_2^V)} \in \mathcal{C}_{\mathbb{Z}_2^2/\mathbb{Z}_2^2}$ to $D_2^{(-)} \in \mathcal{C}_{\mathbb{Z}_2^2/\mathbb{Z}_2^2}$ is

$$\mathcal{C}_{\mathbb{Z}_2^2/\mathbb{Z}_2^2}^{D_2^{(\mathbb{Z}_2^V)}, D_2^{(-)}} = \left\{ D_1^{(\mathbb{Z}_2^V, -; +)}, \quad D_1^{(\mathbb{Z}_2^V, -; -)} \right\}, \tag{C.26}$$

where $D_1^{(\mathbb{Z}_2^V, -; +)}$ descends from 1-morphism $2D_2^{(\mathrm{id})} \to D_2^{(\mathrm{id})} \in \mathcal{C}_{\mathbb{Z}_2^2/\mathbb{Z}_2^V}$ described by a $2 \times 1$ matrix with first entry given by $D_1^{(\mathrm{id})} \in \mathcal{C}_{\mathbb{Z}_2^2/\mathbb{Z}_2^V}$ and second entry given by $D_1^{(V)} \in \mathcal{C}_{\mathbb{Z}_2^2/\mathbb{Z}_2^V}$, and $D_1^{(\mathbb{Z}_2^V, -; -)}$ is described by a $2 \times 1$ matrix with first entry given by $D_1^{(V)} \in \mathcal{C}_{\mathbb{Z}_2^2/\mathbb{Z}_2^V}$ and second entry given by $D_1^{(\mathrm{id})} \in \mathcal{C}_{\mathbb{Z}_2^2/\mathbb{Z}_2^V}$. Similarly, the set of simple 1-morphisms (upto isomorphism) from $D_2^{(-)} \in \mathcal{C}_{\mathbb{Z}_2^2/\mathbb{Z}_2^2}$ to $D_2^{(\mathbb{Z}_2^V)} \in \mathcal{C}_{\mathbb{Z}_2^2/\mathbb{Z}_2^2}$ is

$$\mathcal{C}_{\mathbb{Z}_2^2/\mathbb{Z}_2^2}^{D_2^{(-)}, D_2^{(\mathbb{Z}_2^V)}} = \left\{ D_1^{(-, \mathbb{Z}_2^V; +)}, \quad D_1^{(-, \mathbb{Z}_2^V; -)} \right\}, \tag{C.27}$$

where $D_1^{(-, \mathbb{Z}_2^V; +)}$ is described by a $1 \times 2$ matrix with first entry given by $D_1^{(\mathrm{id})} \in \mathcal{C}_{\mathbb{Z}_2^2/\mathbb{Z}_2^V}$ and second entry given by $D_1^{(V)} \in \mathcal{C}_{\mathbb{Z}_2^2/\mathbb{Z}_2^V}$, and $D_1^{(-, \mathbb{Z}_2^V; -)}$ is described by a $1 \times 2$ matrix with first entry given by $D_1^{(V)} \in \mathcal{C}_{\mathbb{Z}_2^2/\mathbb{Z}_2^V}$ and second entry given by $D_1^{(\mathrm{id})} \in \mathcal{C}_{\mathbb{Z}_2^2/\mathbb{Z}_2^V}$.

The set of simple 1-morphisms (upto isomorphism) from $D_2^{(\mathbb{Z}_2^x)} \in \mathcal{C}_{\mathbb{Z}_2^2/\mathbb{Z}_2^2}$ for $x \in \{S, C\}$ to $D_2^{(\mathrm{id})} \in \mathcal{C}_{\mathbb{Z}_2^2/\mathbb{Z}_2^2}$ is

$$\mathcal{C}_{\mathbb{Z}_2^2/\mathbb{Z}_2^2}^{D_2^{(\mathbb{Z}_2^x)}, D_2^{(\mathrm{id})}} = \left\{ D_1^{(\mathbb{Z}_2^x, \mathrm{id}; +)}, \quad D_1^{(\mathbb{Z}_2^x, \mathrm{id}; -)} \right\}, \tag{C.28}$$

where $D_1^{(\mathbb{Z}_2^x, \mathrm{id}; s)}$ descends from 1-morphism $D_1^{(\mathbb{Z}_2^V, \mathrm{id})} \in \mathcal{C}_{\mathbb{Z}_2^2/\mathbb{Z}_2^V}$ carrying a representation of $\mathbb{Z}_2^S$ described by sign $s$. Similarly, the set of simple 1-morphisms (upto isomorphism) from $D_2^{(\mathrm{id})} \in \mathcal{C}_{\mathbb{Z}_2^2/\mathbb{Z}_2^2}$ to $D_2^{(\mathbb{Z}_2^x)} \in \mathcal{C}_{\mathbb{Z}_2^2/\mathbb{Z}_2^2}$ is

$$\mathcal{C}_{\mathbb{Z}_2^2/\mathbb{Z}_2^2}^{D_2^{(\mathrm{id})}, D_2^{(\mathbb{Z}_2^x)}} = \left\{ D_1^{(\mathrm{id}, \mathbb{Z}_2^x; +)}, \quad D_1^{(\mathrm{id}, \mathbb{Z}_2^x; -)} \right\}, \tag{C.29}$$

where $D_1^{(\mathrm{id}, \mathbb{Z}_2^x; s)}$ descends from 1-morphism $D_1^{(\mathrm{id}, \mathbb{Z}_2^V)} \in \mathcal{C}_{\mathbb{Z}_2^2/\mathbb{Z}_2^V}$ carrying a representation of $\mathbb{Z}_2^S$ described by sign $s$.

Similarly, the set of simple 1-morphisms (upto isomorphism) from $D_2^{(\mathbb{Z}_2^x)} \in \mathcal{C}_{\mathbb{Z}_2^2/\mathbb{Z}_2^2}$ for $x \in \{S, C\}$ to $D_2^{(-)} \in \mathcal{C}_{\mathbb{Z}_2^2/\mathbb{Z}_2^2}$ is

$$\mathcal{C}_{\mathbb{Z}_2^2/\mathbb{Z}_2^2}^{D_2^{(\mathbb{Z}_2^x)}, D_2^{(-)}} = \left\{ D_1^{(\mathbb{Z}_2^x, -; +)}, \quad D_1^{(\mathbb{Z}_2^x, -; -)} \right\}, \tag{C.30}$$

and the set of simple 1-morphisms (upto isomorphism) from $D_2^{(-)} \in \mathcal{C}_{\mathbb{Z}_2^2/\mathbb{Z}_2^2}$ to $D_2^{(\mathbb{Z}_2^x)} \in \mathcal{C}_{\mathbb{Z}_2^2/\mathbb{Z}_2^2}$ is

$$\mathcal{C}_{\mathbb{Z}_2^2/\mathbb{Z}_2^2}^{D_2^{(-)}, D_2^{(\mathbb{Z}_2^x)}} = \left\{ D_1^{(-, \mathbb{Z}_2^x; +)}, \quad D_1^{(-, \mathbb{Z}_2^x; -)} \right\}. \tag{C.31}$$

The set of simple 1-endomorphisms (upto isomorphism) of $D_2^{(\mathbb{Z}_2^V)} \in \mathcal{C}_{\mathbb{Z}_2^2/\mathbb{Z}_2^2}$ is

$$\mathcal{C}_{\mathbb{Z}_2^2/\mathbb{Z}_2^2}^{D_2^{(\mathbb{Z}_2^V)}} = \left\{ D_1^{(\mathbb{Z}_2^V;\mathrm{id}+)}, \quad D_1^{(\mathbb{Z}_2^V;\mathrm{id}-)}, \quad D_1^{(\mathbb{Z}_2^V;-+)}, \quad D_1^{(\mathbb{Z}_2^V;--)} \right\}, \tag{C.32}$$

where $D_1^{(\mathbb{Z}_2^V;\mathrm{id}+)}$ descends from 1-morphism $2D_2^{(\mathrm{id})} \to 2D_2^{(\mathrm{id})} \in \mathcal{C}_{\mathbb{Z}_2^2/\mathbb{Z}_2^V}$ described by a diagonal $2 \times 2$ matrix with both entries given by $D_1^{(\mathrm{id})} \in \mathcal{C}_{\mathbb{Z}_2^2/\mathbb{Z}_2^V}$, $D_1^{(\mathbb{Z}_2^V;\mathrm{id}-)}$ is described by a diagonal $2 \times 2$ matrix with both entries given by $D_1^{(V)} \in \mathcal{C}_{\mathbb{Z}_2^2/\mathbb{Z}_2^V}$, $D_1^{(\mathbb{Z}_2^V;-+)}$ is described by an off-diagonal $2 \times 2$ matrix with both entries given by $D_1^{(\mathrm{id})} \in \mathcal{C}_{\mathbb{Z}_2^2/\mathbb{Z}_2^V}$, and $D_1^{(\mathbb{Z}_2^V;--)}$ is described by an off-diagonal $2 \times 2$ matrix with both entries given by $D_1^{(V)} \in \mathcal{C}_{\mathbb{Z}_2^2/\mathbb{Z}_2^V}$.

The set of simple 1-endomorphisms (upto isomorphism) of $D_2^{(\mathbb{Z}_2^x)} \in \mathcal{C}_{\mathbb{Z}_2^2/\mathbb{Z}_2^2}$ for $x \in \{S, C\}$ is

$$\mathcal{C}_{\mathbb{Z}_2^2/\mathbb{Z}_2^2}^{D_2^{(\mathbb{Z}_2^x)}} = \left\{ D_1^{(\mathbb{Z}_2^x;\mathrm{id}+)}, \quad D_1^{(\mathbb{Z}_2^x;\mathrm{id}-)}, \quad D_1^{(\mathbb{Z}_2^x;-+)}, \quad D_1^{(\mathbb{Z}_2^x;--)} \right\}, \tag{C.33}$$

where $D_1^{(\mathbb{Z}_2^x;\mathrm{id}s)}$ descends from $D_1^{(\mathbb{Z}_2^V;\mathrm{id})} \in \mathcal{C}_{\mathbb{Z}_2^2/\mathbb{Z}_2^V}$ carrying a representation of $\mathbb{Z}_2^S$ described by sign $s$, and $D_1^{(\mathbb{Z}_2^x;-s)}$ descends from $D_1^{(\mathbb{Z}_2^V;-)} \in \mathcal{C}_{\mathbb{Z}_2^2/\mathbb{Z}_2^V}$ carrying a representation of $\mathbb{Z}_2^S$ described by sign $s$.

The set of simple 1-morphisms (upto isomorphism) from $D_2^{(\mathbb{Z}_2^x)} \in \mathcal{C}_{\mathbb{Z}_2^2/\mathbb{Z}_2^2}$ to $D_2^{(\mathbb{Z}_2^y)} \in \mathcal{C}_{\mathbb{Z}_2^2/\mathbb{Z}_2^2}$ for $x \neq y \in \{S, C\}$ is

$$\mathcal{C}_{\mathbb{Z}_2^2/\mathbb{Z}_2^2}^{D_2^{(\mathbb{Z}_2^x)}, D_2^{(\mathbb{Z}_2^y)}} = \left\{ D_1^{(\mathbb{Z}_2^x, \mathbb{Z}_2^y)} \right\}, \tag{C.34}$$

where $D_1^{(\mathbb{Z}_2^x, \mathbb{Z}_2^y)}$ descends from 1-morphism $D_1^{(\mathbb{Z}_2^V;\mathrm{id})} \oplus D_1^{(\mathbb{Z}_2^V;-)} \in \mathcal{C}_{\mathbb{Z}_2^2/\mathbb{Z}_2^V}$.

The set of simple 1-morphisms (upto isomorphism) from $D_2^{(\mathbb{Z}_2^V)} \in \mathcal{C}_{\mathbb{Z}_2^2/\mathbb{Z}_2^2}$ to $D_2^{(\mathbb{Z}_2^x)} \in \mathcal{C}_{\mathbb{Z}_2^2/\mathbb{Z}_2^2}$ for $x \in \{S, C\}$ is

$$\mathcal{C}_{\mathbb{Z}_2^2/\mathbb{Z}_2^2}^{D_2^{(\mathbb{Z}_2^V)}, D_2^{(\mathbb{Z}_2^x)}} = \left\{ D_1^{(\mathbb{Z}_2^V, \mathbb{Z}_2^x)} \right\}, \tag{C.35}$$

where $D_1^{(\mathbb{Z}_2^V, \mathbb{Z}_2^x)}$ descends from $2 \times 1$ matrix with both entries given by $D_1^{(\mathrm{id}, \mathbb{Z}_2^V)} \in \mathcal{C}_{\mathbb{Z}_2^2/\mathbb{Z}_2^V}$. Similarly, the set of simple 1-morphisms (upto isomorphism) from $D_2^{(\mathbb{Z}_2^x)} \in \mathcal{C}_{\mathbb{Z}_2^2/\mathbb{Z}_2^2}$ to $D_2^{(\mathbb{Z}_2^V)} \in \mathcal{C}_{\mathbb{Z}_2^2/\mathbb{Z}_2^2}$ for $x \in \{S, C\}$ is

$$\mathcal{C}_{\mathbb{Z}_2^2/\mathbb{Z}_2^2}^{D_2^{(\mathbb{Z}_2^x)}, D_2^{(\mathbb{Z}_2^V)}} = \left\{ D_1^{(\mathbb{Z}_2^x, \mathbb{Z}_2^V)} \right\}, \tag{C.36}$$

where $D_1^{(\mathbb{Z}_2^x, \mathbb{Z}_2^V)}$ descends from $1 \times 2$ matrix with both entries given by $D_1^{(\mathbb{Z}_2^V, \mathrm{id})} \in \mathcal{C}_{\mathbb{Z}_2^2/\mathbb{Z}_2^V}$.

Additionally, we have

- One simple 1-morphism (upto isomorphism) from $D_2^{(\mathbb{Z}_2 \times \mathbb{Z}_2)}$ to $D_2^{(x)}$ for $x \in \{\mathrm{id}, -\}$ descending from $2 \times 1$ matrix with both entries given by $D_1^{(\mathbb{Z}_2^V, \mathrm{id})} \in \mathcal{C}_{\mathbb{Z}_2^2/\mathbb{Z}_2^V}$.

- One simple 1-morphism (upto isomorphism) from $D_2^{(x)}$ to $D_2^{(\mathbb{Z}_2 \times \mathbb{Z}_2)}$ for $x \in \{\mathrm{id}, -\}$ descending from $1 \times 2$ matrix with both entries given by $D_1^{(\mathrm{id}, \mathbb{Z}_2^V)} \in \mathcal{C}_{\mathbb{Z}_2^2/\mathbb{Z}_2^V}$.

- Two simple 1-morphisms (upto isomorphism) from $D_2^{(\mathbb{Z}_2 \times \mathbb{Z}_2)}$ to $D_2^{(\mathbb{Z}_2^S)}$, with one of them descending from $2 \times 1$ matrix with both entries given by $D_1^{(\mathbb{Z}_2^V;\mathrm{id})} \in \mathcal{C}_{\mathbb{Z}_2^2/\mathbb{Z}_2^V}$, and the other descending from $2 \times 1$ matrix with both entries given by $D_1^{(\mathbb{Z}_2^V;-)} \in \mathcal{C}_{\mathbb{Z}_2^2/\mathbb{Z}_2^V}$.

- Two simple 1-morphisms (upto isomorphism) from $D_2^{(\mathbb{Z}_2^S)}$ to $D_2^{(\mathbb{Z}_2 \times \mathbb{Z}_2)}$, with one of them descending from $1 \times 2$ matrix with both entries given by $D_1^{(\mathbb{Z}_2^V; \mathrm{id})} \in \mathcal{C}_{\mathbb{Z}_2^2/\mathbb{Z}_2^V}$, and the other descending from $1 \times 2$ matrix with both entries given by $D_1^{(\mathbb{Z}_2^V; -)} \in \mathcal{C}_{\mathbb{Z}_2^2/\mathbb{Z}_2^V}$.

- Two simple 1-morphisms (upto isomorphism) from $D_2^{(\mathbb{Z}_2 \times \mathbb{Z}_2)}$ to $D_2^{(\mathbb{Z}_2^C)}$, with one of them descending from $2 \times 1$ matrix with first entry given by $D_1^{(\mathbb{Z}_2^V; \mathrm{id})} \in \mathcal{C}_{\mathbb{Z}_2^2/\mathbb{Z}_2^V}$ and the second entry given by $D_1^{(\mathbb{Z}_2^V; -)} \in \mathcal{C}_{\mathbb{Z}_2^2/\mathbb{Z}_2^V}$, and the other descending from $2 \times 1$ matrix with first entry given by $D_1^{(\mathbb{Z}_2^V; -)} \in \mathcal{C}_{\mathbb{Z}_2^2/\mathbb{Z}_2^V}$ and the second entry given by $D_1^{(\mathbb{Z}_2^V; \mathrm{id})} \in \mathcal{C}_{\mathbb{Z}_2^2/\mathbb{Z}_2^V}$.

- Two simple 1-morphisms (upto isomorphism) from $D_2^{(\mathbb{Z}_2^C)}$ to $D_2^{(\mathbb{Z}_2 \times \mathbb{Z}_2)}$, with one of them descending from $1 \times 2$ matrix with first entry given by $D_1^{(\mathbb{Z}_2^V; \mathrm{id})} \in \mathcal{C}_{\mathbb{Z}_2^2/\mathbb{Z}_2^V}$ and the second entry given by $D_1^{(\mathbb{Z}_2^V; -)} \in \mathcal{C}_{\mathbb{Z}_2^2/\mathbb{Z}_2^V}$, and the other descending from $1 \times 2$ matrix with first entry given by $D_1^{(\mathbb{Z}_2^V; -)} \in \mathcal{C}_{\mathbb{Z}_2^2/\mathbb{Z}_2^V}$ and the second entry given by $D_1^{(\mathbb{Z}_2^V; \mathrm{id})} \in \mathcal{C}_{\mathbb{Z}_2^2/\mathbb{Z}_2^V}$.

- Two simple 1-morphisms (upto isomorphism) from $D_2^{(\mathbb{Z}_2 \times \mathbb{Z}_2)}$ to $D_2^{(\mathbb{Z}_2^V)}$, with one of them descending from diagonal $2 \times 2$ matrix with both entries given by $D_1^{(\mathbb{Z}_2^V, \mathrm{id})} \in \mathcal{C}_{\mathbb{Z}_2^2/\mathbb{Z}_2^V}$, and the other descending from off-diagonal $2 \times 2$ matrix with both entries given by $D_1^{(\mathbb{Z}_2^V, \mathrm{id})} \in \mathcal{C}_{\mathbb{Z}_2^2/\mathbb{Z}_2^V}$.

- Two simple 1-morphisms (upto isomorphism) from $D_2^{(\mathbb{Z}_2^V)}$ to $D_2^{(\mathbb{Z}_2 \times \mathbb{Z}_2)}$, with one of them descending from diagonal $2 \times 2$ matrix with both entries given by $D_1^{(\mathbb{Z}_2^V, \mathrm{id})} \in \mathcal{C}_{\mathbb{Z}_2^2/\mathbb{Z}_2^V}$, and the other descending from off-diagonal $2 \times 2$ matrix with both entries given by $D_1^{(\mathbb{Z}_2^V, \mathrm{id})} \in \mathcal{C}_{\mathbb{Z}_2^2/\mathbb{Z}_2^V}$.

- Four simple 1-endomorphisms (upto isomorphism) of $D_2^{(\mathbb{Z}_2 \times \mathbb{Z}_2)}$, given by either diagonal or off-diagonal $2 \times 2$ matrices with both entries either being $D_1^{(\mathbb{Z}_2^V; \mathrm{id})} \in \mathcal{C}_{\mathbb{Z}_2^2/\mathbb{Z}_2^V}$ or being $D_1^{(\mathbb{Z}_2^V; -)} \in \mathcal{C}_{\mathbb{Z}_2^2/\mathbb{Z}_2^V}$.

One can also compute the composition of 1-morphisms and find them to be the same as those in appendix B.1. As an example, consider computing $D_1^{(\mathrm{id}, -)} \circ D_1^{(-, \mathrm{id})}$ in $\mathcal{C}_{\mathbb{Z}_2^2/\mathbb{Z}_2^2}$. The composition descends from the 1-morphism $2D_1^{(\mathrm{id})} \oplus 2D_1^{(V)} \in \mathcal{C}_{\mathbb{Z}_2^2/\mathbb{Z}_2^V}$. The action of $\mathbb{Z}_2^S$ on this 1-morphism exchanges the two copies of $D_1^{(\mathrm{id})}$ and the two copies of $D_1^{(V)}$, thus leading to the result

$$D_1^{(\mathrm{id}, -)} \circ D_1^{(-, \mathrm{id})} \cong D_1^{(\mathrm{id})} \oplus D_1^{(S)} \oplus D_1^{(C)} \oplus D_1^{(V)}, \tag{C.37}$$

in $\mathcal{C}_{\mathbb{Z}_2^2/\mathbb{Z}_2^2}$, which matches the result found in (B.23).

We leave the matching of other composition rules and fusion rules for 1-morphisms to the interested reader.

## C.2 1-morphisms in the $G = \mathbb{Z}_4$ web

The 1-morphisms and their composition & fusion rules for $\mathcal{C}_{\mathbb{Z}_4/\mathbb{Z}_2}$ are the same as for $\mathcal{C}_{\mathbb{Z}_2^2/\mathbb{Z}_2}$. The derivation from $\mathbb{Z}_2$ gauging of $\mathcal{C}_{\mathbb{Z}_4}$ is also similar to the $\mathbb{Z}_2$ gauging of $\mathcal{C}_{\mathbb{Z}_2^2}$ discussed above.

We thus discuss the computation of 1-morphisms in the subsequent $\mathbb{Z}_2$ gauging

$$\mathcal{C}_{\mathbb{Z}_4/\mathbb{Z}_2} \to \mathcal{C}_{\mathbb{Z}_4/\mathbb{Z}_4}\,, \tag{C.38}$$

in the rest of this subsection.

**1-morphisms.**    Now we compute simple 1-morphisms of $\mathcal{C}_{\mathbb{Z}_4/\mathbb{Z}_4}$ by gauging $\mathbb{Z}_2^S$ 0-form symmetry in $\mathcal{C}_{\mathbb{Z}_4/\mathbb{Z}_2^V}$.

Let us begin by computing simple 1-endomorphisms of $D_2^{(\mathrm{id})} \in \mathcal{C}_{\mathbb{Z}_4/\mathbb{Z}_4}$. They descend from simple 1-endomorphisms of $D_2^{(\mathrm{id})} \in \mathcal{C}_{\mathbb{Z}_4/\mathbb{Z}_2^V}$ which are $D_1^{(\mathrm{id})} \in \mathcal{C}_{\mathbb{Z}_4/\mathbb{Z}_2^V}$ and $D_1^{(V)} \in \mathcal{C}_{\mathbb{Z}_4/\mathbb{Z}_2^V}$. First consider making $D_1^{(\mathrm{id})} \in \mathcal{C}_{\mathbb{Z}_4/\mathbb{Z}_2^V}$ symmetric under $\mathbb{Z}_2^S$, which is done by choosing a local operator $O$ on $D_1^{(\mathrm{id})} \in \mathcal{C}_{\mathbb{Z}_4/\mathbb{Z}_2^V}$ which squares to the identity local operator $D_0^{(\mathrm{id})} \in \mathcal{C}_{\mathbb{Z}_4/\mathbb{Z}_2^V}$ living on the line. There are two possible choices given by $O = D_0^{(\mathrm{id})} \in \mathcal{C}_{\mathbb{Z}_4/\mathbb{Z}_2^V}$ and $O = -D_0^{(\mathrm{id})} \in \mathcal{C}_{\mathbb{Z}_4/\mathbb{Z}_2^V}$, leading to simple 1-endomorphisms of $D_2^{(\mathrm{id})}$ in $\mathcal{C}_{\mathbb{Z}_4/\mathbb{Z}_4}$ that we label as

$$D_1^{(\mathrm{id})}\,, \qquad D_1^{(V)}\,. \tag{C.39}$$

Now consider making $D_1^{(V)} \in \mathcal{C}_{\mathbb{Z}_4/\mathbb{Z}_2^V}$ symmetric under $\mathbb{Z}_2^S$. Due to symmetry fractionalization, we are now looking for an operator $O$ on $D_1^{(V)} \in \mathcal{C}_{\mathbb{Z}_4/\mathbb{Z}_2^V}$ which squares to $-D_0^{(V;\mathrm{id})} \in \mathcal{C}_{\mathbb{Z}_4/\mathbb{Z}_2^V}$ where $D_0^{(V;\mathrm{id})} \in \mathcal{C}_{\mathbb{Z}_4/\mathbb{Z}_2^V}$ is the identity local operator on $D_1^{(V)} \in \mathcal{C}_{\mathbb{Z}_4/\mathbb{Z}_2^V}$. We again have two choices, $iD_0^{(V;\mathrm{id})} \in \mathcal{C}_{\mathbb{Z}_4/\mathbb{Z}_2^V}$ and $-iD_0^{(V;\mathrm{id})} \in \mathcal{C}_{\mathbb{Z}_4/\mathbb{Z}_2^V}$, leading respectively to simple 1-endomorphisms

$$D_1^{(S)}\,, \qquad D_1^{(C)}\,, \tag{C.40}$$

of $D_2^{(\mathrm{id})} \in \mathcal{C}_{\mathbb{Z}_4/\mathbb{Z}_4}$.

We can quickly compute the composition rules of these 1-morphisms. For example, since

$$iD_0^{(V;\mathrm{id})} \otimes iD_0^{(V;\mathrm{id})} = -D_0^{(\mathrm{id})}\,, \tag{C.41}$$

in $\mathcal{C}_{\mathbb{Z}_4/\mathbb{Z}_2^V}$, we learn that

$$D_1^{(S)} \circ D_1^{(S)} \cong D_1^{(V)}\,, \tag{C.42}$$

in $\mathcal{C}_{\mathbb{Z}_4/\mathbb{Z}_4}$. Similarly, since

$$-D_0^{(\mathrm{id})} \otimes -D_0^{(\mathrm{id})} = D_0^{(\mathrm{id})}\,, \tag{C.43}$$

in $\mathcal{C}_{\mathbb{Z}_4/\mathbb{Z}_2^V}$, we learn that

$$D_1^{(V)} \circ D_1^{(V)} \cong D_1^{(\mathrm{id})}\,, \tag{C.44}$$

in $\mathcal{C}_{\mathbb{Z}_4/\mathbb{Z}_4}$. Due to these reasons, we find that $D_1^{(S)} \in \mathcal{C}_{\mathbb{Z}_4/\mathbb{Z}_4}$ generates a $\mathbb{Z}_4$ 1-form symmetry, with the $\mathbb{Z}_2$ element being $D_1^{(V)} \in \mathcal{C}_{\mathbb{Z}_4/\mathbb{Z}_4}$.

The simple 1-morphisms

$$D_1^{(\mathrm{id},\mathbb{Z}_2;+)}\,, \qquad D_1^{(\mathrm{id},\mathbb{Z}_2;-)}\,, \tag{C.45}$$

from $D_2^{(\mathrm{id})} \in \mathcal{C}_{\mathbb{Z}_4/\mathbb{Z}_4}$ to $D_2^{(\mathbb{Z}_2)} \in \mathcal{C}_{\mathbb{Z}_4/\mathbb{Z}_4}$ arise respectively from 1-morphisms from $D_2^{(\mathrm{id})} \in \mathcal{C}_{\mathbb{Z}_4/\mathbb{Z}_2^V}$ to $2D_2^{(\mathrm{id})} \in \mathcal{C}_{\mathbb{Z}_4/\mathbb{Z}_2^V}$ given by row vectors

$$\begin{pmatrix} D_1^{(\mathrm{id})} & D_1^{(\mathrm{id})} \end{pmatrix}\,, \qquad \begin{pmatrix} D_1^{(V)} & D_1^{(V)} \end{pmatrix}\,. \tag{C.46}$$

Similarly, simple 1-morphisms

$$D_1^{(\mathbb{Z}_2,\text{id};+)}, \qquad D_1^{(\mathbb{Z}_2,\text{id};-)}, \tag{C.47}$$

from $D_2^{(\mathbb{Z}_2)} \in \mathcal{C}_{\mathbb{Z}_4/\mathbb{Z}_4}$ to $D_2^{(\text{id})} \in \mathcal{C}_{\mathbb{Z}_4/\mathbb{Z}_4}$ arise respectively from 1-morphisms from $2D_2^{(\text{id})} \in \mathcal{C}_{\mathbb{Z}_4/\mathbb{Z}_2^V}$ to $D_2^{(\text{id})} \in \mathcal{C}_{\mathbb{Z}_4/\mathbb{Z}_2^V}$ given by column vectors

$$\begin{pmatrix} D_1^{(\text{id})} \\ D_1^{(\text{id})} \end{pmatrix}, \qquad \begin{pmatrix} D_1^{(V)} \\ D_1^{(V)} \end{pmatrix}. \tag{C.48}$$

The 1-endomorphisms

$$D_1^{(\mathbb{Z}_2;++)}, \qquad D_1^{(\mathbb{Z}_2;+-)}, \qquad D_1^{(\mathbb{Z}_2;-+)}, \qquad D_1^{(\mathbb{Z}_2;--)}, \tag{C.49}$$

of $D_2^{(\mathbb{Z}_2)} \in \mathcal{C}_{\mathbb{Z}_4/\mathbb{Z}_4}$ arise respectively from 1-endomorphisms of $2D_2^{(\text{id})} \in \mathcal{C}_{\mathbb{Z}_4/\mathbb{Z}_2^V}$ given by $2 \times 2$ matrices

$$\begin{pmatrix} D_1^{(\text{id})} & 0 \\ 0 & D_1^{(\text{id})} \end{pmatrix}, \qquad \begin{pmatrix} 0 & D_1^{(\text{id})} \\ D_1^{(\text{id})} & 0 \end{pmatrix}, \qquad \begin{pmatrix} D_1^{(V)} & 0 \\ 0 & D_1^{(V)} \end{pmatrix}, \qquad \begin{pmatrix} 0 & D_1^{(V)} \\ D_1^{(V)} & 0 \end{pmatrix}. \tag{C.50}$$

There is a single simple 1-morphism

$$D_1^{(\text{id},\mathbb{Z}_4)}, \tag{C.51}$$

from $D_2^{(\text{id})} \in \mathcal{C}_{\mathbb{Z}_4/\mathbb{Z}_4}$ to $D_2^{(\mathbb{Z}_4)} \in \mathcal{C}_{\mathbb{Z}_4/\mathbb{Z}_4}$ coming from the 1-morphism from $D_2^{(\text{id})} \in \mathcal{C}_{\mathbb{Z}_4/\mathbb{Z}_2^V}$ to $2D_2^{(\mathbb{Z}_2^V)} \in \mathcal{C}_{\mathbb{Z}_4/\mathbb{Z}_2^V}$ described by the row vector

$$\begin{pmatrix} D_1^{(\text{id},\mathbb{Z}_2^V)} & D_1^{(\text{id},\mathbb{Z}_2^V)} \end{pmatrix}. \tag{C.52}$$

Similarly, there is a single simple 1-morphism

$$D_1^{(\mathbb{Z}_4,\text{id})}, \tag{C.53}$$

from $D_2^{(\mathbb{Z}_4)} \in \mathcal{C}_{\mathbb{Z}_4/\mathbb{Z}_4}$ to $D_2^{(\text{id})} \in \mathcal{C}_{\mathbb{Z}_4/\mathbb{Z}_4}$ coming from the 1-morphism from $2D_2^{(\mathbb{Z}_2^V)} \in \mathcal{C}_{\mathbb{Z}_4/\mathbb{Z}_2^V}$ to $D_2^{(\text{id})} \in \mathcal{C}_{\mathbb{Z}_4/\mathbb{Z}_2^V}$ described by the column vector

$$\begin{pmatrix} D_1^{(\mathbb{Z}_2^V,\text{id})} \\ D_1^{(\mathbb{Z}_2^V,\text{id})} \end{pmatrix}. \tag{C.54}$$

There are two simple 1-morphisms

$$D_1^{(\mathbb{Z}_2,\mathbb{Z}_4;+)}, \qquad D_1^{(\mathbb{Z}_2,\mathbb{Z}_4;+)}, \tag{C.55}$$

from $D_2^{(\mathbb{Z}_2)} \in \mathcal{C}_{\mathbb{Z}_4/\mathbb{Z}_4}$ to $D_2^{(\mathbb{Z}_4)} \in \mathcal{C}_{\mathbb{Z}_4/\mathbb{Z}_4}$ coming from the 1-morphisms from $2D_2^{(\text{id})} \in \mathcal{C}_{\mathbb{Z}_4/\mathbb{Z}_2^V}$ to $2D_2^{(\mathbb{Z}_2^V)} \in \mathcal{C}_{\mathbb{Z}_4/\mathbb{Z}_2^V}$ described respectively by the $2 \times 2$ matrices

$$\begin{pmatrix} D_1^{(\text{id},\mathbb{Z}_2^V)} & 0 \\ 0 & D_1^{(\text{id},\mathbb{Z}_2^V)} \end{pmatrix}, \qquad \begin{pmatrix} 0 & D_1^{(\text{id},\mathbb{Z}_2^V)} \\ D_1^{(\text{id},\mathbb{Z}_2^V)} & 0 \end{pmatrix}. \tag{C.56}$$

Similarly, there are two simple 1-morphisms

$$D_1^{(\mathbb{Z}_4,\mathbb{Z}_2;+)}, \qquad D_1^{(\mathbb{Z}_4,\mathbb{Z}_2;-)}, \tag{C.57}$$

from $D_2^{(\mathbb{Z}_4)} \in \mathcal{C}_{\mathbb{Z}_4/\mathbb{Z}_4}$ to $D_2^{(\mathbb{Z}_2)} \in \mathcal{C}_{\mathbb{Z}_4/\mathbb{Z}_4}$ coming from the 1-morphisms from $2D_2^{(\mathbb{Z}_2^V)} \in \mathcal{C}_{\mathbb{Z}_4/\mathbb{Z}_2^V}$ to $2D_2^{(\text{id})} \in \mathcal{C}_{\mathbb{Z}_4/\mathbb{Z}_2^V}$ described respectively by the $2 \times 2$ matrices

$$\begin{pmatrix} D_1^{(\mathbb{Z}_2^V,\text{id})} & 0 \\ 0 & D_1^{(\mathbb{Z}_2^V,\text{id})} \end{pmatrix}, \qquad \begin{pmatrix} 0 & D_1^{(\mathbb{Z}_2^V,\text{id})} \\ D_1^{(\mathbb{Z}_2^V,\text{id})} & 0 \end{pmatrix}. \tag{C.58}$$

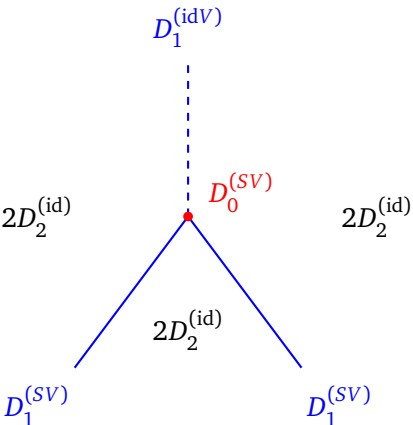

Figure 26: $D_0^{SV}$ is a choice of local operator (shown in red) lying at the junction from $D_1^{(SV)} \otimes_{2D_2^{(\text{id})}} D_1^{(SV)}$ (shown in solid blue) to $D_1^{(\text{id}V)}$ (shown in dashed blue). All these lines and local operators live on the surface $2D_2^{(\text{id})}$.

## C.3 A subtle isomorphism

In this subsection we exhibit an explicit isomorphism between the two objects $D_2^{(\mathbb{Z}_2^V)}$ and $D_2^{(\mathbb{Z}_2^{V'})}$ in the 2-category $\mathcal{C}_{\mathbb{Z}_4/\mathbb{Z}_4}$ found using sequential $\mathbb{Z}_2^S$ gauging of the 2-category $\mathcal{C}_{\mathbb{Z}_4/\mathbb{Z}_2}$ in section 3.1.2.

Before exhibiting the isomorphism, let us first concretely describe the two objects. Both arise by making $2D_2^{(\text{id})} \in \mathcal{C}_{\mathbb{Z}_4/\mathbb{Z}_2}$ symmetric under $\mathbb{Z}_2^S$. First of all, we can implement the $\mathbb{Z}_2^S$ symmetry using the matrix

$$D_1^{(S\text{id})} := \begin{pmatrix} 0 & D_1^{(\text{id})} \\ D_1^{(\text{id})} & 0 \end{pmatrix}, \tag{C.59}$$

leading to the object $D_2^{(\mathbb{Z}_2^V)} \in \mathcal{C}_{\mathbb{Z}_4/\mathbb{Z}_4}$. If we instead implement the $\mathbb{Z}_2^S$ symmetry using the matrix

$$D_1^{(SV)} := \begin{pmatrix} 0 & D_1^{(V)} \\ D_1^{(V)} & 0 \end{pmatrix}, \tag{C.60}$$

then this $\mathbb{Z}_2^S$ symmetry carries a non-trivial anomaly. We can remove the anomaly by choosing more refined information regarding how the $\mathbb{Z}_2^S$ symmetry is implemented on $2D_2^{(\text{id})}$. This refined information is the choice of a 2-morphism

$$D_0^{(SV)}: \ D_1^{(SV)} \otimes D_1^{(SV)} \to D_1^{(\text{id}V)} := \begin{pmatrix} D_1^{(\text{id})} & 0 \\ 0 & D_1^{(\text{id})} \end{pmatrix}, \tag{C.61}$$

which is the choice of a local operator converting two lines implementing the generator of $\mathbb{Z}_2^S$ to the identity line implementing the identity element of $\mathbb{Z}_2^S$. See figure 26. $D_0^{(SV)}$ is a tuple $(D_0^{(SV)(1)}, D_0^{(SV)(2)})$ where

$$\begin{aligned} D_0^{(SV)(1)}: \ & D_1^{(SV)(12)} \otimes D_1^{(SV)(21)} \to D_1^{(\text{id}V)(11)}, \\ D_0^{(SV)(2)}: \ & D_1^{(SV)(21)} \otimes D_1^{(SV)(12)} \to D_1^{(\text{id}V)(22)}, \end{aligned} \tag{C.62}$$

where $D_1^{(SV)(ij)}$ is the element in row $i$ and column $j$ of the matrix $D_1^{(SV)}$, and $D_1^{(\text{id}V)(ii)}$ is the element in row $i$ and column $i$ of the matrix $D_1^{(\text{id}V)}$. Let us choose $D_0^{(SV)(1)}$ to be $D_0^{(\text{id})}$ as a 2-endomorphism of $D_1^{(\text{id})}$, and $D_0^{(SV)(2)}$ to be $-D_0^{(\text{id})}$ as a 2-endomorphism of $D_1^{(\text{id})}$. This relative

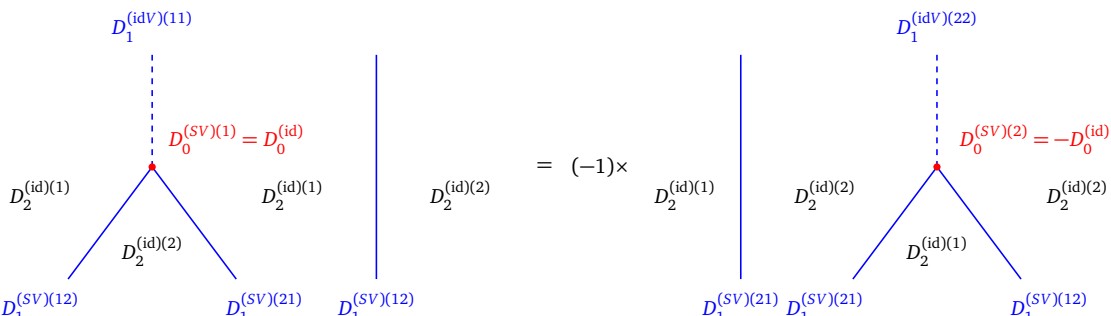

Figure 27: The figure performs an associator move involving the line $D_1^{(SV)}$ (whose components are the lines $D_1^{(SV)(ij)}$), the line $D_1^{(idV)}$ (whose components are the lines $D_1^{(idV)(ii)}$) and the operator $D_0^{(SV)}$ (whose components are the operators $D_0^{(SV)(i)}$). On the right hand side, we obtain an overall minus sign from the non-trivial associator/anomaly involving only the $D_1^{(SV)}$ and $D_1^{(idV)}$ lines, but this sign is canceled by the extra relative sign between the definitions of $D_0^{(SV)(1)}$ and $D_0^{(SV)(2)}$. Thus the total combined associator does not produce any phase and hence the diagrams on the left and right are equal. $D_2^{(id)(i)}$ denotes the different components inside the surface $2D_2^{(id)}$.

choice of sign between $D_0^{(SV)(1)}$ and $D_0^{(SV)(2)}$ cancels the anomaly as shown in figure 27, and we have managed to restore non-anomalous $\mathbb{Z}_2^S$ symmetry, thus obtaining an object in $\mathcal{C}_{\mathbb{Z}_4/\mathbb{Z}_4}$ that we label as

$$D_2^{(\mathbb{Z}_2^{V'})}. \tag{C.63}$$

An alternate way to state the above computation is that we have canceled the $\mathbb{Z}_2^S$ anomaly by stacking with a projective 2-representation of dimension 2, namely a 2d TQFT with two vacua and an anomalous $\mathbb{Z}_2^S$ symmetry, which exchanges the two vacua.

Now let us describe the isomorphism. At the level of lines, the isomorphism is implemented by the matrix

$$I_1 := \begin{pmatrix} 0 & D_1^{(id)} \\ D_1^{(V)} & 0 \end{pmatrix}, \tag{C.64}$$

which intertwines the corresponding lines implementing $\mathbb{Z}_2^S$ symmetry on $D_2^{(\mathbb{Z}_2^{V'})}$ and $D_2^{(\mathbb{Z}_2^V)}$

$$\begin{pmatrix} 0 & D_1^{(id)} \\ D_1^{(V)} & 0 \end{pmatrix}\begin{pmatrix} 0 & D_1^{(V)} \\ D_1^{(V)} & 0 \end{pmatrix} = \begin{pmatrix} 0 & D_1^{(id)} \\ D_1^{(id)} & 0 \end{pmatrix}\begin{pmatrix} 0 & D_1^{(id)} \\ D_1^{(V)} & 0 \end{pmatrix}. \tag{C.65}$$

More precisely we have to specify local operators

$$\begin{aligned} I_0^{(1)} &: I_1^{(12)} \otimes D_1^{(SV)(21)} \to D_1^{(Sid)(12)} \otimes I_1^{(21)}, \\ I_0^{(2)} &: I_1^{(21)} \otimes D_1^{(SV)(12)} \to D_1^{(Sid)(21)} \otimes I_1^{(12)}. \end{aligned} \tag{C.66}$$

Let us choose $I_0^{(1)}$ to be $\beta_1 D_0^{(V;id)}$ where $D_0^{(V;id)}$ is the identity local operator living on $D_1^{(V)}$, and $I_0^{(2)}$ to be $\beta_2 D_0^{(id)}$. For this to be an isomorphism, $I_0^{(1)}$ and $I_0^{(2)}$ should satisfy the two relations shown in figure 28. Naively one of the relations seems to lead to $\beta_1\beta_2 = 1$ while the other relation seems to lead to $\beta_1\beta_2 = -1$, invalidating the existence of isomorphism. However, since we are considering configurations of $D_1^{(V)}$ lines and $D_2^{(S)}$ surfaces, we should take into account symmetry fractionalization, which corrects the first relation to $\beta_1\beta_2 = -1$

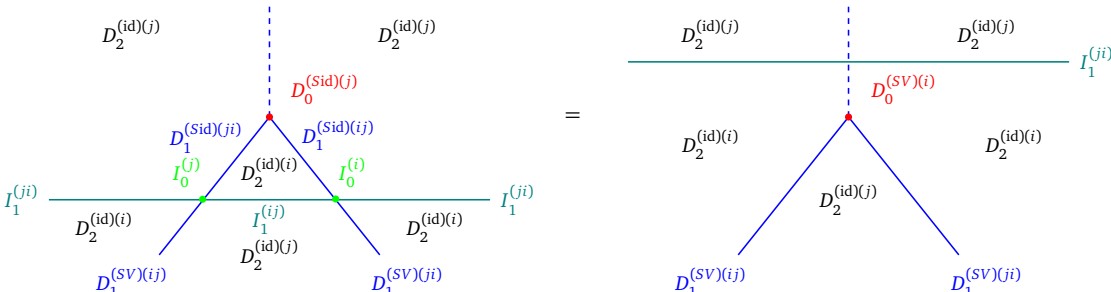

Figure 28: Relations satisfied by isomorphism lines and operators. Here $D_0^{(\text{Sid})(j)} = D_0^{(\text{id})}$. The figure specifies two relations because we have the identification $i = j + 1 \pmod{2}$.

while leaving the second relation unchanged. Thus, picking $\beta_1$ and $\beta_2$ satisfying the relation $\beta_1 \beta_2 = -1$ provides an isomorphism

$$D_2^{(\mathbb{Z}_2)} \cong D_2^{(\mathbb{Z}_2')}. \tag{C.67}$$

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
