# Peer review of "Non-Invertible Symmetry Webs"

_SciPost Physics, doi:SciPost Phys. 15, 160 (2023)_

## Round 1 · Referee Report · Anonymous · 2023-7-14

Report

This paper studies webs of theories with non-invertible symmetries mainly in 3d. There is also a section explaining how the method can be generalized to higher dimensions.

The main idea was to start with a non-anomalous, invertible, symmetry given by the fusion 2-category 2-Vec(G), and perform sequential partial gauging of all possible sub-symmetries. The authors found that the sequential gauging is consistent with gauging 2-Vec(G) in one go. Along the way, many interesting features in the intermediate steps are uncovered, most notably symmetry fractionalization on lines and condensation defects.

One strength of this work is that it contains plenty of examples, which the referee found quite helpful. However, the style of the presentation makes the paper not very readable. There are way too many notations, superscripts, subscripts, with or without parentheses, etc.

Although the paper discussed a very systematic construction of non-invertible symmetries, it seems that the authors still miss some important possibilities:

1. The authors did not discuss the possibility of including discrete theta terms when they gauge 0-form symmetries. For example, including them into the D8-web produces the Pin^- theories, as discussed in (fig 3 of) https://arxiv.org/pdf/1711.10008.pdf. (By the way, please also consider citing this work as it already contains many results regarding the \mathfrak{so} theories revisited by the current paper.)

2. The authors did not discuss the possibility of gauging non-invertible co-dimension 1 defects (even determining the whether they are gaugable is unclear to the referee).

It would be useful to make these points clear: either by making some comments on these points, or by saying that they are beyond the scope of this work and will be left for future work.

Moreover, the paper spent over 100 pages exploring the huge webs of symmetries, and it will be a pity that the paper does not contain possible applications on these symmetries. Do these new symmetries tell us anything about the dynamics of Yang-Mills theories in the webs unknown before? The referee thinks it would be very interesting (and necessary) to at least include a subsection on the physical implications of these symmetries.

Finally, two extremely minor typos:
1. In Table 1: PSU should be \mathrm{PSU}.
2. The notations in eq (2.21) and eq (2.27) conflict. The latter have superscript V while the former doesn’t.

Overall, I would recommend for publication, once the comments are addressed.

---

## Round 1 · Referee Report · Anonymous · 2023-7-15

Report

The manuscript discussed many examples of gauging symmetry 2Vec G, where various non-invertible symmetries/ higher group can occur in various cases.

I recommend publication if the following comments are taken into account (as well as the comments in the first referee report):

- As mentioned in the referee report 1, it would be helpful to have applications such as any observable physical consequence that would be hard to obtain without using the method in the manuscript.

- The manuscript discussed symmetry fractionalizations, which has been extensively studied in the condensed matter literature, e.g. the earliest ones include the ENO paper https://arxiv.org/abs/0909.3140 and https://arxiv.org/abs/1410.4540
What is the relation between the discussion in the manuscript and the literature?

- In the manuscript, the fractionalization classes are related to group extension and described H^2. However, there are more general fractionalization classes when the theories have two form or higher symmetries such as O(N) theory in 3+1d, as described by H^* of higher degree. Do they give more examples?

- a 't Hooft anomaly -> an 't Hooft anomaly

---

## Round 2 · Referee Report · Anonymous · 2023-8-15

Report

The authors have addressed the comments appropriately. I'd like to recommend for publication as it is.

---

## Round 2 · Referee Report · Anonymous · 2023-8-28

Report

The authors have addressed the comments.

---

## Round 2 · Author Response

We thank the referees for their careful reading of our paper and the comments and suggestions. We have implemented the following changes to the manuscript following the referees suggestions:

1) Referee 1 and 2 point out that we have not included any applications at this point. This is true, however the goal of this paper is to first describe the structure of the symmetry and their gauging, and how these are manifest in the context of fusion 2-categories. Applications will appear in subsequent works (e.g. they also have appeared already in the context of lattice versions of these precise constructions in [see for e.g., arXiv:2301.01259 and arXiv:2307.01266] and we currently have some on-going work in this direction).

2) Referee 1's comments about a) adding discrete theta and citation to https://arxiv.org/pdf/1711.10008.pdf is addressed in the introduction.

3) Regarding both referees comments related to connections to the condensed matter literature---we have now added a comment as well as several references in the introduction. While the phenomenon of symmetry fractionalization has been extensively studied in the condensed matter literature, its fusion 2-categorical aspects as well as the general mechanism of how it arises via gauging invertible symmetries has been less appreciated.

4) We have added a footnote in the introduction stating that we plan to develop the gauging of non-invertible co-dimension 1 defects, which typically arise in the categorical symmetry webs as condensation defects or from the gauging of non-normal subgroups, in future works.

5) It is true that gauging non group extensions of 0-form symmetries would yield new examples. For instance gauging 1-form symmetries in 4d or gauging a 1-form symmetry that participates in a 2-group with a non-trivial Postnikov class would deliver symmetry fractionalization patterns controlled by certain H^3 elements. Both these kinds of examples do not appear in the categorical symmetry webs we used to exemplify the approach in this work. However general computations carried out in this work can be used to study such symmetry fractionalizations.

6) We thank the referees for pointing out the typos. These have now been fixed.

---

## Editorial Decision

published